# Continental-scale drivers of soil microbial extracellular polymeric substances

Ke Shi [1,2,3,4], Qing Zheng [2], Baorong Wang [5], Lisa Noll [2], Shasha Zhang[2], Yuntao Hu[2], Honghua Ruan [1] ✉ & Wolfgang Wanek [2] ✉

Extracellular polymeric substances (EPS) are key microbial residues that contribute to soil organic carbon (SOC) and promote soil aggregation. Yet, their abundance and large-scale controls have only begun to be investigated. We conduct extensive soil sampling across a European transect spanning diverse climates, bedrocks, and land uses. Average soil EPS content is $956 \pm 55 \, \mu g \, g^{-1}$ soil ($n = 92$ sites), with EPS-carbon (EPS-C) contributing $1.6 \pm 0.1\%$ to SOC. Bedrock influences EPS content, EPS-C contribution to SOC, and the EPS-C/ microbial biomass carbon (MBC) ratio, whereas land use mainly affects the latter two. The EPS-C/MBC ratio is negatively correlated with microbial growth and carbon use efficiency, and increases under water deficit, while EPS increases with MBC, clay content, and exchangeable calcium. Our results demonstrate that EPS represents a functionally important microbial residue, regulated by climatic, edaphic, microbial, and land-use factors, with significant implications for soil carbon cycling and sequestration.

Soil microbial extracellular polymeric substances (EPS) are essential components of the soil carbon (C) pool[1,2]. In contrast to EPS, microbial necromass mainly consists of microbial cell wall fragments and is widely quantified using amino sugars as biomarkers[3]. They have been extensively investigated, are reasonably well understood[3,4], and contribute approximately 30–60% of the total soil organic carbon (SOC)[3]. However, EPS remains a largely overlooked C pool at regional scales and may play important functional roles in soil aggregation, microbial habitat formation, and short-term C dynamics[1,5]. Its relationships with geological, land use, climatic, edaphic, plant, and microbial factors are not yet fully understood. EPS comprises a heterogeneous mixture of biopolymers (exoproteins, exopolysaccharides) targeted for their extracellular role and secreted by microorganisms that support them to resist environmental stresses and absorb nutrients and other resources[5,6]. The primary components of EPS are polysaccharides (exopolysaccharides) and proteins (exoproteins)[1,6,7]. EPS polysaccharides are composed of neutral sugar, amino sugar, and sugar acid monomers, are highly viscous, and therefore facilitate the attachment of cells to the surfaces of soil particles[8,9]. EPS proteins are comprised of proteinogenic amino acid monomers that perform more complex functions, such as providing structural and enzymatic functions and enabling cell-to-cell communications in response to environmental signals[10]. EPS polysaccharides are important components of biofilms, the strength and structural diversity of which are promoted by EPS proteins. Together, EPS polysaccharides and EPS proteins eventually embed microbial cells and consortia to form millimeter-thick biofilms in aquatic systems, while the extent of biofilm formation in soils is still debated[8,11,12]. Independent of the presence of soil biofilms, EPS plays multifunctional roles in soil ecosystems[1,13]. EPS enhances water retention, mediates nutrient storage, and supports microbial resilience under environmental stress[1]. Beyond favorable effects of EPS on microbial performance, EPS contributes to SOC dynamics by serving as a microbial-derived direct C input and by promoting soil aggregation and mineral-organic associations, which are critical processes for physical and chemical stabilization of SOC[6,13]. Furthermore, EPS facilitates microbial attachment to soil particles,

[1]Department of Ecology, Co-Innovation Center for Sustainable Forestry in Southern China, Nanjing Forestry University, Nanjing, China. [2]Division of Terrestrial Ecosystem Research, Department of Microbiology and Ecosystem Science, Center of Microbiology and Environmental Systems Science, University of Vienna, Vienna, Austria. [3]German Centre for Integrative Biodiversity Research (iDiv) Halle-Jena-Leipzig, Leipzig, Germany. [4]Institute of Biology, Leipzig University, Leipzig, Germany. [5]State Key Laboratory of Soil and Water Conservation and Desertification Control, College of Grassland Agriculture, Northwest A&F University, Yangling, Shaanxi, China. ✉e-mail: hhruan@njfu.edu.cn; wolfgang.wanek@univie.ac.at

potentially enhancing the association between microbial necromass and mineral surfaces following microbial death. Such interactions between EPS and other microbial residues may accelerate soil microbial-derived C stabilization and thus represent an additional, underexplored pathway through which EPS can promote long-term C storage. These diverse functions underscore the ecological relevance of EPS in structuring soils and regulating C cycling.

Given the potentially outstanding role of EPS for microbial function and SOC dynamics, it is interesting to note that quantitative measurements of soil EPS contents remain limited, and the lack of accepted EPS carbon (EPS-C) conversion factors leaves its contribution to SOC largely unknown. So far, the climatic and geological coverage of soil EPS measurements was relatively restricted, with studies confined to smaller-scale and/or local mechanistic questions, e.g., targeting soil EPS responses to current and legacy land use[14], the interaction of land use and geology[15], land reclamation[16], tree plantation age[17,18], agricultural management[19,20], and root-soil aggregate interactions in forests and grasslands[21–23]. Therefore, studies on large-scale patterns and controls of EPS dynamics, including their effects on microbial necromass and SOC stabilization, are much needed.

The content of soil EPS may depend on (i) the microbial secretion of EPS and (ii) its association with soil particles[6,8,13], as well as (iii) its decomposition and recycling. The secretion of EPS is typically stimulated under specific environmental conditions[1]. For instance, in soils rich in labile C, microorganisms tend to stimulate EPS production, which serves both as a structural matrix for effective application of extracellular enzymes and as a potential C resource for subsequent metabolism[1,15,24]. Moreover, when subjected to environmental stresses such as drought, high salinity, pH extremes, and C or nutrient limitations, microorganisms secrete EPS to retain water[25–27], to provide ion and proton buffering, and to acquire nutrients, increasing the metabolite return on investment for the exoenzymes secreted[28]. Finally, as microbial communities expand, EPS secretion enhances cell adhesion on soil particles and helps protect against microbial competition[5,15]. Collectively, these findings underscore that EPS secretion is governed by microbial growth and environmental stress[1], soil properties, nutrient availability[25,26,28,29], and population-level survival strategies[15].

Additionally, the content of soil EPS is influenced by its interactions with soil particles, which offer protection against degradation[30], potentially affecting its contribution to the SOC pool. Soil mineral composition and particle size distribution can therefore affect how EPS associates with soil particles and aggregates. For instance, silt- and clay-rich soils can promote these associations, potentially reducing EPS accessibility to decomposition[11]. Furthermore, soil minerals such as Fe- and Al-oxyhydroxides and exchangeable Ca and Mg interact with EPS functional groups[31,32], promoting the association of EPS with soil particles. In conclusion, EPS secretion, its binding to soil particles, and its decomposition and recycling together determine the contribution of EPS to the SOC pool.

Various hypotheses have been proposed to explain the environmental controls on microbial EPS secretion and soil EPS content, but most have been tested based on laboratory or small-scale field studies. For example, Redmile-Gordon et al.[14] showed that EPS acts as a transient binding agent, with protein levels higher in grasslands than in arable or fallow soils, likely reflecting land-use-driven shifts in microbial EPS secretion. However, forest soils—characterized by distinct microbial communities—were not included. Beyond microbial production, soil EPS content also depends on its interaction with soil minerals[30], shaped by texture and mineralogy derived from bedrock. Soils formed from different bedrocks thus may vary in their capacity to bind with EPS[13], though recently land use was shown to be the more important effector of soil EPS content than bedrock geochemistry in a tropical wet climate[15]. Despite increasing interest, no field study has systematically assessed the combined effects of land use and bedrock across varying climates on soil EPS. Large-scale investigations spanning forests, grasslands, and croplands on contrasting bedrock types across a broad climate gradient are therefore critical to disentangle the ecological and environmental influences on soil EPS-C pools.

Here, we conducted large-scale soil sampling on a transect that spanned ~5500 km across the entire European continent, quantified EPS in soil samples from 92 different sites, and compiled a broad range of related environmental and biotic parameters (i.e., climate, plant, soil, and microbial factors). Thereby, this study addressed three major research questions: (1) What is the distribution of EPS content in soils along a European climate transect, and how do land use and bedrock types influence this distribution? (2) What is the contribution of EPS-C to the SOC pool, and what causes potential differences along a European climate transect? (3) Finally, how does the microbial EPS-C/MBC ratio change on a European climate transect, and how do environmental conditions such as C and nutrient availability, pH, and soil water deficit affect this ratio? By exploring these questions, we aim to gain a deeper understanding of the role of EPS across different soil ecosystems and its influence on soil biogeochemical processes.

## Results
### Soil EPS dynamics and drivers across bedrock and land-use types

Across the European transect, total soil EPS content (sum of EPS polysaccharides and EPS proteins) ranged from 149 to 2495 $\mu g\,g^{-1}$ soil, with a grand mean of $956 \pm 55\,\mu g\,g^{-1}$ soil (SE; $n = 92$), higher than previously reported values (mean: $423\,\mu g\,g^{-1}$ soil) (Figs. 1, 2; Supplementary Table 1; Supplementary Data 1). Bedrock type strongly influenced total EPS content, while land use had no significant effect (Fig. 2a, b; Supplementary Table 2). EPS content was significantly higher in soils on carbonate than on silicate and sediment bedrock, albeit the latter two did not differ significantly (Fig. 2a). Although not statistically significant, total EPS content appeared higher in grasslands than in croplands and woodlands (Fig. 2b). To identify the major environmental and microbial drivers of soil EPS at this large scale, correlation analyses were performed, showing that EPS content was influenced by multiple factors, most strongly by soil physicochemical properties and microbial parameters (Fig. 2c).

Soil EPS content increased with mean annual precipitation (MAP), and showed weaker positive correlations with aridity index (ADI) and elevation, but was not significantly related to latitude, longitude, mean annual temperature (MAT), and potential evapotranspiration (PET) (Fig. 2c). Among plant parameters, EPS content increased with root nitrogen (N), weakly with fine root biomass (FRB), and decreased with root C/N (Fig. 2c). Among the soil properties, EPS content was strongly related to soil C and nutrients, showing a strong positive correlation with soil total nitrogen (TN), positive correlations with SOC, total phosphorus (TP), total organic phosphorus (TOP), total dissolved nitrogen (TDN), exchangeable $Ca^{2+}$ ($Ca_e$), exchangeable $Mg^{2+}$ ($Mg_e$) and weak positive correlations with nitrate ($NO_3^-$) and soil N/phosphorus (P) ratio, while there were no significant correlations with the soil C/N and C/P ratios, and dissolved organic carbon (DOC). Additionally, EPS content was not significantly correlated with soil pH (Supplementary Fig. 3), while it showed positive correlations with soil water holding capacity (WHC) and soil moisture content (SMC). Soil texture, minerals, and cation exchange capacity (CEC) also had strong effects on EPS content. The content of EPS increased with clay and silt content, was positively correlated with soil CEC, but showed no significant relationships with Fe and Al oxyhydroxides, and decreased with sand content (Fig. 2c). Among microbial parameters, EPS content increased with microbial biomass – including MBC, nitrogen (MBN), and phosphorus (MBP) – and decreased with the MBC/MBN ratio. EPS was positively correlated with microbial enzyme activity (exoglucanase-cellobiosidase (CEL), β-glucosidase (BG), and enzyme vector length (VectorL)) and with microbial biomass turnover time (MBT), strongly

negatively correlated with microbial growth normalized to MBC ($q_{Growth}$), and weakly negatively correlated with microbial carbon use efficiency (CUE). Other microbial parameters potentially linked to EPS formation, including nitrogen use efficiency (NUE), leucine aminopeptidase (LAP), and most PLFA-based microbial community composition metrics, were not significantly related to EPS (Fig. 2c). Supplementary Tables 8–10 provide additional results from linear models and linear mixed-effects models assessing the influence of bedrock, land-use types, environmental, and microbial factors on total EPS.

### Drivers of specific soil EPS components

As the primary components of EPS, we investigated whether environmental and microbial drivers of EPS polysaccharides and EPS proteins followed congruent or different patterns. Statistical analysis revealed that bedrock and land use had distinct influences on EPS polysaccharides and EPS proteins across the European transect (Fig. 3; Supplementary Table 2). Specifically, EPS polysaccharides varied significantly with bedrock types (carbonate>silicate/sediment; Fig. 3a)

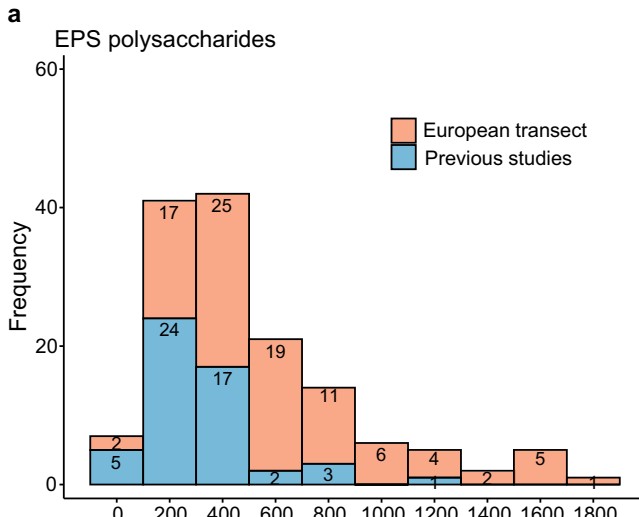

**a**

**EPS polysaccharides**

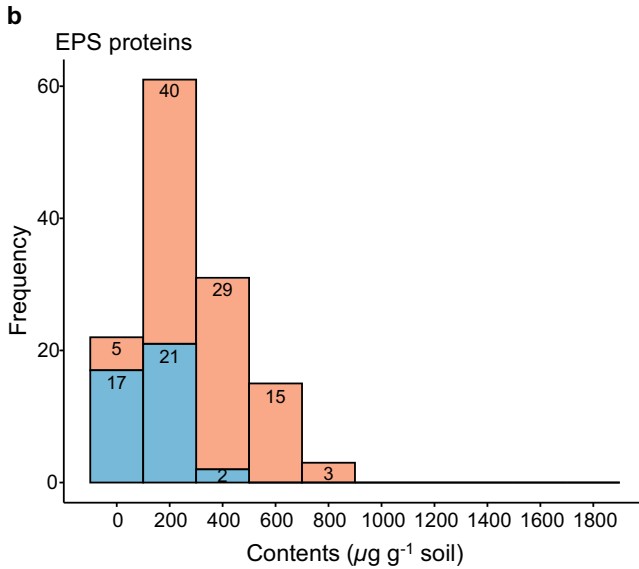

**b**

**EPS proteins**

**Fig. 1 | Comparison of extracellular polymeric substance (EPS) polysaccharides and EPS proteins across the European transect with data derived from previous studies. a** Comparison of EPS polysaccharide contents. **b** Comparison of EPS protein contents. Specific data are listed in Supplementary Table 1, and the complete dataset is provided in Supplementary Data 2.

but not with land use (Fig. 3b). Conversely, EPS proteins differed significantly across land use types (grassland > woodland/cropland; Fig. 3d), while showing no significant variation with bedrock (Fig. 3c).

Principal component analysis (PCA) and random forest analyses revealed broad similarities in environmental and microbial drivers of the two EPS components, i.e., microbial factors (MBN, MBC, $q_{Growth}$, VectorL, CEL, BG, and bacteria to fungi ratio (B/F)), plant traits (FRB and root C/N), climate (ADI), and soil nutrients ($Ca_e$, SOC, TP, and soil C/N) and clay (Fig. 4a). Both EPS compounds were strongly related to soil WHC (Fig. 4a). Notably, $Ca_e$, SOC, and MBN emerged as the most critical factors that significantly influenced EPS polysaccharides (Fig. 4b), while MBN, SOC ($p = 0.059$), and $q_{Growth}$ were the most important drivers of EPS proteins (Fig. 4c). However, we also found distinct patterns among driving factors between EPS components: soil clay content and root C/N influenced EPS polysaccharides, while FRB and B/F affected EPS proteins (Fig. 4b, c). Supplementary Tables 11–16 provide additional results from linear models and linear mixed-effects models assessing the influence of bedrock, land-use types, environmental, and microbial factors on EPS polysaccharides and EPS proteins.

### Contribution of EPS-C and microbial necromass to SOC

Across the European transect, soil EPS-C content ranged from 0.06 – 1.07 g C kg$^{-1}$ soil, with an average of 0.41 ± 0.02 g C kg$^{-1}$ soil (SE; $n = 92$) (Fig. 5a, b). Notably, EPS-C contributed 0.3–3.9% to the SOC, with average contributions of 1.6 ± 0.1% (Fig. 5c, d). The contributions of EPS-C to SOC varied significantly across bedrock and land use types (Fig. 5c, d; Supplementary Table 2). Across bedrock types, EPS-C contributions followed the pattern: carbonate/sediment > silicate (Fig. 5c). In terms of land use, EPS-C contributed 2.0% to SOC in croplands and grasslands, which was significantly higher than the 1.2% contribution in woodlands (Fig. 5d).

Furthermore, we found that across the European transect, microbial necromass carbon (MNC) contributed 20.9% to SOC, with bacterial necromass carbon (BNC) contributing 11.5%, and fungal necromass carbon (FNC) contributing 9.4% (Supplementary Fig. 7). The contribution of MNC to SOC was therefore nearly ten times that of EPS-C. In addition, notably strong positive correlations were observed among SOC, MBC, MNC, and EPS-C (Fig. 6).

### Dynamics and drivers of the EPS-C/MBC ratio

Across the European transect, we observed significant variations in EPS-C/MBC ratios among soils from different bedrock and land-use types (Fig. 7a, b; Supplementary Table 2). EPS-C/MBC ratios declined as follows: sediment / carbonate > silicate (Fig. 7a), while in terms of land use types, it was ranked with cropland / grassland > woodland (Fig. 7b).

Correlation analysis revealed significant negative associations between EPS-C/MBC ratio and microbial factors (microbial growth ($C_{growth}$), CUE, acid phosphatase (AP), and $q_{Growth}$) (Fig. 7c). Soil factors such as clay and silt content, and $Ca_e$ positively affected EPS-C/MBC ratio while others were negatively associated such as soil C/N ratio and sand content (Fig. 7c). Further potential drivers such as climate factors (ADI), plant traits (FRB), soil factors (e.g., soil C/P ratio, SMC, and DOC), and microbial factors (MBC, MBP, MBC/MBN ratio, and β-xylosidase (BX)) showed weak negative correlations with the EPS-C/MBC ratio, while other soil factors (e.g., CEC) and microbial factors (e.g., MBT, MBN/MBP ratio, and VectorL) exhibited weak positive correlations with the EPS-C/MBC ratio (Fig. 7c). Correlations between EPS-C/MBC ratio and soil pH were significant (Supplementary Fig. 3b). Supplementary Tables 17–19 provide additional results from linear models and linear mixed-effects models assessing the influence of bedrock, land-use types, environmental, and microbial factors on EPS-C/MBC ratio.

To assess direct versus indirect effect pathways between environmental and microbial drivers and both soil EPS-C content and EPS-C/

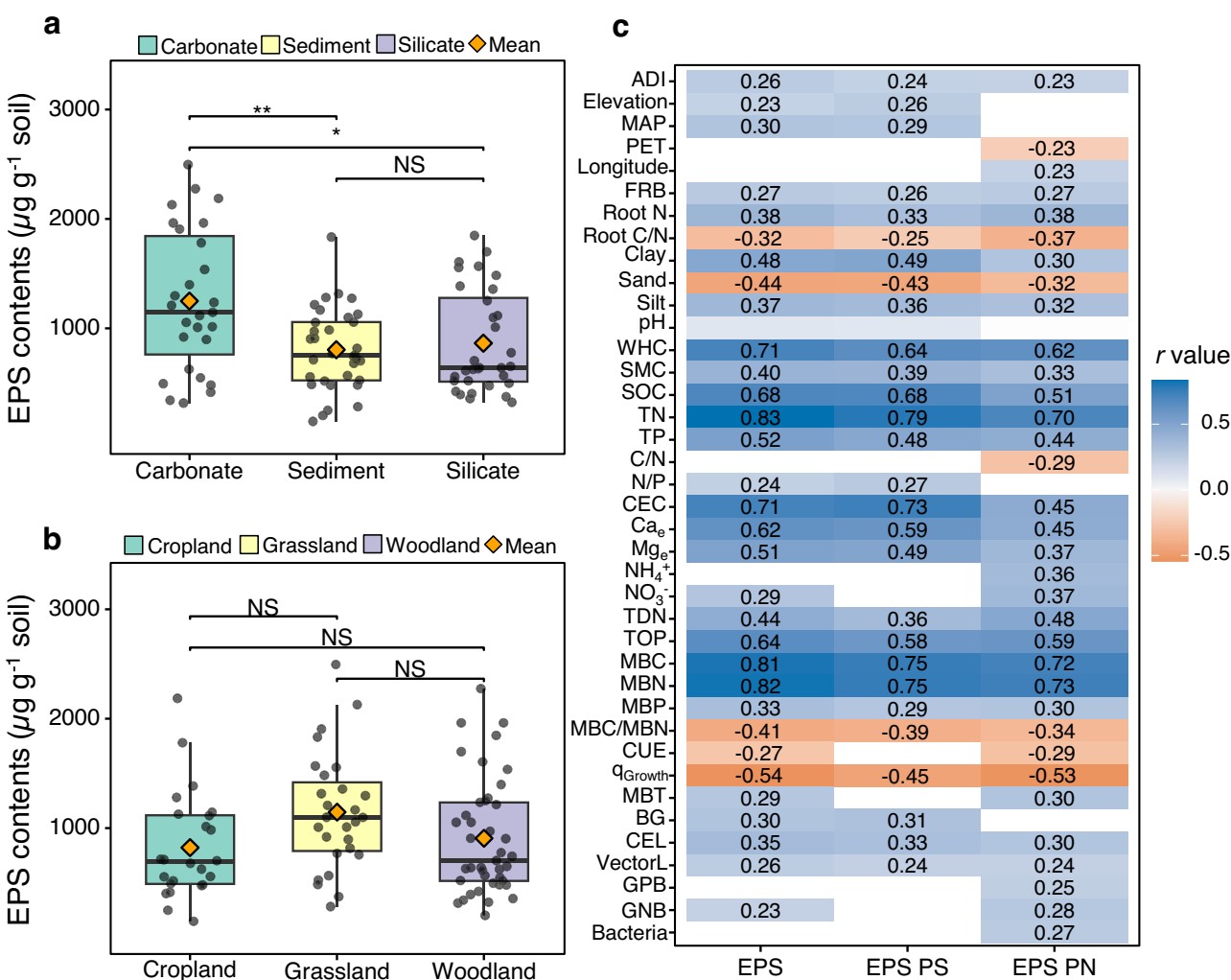

**Fig. 2 | Soil extracellular polymeric substances (EPS) and their correlations with environmental and microbial factors. a** Box plots of EPS contents (sum of EPS polysaccharides and EPS proteins) in soils from different bedrock types. **b** Box plots of EPS contents in soils from different land use types. Individual data points are displayed as dots with jitter to show distribution. Boxes show the median (middle line), 25th and 75th percentiles (interquartile range), and whiskers represent the 5th and 95th percentiles. Significant differences between groups are indicated on the plot. *$p < 0.05$, **$p < 0.01$, ***$p < 0.001$, NS $p > 0.05$. **c** Heatmap showing the correlations between total EPS, EPS polysaccharides, and EPS proteins with environmental and microbial variables. Only significant correlations ($p < 0.05$) are shown, except for pH, which is displayed regardless of significance. Numbers in the heatmap represent correlation coefficients ($r$); variables without numbers are non-

significant ($p > 0.05$). PS, polysaccharides; PN, proteins; ADI, aridity index; MAP, mean annual precipitation; PET, potential evapotranspiration; FRB, fine root biomass; WHC, water holding capacity; SMC, soil moisture content; SOC, soil organic carbon; TN, total nitrogen; TP, total phosphorus; CEC, cation exchange capacity; $Ca_e$, exchangeable $Ca^{2+}$; $Mg_e$, exchangeable $Mg^{2+}$; $NH_4^+$, ammonium; $NO_3^-$, nitrate; TDN, total dissolved nitrogen; TOP, total organic phosphorus; MBC, microbial biomass carbon; MBN, microbial biomass nitrogen; MBP, microbial biomass phosphorus; CUE, microbial carbon use efficiency; $q_{Growth}$, microbial growth normalized to microbial biomass carbon; MBT, microbial biomass turnover time; BG, β-glucosidase; CEL, exoglucanase-cellobiosidase; VectorL, enzyme vector length; GPB, gram-positive bacteria PLFA biomass; GNB, gram-negative bacteria PLFA biomass; Bacteria, bacterial PLFA biomass.

MBC ratio, we performed structural equation modeling (SEM) analysis. Soil physicochemical variables were summarized as a composite factor derived from PC1 of a PCA (Fig. 8b; Supplementary Fig. 1a). Specifically, PC1 (soil) was positively loaded by the soil C/N ratio and near-zero positively loaded by SOC, while clay content and $Ca_e$ had negative loadings. In the SEM, this composite soil factor showed a negative path coefficient to EPS-C, indicating that lower scores on this PCA-derived soil factor (i.e., higher clay, and $Ca_e$; lower soil C/N ratio) were associated with higher EPS-C content (Fig. 8a). In contrast, the microbial composite factor, derived from PCA of CUE, $C_{growth}$, $q_{Growth}$, and MBN (Fig. 8c; Supplementary Fig. 1b), exerted a direct positive effect on EPS-C/MBC ratio and a non-significant negative effect on EPS-C (Fig. 8a). In the PCA loading, $C_{growth}$, $q_{Growth}$, and CUE had negative loadings on PC1 (microbial), while MBN had a near-zero positive loading. Thus, the SEM paths indicate that lower values of this PCA-derived microbial

factor (i.e., higher CUE, $C_{growth}$, and $q_{Growth}$) were associated with lower EPS-C/MBC ratio. Additionally, plant factors (FRB) directly positively affected EPS-C. Climate factors (ADI) exerted direct effects as well, showing a significant positive impact on EPS-C but a negative impact on the EPS-C/MBC ratios. Land use type also directly influenced EPS-C. In contrast, bedrock affected the EPS-C only indirectly by shaping soil factors (Fig. 8a).

## Discussion
Across the European transect, soil EPS contents varied widely (149–2495 μg g⁻¹; 16× difference; Fig. 2), with polysaccharides ranging 79.6–1818 μg g⁻¹ (mean 624 μg g⁻¹) and proteins 14.7–825 μg g⁻¹ (mean 332 μg g⁻¹) (Fig. 3). Polysaccharides therefore accounted for 65% of the total soil EPS, with proteins contributing the remaining 35%. Other EPS compounds like lipids and eDNA were not determined here, but

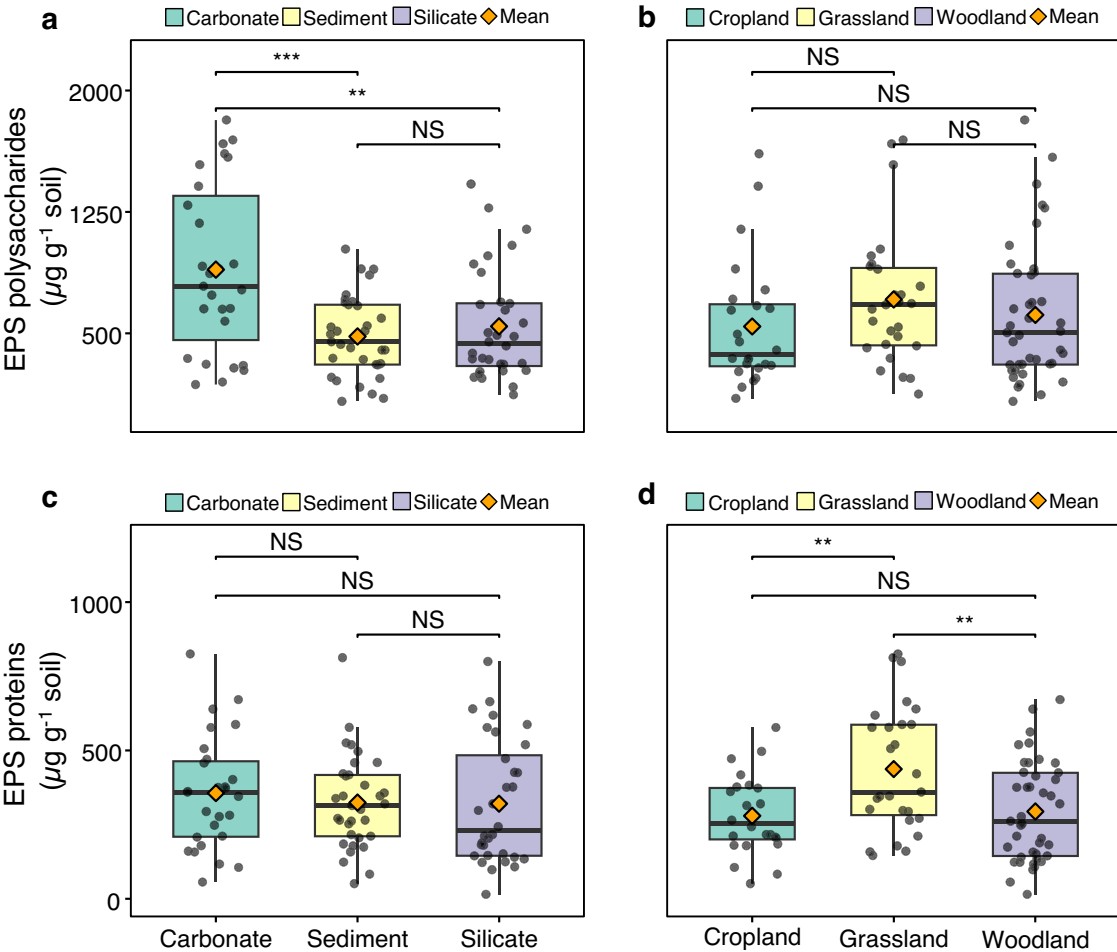

**Fig. 3 | Variations in extracellular polymeric substance (EPS) polysaccharides and EPS proteins across different bedrock and land use types. a** Box plots of EPS polysaccharides in soils from different bedrock types. **b** Box plots of EPS polysaccharides in soils from different land use types. **c** Box plots of EPS proteins in soils from different bedrock types. **d** Box plots of EPS proteins in soils from different land use types. Individual data points are displayed as dots with jitter to show distribution. Boxes show the median (middle line), 25th and 75th percentiles (interquartile range), and whiskers represent the 5th and 95th percentiles. Significant differences between groups are indicated on the plot. $* p < 0.05$, $** p < 0.01$, $*** p < 0.001$, NS $p > 0.05$.

commonly comprise only a minor fraction of EPS ($< 10\%$)[33,34]. An extensive literature survey showed lower EPS contents ($~ 52$ sites in 21 studies differing in climate, geology, and land use), with $423 \mu g\, g^{-1}$ for total EPS, $325 \mu g\, g^{-1}$ for EPS polysaccharides, and $128 \mu g\, g^{-1}$ for EPS proteins (Supplementary Table 1; Supplementary Data 2). To keep sampling efforts manageable at the continental scale, we extracted EPS from air-dried aggregates. It is important to note that air-drying can cause artefacts: triggering cell-lysis[35] and increasing EPS production[1] in the short period before the dry state is reached. However, the majority of increased C-release from soils subjected to drying is removed during the EPS pre-extraction step[36]. While our use of dried aggregates prevents direct comparison with the majority of EPS studies, which extract moist soils (e.g., Zhang et al.[13], Luo et al.[37]; Bölscher et al.[38]), our large dataset enables the relative effects of geological, geographical, and edaphic factors on soil EPS concentrations to be investigated. Other researchers have also performed ad hoc changes to the standard soil EPS extraction protocol. For example, Bublitz et al.[39] claimed that most soils do not contain measurable amounts of non-EPS carbohydrate. However, this unsubstantiated claim is at odds with the majority of soil EPS research where a purification (or pre-extraction) step is included to remove confounding non-EPS proteins and polysaccharides[2]. Zhang et al.[36] deconstructed the EPS extraction process and investigated the quality of both pre-extracts and EPS-extracts. They found that artefacts from EPS extraction of air-dried soil

were much smaller than the large amounts of easily detectable non-EPS proteins and polysaccharides removed during the purifying, pre-extraction step. This emphasizes that while EPS extraction from dry aggregates invites limitations, these are far smaller than the problems occurring when the pre-extraction step is omitted. With some of our soils receiving organic inputs and/or significant root exudates over the continent, confidence between different managements was vital. Therefore, we guarded against EPS overestimation by including the purification steps described by Redmile-Gordon et al.[2], which largely eliminates any errors from overlooking hydration status. Nonetheless, our data should not be compared directly with those of others who ensure full physico-chemical integrity of EPS by avoiding desiccation, as described in the original and complete method[2].

Among the three bedrock types, EPS contents were significantly higher in carbonate than in silicate and sedimentary soils. This pattern was mirrored in microbial (e.g., MBC and MBN) and soil properties (e.g., CEC, clay, and $Ca_e$), known to promote EPS, all of which were significantly influenced by bedrock and peaked in carbonate soils (Supplementary Tables 4–7). Conversely, sand content—which was lowest in carbonate soils—was negatively correlated with EPS contents (Supplementary Tables 5, 7). Based on these findings, we propose that the higher EPS content observed in carbonate soils is primarily associated with two aspects: (1) an indirect pathway, where high soil CEC supports microbial activity[22], increases microbial biomass (e.g., MBC

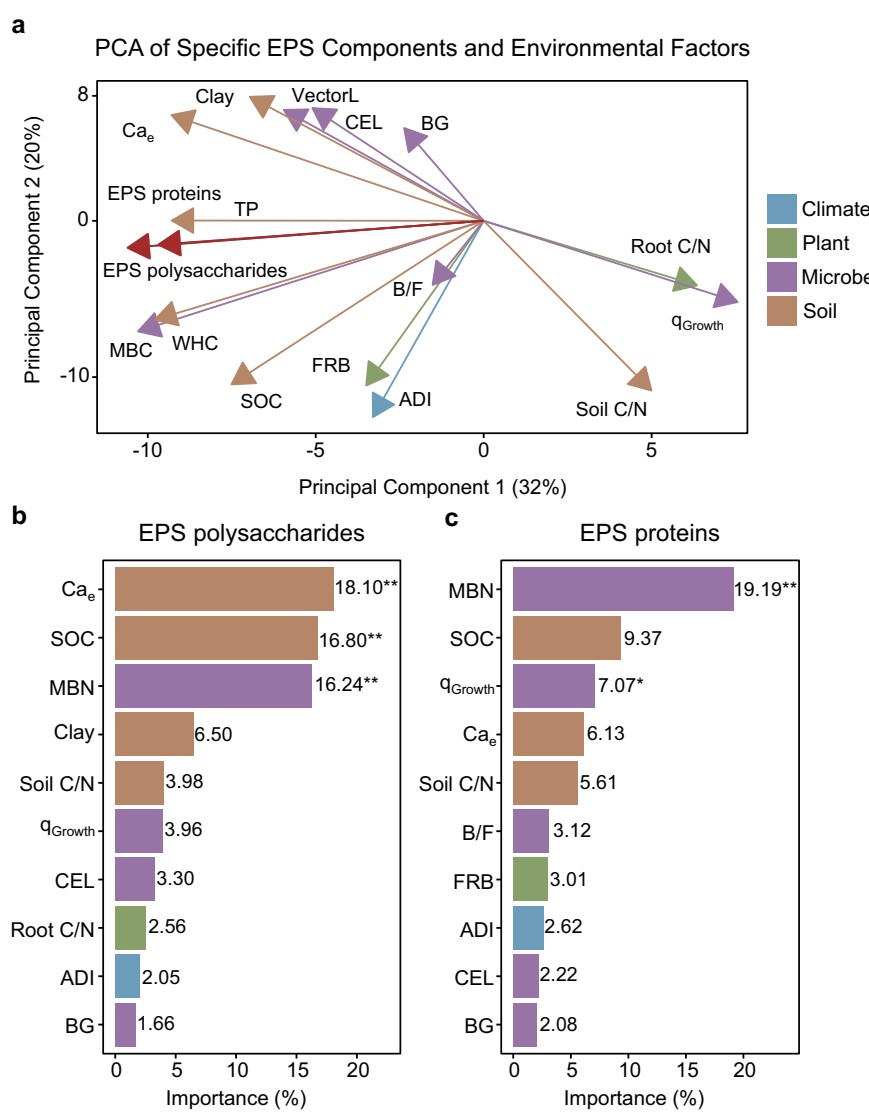

**Fig. 4 | Environmental and microbial drivers of extracellular polymeric substance (EPS) polysaccharides and EPS proteins. a** Principal component analysis (PCA) of environmental and microbial variables associated with EPS polysaccharides and EPS proteins. **b** Importance ranking of predictors for EPS polysaccharides based on random forest analysis. **c** Importance ranking of predictors for EPS proteins based on random forest analysis. *$p < 0.05$, **$p < 0.01$, ***$p < 0.001$.

ADI, aridity index; FRB, fine root biomass; BG, β-glucosidase; VectorL, enzyme vector length; B/F, bacteria to fungi ratio; Ca$_e$, exchangeable Ca$^{2+}$; WHC, water holding capacity; q$_{Growth}$, microbial growth normalized to microbial biomass carbon; SOC, soil organic carbon; TP, total phosphorus; CEL, exoglucanase-cellobiosidase; MBC, microbial biomass carbon; MBN, microbial biomass nitrogen.

and MBN) and thereby promotes the capacity for EPS production[15] (Supplementary Table 3); and (2) soil minerals (clay) and cations (Ca$_e$) may enhance the association of EPS with soil surfaces and within aggregates through multivalent cation bridging between negatively charged sites of EPS and other polymers and minerals[13,40]. CER-based extraction primarily isolates loosely bound or exchangeable EPS[14], associated with soils through reversible Ca$^{2+}$ bridging and electrostatic attraction (Supplementary Discussion 1). Additionally, although our study does not directly include microbial community diversity or composition data, previous research has shown that lithology-driven geochemical processes—such as rock weathering and mineral-associated protection—can indirectly regulate microbial extracellular products (such as glomalin-related soil protein) by affecting microbial diversity and resource elemental stoichiometry[41]. In particular, carbonate soils have been reported to harbor higher microbial diversity[41]. Given the reported positive association between microbial diversity and EPS[42], this pattern aligns with the elevated EPS levels observed in carbonate soils. Interestingly, while carbonate soils exhibited the

highest EPS contents, their EPS-C/MBC ratios followed a different pattern, with soils on sedimentary rocks demonstrating the highest EPS-C/MBC ratios (Fig. 7a). In our study, this ratio does not directly reflect EPS production efficiency—since EPS contents in natural soils result from the balance of microbial production and degradation, modulated by mineral interactions and extraction limitations. Notably, sedimentary soils had the lowest EPS contents, but the highest EPS-C/ MBC ratios, which likely stems from reduced MBC rather than increased EPS contents. Supporting this interpretation, microbial growth-related parameters (e.g., C$_{growth}$, MBC, and MBN) were consistently lowest in sedimentary soils, which were also characterized by the lowest SMC and relatively low ADI in our study (Supplementary Tables 5, 7; Supplementary Fig. 2), potentially reflecting water limitation, a factor suggested by previous studies to be associated with increased soil EPS[27]. Although silicate soils exhibited intermediate levels of both EPS and MBC (Supplementary Table 7), they had the lowest EPS-C/MBC ratio. This suggests that microbes in silicate soils may have allocated proportionally less C toward EPS production or are

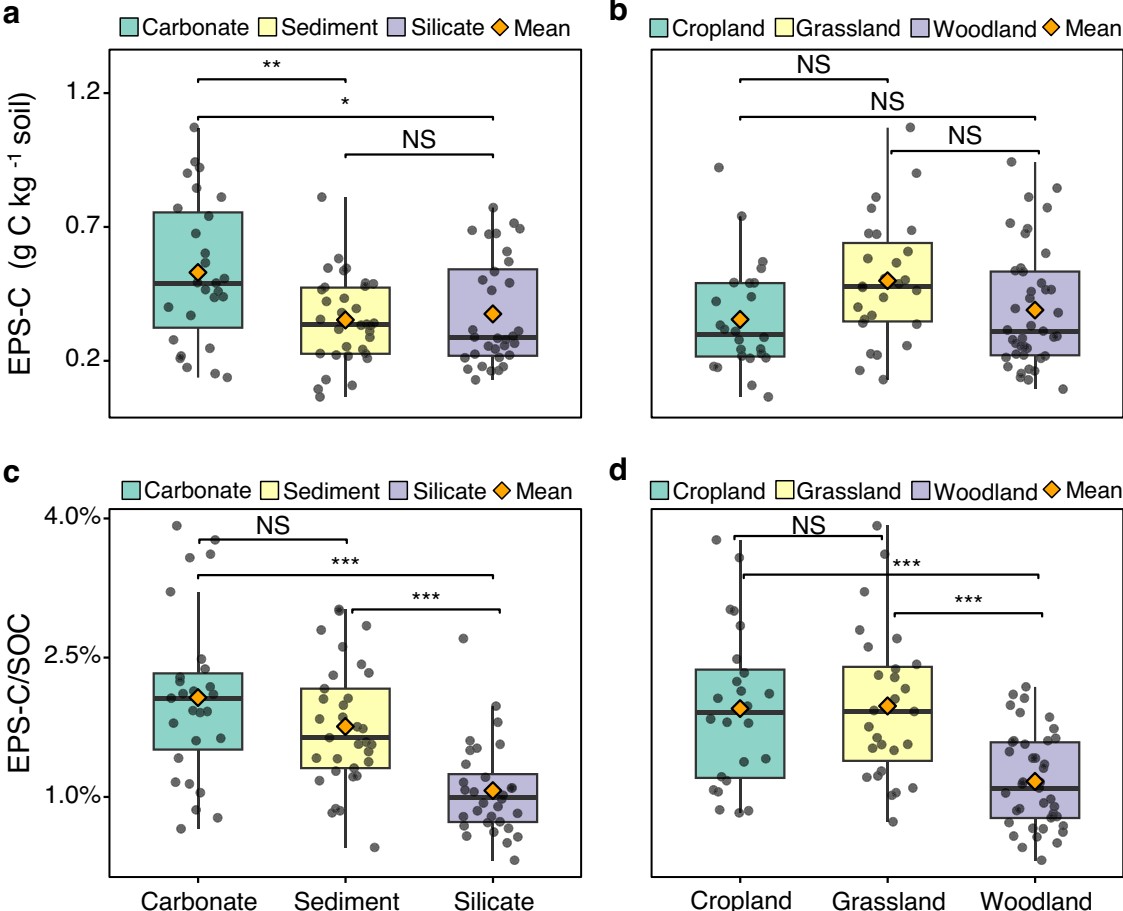

**Fig. 5 | Variations in extracellular polymeric substance carbon (EPS-C) and its contribution to soil organic carbon (EPS-C/SOC) across different bedrock and land use types. a** Box plots of EPS-C in soils from different bedrock types. **b** Box plots of EPS-C in soils from different land use types. **c** Box plots of EPS-C/SOC ratio in soils from different bedrock types. **d** Box plots of EPS-C/SOC ratio in soils from different land use types. Individual data points are displayed as dots with jitter to show distribution. Boxes show the median (middle line), 25th and 75th percentiles (interquartile range), and whiskers represent the 5th and 95th percentiles. Significant differences between groups are indicated on the plot. *$p < 0.05$, **$p < 0.01$, ***$p < 0.001$, NS $p > 0.05$.

less conducive to the association of EPS with the soil surface and within aggregates. Three possible explanations can be proposed: (1) Environmental stress in silicate soils may be relatively mild – for example, these soils exhibited the highest SMC, WHC, and ADI – whereas more severe moisture-related stress in sedimentary soils might have triggered greater EPS production; (2) the lowest levels of $Ca_e$ and clay (highest sand content) in silicate soils may have provided the least protection for EPS (Supplementary Table 7); (3) differences in microbial community composition or metabolic strategies may also contribute to the observed pattern. However, we currently lack direct evidence to confirm the third explanation.

We found land-use was not a significant driver of EPS polysaccharides, but was a highly significant driver of EPS proteins (Fig. 3). This adds important depth to land-use research. Redmile-Gordon et al.[14] found that land-use change to perennial grassland increased soil EPS and aggregate stability, with EPS protein being far more influential for soil structural stability than EPS polysaccharides[14]. Our findings, that both EPS protein and EPS-C are elevated under grassland (across contrasting land management types), support their assertion that CER-extracted EPS protein is more physically influential than EPS polysaccharides in shaping soil aggregates[14] - and thus also C stabilization (Supplementary Discussion 2). Moreover, distinct environmental dependencies and biosynthetic pathways of EPS polysaccharides and EPS proteins caused their contents to be differentially affected by bedrock and land use, respectively (Supplementary Discussion 3).

Across the European transect, soil EPS-C averaged 0.41 g C kg$^{-1}$ (0.06–1.07 g C kg$^{-1}$), contributing 1.6% to SOC (0.3–3.9%). Recent small-scale studies have reported similar values, consistent with our results[27,43]. Although only ~2%, these values are clearly underestimates, because the gentle CER extraction targets a relatively pure EPS fraction with minimal humic interference and microbial lysis, and thus does not fully recover soil EPS[2,44]. Compared to harsher, more complete yet more destructive extraction protocols in soils, CER only extracts 1/2 to 1/20 of total EPS polysaccharides and EPS proteins[2,44,45]. More comprehensive protocols are therefore needed to account for incomplete soil EPS extraction and to achieve fully quantitative EPS estimates. Although EPS-C contributes little to total SOC (not accounting for incomplete extraction), EPS plays a critical role in soil ecosystems[14]. As a dynamic, microbially derived fraction of the soil C pool, CER-extracted EPS likely represents a transient and labile portion of SOC[14], yet it is thought to promote SOC stabilization by enhancing soil aggregation and facilitating mineral-organic associations—functioning as a glue that improves soil structure[13]. EPS may also influence soil C cycling indirectly by enhancing the stability of extracellular enzymes and modulating their activity[29], consistent with the weak positive correlations observed with BG and CEL (Fig. 2c). Despite its constrained quantitative contribution to SOC, EPS may exert a disproportionate influence on soil structural properties, making its role particularly relevant under climate change[14].

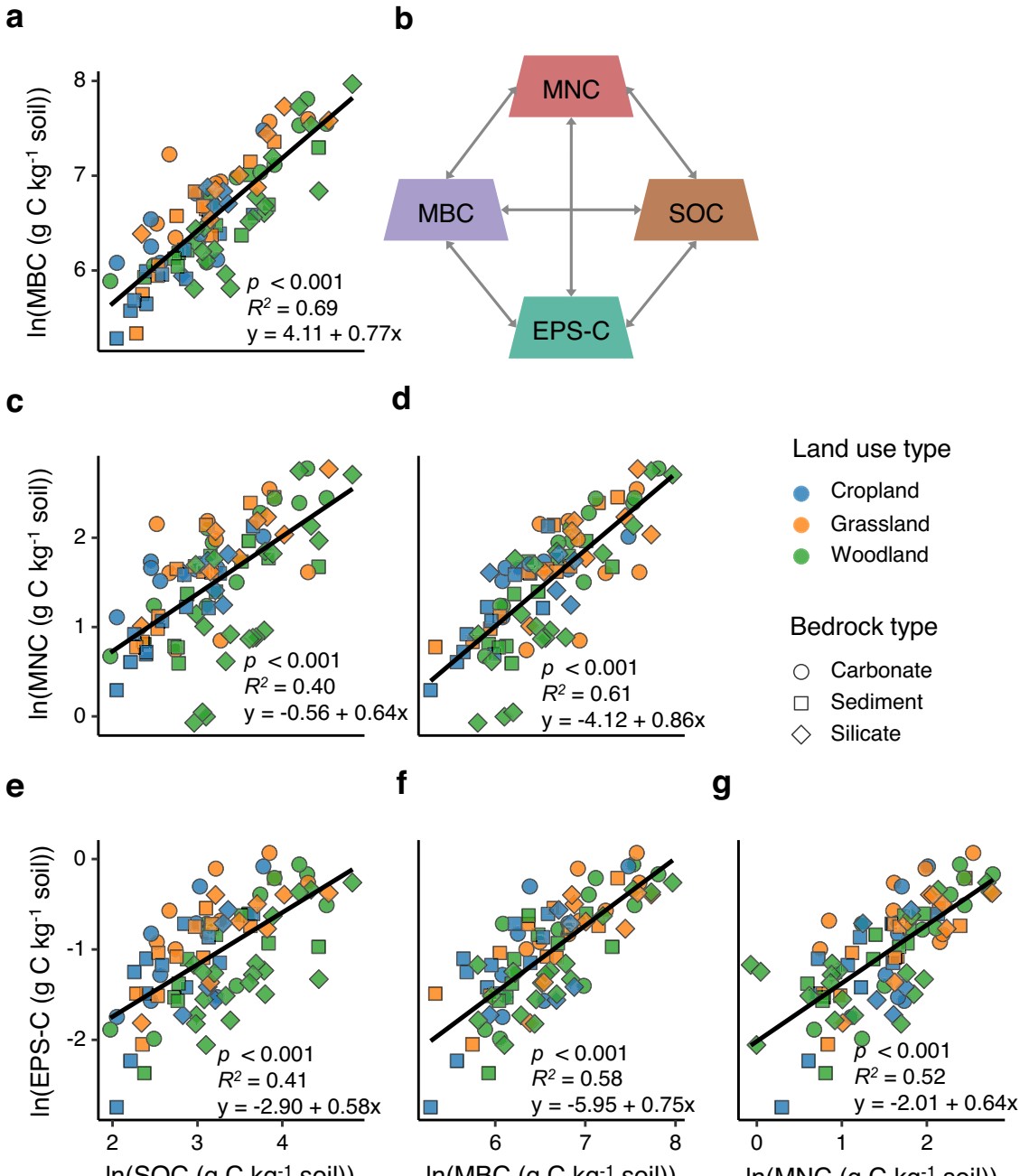

**Fig. 6 | Linear relationships among extracellular polymeric substance carbon (EPS-C), microbial biomass carbon (MBC), microbial necromass carbon (MNC), and soil organic carbon (SOC). a** Relationship between MBC and SOC. **b** Schematic illustration of the relationships among EPS-C, MNC, SOC, and MBC. **c** Relationship between MNC and SOC. **d** Relationship between MNC and MBC. **e** Relationship between EPS-C and SOC. **f** Relationship between EPS-C and MBC. **g** Relationship between EPS-C and MNC. Each point represents an individual soil sample. Black lines indicate power-law relationships fitted as linear models in ln–ln space.

In contrast to extracellular microbial residues, cellular residues (as MNC) contributed roughly 10 times more to SOC than estimated EPS-C across the large-scale transect. Incomplete extraction of EPS partly explains the difference between MNC and EPS-C observed here. Perhaps more importantly, amino sugar biomarkers do not exclusively represent necromass[46]. Galactosamine, for example, can be found in substantial concentrations in plant residues, and other non-necromass materials, even EPS itself[46]. Therefore, while measures of EPS-C are understood to be underestimates, measures of MNC are known to be overestimates[47], and perhaps with a larger margin for error and variability. This means that while on face value our results suggest that

MNC was the larger pool of C, true values might be more comparable in size. More research is needed to elucidate with confidence whether EPS-C or MNC contributes more to SOC formation and stabilization. EPS is a transient pool that is selectively secreted by microorganisms where sufficient labile C is available, with the majority of EPS either stabilizing or being turned over in less than 2.5 years[14]. Interestingly, we found a strong positive relationship between EPS-C and MNC (Fig. 6g), which may be due to them being both outputs of microbial anabolic metabolism, active microbes producing EPS and replicating, with microbial cells turning into necromass via different pathways[48]. In addition, EPS is produced by microbes as a glue for attachment of their

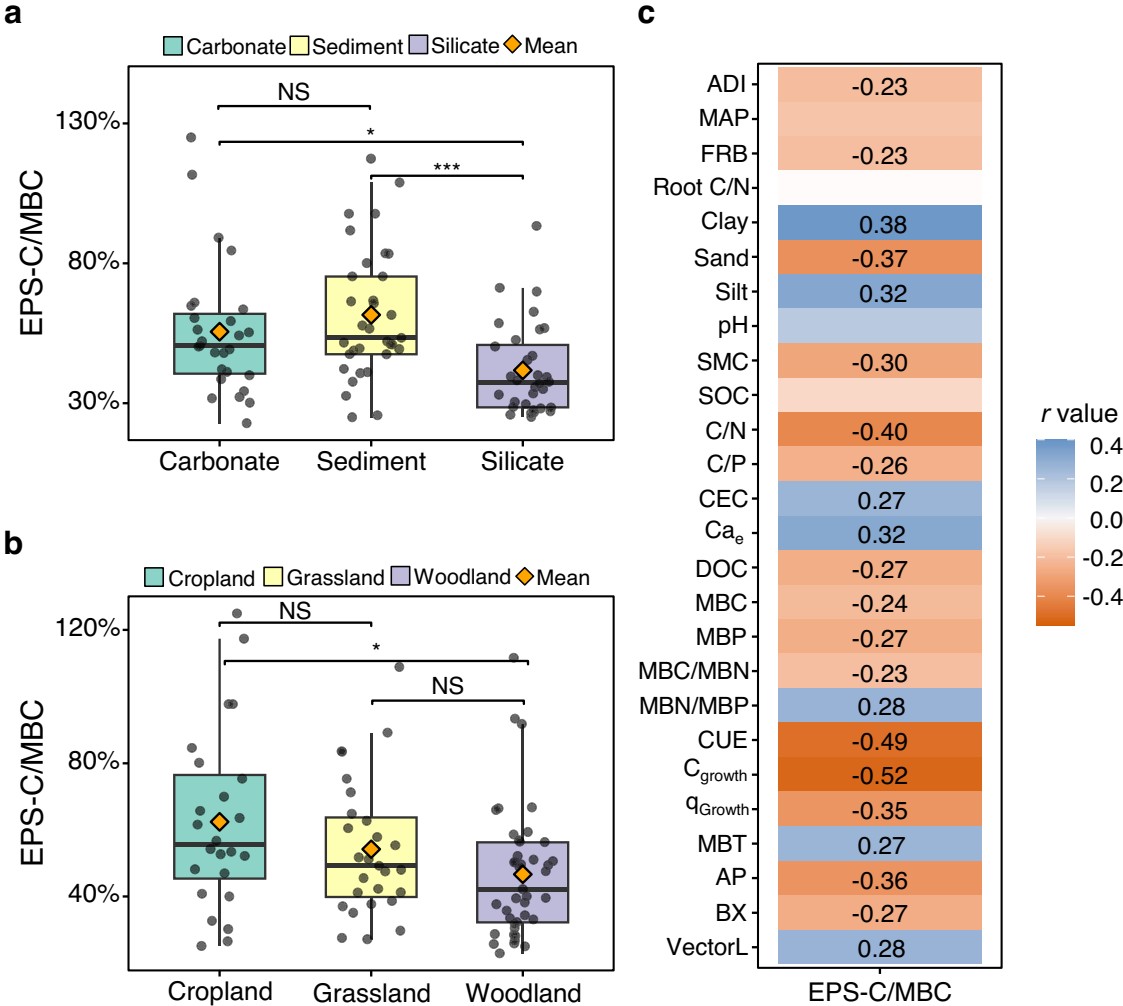

**Fig. 7 | Soil extracellular polymeric substance carbon to microbial biomass carbon (EPS-C/MBC) ratio and its correlations with environmental and microbial factors. a** Box plots of EPS-C/MBC in soils from different bedrock types. **b** Box plots of EPS-C/MBC in soils from different land use types. Individual data points are displayed as dots with jitter to show distribution. Boxes show the median (middle line), 25th and 75th percentiles (interquartile range), and whiskers represent the 5th and 95th percentiles. Significant differences between groups are indicated on the plot. *$p < 0.05$, **$p < 0.01$, ***$p < 0.001$, NS $p > 0.05$. **c** Heatmap showing the correlations between EPS-C/MBC with environmental and microbial variables. Only significant correlations ($p < 0.05$) are shown, except for pH, mean annual precipitation (MAP), Root C/N ratio, and soil organic carbon (SOC), which are displayed regardless of significance. Numbers in the heatmap represent correlation coefficients ($r$); variables without numbers are non-significant ($p > 0.05$). MBC, microbial biomass carbon; ADI, aridity index; FRB, fine root biomass; SMC, soil moisture content; CEC, cation exchange capacity; Ca$_e$, exchangeable Ca$^{2+}$; DOC, dissolved organic carbon; MBP, microbial biomass phosphorus; MBN, microbial biomass nitrogen; CUE, microbial carbon use efficiency; C$_{growth}$, microbial growth; q$_{Growth}$, microbial growth normalized to microbial biomass carbon; MBT, microbial biomass turnover time; AP, acid phosphatase; BX, β-xylosidase; VectorL, enzyme vector length.

cells and colonies to soil surfaces[33], which, after death, cause necromass fragments to be cemented via EPS to soil minerals or organic surfaces, which has been shown microscopically[49,50]. This also promotes the strong collinearity between EPS-C and MNC, MBC and SOC (Fig. 6). Overall, EPS therefore likely plays a double sticky role, on the one hand promoting cell and necromass attachment to (mineral) surfaces stabilizing SOC via necromass accumulation, and on the other hand by stimulating the aggregation of particulate organic matter and of mineral associated organic matter (including MNC-laden mineral particles) into micro- and macroaggregates, thereby also promoting SOC stabilization on a higher level.

Changes in short-term microbial EPS production efficiency were previously calculated by standardizing soil EPS to soil microbial ATP[51], as an independent indicator of the soil microbial biomass. The EPS-C/MBC ratio of soils can also be used, and in natural soils, represents the net result of all historic influences contributing to the current state (whether steady or dynamic). As mentioned above, apparent soil EPS pools reflect a dynamic balance among EPS production, association with soil surfaces and aggregates, and degradation. Distinct bedrock and land-use types have shaped diverse ecological environments and have undoubtedly influenced these EPS dynamics. Therefore, we present EPS-C/MBC ratios as is without inferring EPS production efficiency. SEM analysis indicated that climate, plant, soil, and microbial factors jointly drive the soil EPS-C/MBC ratio. Firstly, environmental conditions are key factors that determine whether and to what extent microorganisms secrete EPS[1,7]. Under harsh conditions, microorganisms produce EPS to enhance survival[8]; for example, EPS helps retain water and protect cells from drought damage[52]. Recent field experiments have shown that drought can significantly increase soil EPS[53]. Moving beyond these localized observations, we show at a large scale that EPS-C/MBC decreases with increasing aridity index (ADI; calculated as MAP/PET), indicating microbial adaptation to long-term water-limited environments. This pattern is further supported by the negative correlation between EPS-C/MBC and SMC (Fig. 9c). Microbial

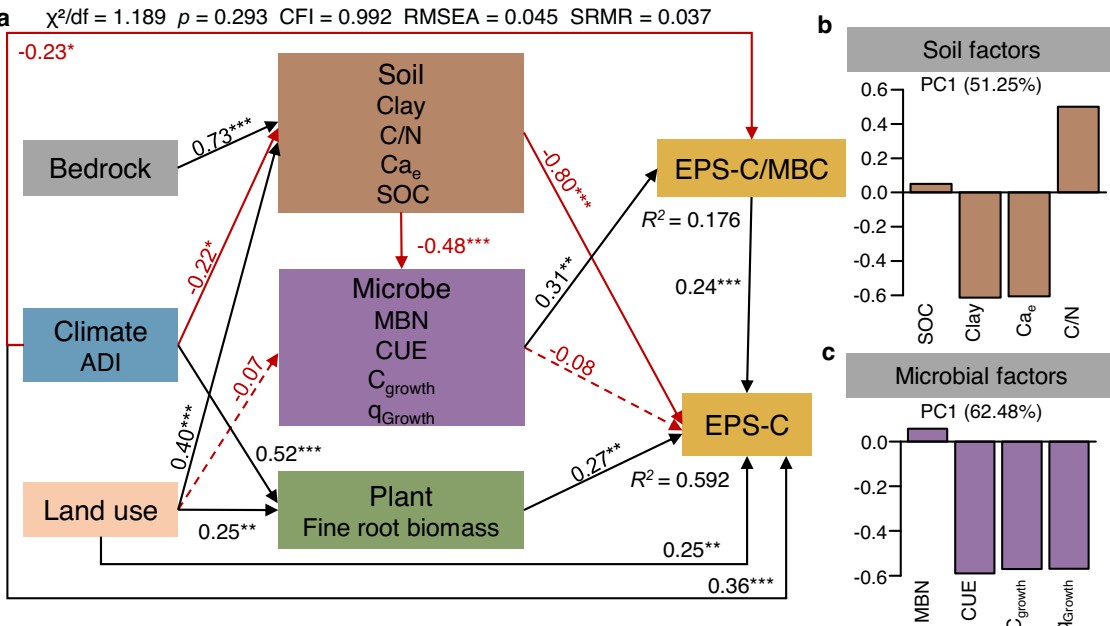

**Fig. 8 | Effects of environmental and microbial factors on soil extracellular polymeric substance carbon (EPS-C) and extracellular polymeric substances carbon to microbial biomass carbon (EPS-C/MBC) ratio. a** Structural equation modeling (SEM) illustrating the direct and indirect pathways by which bedrock type, land use, and climate influence EPS-C and EPS-C/MBC via plant, soil, and microbial factors. Solid lines represent statistically significant paths, and dashed lines indicate non-significant paths. Significance levels are indicated as: *$p < 0.05$, **$p < 0.01$, ***$p < 0.001$. Numbers on the paths are standardized regression coefficients. **b** Bar chart showing the loadings of four soil variables on the soil composite factor (PC1) derived from PCA. SOC, soil organic carbon; C/N, soil carbon to nitrogen ratio; $Ca_e$, exchangeable $Ca^{2+}$. **c** Bar chart showing the loadings of four microbial variables on the microbial composite factor (PC1) derived from PCA. MBN, microbial biomass nitrogen; CUE, microbial carbon use efficiency; $C_{growth}$, microbial growth; $q_{Growth}$, microbial growth normalized to microbial biomass carbon. EPS, extracellular polymeric substances; ADI, aridity index.

communities, therefore, likely invest in greater EPS secretion to retain soil moisture under arid conditions and soil water deficit. However, as measured EPS levels are influenced by production, decomposition, and their association with soil particles and aggregates, the true microbial response requires further investigation. Beyond moisture, environmental pH extremes, particularly high acidity, were also documented to induce EPS production in microbes, where the EPS matrix buffers excessive proton concentrations via its cation-exchange capacity[1,54]. In this study, we found a curvilinear relationship between the EPS-C/MBC ratio and soil pH (Supplementary Fig. 3; ranged from 4.0 to 8.9), with the ratio peaking at neutral pH rather than at the lowest values. This indicates that pH extremes are not a major control of EPS-C/MBC within the studied range.

Although our data show a positive correlation between MBC and total EPS, higher EPS-C/MBC ratios were observed under stressful conditions, such as in soils developed from sedimentary bedrock. These soils are characterized by low WHC and SMC, likely resulting from their relatively low SOC content and relatively high sand content, and by relatively low ADI values, indicating drier climatic conditions (Supplementary Tables 5, 7). This indicates that, despite lower microbial abundance, microbes may allocate a greater proportion of their C to EPS as a survival strategy under environmental stress. Given the limited labile C resources available to microbes in soils, microbes may need to prioritize C allocation to specific processes, which may lead to physiological trade-offs in microbial C allocation[55,56]. In line with EPS secretion being typically stress-induced[13], EPS-C/MBC ratios were lower under non-stressed conditions (Fig. 9), where high microbial growth favors C allocation toward biomass rather than EPS. The negative relationships between EPS-C/MBC and microbial growth ($C_{growth}$) and CUE thus indicate a shift in microbial C allocation from growth to non-growth anabolic processes such as EPS production[38] (Fig. 9e, f). This strategy, adopted by soil microorganisms to secrete EPS or invest in growth or other C allocation processes, is therefore

likely influenced by their physiological state and C allocation trade-offs. However, this pattern does not contradict the positive correlation between EPS and MBC: while the relative allocation to EPS may decrease (lower EPS-C/MBC), absolute EPS production can remain high when microbial biomass is large, as more cells may collectively produce more EPS, forming a matrix for cell attachment and biofilm formation.

Finally, microbial community composition may influence observed EPS patterns, as taxa differ markedly in genomic traits related to EPS synthesis[29] and degradation[15]. Diverse communities may adopt distinct EPS strategies to cope with environmental stress and optimize resource use[57], with some taxa more capable of EPS biosynthesis[58]. Key microbes for soil EPS secretion and degradation can only be identified by deep shotgun metagenomic sequencing and sequencing of key functional genes. However, in this study, we used PLFA profiling to broadly characterize microbial groups, but found no strong link between community structure and EPS content or EPS-C/MBC.

In conclusion, the differences in EPS contents and EPS-C/MBC ratios involve the dynamic interplay among environmental factors, resource availability, microbial metabolic strategies, and soil properties. Addressing both the underestimations of EPS and the over-estimations of necromass, respectively achieved by extraction, will be important in clarifying our understanding of these formative C pools. Further research should focus on quantifying EPS production rates and understanding the dynamics of its formation and turnover using [13]C and [15]N isotope tracer techniques. Complementary to this, optimizing EPS extraction and developing compound-specific analytical techniques is essential for improving the characterization of EPS dynamics. Additionally, linking EPS content and dynamics to its functional implications in soils, including aggregate formation, carbon cycling, water retention, and microbial stress amelioration, will be critical for advancing our knowledge of soil microbial ecology and ecosystem resilience.

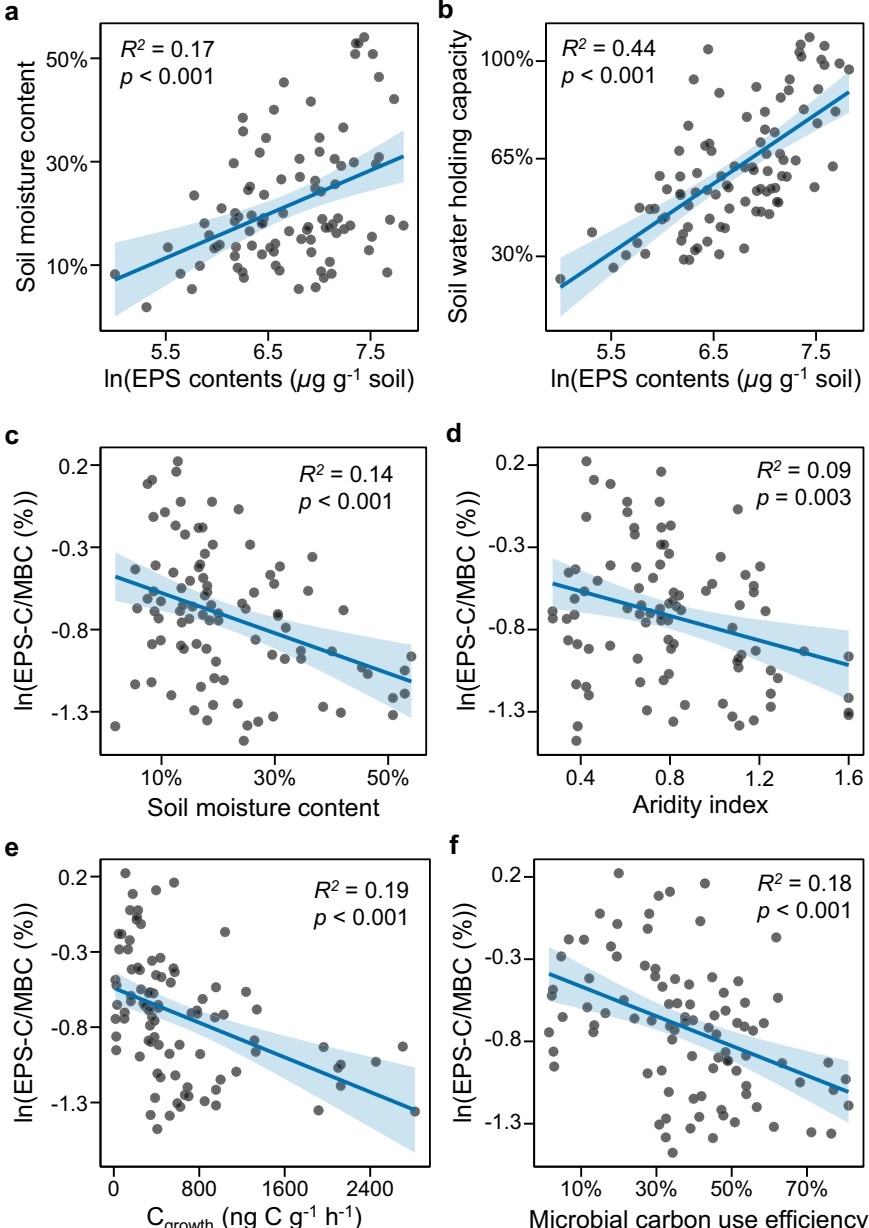

**Fig. 9 | Linear regressions showing relationships between extracellular polymeric substances (EPS) and extracellular polymeric substance carbon to microbial biomass carbon (EPS-C/MBC) ratio and key climatic, soil, and microbial parameters. a** Relationship between EPS and soil moisture content. **b** Relationship between EPS and soil water holding capacity. **c** Relationship between EPS-C/MBC and soil moisture content. **d** Relationship between EPS-C/MBC and aridity index. **e** Relationship between EPS-C/MBC and microbial growth. **f** Relationship between EPS-C/MBC and microbial carbon use efficiency. Blue solid lines indicate linear regression fits and shaded areas represent 95% confidence intervals. The $R^2$ and *p-value* for each regression are shown on each panel. Each point represents an individual soil sample.

## Methods

### Site description
To investigate the distribution of EPS and its drivers on a continental scale, we sampled 92 sites across Europe (36°24′45′ N- 71°01′56″ N, 8°31′56″ W- 29°34′60″ E), spanning from the Mediterranean (Southern Spain) to the subarctic (Northern Norway), and from the Atlantic coast (Western Portugal) to the dry continental steppes (Eastern Romania). The sampled soils were distinct in soil parent material, soil type, land use and vegetation. Sampling sites were selected to represent the natural vegetation as defined in the "Map of the natural vegetation of Europe"[59]. Geographical and climatic data, including latitude, longitude, elevation, mean annual temperature (MAT), mean annual precipitation (MAP) and potential evapotranspiration (PET), were

extracted for each site from the WorldClim database (v.1.4) at 100 m resolution. Bedrock information was obtained from the International Geological Map of Europe (IGME5000, 1:5,000,000)[60], and dominant soil types were derived from the Soil Regions of the European Union and Adjacent Areas map (EUSR5000, 1:5,000,000)[59]. In brief, MAT of the sampling sites ranged from −3 to 18 °C, while MAP ranged from 415 to -1400 mm yr[-1]. An aridity index (ADI) was calculated as the ratio of MAP to mean annual PET[61].

### Sampling
Soil samples were taken from May to August 2017, at the peak of the vegetation season across the European climate transect. The sampled soils differed in terms of soil parent material, land use, and vegetation.

At each site, samples of mineral topsoil (0–15 cm) were taken with a soil corer (5 cm). Each bulk soil sample consisted of five sub-samples taken at ~5 m distance from each other. These five sub-samples were composited and then sieved to give one mixed mineral soil sample per site. Therefore, no within-site replicates were analysed. Where possible, more than one and up to all three land uses (woodland/forest, grassland, and cropland) were sampled in close vicinity, termed a region or site cluster. In total, we sampled 92 mineral topsoils (92 sites) from 39 geographically distinct regions (a region characterized by the same climate and bedrock, but sites with different land use). In 23 regions (site clusters), we were able to sample soils from woodland (forest), grassland, and cropland areas in close proximity (Supplementary Data 1). Bedrock types were aggregated into three large groups for statistical analyses: carbonate (e.g., dolomite, limestone, and marl), sediment (e.g., flysch, molasse, till, and fluvial sand), and silicate (e.g., plutonic, igneous, and metamorphic formations). The soil samples were separated into aliquots, which were either air-dried or stored under moist conditions at 4 °C.

## Characterization of soil, plant, and microbial properties

A comprehensive set of soil physicochemical, plant, and microbial parameters was used in the analyses, compiled from both previously published sources[59] and unpublished datasets within the same project framework. Soil physicochemical properties were analyzed on either air-dried or fresh samples, depending on the parameter. Soil texture (sand, silt, and clay), cation exchange capacity (CEC), exchangeable $Ca^{2+}$ ($Ca_e$), and $Mg^{2+}$ ($Mg_e$) were determined on air-dried soils by the Austrian Agency for Health and Food Safety (AGES) according to European and international standards (ÖNORM). Fe oxyhydroxides and Al oxyhydroxides were determined on air-dried soils by extraction with acid ammonium oxalate ($Fe_o$ and $Al_o$) and Na-dithionite extracts ($Fe_d$ and $Al_d$) at the Institute of Soil Research (IBF, University of Natural Resources and Life Sciences, Vienna, Austria). $Fe_d$ minus $Fe_o$ represented Fe bound in crystalline oxyhydroxides ($Fe_c$). Roots, collected during soil sieving, were washed and oven-dried at 60 °C to determine fine root biomass (FRB). Dried root and air-dried soil samples were ground and analyzed for root carbon (C) and nitrogen (N) contents, and for soil organic carbon (SOC) and total nitrogen (TN) using an elemental analyzer. Root phosphorus (P) was determined by dry ashing, acid extraction, and colorimetric measurement, and root C/N/P ratios were subsequently calculated. Soil pH was measured in water (1:5 (w:v)) using an ISFET pH sensor (Sentron, Leek, Netherlands). Soil total P (TP) and inorganic P (TIP) were determined from 0.5 M $H_2SO_4$ extracts of ignited and control soils, with reactive phosphate quantified colorimetrically using malachite green. Organic P (TOP) was calculated as TP - TIP, and soil C/N/P ratios were subsequently calculated. Dissolved organic C (DOC) and total dissolved N (TDN) were measured in 1 M KCl extracts (1:5, w:v; 1 h) after filtration through ash-free cellulose filters (Whatman) using a TOC/TN analyzer (TOC-VCPH/TNM-1, Shimadzu, Austria). Ammonium ($NH_4^+$) and nitrate ($NO_3^-$) were determined colorimetrically. Soil water holding capacity (WHC) was determined by saturating 10 g of field-moist soil with deionized water and allowing it to drain through an ash-free cellulose filter in a funnel for 2.5 h. Soil moisture content (SMC) was determined gravimetrically as the mass loss upon drying fresh soil.

Soil microbial community composition was analysed using phospholipid fatty acid (PLFA) analysis following Kaiser et al.[62]. Total lipids were extracted with chloroform-methanol-citrate buffer, and phospholipids were separated on silica columns and converted to fatty acid methyl esters prior to gas chromatographic analysis. PLFA biomarkers were assigned to Gram-positive bacteria (GPB, i15:0, a15:0, i16:0, i17:0 and a17:0), Gram-negative bacteria (GNB, 18:1ω7, cy17:0, 16:1ω7, 16:1ω9, cy18:0, cy19:0 and 16:1ω5), fungi (18:2ω6,9, 18:1ω9 and 18:3ω3,6,9), and total bacteria (sum of Gram-positive and Gram-

negative markers together with 18:1ω5, 17:0, 15:0, 17:1ω6 and 17:1ω7)[62]. Total microbial biomass was calculated as the sum of all detected PLFAs. Bacteria-to-fungi (B/F) and Gram-positive-to-Gram-negative bacteria (GPB/GNB) ratios were also calculated accordingly. Microbial process data, such as microbial growth ($C_{growth}$) and carbon use efficiency (CUE), exoenzymes, and microbial C, N, and P, were measured on fresh soils directly after collection. Specifically, microbial biomass carbon (MBC), nitrogen (MBN), and phosphorus (MBP) were determined via chloroform fumigation extraction[63], from which the MBC/MBN/MBP ratios were calculated. Microbial nitrogen use efficiency (NUE) was calculated based on concurrent measurements of microbial growth and gross N mineralization rates[64]. CUE, $C_{growth}$, growth normalized to microbial biomass carbon ($q_{Growth}$), and microbial biomass turnover time (MBT) were estimated from measurements of MBC, basal respiration, and $^{18}O$ incorporation into dsDNA[65]. Soil microbial necromass carbon (MNC), including fungal necromass carbon (FNC) and bacterial necromass carbon (BNC), was quantified using amino sugar biomarkers analysed by acid hydrolysis followed by 1-phenyl-3-methyl-5-pyrazolone (PMP) derivazation and ultra-high performance liquid chromatography (UPLC) coupled to Orbitrap high-resolution mass spectrometry (HRMS)[46]. Glucosamine and muramic acid were used as indicators of fungal and bacterial necromass, respectively. Hydrolytic enzyme activities were measured using microplate assays[65,66]. β-glucosidase (BG), N-acetyl-β-glucosaminidase (NAG), leucine aminopeptidase (LAP), acid phosphatase (AP), cellobiosidase (CEL) and β-xylosidase (BX) were determined fluorometrically using 4-methylumbelliferone (MUB)- or 7-amino-4-methylcoumarin (AMC)-linked substrates. Phenoloxidase (POX) activity was measured colorimetrically using 0.4 mM ABTS (2,2'-azinobis-(3-ethylbenzothiazoline-6-sulfonic acid)) at 420 nm. Enzyme activities were calculated from the increase in fluorescence or absorbance during incubation and expressed as nmol $h^{-1}$ $g^{-1}$ dry soil (or μmol $h^{-1}$ $g^{-1}$ dry soil for POX). Enzymatic vector length and angle were calculated based on enzyme stoichiometric ratios[67].

## Soil EPS analysis

Soil EPS (including polysaccharides and proteins) was extracted from air-dried soil samples in 2024 using cation exchange resin (CER)[2]. Soil EPS binds to negatively charged soil surfaces, such as clay minerals and organic matter, through electrostatic interactions involving $Ca^{2+}$ bridging. CER, in its sodium form, exchanges its $Na^+$ ions with these divalent cations, weakening $Ca^{2+}$ bridging and releasing EPS from the soil matrix. This technique is highly efficient for the extraction of EPS and induces negligible cell lysis, yielding highly pure EPS fractions[2,44].

A preliminary experiment was conducted to optimize EPS extraction using different CER:soil ratios across soils varying in bedrock type and SOC content, showing that 1 g of air-dried soil with 10 g of CER provided a good balance for determining both EPS polysaccharides and EPS proteins (Supplementary Method 1; Supplementary Fig. 4). Before extraction, we also calculated the required amount of CER for each soil sample based on its SOC content following the method proposed by Redmile-Gordon et al.[2]. These calculations showed that 10 g CER per 1 g dry soil provided sufficient ion-exchange capacity even for the samples with the highest SOC levels, and importantly for soils with the highest exchangeable $Ca^{2+}$ concentrations. Therefore, we applied this amount uniformly across all soils to ensure methodological consistency across SOC-rich and SOC-poor soils and limestone and non-limestone soils.

In the formal EPS extraction procedure, the extraction buffer was prepared (2 mM $Na_3PO_4 \cdot 12H_2O$, 4 mM $NaH_2PO_4 \cdot H_2O$, 9 mM NaCl, 1 mM KCl, pH adjusted to 7.4 with 1 M HCl), and the solution cooled to 4 °C in advance. CER (Amberlite™ IR120 Ion Exchange Resin, $Na^+$ form, 15–50 mesh, Sigma Aldrich) was thoroughly hydrated and washed using the

extraction buffer. Aliquots of air-dried soils (1 g) were weighed into 50 ml centrifuge tubes and amended with 25 ml 10 mM $CaCl_2$ solution. The suspension was then shaken gently for 30 min, and centrifuged at 4 °C at 4000 g for 30 min, after which the supernatant was carefully poured off to remove interfering extractable non-EPS microbial products such as free sugars and amino acids. Subsequently, EPS was extracted by adding 25 ml of extraction buffer and 10 g CER, and the mixture was shaken for 2 h before centrifugation at 4 °C at 4000 g for 30 min. The supernatant was then filtered through 0.45 μm filters (VWR® Syringe Filters, Nylon, 25 mm, Avantor), and the obtained solution was stored at 4 °C and used for the determination of EPS polysaccharides and EPS proteins within 4 d.

EPS polysaccharides were measured by acid hydrolysis and anthrone reaction using glucose as a standard[68]. The color reagent was produced by dissolving 0.2 g anthrone in 100 ml concentrated $H_2SO_4$ (previously cooled to 0 °C). For colorimetric assays, 200 μl of sample, standard, or blank (MQ) was added to 10 ml glass tubes, followed by the addition of 1 ml anthrone reagent. The mixture was then mixed well and heated to 100 °C in a water bath for 10 min. After rapid cooling to room temperature, the absorbance was measured at a wavelength of 625 nm.

EPS proteins were determined by a modified Lowry method[69], based on Cu-binding to polypeptides and the Folin-Ciocalteu reaction, with modifications to account for humic compound interferences in soil extracts. Bovine serum protein was used as the standard. Samples, standards, and blanks (MQ) were pipetted into microtiter plates, mixed with copper-containing or copper-free NaK tartrate/$Na_2CO_3$-NaOH buffer for 10 min, supplemented with Folin-Ciocalteu reagent, and after 30 min, EPS protein absorbance was measured at a wavelength of 750 nm.

Total EPS was approximated as the sum of EPS polysaccharides and EPS proteins[70,71]. This approximation is based on the fact that EPS polysaccharides and EPS proteins are the major components of microbial EPS, together contributing approximately 90% of total EPS[33,34]. This approach has been widely employed for estimating total EPS levels. However, we acknowledge that this method may slightly overestimate total EPS due to potential overlaps between components —for example, measurement of glycoproteins[33], which are partially detected in both colorimetric protein and polysaccharide assays. Despite this, the potential bias is likely small and considered acceptable for comparative purposes across sites. The EPS data used in this study have been provided as Supplementary Data 1.

## Estimation of EPS-C
The C content of EPS polysaccharides (39.1% C) was estimated by elemental analysis in our laboratory as the average of four microbial and plant exopolysaccharide standards, including hyaluronic acid (37.5% C), xylan (40.5% C), pectin (38.3% C), and arabinogalactan (40.2% C).

The C content of EPS proteins (50.7% C) was estimated from the global average protein N content (16% N), multiplied by a weighted mean C/N ratio of proteinogenic amino acids. Given that individual proteinogenic amino acids show variable C/N ratios, ranging from 1.5:1 to 9:1, and since proteins show marked deviations in their amino acid composition, we refrained from the use of a mean proteinogenic amino acid C/N ratio. Instead, we here applied a weighted mean amino acid C/N ratio calculated from the amino acid sequences of all genome-encoded polypeptides comprising the full proteome of the yeast *Saccharomyces cerevisiae* as an environmental microbe representative[72]. This gives a proteome-average of 5.03 mol C and 1.36 mol N per mol protein-derived amino acids[72], a molar C/N ratio of 3.70 (5.03 / 1.36 = 3.70), and a mass-based C/N ratio of 3.17 ((5.03 mol C × 12) / (1.36 mol N × 14) = 3.17). Using a global average protein N content of 16% by mass[73,74], and a weighted mass-based amino acid C/N in microbial proteins of 3.17, the corresponding protein C content can be

approximated by formula (1):

$$\text{protein C content}[\%C] = 16.0(\%N) \times 3.17\,(C/N) = 50.7\% \qquad (1)$$

These conversion factors were applied to measured EPS polysaccharide and EPS protein contents (in mg g⁻¹ soil dw) to estimate their respective C contents.

The final formula (2) for summed EPS-C is:

$$\text{EPS-C}\left(\text{g C kg}^{-1}\text{soil dw}\right) = \\ (\text{EPS polysaccharides} \times 39.1\%) + (\text{EPS proteins} \times 50.7\%) \qquad (2)$$

which provides an approximate estimate based on representative C content. To our knowledge, this represents an important attempt to quantify soil EPS-C contents, as such estimations were previously hindered by the lack of representative data on the C content of EPS components. In this study, we addressed this gap by determining average C content values from standard compounds representative of microbial EPS polysaccharides and by calculating protein C from proteome-averaged protein C/N ratios and published global protein N contents. While these estimates provide a useful approximation, we acknowledge that they do not fully capture the chemical diversity of EPS in soils. Based on EPS-C, we calculated the ratio of EPS-C to MBC. This EPS-C/MBC ratio has previously been termed EPS production efficiency, referring to the amount of EPS secreted by microorganisms in soils[15]. However, post-production processes, including EPS-soil associations and decomposition, can modify this ratio. While the concept is useful for characterizing short-term microbial EPS responses under conditions where these processes are negligible[51], in natural soils, the metric is more appropriately referred to as the EPS-C/MBC ratio and should be interpreted with caution. Additionally, we calculated the EPS-C/SOC ratio to represent the contribution of EPS-C to total SOC.

## Statistical analyses
Statistical analyses were performed and visualized using R version 4.1.2[75]. One-way ANOVA was applied to examine the effects of bedrock and land use on the target EPS variables, namely total soil EPS content, EPS polysaccharide and EPS protein contents, the EPS protein/EPS polysaccharide ratio (Supplementary Fig. 5), EPS-C content, the EPS-C/SOC ratio, and EPS-C/MBC ratio. Subsequently, we used linear regression models to analyze the relationships between EPS variables (total soil EPS content, EPS polysaccharide content, EPS protein content, and EPS-C/MBC ratio) and environmental and microbial factors. Three models were applied: (1) simple linear regressions to assess the effect of environmental and microbial factors on EPS variables, (2) multiple linear regressions with an interaction term to examine the combined effects of environmental and microbial factors and land use type, and (3) multiple linear regressions to evaluate the interaction between environmental and microbial factors and bedrock type (*lm* function in R). To assess the influence of environmental and microbial variables on EPS dynamics, we performed Spearman's correlation analysis. Moreover, principal component analysis (PCA) was employed to examine the impacts of environmental and microbial factors on EPS polysaccharides and EPS proteins (R package *stats*). Random forest models were used to rank the importance of controlling factors of EPS polysaccharides, EPS proteins, EPS-C, EPS-C/SOC, and EPS-C/MBC ratios (Supplementary Fig. 6) (R package *randomForest*). We examined the relationships between EPS-C, MNC, MBC, and SOC by fitting linear regressions to ln-transformed variables using the *lm* function in R. To investigate the effects of bedrock and land use type on environmental and microbial factors, we performed two-way ANOVA for each environmental and microbial variable. Variance partitioning analysis (VPA) was then conducted to quantify the relative contributions of bedrock

and land use type to the variation in these factors. To evaluate the effects of bedrock, land use type, and environmental and microbial factors on soil EPS-C content and the EPS-C/MBC ratio, we performed structural equation modeling (SEM) analysis (R package *lavaan*). Before constructing the SEM, we first examined collinearity among all environmental variables and removed highly correlated factors to reduce multicollinearity. After this selection process, the retained variables included ADI, FRB, MBN, $C_{growth}$, $q_{Growth}$, CUE, SOC, clay content, soil C/N, and $Ca_e$. The environmental variables were grouped into four categories: climate (ADI), plant (FRB), microbe, and soil. Among them, SOC, soil clay content, soil C/N, and $Ca_e$ were combined into a soil composite variable, while MBN, $C_{growth}$, $q_{Growth}$, and CUE were combined into a microbial composite variable, for both using PCA. We used the first principal component (PC1) to represent each composite variable, as it captured a substantial proportion of the variance (51.3% for soil, 62.5% for microbes) (Supplementary Fig. 1). Using this categorization, we developed an SEM model to explore how these four environmental and microbial components, along with bedrock type and land use, directly or indirectly influence EPS-C content and EPS-C/MBC ratio. Model fit was evaluated using multiple fit indices, including the Chi-square test ($\chi^2$), Comparative Fit Index (CFI), Root Mean Square Error of Approximation (RMSEA), Standardized Root Mean Square Residual (SRMR), and Akaike Information Criterion (AIC). In the best-fitting model, we calculated direct and indirect pathway coefficients to determine the key drivers of EPS-C/MBC ratio and EPS-C variation across climatic, geological, and land-use gradients.

## Data availability

The EPS data generated in this study are provided in Supplementary Data 1, including measurements of EPS polysaccharides, EPS proteins, total EPS, EPS polysaccharides-C, EPS proteins-C, EPS-C, and associated site information (site, region, land use, bedrock, latitude, longitude and elevation). A compiled dataset of EPS data extracted from previous studies is provided in Supplementary Data 2. Raw data underlying all figures are available in the Source Data files accompanying this paper. A subset of additional environmental variables and microbial data used in this study is available under restricted access due to data ownership and sharing restrictions; access may be granted by the corresponding author and with permission from the data owners. The raw environmental datasets cannot be made publicly available for this reason. Summary statistics of climatic, edaphic, plant and microbial variables (mean, SD and sample size, n) are provided in Supplementary Table 7. Processed datasets used for figure generation, together with summary statistics necessary to interpret the analyses and reproduce the presented figures, are included within the Supplementary Information and Source Data files. Source data are provided with this paper.

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

## Acknowledgements

This research was funded in whole or in part by the Austrian Science Fund (FWF) [grant DOI: 10.55776/P28037] (W.W.). This study was further supported by the National Key Research and Development Program of China (No. 2023YFD2200404 and No. 2021YFD2200402/3) (H.R.), and the program of China Scholarship Council (No. 202308320320) (K.S.). For open access purposes, the author has applied a CC BY public copyright license to any author-accepted manuscript version arising from this submission.

## Author contributions

K.S. drafted the manuscript, conducted laboratory work, and analyzed and interpreted the data. Q.Z., B.W., L.N., S.Z., and Y.H. performed laboratory work, contributed to data analysis, and assisted in manuscript editing. H.R. reviewed and revised the manuscript. W.W. designed the study, interpreted the data, and contributed to manuscript editing. All authors contributed to the manuscript and approved the final version for publication.

## Competing interests

The authors declare no competing interests.
