## [Peer Review file · Nature Communications]

Continental-scale drivers of soil microbial extracellular polymeric substances

Corresponding Author: Dr Wolfgang Wanek

Version 0:

Reviewer comments:

Reviewer #1

(Remarks to the Author)

Reviewer's Response:

This impactful and interesting study is centred in an emerging area of great relevance for soil health and climate science. Understanding the drivers of soil microbial EPS in Europe (continental scale) is an important step forwards in understanding the controls and relevance of this functional pool of soil C. The abstract, methods, and results are well written in the main. However, there are a few comments which need attention before publication, and I would not hesitate to recommend publication if these were resolved. Most importantly, the authors need to resolve a current conflict in their original manuscript:

- 1) acknowledging decomposition and stabilisation occur, but
- 2) confounding this within the (unnecessary) assumption that the observed EPS-C/MBC ratio = "EPS production efficiency". This is easily resolved.

Explanation:

The processes of decomposition and stabilisation occur after production, and are therefore potentially independent of the EPS production efficiency. Therefore, the units used throughout the manuscript (EPS-C/MBC) require limiting to this unit, and not interpreting as "EPS production efficiency". This change complements the thinking presented elsewhere in the manuscript (e.g line 323) - clarifying that the EPS pool measured using the standard approach (GER) is transient (subject to decomposition and stabilisation), as expected from i) established understanding on the nature of biofilms and EPS₁₃, and ii) as previously observed in grassland, cultivated, and fallow soils¹⁶ Nonetheless, as revealed in the present study: EPS is still very important as a (direct) carbon pool in its own right, and contributes manifold (indirect) effects relevant for C stabilisation in soil, as already communicated clearly in lines: 441-446.

Specific comments:

Line 25-26, This sentence is vague. Many studies have successfully rejected various null hypotheses and 'highly limited' can suggest poor (when I believe 'scarce' was meant). I suggest the authors keep to the main subject of the study (EPS-C and EPS-C/MBC ratio) and replace this sentence with "However, systematic evaluation of this claim has yet to be reported."

Line 27, unnecessary preamble. I suggest to start the sentence at "We conducted extensive soil sampling..."

Line 31 (and throughout manuscript - besides noted exceptions): "EPS production efficiency" should be independent of decomposition or stabilisation (as stated on Line 323). The EPS in these natural soils (presumably at approximate equilibrium) are subject to unknown and contrasting decomposition and stabilisation processes. Please present all as "EPS-C/MBC ratio" which more accurately reflects the measurements made.

Line 42-43: it is circular to explain EPS accumulation by the compound EPS-C/MBC ratio. While it would not be circular to say accumulation was governed by EPS production efficiency (suggesting decomposition and stabilisation were negligible, or not variable) this is not what was measured.

Line 43: replace “promoting” with “thought to promote”
Introduction

Line 63: meaning of “backbone” is unclear. This would suggest imparting physical strength but that is a role more typical of EPS proteins, please describe in clear terms.

Line 72: please revise “tested” to “reported”.

Line 76-77: EPS contribution to SOC depends on more than this: both stable and unstable. Replace “contribution” with “stabilisation”. Only long-term stable SOC relies on EPS stabilisation (not measured here). Reference 16 showed that stabilised EPS (persisting in soil for more than 2.5 years) was not extractable using the CER method. The fraction measured in the current study also represents non-stabilised EPS (as the method was intended 39). Stabilised, and/or partly decomposed materials are altered to the point at which they become difficult to characterise and quantify.

Line 79: please replace “and is calculated as EPS” with “and was calculated as EPS” to accurately reflect the cited works 17, 48

Line 80: Please insert the following sentence before “For instance” to clarify [if the authors agree?]

“However, we assume variable stabilisation and decomposition processes occur after EPS production. Thus while “EPS production efficiency” is a useful concept for characterising short-term EPS responses of the soil microbial biomass (where stabilisation/decomposition processes are assumed to be negligible)48 in studies of natural soils in comparative equilibrium (with time for soil-specific stabilisation and decomposition processes to occur) we present this metric simply as the EPS:MBC ratio. This not only represents the current state of equilibrium more accurately, but is consistent with the findings that current methods of soil EPS measurement (by CER extraction) do not quantify the decomposed or stabilised EPS fractions remaining after 2.5 years or more, and instead represent a transient -but more active- fraction of SOC16

Lines 80-86 continue smoothly: illustrating the point further. Except for: Line 80 “nutrient-rich” is vague. EPS production is known to depend on C availability 7, 17, 48 . Please re-examine and replace “nutrient-rich soil” with “soils rich in labile C”.

Line 82: what is “high microbial biomass”? The point missed here is that EPS can be produced without large increases in microbial biomass to acquire nutrients7, 17, 48. Please edit Line 81 to make clear: “For instance, in soils with free labile C, microorganisms tend to stimulate EPS production, serving as both a template for effective application of enzymes, and as a potential carbon source for subsequent metabolism7, 17, 48

Lines 81-83: accordingly, please delete the conflicting sentence which follows at Lines 81-83. (The text then leads on smoothly to the next good statement at lines 83-85).

Line 118: something wrong here. Please rewrite as “and in the latter case, what causes potential differences at the continental scale” if I understand correctly?

Line 120: please replace EPS production efficiency with simple EPS/MBC units (as with rest of manuscript) (I advise a full search of all figures and tables too).

Line 251: please replace EPS production efficiency in heading.

276-277: This is circular. Of course EPS-C scales positively with EPS-C/MBC. Please remove.

Figure 7: Axes are labelled correctly, titles and captions need production efficiency removing.

Figure 8: titles and captions need production efficiency removing.

322-323: Correct. Generation = production, so efficiency of EPS production is best not inferred from EPS and MBC concentrations in natural soils (or soils incubated for a long time). See comments above.

326-327: There is no evidence in the literature that stabilised EPS is extractable using CER. The evidence suggests the opposite is true16, 48. Perhaps the high EPS concentrations in carbonate soils suggests instead that carbonate soils (with their naturally improved structural stability, porosity, and heterogeneity) stabilise less EPS: representing instead reduced stabilisation or reduced “loss” to organo-mineral associations. This would make sense logically as a more homogenous distribution of (structureless) fine minerals (from non-carbonate soils) is likely to increase the formation of “stabilised” organo-mineral associations, and soil microbial communities do well in more structured media that can be expected from carbonate soils, with EPS being positively correlated with biodiversityX, Y (Wu et al., 2019; Zhang et al., 2024) Would the authors concur?

Recommended references X, Y:

Wu, Y., Cai, P., Jing, X., Niu, X., Ji, D., Ashry, N.M., Gao, C. and Huang, Q. (2019) Soil biofilm formation enhances microbial community diversity and metabolic activity. *Environment International*, 132, 105116. CrossRef Google Scholar PubMed

Wu, Y., Cai, P., Zhang, M., Qu, C., Huang, Q., and Cai, P. (2024). Microbial extracellular polymeric substances (EPS) in soil:

From interfacial behaviour to ecological multifunctionality. *Geo-Bio Interfaces*, 1, 2024. DOI: 10.1180/gbi.2024.4.

Line 333-335: As above, this is counter-intuitive, and likely arises from the previous view that stabilised EPS is extracted with CER (when existing evidence and theory suggests it is not). Please change this statement to also reflect the rationale above.

Line 335: "Sedimentary rocks can be rich..."

Line 340: "properties thought to promote..."

Line 367: "possibly due to relaxed..."

Line 376-379: to refine and shorten the narrative here I would recommend reconciling the previously mentioned literature in context. I suggest replacing the unnecessary sentence: "Nevertheless, these microbial diversity aspects were not effectively investigated in this study and warrant further exploration in the future". AND In conclusion, it is the interplay...." (repetition of land-use)(nonspecific EPS fractions).

With

"While microbial diversity was not investigated in this study, in accordance with investigations of legacy vs current management effects on EPS16, it is primarily the interplay between plant inputs and anthropogenic management which causes these differences in total and proteinaceous EPS concentrations".

Line 385: "Stabilization" was not measured, please remove.

Lines 385-388: Accordingly, please refine, or remove 'stabilization' from this sentence.

Line 422: Please insert a full stop after "on a continental scale".

Line 422: Please delete "which is mainly due to differences in their lability or stability" and deal with the main known artefacts first. We do not yet know how much of either fraction is stabilized in soils.

For a more conservative assessment, please insert "Incomplete extraction of EPS partly explains the difference between MNC and EPS-C observed here. Perhaps more importantly, amino sugar biomarkers do not exclusively represent necromass [reference Salas et al]. GalN for example can be found in substantial concentrations in plant residues, and other non-necromass materials, even EPS itself [reference Salas et al]. Therefore, while measures of EPS-C are understood to be underestimates, measures of MNC are known to be overestimates and perhaps with a larger margin for error and variability. This means that while on face value our results suggest that MNC was the larger pool of C, true values might be comparable. More research is needed to elucidate with confidence whether EPS-C or MNC contributes more to stable SOC formation. Nonetheless, our data for soils sampled at the continental scale show that both pools are important for (direct) carbon storage in soils – with EPS-C having been largely overlooked until now.

For information: From Salas (Salas et al., 2024): "Galactosamine (GalN) and mannosamine (ManN) are nonspecific amino sugars which can be found in plants (Indorf et al., 2011), fungal cell walls (Beauvais and Latgé, 2018), EPS of bacteria (Caruso et al., 2018; Cheah and Chan, 2022; Vaningelgem et al., 2004), and in the cell envelope of some archaeal species (Albers and Meyer, 2011).

Line 424: Please start new paragraph with "EPS is an active pool that is..."

Line 428-429: Please delete speculation "This passivity makes the contribution of MNC significantly higher than EPS-C" There is no soils data available to confirm or deny this. Stabilization of EPS in soils is not yet understood, and certainly not measured/reported.

Line 448: Please change title to reflect content - removing reference to EPS production efficiency, and instead reflect Drivers of EPS-C/MBC ratio, and EPS accumulation.

Line 449: To more accurately reflect the cited work, please change first line regarding reference 48 to: Changes in short-term microbial EPS production efficiency was previously calculated by standardizing soil EPS concentrations to concentrations of soil microbial ATP 48.

Line 451: Insert "However," before "as mentioned above"

453: Please replace this sentence with "Distinct bedrock and land-use types have shaped diverse ecological environments, and no doubt influenced decomposition and accumulation of EPS".

454: Before "a more detailed..." please insert "Therefore, we present EPS-C/MBC ratios "as is" without inferring EPS production efficiency". The rest of the paragraph should now flow smoothly.

Line 475: "Given the limited labile C..."

Lines 480-482: please simplify and abbreviate this circular sentence

Line 480-484: please reconsider and present in context of Bölscher et al., 2024: Bölscher, T., Vogel, C., Olagoke, F.K.,

Meurer, K.H.E., Herrmann, A.M., Colombi, T., Brunn, M., and Domeignoz-Horta, L.A. (2024). Beyond growth: The significance of non-growth anabolism for microbial carbon-use efficiency in the light of soil carbon stabilisation. *Soil Biology and Biochemistry*, 193, 109400. DOI: 10.1016/j.soilbio.2024.109400

487-489: I do not believe the evidence is strong enough to support this (complex) statement which seems to run counter to established understanding, please remove, or if the authors feel this is vital, contest this request?

487: Please replace the sentence here with information consistent with the referenced works "This trade-off means that C investment into EPS can only be maintained if sufficient labile C is available 17, 48" (Note that labile C cannot be inferred from SOC).

490: This bold statement may apply to the investigated soils, but is too sweeping as a universal statement. I urge the authors to reconsider, or simply remove.

492: This is clearly a mistake. Please remove this line and finish the paragraph as the rest (to line 494) is repetition/not necessary.

508: please remove "stabilization"

510: replace "can assist" with "may assist"

511: please delete "enforcing"

514: change 'production efficiency'

516-517: "Essential" is subjective and a little strong. For balance, please replace the sentence "it is essential to improve..." with "Addressing both the underestimations of EPS, and the overestimations of necromass respectively achieved by extraction will be important in clarifying our understanding of these formative carbon pools."

584: please remove "s" from "resins". The rest of the method section looks good! Thank-you for all the hard work that went into this interesting and very useful manuscript.

Best wishes and regards,
Dr. Marc Redmile-Gordon

Reviewer #3

(Remarks to the Author)

Review by Stefano Manzoni, Stockholm University

The manuscript by Ke Shi and co-workers explores spatial variations in extracellular polymeric substances (EPS) across Europe. This is a timely topic, as EPS plays important roles in modulating microbial activity in soils, with potential consequences for carbon and nutrient cycling. EPS spatial patterns and their drivers are largely unknown, so this is a welcome contribution. I should add that my background is in engineering (and modelling) and I don't feel I can evaluate the methodology developed for EPS analysis. Therefore, my review focuses on conceptual aspects and data analysis.

Overall, this is a well-written manuscript (though I would suggest some changes especially in the Discussion—see detailed comments below), with a clear structure and neatly-presented results. I like the progression from simple soil groupings (by land use or geology) to more detailed analyses of EPS dependence on various climatic, soil, and biological predictors. Nevertheless, I have some (relatively minor) comments on the interpretation the results.

Main comments

EPS production efficiency is defined as the ratio of EPS carbon (EPS-C) to microbial biomass carbon (MB-C). In my view, this is not a production efficiency, which would be rather defined as a rate of EPS production per unit microbial growth (or carbon uptake). My impression is that calling the EPS-C/MB-C ratio 'production efficiency' can be misleading. Indeed, some of the interpretations of this ratio in the Discussion mix up aspects related to EPS production and decay, which I agree are both important. I suggest to change the terminology to 'biomass specific EPS' or something along those lines. For example, in L276-277 it is stated that 'EPS production efficiency promotes EPS accumulation', but the data only indicate that EPS and EPS per unit biomass are correlated, suggesting that microbial biomass does not affect much the variation in EPS (if I interpret correctly). I agree with the statement in L322, which points to EPS as result of production and decay. But then the following statements introduce some confusion by interpreting production efficiency as the result of production and decay—I would clarify this part of the Discussion.

I struggle to reconcile the function of EPS and the activity of extracellular enzymes. Are extracellular enzymes targeting EPS? This is a general comment, and not meant to criticize this work—still, here both enzymatic activity and EPS are measured, raising the question as to how the two interact in this dataset. The activity of some enzymes appears to be positively correlated with EPS contents, suggesting that microbes produce both and that enzymes do not affect negatively EPS (or the negative interaction is less strong than the coordinated production). Would it be worth discussing these interactions?

Statistical analyses: even random forest approaches handle well collinearity, some of the variables included are very strongly correlated, so I wonder results would be more robust or easier to explain by selecting variables a bit more before running the random forest model and evaluating the importance rankings. Moreover, it is not clear how PC1 and PC2 were combined (L658-660).

Data availability: it would be useful to be able to access the data in an open access repository.

Other comments

General: EPS is expressed as carbon mass per unit mass of soil, so I would call it 'content' (mass/mass ratio) rather than 'concentration' (mass/volume ratio) (McNaught, A.D., Wilkinson, A., 1997. IUPAC Compendium of Chemical Terminology, Blackwell Scientific Publications). I realize the terminology in the literature is inconsistent to this regard, so I do not feel strongly in either direction.

Section 2.1: please define acronyms in the Results section, otherwise it is hard to read the text while continuously flipping pages to get to the definitions in the Methods.

Figure 2: this figure is very dense of information, so I wonder if it could be simplified e.g., by removing the correlation heatmap for the ratio of EPS polysaccharide to EPS protein. I might have missed the importance of this ratio (I think it is only briefly mentioned in the Discussion), and in any case it does not seem to be highly correlated with many drivers.

Figure 8: 'bedrock' text is not visible (black on black).

L211 and L417: where is the evidence of the strong association between WHC and EPS shown? WHC does not appear among the highly ranked drivers.

L319: was the methodology to estimate EPS in previous papers the same as used here? Can different measurement approaches explain the differences?

L329: the cited article compared two soils only, so I wonder if the evidence that in soils originated from sedimentary rocks microbes are more adapted to decompose EPS is really strong and worth highlighting here in the Discussion (but admittedly I only scanned the cited article...)

L368-369: alternatively, low MB-C can be linked to depleted soil organic matter or degraded soil after intensive agricultural practices.

L398: rather than 'contents' I would call these 'fractions' (mass or EPS per unit mass of soil organic carbon).

L400-404: this material was reported in the Methods and does not add much to the Discussion.

L416: not clear what 'amplifying effect' means.

L420: suggested change 'climate change effects in agricultural systems'

L426-427: please add rough time scales for decomposition of EPS and microbial necromass, as these statements are now quite vague.

L463: aridity is not a measure of drought severity or frequency of occurrence; to discuss drought effects, perhaps consider a climatic index that describe the intensity and/or frequency of dry periods.

L489-492: the logic here is not clear—you start by pointing to a difference between aquatic and terrestrial systems, but then you conclude that carbon and nutrient limitation promote EPS production in both.

L493-494: I did not understand how exoenzymes are related to the previous sentences on EPS.

L518: I agree this would be great to actually estimate EPS production rates.

Supplementary information: it would be helpful to provide captions of the supplementary Excel spreadsheets after the statement: 'Tables S3-S14 are provided as an attached Excel file'. If data is not available in an open access repository, perhaps it could be included as supplementary information.

Reviewer #4

(Remarks to the Author)

Overall, the authors tackle an interesting topic which highly discussed within the community. However, in my point of view that authors do not help in going a step further.

Please see my comments below.

There is no questioning of the data in relation to a mechanistic understanding. In my opinion, the authors should have an idea of possible connections, which they then test in order to really take a step towards a better understanding.

Everything was related to everything, but I miss the mechanistic idea behind it and its explanation. It would be better if the authors state clear hypothesis and test them.

Furthermore, I have several doubts in terms of the methodology used. Some examples

In the methodology section it is not becoming clear, which data have been taken from other studies and which data were analyzed on which kind of samples (Stored over how long!?) with which kind of purpose, several methods are just mentioned and the methodology is just referenced. But as a reader I expect to find the most important information within the paper.

As the authors did EPS extractions on dry soil stored for an unknown period of time between 2017 and 2025, which I would highly question to produce the same results as other studies of EPS usually extracting EPS from fresh soil samples (could be one reason why the authors gained higher EPS concentrations!?), I am also questioning how and under which conditions the other data have been gained, e.g. CUE!? Same with the turnover time (microbial biomass turnover time) – how was determined on the sample taken along the transect and why was it related to EPS – what is the expectation behind?

Furthermore, important calculations such as EPS-C is not described in detail – but used as a kind of indicator for EPS stabilization or accumulation (all of the terms used vague without clear definition). Since the calculation is not clearly explained in the method section, it feels to me as if the authors interpret the concentration as one or the other, depending on their discretion - as it suits the story better.

Overall, the authors draw conclusions that they do not prove and rely on speculation and, force and back correlation. In addition, they use words without defining them and defining their analyses and are therefore appearing very very vague.

Specific comments:

Abstract:

Line 24-25: EPS is part of the SOC, or? What do you want to say in regard to the “contribution”? Specifically, as you relate to it within the next sentence.

Line 26-27: At that point in time it is not clear to which theories you refer too. Please rewrite. What is the specific knowledge gap you want to fulfill, “contribution” to SOC storage – drivers/controls of EPS?

Line 29: Do you think that large scale controls matter for EPS? What is meant here by large-scale controls?

Line 42: Unclear, what the authors mean with EPS accumulation!?! Include as for EPS production efficiency... I don't think that prompted is the right verb here – rewrite

Line 45: In my opinion the authors should not write “critical component of the soil-stable C pool” – as at least to my knowledge it is not proven that it is part of the soil-stable C pool – and the authors do not prove that either in their manuscript – rewriting necessary

Introduction:

Line 50 “elements” - remove by components

Line 52-53: When I came across the number of 30-60% of SOC for amino sugars, is a contribution of 1.6% as mentioned for EPS substantial?

Line 55: “a natural mixed polymer” – as I understood EPS composition, it is composed by a variety of different constituents – please adapt

Line 70: I would wish that the authors elaborate a bit more on the “potential outstanding role of EPS... for SOC dynamics...” it is mentioned for the microbial functioning but for SOC dynamics it remains unclear to me at that point in time, specifically if their existence is debated overall in soil.

That points need more preparation in my point of view, specifically if that is a main topic for the whole manuscript.

Line 72: In regard of the existing literature the authors should check the current literature a bit stronger – e.g. Zhang et al. 2023 Chemical geology a whole table summarizing studies of EPS measurements from different soil types, including different land uses as well.

Line 74-75: The authors should introduce microbial necromass in regard to the before mentioned cellular and extracellular residues. Please better pave the way why EPS could/should effect necromass - it is not becoming clear to me where the differences between the components are.

Line 76-77: Is this statement assumed or verified by the papers referenced? I screened the article shortly and I could find a prove of it – Please be as precise as possible in your statements ... throughout the manuscript and adapt accordingly.

Line 87-95: Repetitive – can be more condensed and targeted

Line 97: As indicated beforehand, there is no prove of long-term preservation of EPS in soils – please add the prove to be able to make that statement or adapt accordingly

Screened shortly - Craig et al. did not include EPS measurements and the article by Chenu – no prove of preservation – assumption due was done due to the formed structures... and that needs be cleared up....

Line 106: The authors need to clarify for my better understanding what they mean by “accumulation” of EPS – are the authors using it simultaneously for preservation, stabilization and why do the authors think so.... EPS are from their composition relatively easy to degrade (proteins / sugars) – I would assume that EPS occurrence in soils is quite dynamic (as microbes use it as well as nutrient storage), so why an accumulation!?! Terms need to be clarified.

Line 107-108: What is the relation of the reference to the sentence? – it's starting to get exhausting – the authors studied one bedrock – what do the authors want to say – and is it really the best reference?

105-111: It would be preferred if the authors stated hypotheses which they tested in their study instead of saying that there are several out there. What specific hypotheses and procedures (compared to the other studies) guided their science? – What is new?

Line 113: Does that mean that the samples were stored from this study frozen?

Line 115-116: “What is the distribution of EPS on global scale” – this question is not well prepared in my point of view. I would assume that the EPS concentrations and compositions change with the microbes present in soil and their abundance – probably major differences on the small scale – so what would be the gain of information from a global distribution?

Line 118-119: Again I don't get the specification on the continental scale – needs clarification!

Results:

Line 136-137: What was analyzed, why and what was the result? Instead of referring that more detailed analysis can be found in the supplement....

Line 139: What is the soil total EPS? Why did the authors include it? is it sound to just sum up sugars and proteins? Could they not overlap? Please elaborate on that!

Line 145: turnover time!?! Turnover time of what!?! Same with the figures – clear that up!

Line 139-147: Instead of writing that you related everything to everything – it would be good if the authors stick to what they expected and what they found. A sentence over 9 line without any information – move to methods

By the way, Ca and Mg are nutrients and not really soil minerals!

Line 196-198: Might that be due to the EPS extraction method. There also different extraction method?

Line 211: The authors should be bit careful in their statements. For me it is not becoming clear how the authors come from a correlation to the guess that “indicating that EPS promotes soil water holding capacity more than bulk SOC or texture” – please revise or explain

Wild guesses and interpretations belong in the discussion – move it.

Line 227-228: Unclear how this numbers have been generated!

Line 235-237: Could it be that the authors extracted necromass, due to the soil storage and EPS extraction from dry soil?

Line 237-239: Interpretation into the discussion

Line 267: How did the authors investigate EPS stabilization!?!?

Line 268-270: Sorry, I don't get it. Ca was positively correlated with EPS and now it has a negative effect on EPS-C!?

Clarify! Heatmap of EPS driving factors – for both EPS polysaccharides and proteins compared to SEM? – to me it makes no sense

The authors base all of the presented data on EPS concentrations determined at one point in time. How was an EPS-C accumulation or stabilization determined? As it is not clearly

Line 270: Which kind of expectations? And what do the authors mean by lower PC (soil) values?

Line 276-277: refer to the data

Discussion:

It would be overall better if the authors stick to their data and what they can prove or reject instead of wild interpretations and assumptions!

Line 309: The author don't do a spatial analysis, so please be precise and write that the study has been conducted along a transect (not even covering Europe, but just some parts see Map of Noll et al). Clarify!

Line 315-316: See my comment above – the authors overlooked quite some studies....

Line 319-320: Maybe due to differences in EPS extraction from dry/wet soil and storage and EPS polysaccharide determinations!? Please include

If referring to explanations of different studies (I assume that previous studies can be checked in more detail in regard to your studied soils (ADI, carbonates and SOC – for example to check if your message can be underpinned by the other studies or not, instead of referring to a bunch of potential differences without a closer look, which is not advancing our knowledge.

Line 322-323: Ok, and now go a step further – what does that mean for your interpretations!? Formatting of the reference!

Line 326-327: The authors should elaborate a bit more on their assumption.

Line 337-339: But an overall higher microbial biomass would also mean more potential EPS producers, or? The more MBC the more EPS-C see your own figure – clarify

Line 339-341: To be able to follow your train of thoughts I would like to check the statement that the soils developed on sediment bedrock “were intermediate in properties promoting EPS stabilization” – I guess meaning low clay, low silt, Ca and Fe contents – that's what I would like to prove it.

Line 335ff: I would assume that the larger difference in nutrient can be found based on the land use instead of the bedrocks. Usually cropland and even grassland is fertilized. So, I would expect to find lower nutrients mainly in woodland. You have the data for the parent material and the land use – prove it, instead of referring to some references

Line 341-343: Also in the croplands, I would highly doubt that and you have data to prove it – do it!

Line 344: The authors should precisely use and explain which parameters they assume to influence EPS stabilization and then use them all accordingly, not once including more (Line 340), next time less (Line 344)

Line 345-346: Prove it! Is a low pH going along with reduced microbial parameters lower MBC, lower CUE,

Line 346-349: The authors did not prove the first step in my point of view, that the bedrock is making the difference in soil properties (not differences in bedrock chemistry – please be precise) – I can find that data. The authors found significant different EPS concentrations, the rest is speculation.... Revise it accordingly

Line 462-463: The authors found a weak negative correlation line 261- 262 – so is it worse pushing it here – or would be better to try to discuss instead of refreshing what others propagated but have been seldom proven.

Line 467-468: refer to the data – where can I find it as a reader – such a figure with this correlation would be nice to see in the supplement!

Line 331-333: What do the authors mean by “promote adsorption”? I thought the EPS is bound via Ca-bridging!? Clarify

Line 33-335: How did the authors study sorption and stabilization controls on EPS concentrations (maybe I missed that point?) – it can't be evident from a correlation!!!! Refer to the data that all these properties are highest in the carbonate soils – I would wish to prove that.

Line 337: The authors should better let us participate in their nebulous explanations, what is meant here by “positive feedback mechanisms”.

Line 350-364: Condense and stick to the data you have.

366-368: Please prove it and how does that fit with the discussion beforehand? Did you have cropland on silicate bedrock!?

Line 368-370: Do you prove if the lowest plant root input in cropland caused the lowest MBC levels!? I always thought it is due to the overall management (mechanical and chemical interventions)!

Maybe the croplands are just lowest in EPS stabilization properties? Just sarcastic joke... please revise your discussion along your data and what you can prove and not prove. Instead of interpreting your data once like that and then the other way around as it better suits your story.

Line 373-374: How is agricultural management reducing stabilization properties? And if you find overall lower MBC, how will the management increase EPS degradation?

Line 382-385: Which difference reflects what! Explain!

Line 386: How can the authors come up with an EPS accumulation!? You did not study that, or?

Line 388-393: here the authors open up another discussion box – which they also can't prove – delete – unnecessary for the study

Line 398-400: It would help if the authors include a precise description of the calculation.

Line 401: Prove that number!

Line 401-404: But microbes produce a variety of EPS sugars and proteins – even single microbes can produce different EPS constituents – how does that relate to your estimation based on one species!?

Line 404: Still unclear how you then come up with EPS-C (summing up!)?

Line 413: what is meant by highly active soil C pool? How can a highly active C pool promote SOC stabilization – seem contradicting to me?

Line 415-416: “amplifying effect on the soil C cycle”? – please stick to your data!

Line 416-419: Sorry, I can't follow this line – I can't find a prove of this statement within this manuscript!

Line 419- 420: Coming up with such statement after demonstrating such a low contribution to SOC is really confusing!
Line 421-440: Why is this part of the discussion here – separate section – if the authors want to point out that necromass is important!

Line 421-446: Stick to your data and not conclude assumptions! Rewrite and condense

Line 477-478: Why do the authors think so? If EPS is a kind of survival strategy C would rather allocate into EPS, then growth, or? How do the authors come up with their assumption – based on what?

Line 484-485 and Line 492: Prove that – refer to data and how do the authors interpret the positive correlation between MBC and EPS, the higher the microbial biomass the higher EPS – as shown by your data – how is that fitting to your argumentation?

Material and Methods:

4.2. Sampling:

Line 544-546: It is still unclear how many samples per site are analyzed. What forms the replicates? How where the cluster build, based on which assumptions? It is not enough to give citation in regard to its importance.

Line 550: Which data have been taken within these study and which data come from other studies? Needs clarification.

Line 543-544: The authors need to clarify when the analyzes were done (sampling in 2017) and if the storage of the soil samples can change EPS concentrations. Microbial parameters are variable depending on the soil storage. Please clarify! How comparable are EPS extraction from dry and wet soil samples? For dry samples usually provide higher SOC extracted due to the dead organisms – don't you extract completely different substances?

Line 592-596: I shortly checked the method by Redmile-Gordon and the amount of CER added to fresh soil is depending on the SOC content. Is it correct that the authors haven't done that!?

Line 622-626: The estimation of EPS-C is a central part of the manuscript. The authors need to give more informations on how the EPS-C estimation (calculation) has been done instead of referring to a reference. It is necessary to a reader to be able to follow and understand the manuscript.

Version 1:

Reviewer comments:

Reviewer #1

(Remarks to the Author)

Dear Authors

Many thanks for your responses and updates to the manuscript. I look forward to seeing this excellent and extensive work published - after the following elements are actioned or discussed:

Line 27: To say factors influencing EPS are "poorly quantified" can be interpreted in several ways. To avoid the derogatory sense and misunderstanding, I would suggest a change to "Yet, their abundance and large-scale environmental and microbial controls have only begun to be investigated." ... the subsequent sentence can then simply continue with "We conducted extensive soil sampling..."

This solves the problems, and emphasises the novelty of your work - without adding potential for misunderstanding.

Line 92: No need to be negative here: "which has hindered understanding of large-scale controls of soil EPS". These works have not "hindered" understanding, on the contrary, your study is a logical progression from these sensible forerunning investigations. Please delete this unnecessary critique (also, the scale limitations of others' works are already made clear in lines 86 to 89).

Line 107: There are several studies underway. This sentence does not make sense – presumably after the preceding changes were made. To correct, please change "remain largely unconstrained" to "are much needed" – highlighting the importance of this body of work.

Lines 183 to 184 state: "EPS proteins differed significantly across land use types (grassland > woodland 184 / cropland; Fig. 3d), while showing no significant variation with bedrock (Fig. 3c)". Correct. However, currently the discussion appears to suggest that land-use has *no effect* which contradicts the present statement, and the state-of-the-art. This result (effect of land-use on EPS protein content) is significant and should be presented as being in accordance with ref#16 (effects of land-use on EPS protein and soil properties) in the discussion (proposed below):

Line 308 – 309: currently in error, and can be improved. "Meanwhile, EPS contents did not differ significantly among land use types but tended to be highest in grasslands". This is followed by speculation on mechanisms and a link to Supplementary Discussion 5 without clarifying the main effects (or comparing to relevant literature in the main discussion). Figure 3 shows 1) a statistically significant effect of [bedrock] on [EPS polysaccharide], and 2) a statistically significant effect of [land-use] on [EPS protein]. This should be much clearer in the discussion.

To resolve all of the issues above, please replace the following text "Meanwhile, EPS contents did not differ significantly among land use types but tended to be highest in grasslands, with microbial biomass, nutrient availability, and plant inputs mediating the land-use based variation; in contrast," with "We found land-use was not a significant driver of EPS

polysaccharide, but was a highly significant driver of EPS protein (Figure 3). This adds important depth to land-use research. Redmile-Gordon et al., (2020) 16 found land-use change to perennial grassland increased soil EPS and aggregate stability with EPS protein being far more influential for soil structural stability than EPS polysaccharide¹⁶. Our findings, that both EPS protein and EPS-C are elevated under grassland (across contrasting soil bedrocks) supports their assertion that CER-extracted EPS protein is more physically influential than polysaccharides in the stabilisation of soil aggregates¹⁶ - and thus also C stabilisation.” The narrative can then smoothly continue with “EPS-C/MBC ratios peaked in croplands, likely reflecting enhanced microbial EPS investment under water limitation (Supplementary Discussion 4)...

Line 343 to 344: There has been a small mistake, currently misleading: “EPS is a transient pool that is selectively secreted by microorganisms in response to adverse environmental conditions, with no measurable legacy effects observed beyond 2.5 years¹⁶. The cited work (16) found higher EPS protein where there was no disturbance, certainly not “adverse conditions” – however, the study was not set up to determine effects of adversity. Also, there were no measurements conducted between 2.5 years and 50 – so these EPS could still have been measurable after 10 years, 20 years (we simply don’t know more than they are undetectable after 50). So the above statement is incorrect. Accordingly, please delete all of the following, including the unfounded speculation about MNC being more resistant to decomposition (we simply don’t know yet until measurements of EPS at 10, 20, 30 years are conducted):

“EPS is a transient pool that is selectively secreted by microorganisms in response to adverse environmental conditions, with no measurable legacy effects observed beyond 2.5 years¹⁶. In contrast, MNC as the inevitable product of microbial turnover and death, is more resistant to degradation, with turnover times ranging from multiple years to decades^{43, 44}”.

Please also delete the paragraph break so that this discussion on EPS-C, MNC, and SOC formation is together.

Finally: Supplementary discussion 5 has unsupported speculation on roles in soil structure: “EPS polysaccharides are closely associated with soil Ca²⁺, highlighting their role in reinforcing soil structure, whereas EPS proteins are more strongly linked to microbial biosynthetic activity (qGrowth)”.

While the association might be suggestive of something, such as dependency on Ca²⁺ (for production, structure, or otherwise) proteins are typically more formative in soil structural stability - and don’t require increased Ca²⁺ availability to fulfil that function¹⁶. For simplicity, please just delete the speculation “highlighting their role in soil structure” as it would not be appropriate to use an association with Ca²⁺ to infer structural relevance of EPS polysaccharide over EPS protein in this way.

That’s all.

Good luck with the other referees, and thanks for the interesting study!

Reviewer #3

(Remarks to the Author)

I reviewed an earlier version of this work and find the revisions have improved it. I have three remaining comments (and a few specific ones), listed below:

- Soil moisture content (SMC) as predictor of EPS content: SMC at the time of measurement (L469-470) might not be representative of the soil moisture regime, or even the moisture levels in the previous week, so I am not sure it is a meaningful predictor of EPS. Also, SMC likely correlates with silt and clay content, as fine texture soils hold more water than sandy soils. I suspect the correlations found between EPS and SMC might be mirroring those with clay content (or water holding capacity, which might also be correlated with clay content). I get the impression that removing the EPS-SMC correlations (Figures 9-10) would not change the main messages.

- Aridity index: since potential evapotranspiration (PET) has already been calculated for all the sites, I wonder why not defining aridity index in the usual way as ratio of mean annual precipitation (MAP) to mean annual PET? The current definition ($ADI = MAP / (MAT + 33)$) yields a quantity that has strange dimensions of length per degree of temperature. (Please also note that the units should be reported when values of ADI defined in this way are reported or shown, as it is not a non-dimensional quantity.) I would suggest to use the usual ADI definition instead, to make comparisons with other studies simpler—though I doubt such a change would alter the results.

- Figure 6: I wonder if power law relations might better represent the patterns in this figure, while also reducing heteroscedasticity issues (at least by eye it seems variations are higher at higher values of SOC, MBC, MNC). Also, power laws $y = a \cdot x^b$ pass through the origin of the axes, while the linear regressions now predict non-zero MBC, MNC, and EPS-C even at zero SOC, which is not very meaningful.

Specific comments:

L83-93: this part of the paragraph seems out of place, as it presents definitions of EPS metrics, while the rest of the paragraph is on EPS functions

L94: not sure what “template” means in this context

L152: typo “not significantly”

L277: language here is a bit vague—what are the functional traits involved? They are listed thereafter, but as written, the statement ending in L277 appears not conclusive

L358: I would refer to EPS-C/MBC ratio instead of “EPS production efficiently” as elsewhere in the manuscript

L384: why would conditions be more stressful in soils developed from sedimentary bedrock? I would remind of the possible reasons

L570: even at steady state the ratio of the EPS and microbial biomass contents does not always reflect the ratio of the rates of EPS production over biomass production (unless the turnover rates of EPS and biomass are exactly the same)

Reviewer #4

(Remarks to the Author)

The authors substantially reworked their manuscript and improved it.

However, I still have doubts about the results and outcomes of the study.

First of all, and most importantly, the methodology. EPS has been extracted from dry soil samples, usually done on fresh soil samples, although it is known that microbial processes change on short-timescale and also based on sample preparation and storage. Furthermore, the EPS extraction method used is not solely specific in its extraction and we know that soils flash C, getting available, after drying soil samples. Overestimation is quite likely; authors show higher EPS-sugar data than other studies (Figure 1). So, what do the authors study!?

Sorry, but I still don't find the response convincing: In a reference used in the response to my caution with regard to the method, the authors showed 25 % more sugar's – so a strong overestimation of C within the EPS after extracting from dry soil material – authors tried to convince that the reduction in the other measured components (e.g. proteins going down) would compensate for this, as the sum is not changing much – not convincing; Their own analysis (no reference to the data in the manuscript – see below) did not show large variation, in the response the results are given as mean and SE, could you give it as SD and refer to data and discussion in the supplement

Overall, I strongly question whether it makes sense to combine the various EPS components, as there could also be overlaps between the different components.

I would recommend discussing the topic even more openly. Line 318..... : This differences could either come from... or could be due to sample treatment....

Line 323: Refer to data in the supplement

Secondly, the authors still use vague expressions – which are misleading and new terms without real definitions - and thus without background and verifiability. In addition, it remains unclear to an inexperienced reader what has been verified or is just assumptions, as this is not made clear. Just some examples from the Abstract: "EPS retention" – some parameters increase EPS retention, but what is EPS retention!? "environmental filters" – regulated by environmental filters – no clue; it is still not clear if EPS has a significant implication on Soil carbon cycling and sequestration – that should be formulated like that and not just claimed - The authors show themselves, EPS-C contribution of 0,3-3,9% of SOC against 30-60% necromass contribution to SOC – significance?

I would recommend the authors to critically review their text once again in this regard.

Minor comments:

Line 338-339 – prove it for MBC/MBN – you have the data

Line 351-353: It's going too far in my point of view; WHC is influenced by SOC and texture – both parameters driving EPS – so is it not just a false correlation; which occurs when two variables show a statistical relationship, but one does not cause the other; this can be due to a third confounding variable influencing both, such as SOC and texture!?

Line 364: double prove your interpretation by checking if the sedimentary soils are more often found in arid conditions – I would have thought that carbonate rich soils are more often found under arid conditions – thus it is counterintuitive – please prove

Line 390-391: Is water limitation the only stress potentially triggering microbes to produce EPS!?

Line 408-409: see my comment above about false correlations – prove what is driving WHC – you have the data, and check autocorrelations

Line 451: Check the Bublitz et al.

Overall, I would recommend subheadings in the discussion based on the questions set now for the manuscript, which gives the authors the chance to come back to them more clearly in the discussion.

I would also recommend to reduce the amount of figures (10 until now) within the manuscript and move figures which might not as necessary to the supplement.

Reviewer #1 (Comments to the Author):

[Comment 1] This impactful and interesting study is centred in an emerging area of great relevance for soil health and climate science. Understanding the drivers of soil microbial EPS in Europe (continental scale) is an important step forwards in understanding the controls and relevance of this functional pool of soil C. The abstract, methods, and results are well written in the main. However, there are a few comments which need attention before publication, and I would not hesitate to recommend publication if these were resolved. Most importantly, the authors need to resolve a current conflict in their original manuscript:

1) acknowledging decomposition and stabilisation occur, but

2) confounding this within the (unnecessary) assumption that the observed EPS-C/MBC ratio = “EPS production efficiency”. This is easily resolved.

Explanation: The processes of decomposition and stabilisation occur after production, and are therefore potentially independent of the EPS production efficiency. Therefore, the units used throughout the manuscript (EPS-C/MBC) require limiting to this unit, and not interpreting as “EPS production efficiency”. This change complements the thinking presented elsewhere in the manuscript (e.g line 323) - clarifying that the EPS pool measured using the standard approach (CER) is transient (subject to decomposition and stabilisation), as expected from i) established understanding on the nature of biofilms and EPS13, and ii) as previously observed in grassland, cultivated, and fallow soils¹⁶ Nonetheless, as revealed in the present study: EPS is still very important as a (direct) carbon pool in its own right, and contributes manifold (indirect) effects relevant for C stabilisation in soil, as already communicated clearly in lines: 441-446.

[Response 1] Thank you very much for your constructive and insightful comment. We fully agree that EPS in soils represents the dynamic balance between production, decomposition, and stabilization processes. Current extraction methods do not allow for accurate quantification of either actual EPS stabilization or its production and degradation dynamics. To investigate the formation, stabilisation, and decomposition processes of EPS, isotope (e.g., ¹³C) tracing approaches coupled to EPS-specific biomarker analysis are required but not yet available or validated ¹. Based on current extraction methods, it is indeed inappropriate to interpret the EPS-C/MBC ratio as a direct indicator of “EPS production efficiency” in natural soils. In response, we have revised the manuscript accordingly and now refer to this metric simply as the “EPS-C/MBC ratio”. Additionally, we provide a more accurate and cautious interpretation of its ecological relevance, as follows:

“Although this ratio does not directly reflect EPS production efficiency – since EPS contents in natural soils result from the balance of microbial production and degradation,

modulated by mineral interactions and extraction limitations – it may serve as a proxy for the relative allocation of microbial C assimilates to extracellular non-growth products under steady state conditions” (Pages 10, Lines 283-287; all page and line numbers refer to the revised clean version without track changes).

“Changes in short-term microbial EPS production efficiency were previously calculated by standardizing soil EPS to soil microbial ATP², a proxy for microbial biomass. This EPS-C/MBC ratio may be used to assess the potential of microorganisms to secrete EPS under various environmental conditions². However, as mentioned above, the accumulation of soil EPS is contingent on the dynamic balance between its generation, stabilization, and degradation” (Page 12, Lines 358-362).

At the same time, we have added relevant content in the Introduction (see our response to Reviewer 1’s comment 11, R1-11; the same applies below). Throughout the manuscript, “EPS production efficiency” has been replaced with “EPS-C/MBC ratio”, consistent with our response R1-4. Additional modifications related to the EPS-C/MBC ratio can be found in our response R3-2.

[Comment 2] Line 25-26, This sentence is vague. Many studies have successfully rejected various null hypotheses and ‘highly limited’ can suggest poor (when I believe ‘scarce’ was meant). I suggest the authors keep to the main subject of the study (EPS-C and EPS-C/MBC ratio) and replace this sentence with “However, systematic evaluation of this claim has yet to be reported.”

[Response 2] Thank you for your helpful comment. We have removed the original sentence “However, despite various conjectures and hypotheses regarding soil EPS controls, empirical research and experimental evidence to validate these theories have remained highly limited” and, following your suggestion, further condensed it to: “**Yet, their abundance and large-scale environmental and microbial controls remain poorly quantified**” (Page 3, Lines 26-27).

[Comment 3] Line 27, unnecessary preamble. I suggest to start the sentence at “We conducted extensive soil sampling...”

[Response 3] We have revised this sentence from “In this study, we addressed this knowledge gap by conducting extensive soil sampling across Europe, encompassing diverse climates and bedrock and land use types, to systematically investigate soil EPS contents and large-scale controls” to “**We addressed this gap by conducting extensive soil sampling across a European transect spanning diverse climates, bedrock types, and land uses**” (Page 3, Lines 27-29).

[Comment 4] Line 31 (and throughout manuscript - besides noted exceptions): “EPS production efficiency” should be independent of decomposition or stabilisation (as

stated on Line 323). The EPS in these natural soils (presumably at approximate equilibrium) are subject to unknown and contrasting decomposition and stabilisation processes. Please present all as “EPS-C/MBC ratio” which more accurately reflects the measurements made.

[Response 4] Thank you for your insightful comment. We have recognized that the term “EPS production efficiency” was inaccurately used in the manuscript. In response, we have revised the text to adopt the more precise term “**EPS-C/MBC ratio**” in the abstract and consistently throughout the manuscript to ensure clarity and accuracy. See also our responses **R1-1**, **R1-11**, **R-16**, and **R3-2**.

***[Comment 5]** Line 42-43: it is circular to explain EPS accumulation by the compound EPS-C/MBC ratio. While it would not be circular to say accumulation was governed by EPS production efficiency (suggesting decomposition and stabilisation were negligible, or not variable) this is not what was measured.*

[Response 5] Thank you for your helpful comment. We have revised this sentence from “On a large scale, soil EPS accumulation was promoted by its production efficiency and by soil factors promoting the sorption and stabilization of EPS, such as clay content, exchangeable Ca and Fe oxides” to “**Across the transect, EPS increased with microbial biomass as a proxy for EPS production potential, while soil properties such as clay content and exchangeable calcium enhanced EPS retention**” to avoid circular reasoning and provide a more mechanistic and cautious explanation of the drivers of soil EPS (**Page 3, Lines 34-36**).

***[Comment 6]** Line 43: replace “promoting” with “thought to promote”.*

[Response 6] The wording has been revised to improve clarity. We observe a strong positive relationship between microbial biomass carbon (MBC) and EPS, indicating that microbial community size scales with EPS production potential. Therefore, we have revised this sentence from “soil EPS accumulation was promoted by its production efficiency” to “**EPS increased with microbial biomass as a proxy for EPS production potential**” (**Page 3, Lines 34-35**).

***[Comment 7]** Line 63: meaning of “backbone” is unclear. This would suggest imparting physical strength but that is a role more typical of EPS proteins, please describe in clear terms.*

[Response 7] In response, we have revised the inaccurate term and changed the original sentence from “EPS polysaccharides are important backbone components of biofilms, the strength and structural diversity of which are promoted by EPS proteins” to “**EPS polysaccharides are important components of biofilms, the strength and structural**

diversity of which are promoted by EPS proteins” to improve clarity and accuracy (Page 4, Lines 56-57).

[Comment 8] Line 72: please revise “tested” to “reported”.

[Response 8] The sentence has been removed as part of the revisions to refine the introduction.

[Comment 9] Line 76-77: EPS contribution to SOC depends on more than this: both stable and unstable. Replace “contribution” with “stabilisation”. Only long-term stable SOC relies on EPS stabilisation (not measured here). Reference 16 showed that stabilised EPS (persisting in soil for more than 2.5 years) was not extractable using the CER method. The fraction measured in the current study also represents non-stabilised EPS (as the method was intended 39). Stabilised, and/or partly decomposed materials are altered to the point at which they become difficult to characterise and quantify.

[Response 9] Thank you for your important and insightful comments. We agree that the term “contribution” was too broad and potentially misleading in this context. As suggested and in response to reviewer 4’s comment 17 (R4-17), we have revised this sentence from “The contribution of EPS to SOC depends on (i) the microbial secretion of EPS and (ii) its long-term protection by binding with soil particles^{3, 4}” to “**The content of soil EPS may depend on (i) the microbial secretion of EPS and (ii) its association with soil particles^{5, 6, 7}, as well as (iii) its decomposition and recycling**” to more accurately reflect the role of EPS in SOC dynamics (Page 5, Lines 80-81). See also our response R1-22 on EPS stabilisation mechanisms, and that the CER approach only extracts a fraction of EPS that is weakly bound and intermittently stabilized.

[Comment 10] Line 79: please replace “and is calculated as EPS” with “and was calculated as EPS” to accurately reflect the cited works 17, 48.

[Response 10] We revised the sentence as suggested to improve accuracy (Page 5, Line 84).

[Comment 11] Line 80: Please insert the following sentence before “For instance” to clarify [if the authors agree?] “However, we assume variable stabilisation and decomposition processes occur after EPS production. Thus while “EPS production efficiency” is a useful concept for characterising short-term EPS responses of the soil microbial biomass (where stabilisation/decomposition processes are assumed to be negligible)48 in studies of natural soils in comparative equilibrium (with time for soil-specific stabilisation and decomposition processes to occur) we present this metric simply as the EPS:MBC ratio. This not only represents the current state of equilibrium more accurately, but is consistent with the findings that current methods of soil EPS

measurement (by CER extraction) do not quantify the decomposed or stabilised EPS fractions remaining after 2.5 years or more, and instead represent a transient -but more active- fraction of SOC¹⁶.”

[Response 11] Thank you for your kind contribution of this text block that clarifies the weaknesses of using EPS-C/MBC as EPS production efficiency, related to downstream EPS processing and incomplete determination of soil EPS by the CER method. We agree that using the term EPS-C/MBC ratio provides a more accurate representation of the current state condition, rather than referring to it as EPS production efficiency. We have revised the manuscript accordingly and incorporated this text block into the main text (**Page 5, Lines 84-93**). See also response **R1-1** and **R3-2**.

[Comment 12] *Lines 80-86 continue smoothly: illustrating the point further. Except for: Line 80 “nutrient-rich” is vague. EPS production is known to depend on C availability 7, 17, 48. Please re-examine and replace “nutrient-rich soil” with “soils rich in labile C”.*

[Response 12] The sentence “For instance, in nutrient-rich soils, microorganisms tend to stimulate EPS secretion to store it as an additional C resource⁸” has been revised to “**For instance, in soils rich in labile C, microorganisms tend to stimulate EPS production, which serves both as a template for effective application of extracellular enzymes, and as a potential C resource for subsequent metabolism^{8, 9, 10}**” as suggested to improve accuracy (**Page 5, Lines 93-96**).

[Comment 13] *Line 82: what is “high microbial biomass”? The point missed here is that EPS can be produced without large increases in microbial biomass to acquire nutrients 7, 17, 48. Please edit Line 81 to make clear: “For instance, in soils with free labile C, microorganisms tend to stimulate EPS production, serving as both a template for effective application of enzymes, and as a potential carbon source for subsequent metabolism 7, 17, 48*

[Response 13] We apologize for the earlier expression. In response, we have revised the sentence from “For instance, in nutrient-rich soils, microorganisms tend to stimulate EPS secretion to store it as an additional C resource⁸” to “**For instance, in soils rich in labile C, microorganisms tend to stimulate EPS production, which serves both as a template for effective application of extracellular enzymes, and as a potential C resource for subsequent metabolism^{8, 9, 10}**” to reflect your suggestion (**Page 5, Lines 93-96**).

[Comment 14] *Lines 81-83: accordingly, please delete the conflicting sentence which follows at Lines 81-83. (The text then leads on smoothly to the next good statement at lines 83-85).*

[Response 14] As suggested, we have deleted the conflicting sentence: “This means that when soil microbial biomass is high and microbial C and nutrient demands are met, microorganisms promote the secretion of EPS after adequate self-synthesis” to improve clarity and ensure logical flow.

[Comment 15] Line 118: something wrong here. Please rewrite as “and in the latter case, what causes potential differences at the continental scale” if I understand correctly?

[Response 15] Thank you for your careful observation. After careful consideration, and to maintain focus and clarity of the manuscript, we have removed this sentence “Do the two primary components of EPS, EPS polysaccharides and EPS proteins, follow the same or different drivers, in the latter case causing EPS composition to diverge?” from the text. We believe this does not affect the overall logic or conclusions of the study.

[Comment 16] Line 120: please replace EPS production efficiency with simple EPS/MBC units (as with rest of manuscript) (I advise a full search of all figures and tables too).

[Response 16] In response, we have replaced the term “EPS production efficiency” with the consistent use of “EPS-C/MBC ratio” throughout the manuscript, as well as in all relevant figures and tables, as advised.

[Comment 17] Line 251: please replace EPS production efficiency in heading.

[Response 17] We have modified the heading accordingly (**Page 8, Line 210**).

[Comment 18] 276-277: This is circular. Of course, EPS-C scales positively with EPS-C/MBC. Please remove.

[Response 18] We have removed the sentence “Moreover, EPS-C scaled positively with EPS production efficiency, showing that EPS production efficiency promotes EPS accumulation” accordingly.

[Comment 19] Figure 7: Axes are labelled correctly; titles and captions need production efficiency removing.

[Response 19] We have revised **Fig. 7** and its caption accordingly.

[Comment 20] Figure 8: titles and captions need production efficiency removing.

[Response 20] We have revised **Fig. 8** and its caption accordingly.

[Comment 21] 322-323: Correct. Generation = production, so efficiency of EPS production is best not inferred from EPS and MBC concentrations in natural soils (or soils incubated for a long time). See comments above.

[Response 21] Thank you for your insights. We now understand that EPS production efficiency cannot be reliably inferred from EPS and MBC concentrations in natural soils or soils stored for extended periods. Consequently, we recognize that our previous explanation—attributing higher EPS concentrations in carbonate soils to EPS stabilization, and higher EPS-C/MBC in sedimentary soils to increased microbial production efficiency—was not sufficiently supported and may be misleading. To avoid potential overinterpretation, we have removed the sentence “Generally, soil EPS concentrations are governed by the dynamic balance between its generation, stabilization and decomposition” which was previously used to summarize this paragraph. The relevant paragraph has been revised accordingly; please refer to our response **R1-1**, **1-11**, **R1-22**, and **R3-2** for further details.

[Comment 22] 326-327: There is no evidence in the literature that stabilised EPS is extractable using CER. The evidence suggests the opposite is true^{16, 48}. Perhaps the high EPS concentrations in carbonate soils suggests instead that carbonate soils (with their naturally improved structural stability, porosity, and heterogeneity) stabilise less EPS: representing instead reduced stabilisation or reduced “loss” to organo-mineral associations. This would make sense logically as a more homogenous distribution of (structureless) fine minerals (from non-carbonate soils) is likely to increase the formation of “stabilised” organo-mineral associations, and soil microbial communities do well in more structured media that can be expected from carbonate soils, with EPS being positively correlated with biodiversity^{X, Y} (Wu et al., 2019; Zhang et al., 2024) Would the authors concur?

Recommended references X, Y:

Wu, Y., Cai, P., Jing, X., Niu, X., Ji, D., Ashry, N.M., Gao, C. and Huang, Q. (2019) Soil biofilm formation enhances microbial community diversity and metabolic activity. Environment International, 132, 105116. CrossRefGoogle ScholarPubMed

Wu, Y., Cai, P., Zhang, M., Qu, C., Huang, Q., and Cai, P. (2024). Microbial extracellular polymeric substances (EPS) in soil: From interfacial behaviour to ecological multifunctionality. Geo-Bio Interfaces, 1, 2024. DOI: 10.1180/gbi.2024.4.

[Response 22] Thank you for your important and valuable comments. We now understand that the assumption that stabilized EPS can be extracted by cation exchange resin (CER) is not supported by the literature, and the statement “This suggested that high EPS concentrations in carbonate soils may depend on their stabilization processes” has been removed to avoid potential misinterpretation. To clarify, EPS in soils is stabilized via multiple mechanisms^{5, 8, 11}, including (i) physical occlusion inside

microaggregates, (ii) co-precipitation and complexation with Fe/Al, (iii) hydrogen bonding, (iv) hydrophobic interactions, (v) inner-sphere ligand exchange to Fe/Al (oxyhydr)oxides and clays, (vi) outer-sphere electrostatic attraction, and (vii) multivalent cation bridging between negatively charged sites of EPS and those on other polymers and minerals. The importance of these mechanisms differs between main EPS compounds, with EPS polysaccharides mainly bound/stabilized by (ii, iii, v, vii) and EPS proteins via (ii, iii, iv, v, vi, vii). The cation bridging mechanism via Ca^{2+} (vii) and electrostatic attraction (vi) causes a loosely bound (not strongly stabilized) association, which is reversible; it is this EPS fraction that is targeted by the CER method due to removing Ca^{2+} by CER and the extraction buffer, weakening electrostatic forces. Binding of EPS by other mechanisms causes weaker binding and stabilization (iii) or stronger binding/immobilization and stabilization (ii, iv, v), but these latter EPS fractions are not targeted or extracted by CER. Given that this certainly is too much detail, we did not add more of this in the manuscript, also because comprehensive studies on these across EPS components and sources, and soil types are yet lacking. In cultures, it is also this loosely bound fraction of EPS that is extracted by CER, while the tightly bound EPS fraction remains unextracted. For a comparison of extraction methods, please also refer to Huang et al. ¹¹.

In light of your suggestion, we have also revised the **Discussion Section** to reflect an alternative interpretation: that the elevated EPS contents observed in carbonate soils likely reflect increased microbial production and short-term retention, rather than long-term stabilization. We further included a two-way ANOVA and variance partitioning analysis to examine the relative contributions of bedrock type and land use to key environmental and microbial factors, and linked these to EPS levels via correlation analyses (Supplementary Table 5). By integrating these relationships and comparing the absolute values of environmental and microbial factors across bedrock types, we found that carbonate soils had significantly higher clay content (Supplementary Table 5). This may suggest a more compact and homogeneous soil structure, which does not fully support the hypothesis that EPS loss via organo-mineral stabilization is lower due to greater structural heterogeneity in carbonate soils. However, our literature review indicates that carbonate soils often provide more favourable conditions for microbial growth and diversity ¹². Given that EPS production has been positively correlated with microbial community diversity ¹³, we adopted this explanation in our revised discussion. Accordingly, we propose that the elevated EPS contents in carbonate soils may be primarily attributed to enhanced microbial diversity and activity under more suitable environmental conditions. We have cited the suggested studies and revised the relevant paragraph in the **Discussion Section**, please see below for the updated text:

Original paragraph: Among the three bedrock types, EPS concentrations were significantly higher in carbonate soils than in soils on silicate and sedimentary bedrock.

Generally, soil EPS concentrations are governed by the dynamic balance between its generation, stabilization and decomposition (Shi et al., 2024). Interestingly, while carbonate soils exhibited the highest EPS concentrations, EPS production efficiencies followed a different pattern, with soils on sedimentary rocks demonstrating the highest EPS production efficiency. This suggested that high EPS concentrations in carbonate soils may depend on their stabilization processes¹⁴, whereas the EPS in sedimentary rocks (despite their high EPS production efficiencies) may be rapidly decomposed due to the presence of adapted microbial communities efficiently utilizing EPS¹⁵. Comparing the three studied bedrock types, carbonate rocks are rich in base cations such as Ca^{2+} and Mg^{2+} ¹⁶, weather rapidly¹⁷ and have neutral soil pH (Supplementary Table S15)^{18, 19}. This provides a suitable living environment for microorganisms, but also promotes the adsorption of EPS by minerals and thereby aids with its stabilization and accumulation^{20, 21}. Sorption and stabilization controls on EPS concentrations were also evident from positive correlations between EPS concentration and Ca_e , clay, silt and Fe_d , and these properties were also highest in carbonate soils. Sedimentary rocks are rich in various nutrients (such as NH_4^+)²² that may stimulate the microbial secretion of EPS. This, in turn, can stimulate microbial resource utilization efficiency and microbial activities through a positive feedback mechanism²³. However, greater microbial activities may result in the recycling and degradation of EPS¹⁵, thus, eventually limiting its accumulation²⁴. In the studied soils, those deriving from sedimentary bedrock were intermediate in properties promoting EPS stabilization (e.g. clay, silt, Ca_e , Fe_d) but had highest EPS production efficiencies, which may explain the second highest soil EPS concentrations. In contrast, silicate rocks, due to their slower weathering rates and limited nutrient release capacities, may lack certain essential soil microbial elements, thus, limiting microbial activities and the generation of EPS^{25, 26}. Silicate soils had the lowest levels of potentially EPS-binding properties (clay, silt, and, Ca_e), and the lowest EPS production efficiencies which were accompanied by lowest soil pH values eventually reducing microbial activities and decomposition processes.

After revision: Among the three bedrock types, EPS contents were significantly higher in carbonate than in silicate and sedimentary soils. This pattern was mirrored in microbial (e.g., MBC and MBN) and soil properties (e.g., CEC, clay, and Ca_e), known to promote EPS, all of which were significantly influenced by bedrock and peaked in carbonate soils. Conversely, sand content – which was lowest in carbonate soils – was negatively correlated with EPS contents (Supplementary Table 5). Based on these findings, we propose that the highest EPS content in carbonate soils is primarily driven by two aspects: (1) an indirect pathway, where highest soil pH and CEC (particularly Ca_e) modulates the soil microenvironment in ways that stimulate microbial activity²⁷, increase microbial biomass (e.g., MBC and MBN) and thereby causes greater overall capacity for EPS production⁹; and (2) soil minerals (clay) and cations (Ca_e) may

contribute to the selective retention of EPS within the soil matrix through multivalent cation bridging between negatively charged sites of EPS and other polymers and minerals^{5, 11}. However, CER-based extraction primarily isolates loosely bound or exchangeable EPS²⁸, stabilized through reversible Ca²⁺ bridging and electrostatic attraction (Supplementary Discussion 3). These fractions are unlikely to represent long-term stabilization but may provide transient protection from microbial degradation, thereby promoting short-term EPS persistence²⁸. Additionally, although our study does not directly include microbial community diversity or composition data, previous research has shown that lithology-driven geochemical processes – such as rock weathering and mineral-associated protection – can indirectly regulate the accumulation of microbial extracellular products (such as glomalin-related soil protein) via microbial functional traits¹². In particular, carbonate soils have been reported to harbor higher microbial diversity¹², which, given the positive relationship between diversity and EPS production¹³, may reflect a greater EPS secretion potential¹². Moreover, the relatively higher WHC in carbonate soils – may reflect the ecological function of soil EPS in improving soil water retention⁵ (Fig. 9). Interestingly, while carbonate soils exhibited the highest EPS contents, their EPS-C/MBC ratios followed a different pattern, with soils on sedimentary rocks demonstrating the highest EPS-C/MBC ratios (Fig. 7a). Although this ratio does not directly reflect EPS production efficiency – since EPS contents in natural soils result from the balance of microbial production and degradation, modulated by mineral interactions and extraction limitations – it may serve as a proxy for the relative allocation of microbial C assimilates to extracellular non-growth products under steady state conditions. Notably, sedimentary soils had the lowest EPS contents, but the highest EPS-C/MBC ratios, which likely stems from reduced MBC rather than increased EPS contents. Supporting this interpretation, microbial growth-related parameters (e.g., C_{growth}, MBC, and MBN) were consistently lowest in sedimentary soils, which also had the lowest WHC and SMC (Supplementary Table 5). Given the decrease in MBC at lower WHC and SMC (Supplementary Fig. 2), we infer that limited water availability in sedimentary soils may constrain microbial growth and shift C allocation towards EPS formation. This shift may reflect an adaptive microbial strategy to enhance survival and maintain extracellular protection under water-stressed conditions⁵. Although silicate soils exhibited intermediate levels of both EPS and MBC (Supplementary Table 5), they had the lowest EPS-C/MBC ratio. This suggests that microbes in silicate soils may have allocated proportionally less C toward EPS production or are less conducive for intermittent EPS retention. Three possible explanations can be proposed: (1) Environmental stress in silicate soils may be relatively mild – for example, these soils exhibited the highest SMC and WHC – whereas more severe moisture-related stress in sedimentary soils might have triggered greater EPS production; (2) the lowest levels of

Ca_e and clay (highest sand content) in silicate soils may have provided the least protection for EPS (Supplementary Table 5); (3) differences in microbial community composition or metabolic strategies may also contribute to the observed pattern. However, we currently lack direct evidence to confirm the third explanation. Overall, bedrock influenced EPS through (1) promoting microbial biomass and EPS secretion via bedrock-driven environmental conditions, and (2) shaping soil texture and mineralogy, which governs EPS retention and intermittently protects it from microbial degradation” (Pages 9-11, Lines 259-307). Related discussion on bedrock effects on EPS and stabilisation mechanisms can be found in responses R4-45, R4-50/51, and R4-54/55.

We have also added a discussion on the stabilization mechanisms of EPS in soils. Owing to the word limit imposed by *Nature Communications*, this part is now provided in **Supplementary Discussion 3**:

Supplementary Discussion 3 - Mechanisms of EPS stabilization in soils

EPS in soils is stabilized via multiple mechanisms^{5, 8, 11}, including (i) physical occlusion inside microaggregates, (ii) co-precipitation and complexation with Fe/Al, (iii) hydrogen bonding, (iv) hydrophobic interactions, (v) inner-sphere ligand exchange to Fe/Al (oxyhydr)oxides and clays, (vi) outer-sphere electrostatic attraction, and (vii) multivalent cation bridging between negatively charged sites of EPS and those on other polymers and minerals. The importance of these mechanisms differs between main EPS compounds, with EPS polysaccharides mainly bound/stabilized by (ii, iii, v, vii) and EPS proteins via (ii, iii, iv, v, vi, vii). The cation bridging mechanism via Ca²⁺ (vii) and electrostatic attraction (vi) causes a loosely bound (not strongly stabilized) association, which is reversible; it is this EPS fraction that is targeted by the CER method due to removing Ca²⁺ by CER and the extraction buffer, weakening electrostatic forces. Binding of EPS by other mechanisms causes weaker binding and stabilization (iii) or stronger binding/immobilization and stabilization (ii, iv, v), but these latter EPS fractions are not targeted or extracted by CER. Since comprehensive studies across EPS components, sources, and soil types are still lacking, this remains poorly understood. In culture experiments, it is also the loosely bound fraction of EPS that is extracted by CER, whereas the tightly bound fraction remains unextracted.

[Comment 23] Line 333-335: As above, this is counter-intuitive and likely arises from the previous view that stabilised EPS is extracted with CER (when existing evidence and theory suggests it is not). Please change this statement to also reflect the rationale above.

[Response 23] Thank you for pointing this out. We have revised the statement to align with current evidence and theory, clarifying that the elevated EPS contents observed in

carbonate soils likely reflect increased microbial production and short-term retention, rather than long-term stabilization. Please see the revised **Discussion Section** in the manuscript and our response **R1-22 (Pages 9-11, Lines 259-307)**.

[Comment 24] Line 335: “Sedimentary rocks can be rich...”

[Response 24] The sentence has been removed as part of the revisions to refine the discussion section.

[Comment 25] Line 340: “properties thought to promote...”

[Response 25] The sentence has been removed as part of the revisions to refine the discussion section.

[Comment 26] Line 367: “possibly due to relaxed...”

[Response 26] The sentence has been removed as part of the revisions to refine the discussion section.

[Comment 27] Line 376-379: to refine and shorten the narrative here I would recommend reconciling the previously mentioned literature in context. I suggest replacing the unnecessary sentence: “Nevertheless, these microbial diversity aspects were not effectively investigated in this study and warrant further exploration in the future”. AND In conclusion, it is the interplay....” (repetition of land-use) (nonspecific EPS fractions). With “While microbial diversity was not investigated in this study, in accordance with investigations of legacy vs current management effects on EPS16, it is primarily the interplay between plant inputs and anthropogenic management which causes these differences in total and proteinaceous EPS concentrations”.

[Response 27] Thank you for your valuable suggestion. We have revised the sentence from “Nevertheless, these microbial diversity aspects were not effectively investigated in this study and warrant further exploration in the future. In conclusion, it is the interplay between plant inputs and anthropogenic management which cause land use driven differences in EPS concentrations” to “**While microbial diversity was not investigated in this study, in accordance with investigations of legacy vs current management effects on EPS²⁸, it is primarily the interplay between plant inputs and anthropogenic management that causes these differences in total EPS and EPS protein contents**” as suggested to improve clarity and conciseness. However, since the effect of land use on soil EPS was not significant (Fig. 2b), this part of the discussion has been moved to **Supplementary Discussion 4**. More responses on land use effects on EPS can be found in **R3-13, R4-57, R4-58, and R4-59**.

[Comment 28] Line 385: “Stabilization” was not measured, please remove.

[Response 28] Done. The original sentence has been moved from main text to **Supplementary Discussion 5 - Bedrock vs. land use: drivers of specific EPS components.**

[Comment 29] Lines 385-388: Accordingly, please refine, or remove ‘stabilization’ from this sentence.

[Response 29] Done. The original sentence has been moved from main text to **Supplementary Discussion 5 - Bedrock vs. land use: drivers of specific EPS components.**

[Comment 30] Line 422: Please insert a full stop after “on a continental scale”.

[Response 30] Done (**Page 11, Line 334**)

[Comment 31] Line 422: Please delete “which is mainly due to differences in their lability or stability” and deal with the main known artefacts first. We do not yet know how much of either fraction is stabilized in soils.

For a more conservative assessment, please insert “Incomplete extraction of EPS partly explains the difference between MNC and EPS-C observed here. Perhaps more importantly, amino sugar biomarkers do not exclusively represent necromass [reference Salas et al]. GalN for example can be found in substantial concentrations in plant residues, and other non-necromass materials, even EPS itself [reference Salas et al]. Therefore, while measures of EPS-C are understood to be underestimates, measures of MNC are known to be overestimates and perhaps with a larger margin for error and variability. This means that while on face value our results suggest that MNC was the larger pool of C, true values might be comparable. More research is needed to elucidate with confidence whether EPS-C or MNC contributes more to stable SOC formation. Nonetheless, our data for soils sampled at the continental scale show that both pools are important for (direct) carbon storage in soils – with EPS-C having been largely overlooked until now.

For information: From Salas (Salas et al., 2024): “Galactosamine (GalN) and mannosamine (ManN) are nonspecific amino sugars which can be found in plants (Indorf et al., 2011), fungal cell walls (Beauvais and Latgé, 2018), EPS of bacteria (Caruso et al., 2018; Cheah and Chan, 2022; Vaningelgem et al., 2004), and in the cell envelope of some archaeal species (Albers and Meyer, 2011).

[Response 31] Thank you for your constructive and highly valuable comments. As suggested, we have revised the original sentence “In contrast to extracellular microbial residues, the contribution of cellular residues in the form of MNC to SOC was ~10× that of estimated EPS-C on a continental scale. This is mainly due to differences in C allocation to extracellular versus cellular residue formation mechanisms and

differences in their lability or stability” and introduced your suggested text as “Incomplete extraction of EPS partly explains the difference between MNC and EPS-C observed here. Perhaps more importantly, amino sugar biomarkers do not exclusively represent necromass²⁹. Galactosamine, for example, can be found in substantial concentrations in plant residues, and other non-necromass materials, even EPS itself²⁹. Therefore, while measures of EPS-C are understood to be underestimates, measures of MNC are known to be overestimates³⁰ and perhaps with a larger margin for error and variability. This means that while on face value our results suggest that MNC was the larger pool of C, true values might be more comparable in size. More research is needed to elucidate with confidence whether EPS-C or MNC contributes more to SOC formation and stabilization” (Pages 11-12, Lines 334-342). See also responses R4-12 and R4-33.

[Comment 32] Line 424: Please start new paragraph with “EPS is an active pool that is...”

[Response 32] The revision has been made as suggested to improve the structure and readability (Page 12, Line 343)

[Comment 33] Line 428-429: Please delete speculation “This passivity makes the contribution of MNC significantly higher than EPS-C” There is no soils data available to confirm or deny this. Stabilization of EPS in soils is not yet understood, and certainly not measured/reported.

[Response 33] The contentious statement has been removed as suggested.

[Comment 34] Line 448: Please change title to reflect content - removing reference to EPS production efficiency, and instead reflect Drivers of EPS-C/MBC ratio, and EPS accumulation.

[Response 34] Thank you for the suggestion. Since subtitles cannot be used in the Discussion section according to the journal format, the original subtitle has been removed.

[Comment 35] Line 449: To more accurately reflect the cited work, please change first line regarding reference 48 to: Changes in short-term microbial EPS production efficiency was previously calculated by standardizing soil EPS concentrations to concentrations of soil microbial ATP 48.

[Response 35] Thank you very much for your helpful comment. We have revised the sentence from “Microbial EPS production efficiency is calculated by standardizing EPS-C concentrations to microbial biomass C” to “Changes in short-term microbial EPS production efficiency were previously calculated by standardizing soil EPS to soil

microbial ATP², a proxy for microbial biomass” to more accurately reflect the findings of the cited work (Page 12, Lines 358-359).

[Comment 36] Line 451: Insert “However,” before “as mentioned above”.

[Response 36] Done as suggested (Page 12, Line 361).

[Comment 37] 453: Please replace this sentence with “Distinct bedrock and land-use types have shaped diverse ecological environments, and no doubt influenced decomposition and accumulation of EPS”.

[Response 37] Thank you for your suggestion. We have revised the sentence from “Distinct bedrock and land-use types have shaped diverse ecological environments and strongly influenced the production and accumulation of EPS” to “**Distinct bedrock and land-use types have shaped diverse ecological environments, and no doubt influenced the decomposition and accumulation of EPS**” as suggested (Page 12, Lines 362-364).

[Comment 38] 454: Before “a more detailed...” please insert “Therefore, we present EPS-C/MBC ratios “as is” without inferring EPS production efficiency”. The rest of the paragraph should now flow smoothly.

[Response 38] Thank you very much for your helpful comment. The sentence has been added as recommended, and the paragraph now flows really smoothly (Page 12, Lines 364-365).

[Comment 39] Line 475: “Given the limited labile C...”

[Response 39] Done (Page 13, Line 386).

[Comment 40] Lines 480-482: please simplify and abbreviate this circular sentence.

[Response 40] We have revised the sentence from “The strategy adopted by soil microorganisms to secrete EPS is therefore likely strongly influenced by their physiological state and by C trade-offs governing (extra)cellular C allocation” to “**This strategy, adopted by soil microorganisms to secrete EPS or invest in growth or other C allocation processes is therefore likely influenced by their physiological state and C allocation trade-offs**” for clarity and conciseness (Page 13, 392-394).

[Comment 41] Line 480-484: please reconsider and present in context of Bölscher et al., 2024: Bölscher, T., Vogel, C., Olagoke, F.K., Meurer, K.H.E., Herrmann, A.M., Colombi, T., Brunn, M., and Domeignoz-Horta, L.A. (2024). Beyond growth: The significance of non-growth anabolism for microbial carbon-use efficiency in the light of soil carbon stabilisation. *Soil Biology and Biochemistry*, 193, 109400. DOI: 10.1016/j.soilbio.2024.109400

[Response 41] Thank you very much for your valuable comment. We have revised this sentence from “In this study, EPS production efficiency scaled negatively with microbial growth (C_{growth}), and therefore also negatively with microbial CUE and with microbial biomass. This provides the first and strong hint for a microbial trade-off in C allocation between EPS secretion and growth processes and CUE, based on *in vivo* process measurements” to “The negative relationships between EPS-C/MBC and microbial growth (C_{growth}) and CUE thus indicate a shift in microbial C allocation from growth to non-growth anabolic processes such as EPS production³¹” (Page 13, Lines 390-392). See also the following responses on microbial C allocation between EPS and growth, and stress-induced phenomena: **R4-3**, **R4-74**, and **R4-75**.

[Comment 42] 487-489: I do not believe the evidence is strong enough to support this (complex) statement which seems to run counter to established understanding, please remove, or if the authors feel this is vital, contest this request?

[Response 42] Thank you for your valuable comment. We agree that the current evidence may not be sufficient to support the complexity of this statement, and we have therefore removed the sentence “This trade-off also allows to test the EPS production relationship to resource availability. For instance, we found that EPS production efficiency was high when microbial growth was low due to microbial C limitation as indicated by high enzyme vector length and low DOC” as suggested.

[Comment 43] 487: Please replace the sentence here with information consistent with the referenced works “This trade-off means that C investment into EPS can only be maintained if sufficient labile C is available 17, 48” (Note that labile C cannot be inferred from SOC).

[Response 43] Thank you for your suggestion. After careful consideration, we have removed this sentence to improve the logical flow of the manuscript. This revision does not affect the accuracy of the discussion on microbial C allocation trade-offs, which can be found on **Page 13, Lines 382-398**. We hope this clarifies our revision and appreciate your understanding.

[Comment 44] 490: This bold statement may apply to the investigated soils, but is too sweeping as a universal statement. I urge the authors to reconsider, or simply remove.

[Response 44] Thank you very much for your valuable suggestion. In response, we have removed the statement of “Different from aquatic systems, EPS production is obviously not stimulated as an intermittent extracellular store of C and N by secreting EPS polysaccharides and EPS proteins³², under ample C and nutrient supply in soils” from the manuscript.

[Comment 45] 492: This is clearly a mistake. Please remove this line and finish the paragraph as the rest (to line 494) is repetition/not necessary.

[Response 45] Thank you very much for your valuable suggestion. We have removed the sentence “Instead, C and nutrient limitation promote EPS production in aquatic and soil microbial communities, though the mechanistic underpinning remains to be resolved in soils. In aquatic biofilms resource limitation has been called for maximizing the microbial return on investment for exoenzymes^{8, 23}” as requested.

[Comment 46] 508: please remove “stabilization”.

[Response 46] The sentence has been removed as part of the revisions to refine the discussion section.

[Comment 47] 510: replace “can assist” with “may assist”.

[Response 47] The sentence has been removed as part of the revisions to refine the discussion section.

[Comment 48] 511: please delete “enforcing”.

[Response 48] The sentence has been removed as part of the revisions to refine the discussion section.

[Comment 49] 514: change ‘production efficiency’

[Response 49] Done **(Page 13, Line 406)**.

[Comment 50] 516-517: “Essential” is subjective and a little strong. For balance, please replace the sentence “it is essential to improve...” with “Addressing both the underestimations of EPS, and the overestimations of necromass respectively achieved by extraction will be important in clarifying our understanding of these formative carbon pools.

[Response 50] Thank you very much for your valuable suggestion. As recommended, we have replaced the original sentence “It is essential to improve EPS quantification in soils, addressing potential underestimations by current methods” with “Addressing both the underestimations of EPS and the overestimations of necromass, respectively achieved by extraction, will be important in clarifying our understanding of these formative carbon pools” to improve the balance and objectivity of the statement **(Pages 13-14, Lines 408-410)**.

[Comment 51] 584: please remove “s” from “resins”.

[Response 51] The sentence has been revised as suggested to improve accuracy **(Page 16, Line 491)**.

Reviewer #3 (Comments to the Author):

[Comment 1] The manuscript by Ke Shi and co-workers explores spatial variations in extracellular polymeric substances (EPS) across Europe. This is a timely topic, as EPS plays important roles in modulating microbial activity in soils, with potential consequences for carbon and nutrient cycling. EPS spatial patterns and their drivers are largely unknown, so this is a welcome contribution. I should add that my background is in engineering (and modelling) and I don't feel I can evaluate the methodology developed for EPS analysis. Therefore, my review focuses on conceptual aspects and data analysis. Overall, this is a well-written manuscript (though I would suggest some changes especially in the Discussion—see detailed comments below), with a clear structure and neatly-presented results. I like the progression from simple soil groupings (by land use or geology) to more detailed analyses of EPS dependence on various climatic, soil, and biological predictors. Nevertheless, I have some (relatively minor) comments on the interpretation the results.

[Response 1] We sincerely thank you for the thoughtful and constructive comments, and for recognizing the value and timeliness of our study on the distributions of soil EPS across Europe. We truly appreciate your positive feedback. In response to your insightful suggestions, we have carefully revised the manuscript and provided detailed point-by-point responses below.

[Comment 2] EPS production efficiency is defined as the ratio of EPS carbon (EPS-C) to microbial biomass carbon (MB-C). In my view, this is not a production efficiency, which would be rather defined as a rate of EPS production per unit microbial growth (or carbon uptake). My impression is that calling the EPS-C/MB-C ratio 'production efficiency' can be misleading. Indeed, some of the interpretations of this ratio in the Discussion mix up aspects related to EPS production and decay, which I agree are both important. I suggest to change the terminology to 'biomass specific EPS' or something along those lines. For example, in L276-277 it is stated that 'EPS production efficiency promotes EPS accumulation', but the data only indicate that EPS and EPS per unit biomass are correlated, suggesting that microbial biomass does not affect much the variation in EPS (if I interpret correctly). I agree with the statement in L322, which points to EPS as result of production and decay. But then the following statements introduce some confusion by interpreting production efficiency as the result of production and decay—I would clarify this part of the Discussion.

[Response 2] Thank you very much for your thoughtful and important comments. We fully agree that the EPS-C/MBC ratio does not represent true “production efficiency” in a physiological sense, as it does not account for rates of EPS formation relative to microbial growth or carbon uptake, nor does it separate EPS production from

degradation processes. Following your suggestion and those of Reviewer 1 (**see also our responses to Reviewer 1's comment 4, R1-4, and comment 16, R1-16; the same applies below**), we have replaced the term “EPS production efficiency” throughout the manuscript with “EPS-C/MBC ratio” to avoid misinterpretation.

We also revised the relevant parts of the **Introduction** and **Discussion Section** to clarify that this ratio reflects the relation between microbial extracellular / non-growth carbon (EPS) and microbial (cellular) biomass carbon (MBC) at steady state, both underlying concurrent formation and decomposition (turnover) processes, and therefore the relative allocation of microbial biomass C to EPS at any given time point. At the same time, we acknowledge that the observed EPS levels are the net result of production, degradation, and potential stabilization in the soil; this is detailed in the main text on **Pages 5 (Lines 84-93; all page and line numbers refer to the revised clean version without track changes), 10 (Lines 283-287) and 12 (Lines 358-362)**, and also addressed in our response to **R1-1** and **R1-11**. Moreover, we removed or rephrased previous statements that implied causal relationships between the ratio and EPS accumulation.

[Comment 3] I struggle to reconcile the function of EPS and the activity of extracellular enzymes. Are extracellular enzymes targeting EPS? This is a general comment, and not meant to criticize this work—still, here both enzymatic activity and EPS are measured, raising the question as to how the two interact in this dataset. The activity of some enzymes appears to be positively correlated with EPS contents, suggesting that microbes produce both and that enzymes do not affect negatively EPS (or the negative interaction is less strong than the coordinated production). Would it be worth discussing these interactions?

[Response 3] Thank you very much for this insightful comment. Proteins are an integral part of EPS, the extracellular proteins (exoproteins, exopeptides) serving structural roles as well as mediating enzymatic functions. This has for instance been shown using meta-proteomics analysis of environmental biofilms such as in activated sludge flocs^{33,34}, though not yet for soils. Since a part of the EPS proteins and EPS polysaccharides are not fully stabilized, they have also been considered a transient extracellular storage pool of C and N where microbial exoenzymes are secreted for their mobilization under C and/or N limiting conditions. We therefore agree that the relationship between EPS and extracellular enzyme activity is complex and deserves attention. However, in our study, we observed slight positive correlations between EPS contents and enzyme activities such as β -glucosidase (BG) and exoglucanase-cellobiosidase (CEL) (Fig. 2c). Moreover, carbonate soils exhibited the highest EPS contents as well as the highest BG and CEL activities (Supplementary Table 5), which may suggest a potential link between EPS production and enzyme secretion. Although this pattern is in line with

previous findings showing that both EPS and enzymes can be co-expressed as part of microbial responses to environmental conditions, especially under nutrient-limited or stressful environments³⁵, our results do not provide strong evidence for a causal relationship.

To address this, we have now added the following statement to the **Discussion Section**:

“EPS may also influence soil C cycling indirectly by enhancing the stability of extracellular enzymes and modulating their activity³⁵, consistent with the weak positive correlations observed with BG and CEL” (Page 11, Lines 326-328).

We hope this addition better explains the observed pattern. See also EPS-exoenzyme relations in responses **R1-13**, **R3-16**, and **R4-69**.

[Comment 4] Statistical analyses: even random forest approaches handle well collinearity, some of the variables included are very strongly correlated, so I wonder results would be more robust or easier to explain by selecting variables a bit more before running the random forest model and evaluating the importance rankings. Moreover, it is not clear how PC1 and PC2 were combined (L658-660).

[Response 4] Thank you for your critical comment. Although random forest models are relatively robust to multicollinearity, we agree that reducing collinearity among predictors can improve interpretability and model robustness. In our original analysis, we included aridity index (ADI), fine root biomass (FRB), C/N ratio of root (Root C/N), microbial biomass nitrogen (MBN), microbial growth (C_{growth}), microbial growth normalized to microbial biomass carbon (q_{Growth}), enzyme vector length (VectorL), soil pH, soil C/N, β -glucosidase (BG), soil bacteria to fungi ratio (B/F ratio), clay content, exchangeable Ca^{2+} (Ca_e), Fe oxyhydroxides extracted with Na-dithionite (Fe_d), and soil organic carbon (SOC). While the variance inflation factor (VIF) values were all below 10, we identified some variables with high pairwise correlations (e.g., pH with Ca_e , $r > 0.7$; q_{Growth} with C_{growth} , $r > 0.6$). Based on this, we removed pH and C_{growth} . In addition, due to further optimization of the dataset, Fe_d no longer emerged as an important driver and was therefore excluded. At the same time, we found that exoglucanase-cellobiosidase (CEL) showed slight positive relationships with both EPS proteins and EPS polysaccharides (Fig. 2c), and thus included CEL as an additional predictor. We then re-ran the random forest model using a more refined set of predictors, including:

- Climatic factor: ADI
- Plant factors: FRB (biomass), Root C/N (nutrient content)
- Soil nutrients: SOC, Ca_e , Soil C/N
- Soil texture: Clay
- Microbial factors: MBN, q_{Growth} , BG, B:F ratio, CEL, VectorL

All predictors in this refined model showed VIF values = 7.76, indicating that multicollinearity was not a concern.

The results of this optimized random forest model are now presented as the new **Fig. 4**, which replaces the original version.

Original Fig. 4:

New Fig. 4:

Figure 4. Environmental and microbial drivers of EPS polysaccharides and EPS proteins. **a** Principal component analysis (PCA) of environmental and microbial variables associated with EPS polysaccharides and EPS proteins. The first and second principal components (PC1 and PC2) explain 32% and 20% of the total variance, respectively. **b** Importance ranking of predictors for EPS polysaccharides based on

random forest analysis. **c** Importance ranking of predictors for EPS proteins based on random forest analysis. * $p < 0.05$, ** $p < 0.01$, *** $p < 0.001$. EPS, extracellular polymeric substances; ADI, aridity index; FRB, fine root biomass; BG, β -glucosidase; VectorL, enzyme vector length; B:F, bacteria to fungi ratio; Ca_e , exchangeable Ca^{2+} ; WHC, water holding capacity; q_{Growth} , microbial growth normalized to microbial biomass carbon; SOC, soil organic carbon; TP, total phosphorus; CEL, exoglucanase-cellobiosidase; MBC, microbial biomass carbon; MBN, microbial biomass nitrogen.

Moreover, we agree that the method for combining PC1 and PC2 in constructing the soil and microbial composite variables should be described more clearly. In the revised analysis, we simplified the approach by using only the first principal component (PC1) to represent soil and microbial variables, rather than combining PC1 and PC2. This decision was based on the fact that PC1 alone captured the majority of the variance (51.25% for soil, 62.48% for microbes) and yielded well-fitting SEM models ($p = 0.433$; see Fig. 8a and Supplementary Fig. 1).

We have revised the original sentence from “Among them, SOC, soil clay content, soil C/N ratio, Ca_e , and Fe_d were combined into a soil composite variable, while MBN, C_{growth} , g_{Growth} , and CUE were combined into a microbial composite variable using PCA. Instead of using only the first principal component (PC1) or the second principal component (PC2) alone, soil and microbial variables were constructed by combining the PC1 and PC2 based on their explained variance, and this was necessary because SEM models using only PC1 or PC2 did not yield satisfactory results ($p < 0.05$) (Supplementary Fig. S6)” to “Among them, SOC, soil clay content, soil C:N, and Ca_e were combined into a soil composite variable, while MBN, C_{growth} , q_{Growth} , and CUE were combined into a microbial composite variable, for both using PCA. We used the first principal component (PC1) to represent each composite variable, as it captured a substantial proportion of the variance (51.3% for soil, 62.5% for microbes) (Supplementary Fig. 1)” (Page 19, Lines 601-605). This simplified approach provided a more transparent representation of the underlying variation compared with the previous weighted combination of PC1 and PC2.

[Comment 5] Data availability: it would be useful to be able to access the data in an open access repository.

[Response 5] Thank you for your suggestion. We have now uploaded the EPS-related variables (including EPS polysaccharides, EPS proteins, total EPS, EPS-carbon, and site information such as bedrock type, land use type, latitude, longitude, and elevation) as **Supplementary Dataset 1**, and the data we compiled from previously published studies on soil EPS as **Supplementary Dataset 2**. Other related datasets (e.g., microbial necromass carbon, microbial carbon use efficiency, and additional microbial variables)

are not yet published and therefore are not included at this stage; however, their mean values across different bedrock and land-use types are provided in **Supplementary Table 5**.

[Comment 6] General: EPS is expressed as carbon mass per unit mass of soil, so I would call it ‘content’ (mass/mass ratio) rather than ‘concentration’ (mass/volume ratio) (McNaught, A.D., Wilkinson, A., 1997. IUPAC Compendium of Chemical Terminology, Blackwell Scientific Publications). I realize the terminology in the literature is inconsistent to this regard, so I do not feel strongly in either direction.

[Response 6] Thank you for the important comment and for pointing us to the IUPAC terminology. We agree that “content” is more accurate in this context (mass of carbon per mass of soil), and therefore have replaced “concentration” with “**content**” throughout the manuscript where applicable.

[Comment 7] Section 2.1: please define acronyms in the Results section, otherwise it is hard to read the text while continuously flipping pages to get to the definitions in the Methods.

[Response 7] We agree that defining acronyms upon first use in the **Results section** improves readability. Accordingly, we have ensured that each variable is introduced with its full name at first mention, followed by the acronym in parentheses throughout the **Results section**.

[Comment 8] Figure 2: this figure is very dense of information, so I wonder if it could be simplified e.g., by removing the correlation heatmap for the ratio of EPS polysaccharide to EPS protein. I might have missed the importance of this ratio (I think it is only briefly mentioned in the Discussion), and in any case it does not seem to be highly correlated with many drivers.

[Response 8] We have revised **Fig. 2** to improve clarity by moving the correlation heatmap for the ratio of EPS proteins to EPS polysaccharides to **Supplementary Fig. 5**. As this ratio was only briefly mentioned in the Discussion and showed limited correlation with most drivers, we agree that its exclusion helps simplify the figure and better highlight the key findings.

New **Fig. 2**:

Figure 2. Soil EPS contents and their correlations with environmental and microbial factors. **a** Box plots of total EPS contents (sum of EPS polysaccharides and EPS proteins) in soils from different bedrock types. **b** Box plots of total EPS contents in soils from different land use types. Individual data points are displayed as black dots with jitter to show distribution. Boxes show the median (middle line), 25th and 75th percentiles (interquartile range), and whiskers represent the 5th and 95th percentiles. The orange diamond in each box represents the mean value. Significant differences between groups are indicated on the plot. * $p < 0.05$, ** $p < 0.01$, *** $p < 0.001$, NS $p > 0.05$. **c** Heatmap showing the correlations between total EPS, EPS polysaccharides, and EPS proteins with environmental and microbial variables. Only significant correlations ($p < 0.05$) are shown, except for pH, which is displayed regardless of significance. Numbers in the heatmap represent correlation coefficients (r); variables without numbers are non-significant ($p > 0.05$). The color intensity indicates the strength of correlation, with blue representing positive correlations and orange representing negative correlations. EPS, extracellular polymeric substances; EPS PS, EPS polysaccharides; EPS PN, EPS proteins; ADI, aridity index; MAP, mean annual precipitation; PET, potential evapotranspiration; FRB, fine root biomass; WHC, water holding capacity; SMC, soil moisture content; SOC, soil organic carbon; TN, total nitrogen; TP, total phosphorus; CEC, cation exchange capacity; Ca_e, exchangeable Ca²⁺; Mg_e, exchangeable Mg²⁺; NH₄⁺, ammonium; NO₃⁻, nitrate; TDN, total dissolved nitrogen; TOP, total organic phosphorus; MBC, microbial biomass carbon; MBN, microbial biomass nitrogen; MBP, microbial biomass phosphorus; CUE, microbial carbon use efficiency; q_{Growth}, microbial growth normalized to microbial biomass carbon; MBT, microbial biomass turnover time; BG, β-glucosidase; CEL, exoglucanase-cellobiosidase; VectorL, enzyme vector length; GPB, gram-positive bacteria PLFA biomass; GNB, gram-negative bacteria PLFA biomass; Bacteria, bacterial PLFA biomass.

New **Supplementary Fig. 5:**

Supplementary Figure 5 | Soil EPS proteins to EPS polysaccharides (EPS PN:PS) ratio and their correlations with environmental and microbial factors. a Box plots of EPS PN:PS ratio in soils from different bedrock types. **b** Box plots of EPS PN:PS ratio in soils from different land use types. Individual data points are displayed as black dots with jitter to show distribution. Boxes show the median (middle line), 25th and 75th percentiles (interquartile range), and whiskers represent the 5th and 95th percentiles. The orange diamond in each box represents the mean value. Significant differences between groups are indicated on the plot. * $p < 0.05$, ** $p < 0.01$, *** $p < 0.001$, NS $p > 0.05$. **c** Heatmap showing the correlations between EPS PN:PS ratio with environmental and microbial variables. Only significant correlations ($p < 0.05$) are shown, except for pH, leucine aminopeptidase (LAP), and bacterial to fungal ratio (B:F), which are displayed regardless of significance. Numbers in the heatmap represent correlation coefficients (r); variables without numbers are non-significant ($p > 0.05$). The color intensity indicates the strength of correlation, with blue representing positive correlations and orange representing negative correlations. EPS, extracellular polymeric substances; Root C, root carbon content; SOC, soil organic carbon; CEC, cation exchange capacity; NUE, microbial nitrogen use efficiency; NAG, N-acetyl- β -glucosaminidase; VectorA, enzyme vector angle.

[Comment 9] Figure 8: 'bedrock' text is not visible (black on black).

[Response 9] Fig. 8 has been revised to improve clarity — the color of the pattern representing “bedrock” has been adjusted to ensure better visibility.

Original **Fig. 8:**

New Fig. 8:

Figure 8. Effects of environmental and microbial factors on soil EPS-C and EPS-C/MBC ratio. **a** SEM illustrating the direct and indirect pathways by which bedrock type, land use, and climate influence EPS-C and EPS-C/MBC via plant, soil, and microbial factors. Black lines indicate positive effects, and red lines indicate negative effects. Solid lines represent statistically significant paths, and dashed lines indicate non-significant paths. Significance levels are indicated as: * $p < 0.05$, ** $p < 0.01$, *** $p < 0.001$. Numbers on the paths are standardized regression coefficients. **b** Bar chart showing the loadings of four soil variables on the soil composite factor (PC1) derived from PCA. SOC, soil organic carbon; C:N, soil carbon to nitrogen ratio; Ca_e , exchangeable Ca^{2+} . **c** Bar chart showing the loadings of four microbial variables on the microbial composite factor (PC1) derived from PCA. MBN, microbial biomass nitrogen; CUE, microbial carbon use efficiency; C_{growth} , microbial growth; q_{growth} , microbial growth normalized to microbial biomass carbon. EPS, extracellular polymeric substances; EPS-C, extracellular polymeric substance carbon; ADI, aridity index; MBC, microbial biomass carbon.

[Comment 10] L211 and L417: where is the evidence of the strong association between WHC and EPS shown? WHC does not appear among the highly ranked drivers.

[Response 10] We apologize for the confusion. In the original version of the manuscript, the relationship between EPS and soil water holding capacity (WHC) was only presented in the Supplementary Information, which made it difficult to support the relevant statements in the main text. To address this, we have now included them in the heatmap of Fig. 2c. Moreover, PCA revealed a strong positive correlation between WHC and both EPS polysaccharides and EPS proteins (Fig. 4a). However, WHC was not included in the random forest analysis because the model is designed to identify drivers rather than outcomes of EPS. Since WHC itself is largely influenced by EPS, it should not be treated as a predictor. Accordingly, WHC did not appear among the top-ranked predictors in the random forest results (Fig. 4b, 4c). Given the potential importance of EPS for soil water retention, we additionally included Fig. 9 to show the significant linear relationships between soil WHC, soil moisture content (SMC), and EPS, which further support the Discussion. To avoid overstatement, we have revised the sentence from “At the same time, both EPS compounds were strongly positively related to soil water holding capacity (particularly EPS polysaccharides), indicating that EPS promotes soil water holding capacity more than bulk SOC or texture do (Fig. 4)” to “**Both EPS compounds were strongly related to soil WHC (Fig. 4a)**” (Page 8, Lines 188-189) and the sentence in Discussion section from “One important functional aspect of EPS was shown here to be the key positive driver of soil water holding capacity, compared to weaker associations to SOC or soil texture (in the absence of direct measurements of soil porosity and pore size distribution)” to “**A key functional role was evident in the strong positive association between EPS and soil water holding capacity (Fig. 9a), highlighting its contribution to soil moisture retention**” (Page 11, Lines 328-330). See also our responses R4-53.

Figure 9. Linear regressions between EPS and soil water properties. a Relationship between EPS and water holding capacity. **b** Relationship between EPS and soil moisture content. Blue lines indicate linear regression fits with 95% confidence intervals (shaded). The R^2 and p -value for each regression are shown on each panel. Each point represents an individual soil sample. EPS, extracellular polymeric substances.

[Comment 11] L319: was the methodology to estimate EPS in previous papers the same as used here? Can different measurement approaches explain the differences?

[Response 11] The studies we compiled for comparison all used the cation exchange resin (CER) method to extract EPS from soils following the soil CER protocol published by Redmile-Gordon et al. ³⁶, ensuring consistency in the core extraction approach. However, it is true that almost all of these studies used fresh (moist) soils, whereas our study employed air-dried soils. This difference in sample preparation may contribute to methodological variation. To further address this, we reviewed the literature and found that Zhang et al. ³⁷ reported lower EPS yields from air-dried soils after rewetting and 2-week pre-incubation compared to moist soils. Additionally, we conducted a small experiment to directly compare EPS extraction from dry and moist soil samples. We found that for EPS polysaccharides, extraction from air-dried soils showed a mean difference of $-1\% \pm 3\%$ (mean \pm SE, $n = 7$; range: -10% to $+11\%$) relative to fresh soils. In contrast, EPS protein extraction from air-dried soils showed a mean difference of $+9\% \pm 1\%$ (mean \pm SE, $n = 5$; range: $+7\%$ to $+14\%$) across cropland, grassland, and forest soils (**Supplementary Table 3**). See also our response **R4-3** and **R4-78** for more detail.

In response, we have revised the sentence from “This difference may be due to variations in geographical area, soil management practices, microbial communities, climate, and soil organic matter content” to “**This difference partly reflects differences in bedrock and land-use composition, sampling season, and methodological factors (Supplementary Discussion 1). Due to sample availability, we used air-dried soils, whereas most previous studies used fresh soils. Our preliminary experiment showed minor effects of air-drying: EPS polysaccharides decreased by $1 \pm 3\%$, EPS proteins increased by $9 \pm 1\%$, and total EPS (polysaccharides + proteins) increased by $4 \pm 3\%$. Long-term soil storage had similarly no impact on EPS content (Supplementary Discussion 2)**” (**Page 9, Lines 253-258**).

To provide additional context and a more comprehensive interpretation of our findings, we have included more detailed discussions in **Supplementary Discussions 1 and 2** (please see below).

Supplementary Discussion 1 - Comparison with published EPS datasets

To compare our EPS data with previously published values, we compiled 216 EPS data points from 22 studies using the CER extraction method, with all data available in

Supplementary Dataset 2. After screening for surface bulk soils and excluding data with added substrates, 83 data points from 18 studies were retained. Across ~30 sites differing in climate, geology, and land use, previously reported mean values were 429 $\mu\text{g g}^{-1}$ for total EPS (sum of EPS polysaccharides and EPS proteins), 335 $\mu\text{g g}^{-1}$ for EPS polysaccharides, and 131 $\mu\text{g g}^{-1}$ for EPS proteins (Supplementary Table 1), which were generally lower than the levels observed in our study^{2, 4, 9, 27, 28, 36, 37, 38, 39, 40, 41, 42, 43, 44, 45, 46, 47, 48, 49, 50}.

The higher EPS contents in our dataset likely reflect a combination of ecological and methodological factors. Ecologically, our study included a higher proportion of grassland sites (27/92 vs. 7/46 in previous studies) and calcareous bedrock (27/92 vs. 2/46), both associated with higher EPS contents. In addition, ~67% of previously published data were sampled in January–April and October, whereas our samples were collected from May to August, when microbial activity is elevated, potentially enhancing EPS production. Methodologically, soil storage conditions may influence the amount of extractable EPS. Together, these factors likely account for the elevated mean and median EPS levels observed in our study compared to the literature.

Supplementary Discussion 2 - Effects of soil drying and long-term storage on EPS

Due to sample availability, we used air-dried soils, whereas most previous studies used fresh soils. Zhang et al.³⁷ reported that EPS contents of air-dried remoistened soils typically differed by -13% to +26% from those of fresh soils, depending on EPS composition, which may contribute to minor differences in absolute values. In our additional experiment comparing seven forest, cropland, and grassland soils, air-drying caused only minor changes: EPS polysaccharides decreased by $1\% \pm 3\%$, EPS proteins increased by $9\% \pm 1\%$, and total EPS (sum of EPS polysaccharides and EPS proteins) changed by $4\% \pm 3\%$ (Supplementary Table 3).

EPS was not measured when the original samples were collected (2017); the soils were air-dried and stored, and EPS was later determined in 2024. As a result, direct assessment of long-term storage effects on these soils was not possible. To evaluate potential storage impacts on EPS, we conducted a comparison experiment using two forest soils collected in 2021 in China. The first soil was sampled in Dongtai City (reported previously⁴), and the second in Nanjing in autumn 2021 for an unpublished earthworm microcosm experiment. After four years of air-dry storage (2021 - 2025), both soils showed only minor changes in EPS contents: EPS polysaccharides increased by 4% on average, EPS proteins increased by 9%, and total EPS changed by 6% on average (Supplementary Table 4). These results indicate that long-term air-dry storage had negligible effects on EPS measurements.

Moreover, necromass contamination appears to have only a minor impact on EPS quantification, as shown by limited muramic acid carryover from bacterial cellular

necromass (~10%)¹, while other amino sugars such as galactosamine and mannosamine were shown to be integral constituents of EPS, and glucosamine distributes between EPS and necromass, which complicates clear attribution at compound-specific levels.

[Comment 12] L329: the cited article compared two soils only, so I wonder if the evidence that in soils originated from sedimentary rocks microbes are more adapted to decompose EPS is really strong and worth highlighting here in the Discussion (but admittedly I only scanned the cited article...)

[Response 12] Thank you for your valuable comment. We agree that the evidence from the cited article may not be sufficient to strongly support the idea that microbes in soils derived from sedimentary rocks are more adapted to decompose EPS. Moreover, since we only measured EPS at a single time point, we are unable to assess its stability or decomposition directly.

Based on your suggestion and a more in-depth analysis of our data (including two-way ANOVA assessing the effects of bedrock and land use on environmental and microbial factors, variance partitioning, and correlations between EPS and environmental parameters (Supplementary Table 5), we revised our interpretation regarding EPS differences among bedrock types. See also our response **R1-22** for further detail.

Accordingly, we deleted the original sentence: “Comparing the three studied bedrock types, carbonate rocks are rich in base cations such as Ca²⁺ and Mg²⁺ ¹⁶, weather rapidly ¹⁷ and have neutral soil pH (Supplementary Table S15) ^{18, 19}”. In the revised **Discussion Section**, we now suggest that the lower EPS content in sedimentary rock soils may be attributed to lower microbial biomass (e.g., MBC and MBN), rather than specific microbial adaptations for EPS decomposition (**Pages 9-11, Lines 259-307**).

[Comment 13] L368-369: alternatively, low MBC can be linked to depleted soil organic matter or degraded soil after intensive agricultural practices.

[Response 13] Thank you very much for your important comment. In response, we have revised the sentence to better reflect the potential role of long-term intensive agricultural management in reducing microbial biomass carbon (MBC) through soil degradation or organic matter depletion. This change aligns more closely with our data and current understanding of soil processes under cropland management. We have revised the sentence from “However, low plant root C inputs in croplands caused the lowest MBC levels and therefore, even at the highest microbial EPS production efficiencies, the lowest EPS concentrations” to “**Conversely, the lowest MBC and SOC observed in cropland soils may reflect soil degradation or organic matter depletion**”

caused by long-term intensive management practices such as tillage, which could in turn contribute to the reduced EPS levels”. However, since the effect of land use on soil EPS was not significant (Fig. 2b), this part of the discussion has been moved to **Supplementary Discussion 4**. See also land use related responses **R4-57** and **R4-59**.

[Comment 14] L398: rather than ‘contents’ I would call these ‘fractions’ (mass or EPS per unit mass of soil organic carbon).

[Response 14] The sentence has been removed as part of the revisions to refine the discussion section.

[Comment 15] L400-404: this material was reported in the Methods and does not add much to the Discussion.

[Response 15] We recognize that this repetition may lead to redundancy in the manuscript. As this information has already been described in the Methods section (**Pages 17-18, Lines 542-557**), we have removed the following sentences from the **Discussion Section**: “Based on literature data and own measurements of model EPS polysaccharide compounds, we derived an average 39.1% C for EPS polysaccharides. In addition, a representative EPS protein C content was derived from genomic data, here the yeast *Saccharomyces cerevisiae*, to predict the amino acid compositions of all potentially expressed polypeptides, which produced a protein C content of 50.7%⁵¹”.

[Comment 16] L416: not clear what ‘amplifying effect’ means.

[Response 16] Thank you for raising this important question. We acknowledge that the original expression “amplifying effect on the soil C cycle” was vague. Our intention was to highlight a possible functional role of EPS in modulating soil enzymatic activity, which may in turn influence C cycling processes. For instance, in our study, we found slight positive correlations between EPS and β -glucosidase (BG) activity, as well as with cellobiohydrolase (CEL, exoglucanase) activity (Fig. 2c). In line with this, a review by De Beeck et al.³⁵ discusses how fungal EPS matrices can enhance the stability and activity of extracellular enzymes under varying soil conditions (e.g., pH, temperature, and denaturing agents), potentially affecting biogeochemical processes. However, our results do not provide strong evidence for a causal relationship, and we have therefore only briefly discussed this phenomenon in the manuscript (**Page 11, Lines 326-328**); please see our response **R3-3** for the specific revision. For further revisions and detailed discussion on enzyme–EPS interactions, please refer to our responses **R1-13**, **R3-3**, and **R4-69**.

[Comment 17] L420: suggested change ‘climate change effects in agricultural systems’

[Response 17] We have revised the sentence to emphasize its relevance under climate change (Page 11, Line 332).

[Comment 18] L426-427: please add rough time scales for decomposition of EPS and microbial necromass, as these statements are now quite vague.

[Response 18] Thank you very much for your valuable suggestion. We agree that adding time scale for the turnover of EPS and microbial necromass helps to clarify the contrast between their persistence in soil. We also revisited the literature for experiments where ¹³C- or ¹⁴C-labeled EPS or EPS-mimics (soluble proteins or polysaccharides) or killed isotopically labelled microbial cells (as necromass-mimics) were added to soils and their decomposition tracked. Extractable microbial/plant polysaccharides and proteins as EPS-like compounds added to soils decomposed fast (60-80% lost as CO₂ in 4-6 months) while fungal ¹⁴C-melanin (<15% loss in one year)^{52, 53, 54} and bacterial and fungal ¹³C-labeled cells decomposed slowly, with mean residence times of 2-8 years⁵⁵. Clearly, such experiments are highly artificial, and decomposition of EPS and necromass formed autochthonously in soils, bound to soil particle surfaces, likely turns over more slowly than dissolved or particulate matter added allochthonous. We have now revised this sentence from “EPS is an active pool that is selectively secreted by microorganisms in response to adverse environmental conditions, being limited in quantity, but otherwise exhibits relatively rapid decomposition^{6, 56}. As the inevitable product of microbial turnover and death, MNC is rather recalcitrant against degradation, and by binding to soil mineral surfaces is preserved in soil for prolonged periods to become an important source of SOC^{57, 58, 59}” to “EPS is a transient pool that is selectively secreted by microorganisms in response to adverse environmental conditions, with no measurable legacy effects observed beyond 2.5 years²⁸. In contrast, MNC as the inevitable product of microbial turnover and death, is more resistant to degradation, with turnover times ranging from multiple years to decades^{55, 60}” (Page 12, Lines 343-346).

[Comment 19] L463: aridity is not a measure of drought severity or frequency of occurrence; to discuss drought effects, perhaps consider a climatic index that describe the intensity and/or frequency of dry periods.

[Response 19] Thank you very much for your helpful comment. We agree that aridity is not the best index for capturing the severity or frequency of drought events. Due to data limitations (i.e., we only had access to long-term mean values such as mean annual precipitation (MAP), mean annual temperature (MAT), and potential evapotranspiration (PET), we used the ADI (aridity index) to reflect long-term climatic dryness rather than short-term drought conditions. To avoid conceptual confusion, we have revised the sentence from “Although this has not yet been demonstrated for soil

microbial communities, we demonstrate this adaptive phenomenon on a large scale i.e. reflected by increasing EPS production efficiency with increasing aridity (indicated by a decrease in the aridity index, ADI) and with worsening drought conditions” to “Although not previously demonstrated for soil microbial communities, we show this adaptive phenomenon at a large scale, reflected by increasing EPS-C/MBC along an aridity (ADI) gradient, suggesting microbial adaptation to long-term water-limited environments. This pattern is further supported by the negative correlation between EPS-C/MBC and SMC (Fig. 10)” (Page 12, Lines 369-372). We have also added a new Fig. 10 to illustrate the relationships between EPS-C/MBC and key driving factors. See also response R4-52 on drought and aridity impacts on EPS.

Figure 10. Linear regressions between EPS-C/MBC and key soil and microbial parameters. **a** Relationship between EPS-C/MBC and soil moisture content. **b** Relationship between EPS-C/MBC and aridity index. **c** Relationship between EPS-C/MBC and microbial growth. **d** Relationship between EPS-C/MBC and microbial carbon use efficiency. Blue lines indicate linear regression fits with 95% confidence intervals (shaded). The R^2 and p -value for each regression are shown on each panel.

Each point represents an individual soil sample. EPS, extracellular polymeric substances; MBC, microbial biomass carbon; C_{growth} , microbial growth.

[Comment 20] L489-492: the logic here is not clear—you start by pointing to a difference between aquatic and terrestrial systems, but then you conclude that carbon and nutrient limitation promote EPS production in both.

[Response 20] Thank you for pointing out the unclear logic in this sentence. We have removed the sentence “Different from aquatic systems, EPS production is obviously not stimulated as an intermittent extracellular store of C and N by secreting EPS polysaccharides and EPS proteins under ample C and nutrient supply in soils”.

[Comment 21] L493-494: I did not understand how exoenzymes are related to the previous sentences on EPS.

[Response 21] We acknowledge that the original sentence was logically disconnected from the preceding discussion of EPS, which may have caused confusion. To improve clarity, we have now removed the sentence “Instead, C and nutrient limitation promote EPS production in aquatic and soil microbial communities, though the mechanistic underpinning remains to be resolved in soils. In aquatic biofilms resource limitation has been called for maximizing the microbial return on investment for exoenzymes^{8, 23}” referring to aquatic ecosystems and extracellular enzymes.

[Comment 22] L518: I agree this would be great to actually estimate EPS production rates.

[Response 22] We agree that quantifying EPS production rates is crucial for understanding microbial carbon allocation and its implications for soil carbon cycling. We have now revised this sentence from “Further research should focus on understanding the dynamics of EPS formation, stabilization, and turnover using isotope tracer methods” to “**Further research should focus on quantifying EPS production rates and understanding the dynamics of its formation and turnover using ^{13}C and ^{15}N isotope tracer techniques. Complementary to this, optimizing EPS extraction and developing compound-specific analytical techniques is essential for improving characterization of EPS dynamics**” (Page 14 , Lines 410-413).

[Comment 23] Supplementary information: it would be helpful to provide captions of the supplementary Excel spreadsheets after the statement: ‘Tables S3-S14 are provided as an attached Excel file’. If data is not available in an open access repository, perhaps it could be included as supplementary information.

[Response 23] Thank you for your helpful suggestion. To improve clarity and better integrate the supplementary information with the relevant main text sections, we have split the original sentence “More detailed analyses of bedrock, land-use types, and environmental factors influencing EPS variables can be found in Supplementary Tables S3–S14” into three separate and context-specific sentences:

- (1) “**Supplementary Tables 6-8 provide additional results from linear models and linear mixed-effects models assessing the influence of bedrock, land-use types, environmental, and microbial factors on total EPS**” (Page 7, Lines 174-176).
- (2) “**Supplementary Tables 9-14 provide additional results from linear models and linear mixed-effects models assessing the influence of bedrock, land-use types, environmental, and microbial factors on EPS polysaccharides and EPS proteins**” (Page 8, Lines 193-196).
- (3) “**Supplementary Tables 15-17 provide additional results from linear models and linear mixed-effects models assessing the influence of bedrock, land-use types, environmental, and microbial factors on EPS-C/MBC ratio**” (Pages 8-9, Lines 224-226).

In addition, we have provided a “**Description of Additional Supplementary Files**” document, which lists the names and main contents of all supplementary tables and datasets, to improve clarity and accessibility. The EPS-related variables—including EPS polysaccharides, EPS proteins, total EPS, EPS-carbon, and site information such as bedrock type, land use type, latitude, longitude, and elevation—are provided as **Supplementary Dataset 1**, while data compiled from previously published studies on soil EPS are included as **Supplementary Dataset 2**. Other related datasets (e.g., microbial necromass carbon, microbial carbon use efficiency, and additional microbial variables) are not yet published and therefore are not included at this stage; however, their mean values across different bedrock and land-use types are provided in **Supplementary Table 5**. We also added a sentence “**All core data supporting the findings of this study are included in the Supplementary Information**” in the **Data Availability** to clarify this (Page 20, Line 629). See also our response **R3-5**.

Reviewer #4 (Comments to the Author):

[Comment 1] Overall, the authors tackle an interesting topic which highly discussed within the community. However, in my point of view that authors do not help in going a step further.

[Response 1] We sincerely thank you for this insightful comment and for acknowledging the relevance of the topic. We understand your concern regarding the need to move the field forward. We apologize if this was not sufficiently clear in our

original manuscript and would like to take this opportunity to clarify the novel aspects of our work. We believe our study advances the field in several key ways:

(1) Large-scale survey: While EPS is increasingly recognized for its role in soil health and soil organic carbon (SOC) cycling, most previous studies have been limited to laboratory experiments or small-scale field surveys. Large-scale investigations, particularly across climatic and geological gradients, have been lacking. Our study represents the systematic survey of soil EPS across a European transect, covering diverse climatic and geological conditions.

(2) Comprehensive assessment of drivers: Previous studies have typically focused on a narrow set of potential drivers. In contrast, we evaluated 66 climatic, vegetation, microbial, and soil variables to identify the key factors controlling EPS across the European transect.

(3) Differentiation of EPS components: We explicitly explored whether EPS polysaccharides and EPS proteins follow similar or distinct environmental drivers, providing new insights into the ecological roles of different EPS fractions.

(4) Quantification of EPS-C: Using conversion factors, we provided an estimate of EPS-carbon (EPS-C). While this approach does not allow precise quantification of EPS-C, it enables comparison with microbial necromass carbon (MNC) and places EPS-C into a broader soil carbon context.

We have also revised the **Introduction** to better emphasize the research gap and questions addressed in this study (**Pages 4-5, Lines 72-79; Pages 5-6, Lines 113-125; all page and line numbers refer to the revised clean version without track changes**). In addition, in this revised version, we have undertaken major efforts to strengthen our contributions:

(1) refined wording throughout to replace vague expressions with precise and professional terminology (e.g., replacing accumulation and contribution with EPS content, and using EPS-C/MBC ratio—cautiously proposed as a proxy for the relative allocation of microbial carbon assimilates to extracellular non-growth products under steady-state conditions—rather than “EPS production efficiency”);

(2) optimized the methodological section by providing a more detailed description of the procedure used to estimate EPS-C (including conversion factors and formulas) (**Pages 17-18, Lines 541-571**) and by elaborating other methodological details for greater clarity and reproducibility (**Pages 15-16, Lines 445-482**);

(3) deepened the statistical analyses by incorporating two-way ANOVA, variance partitioning analysis (VPA), and redundancy analysis (RDA) to better assess the drivers of EPS (**Supplementary Table 5**), and added **Figs. 9** and **10** to illustrate the linear relationships of EPS and the EPS-C/MBC ratio with key factors.;

(4) substantially revised the Discussion, offering an in-depth interpretation of how EPS responds to bedrock type and land use based on more comprehensive data analyses (**Pages 9-11, Lines 259-307 and Supplementary Discussion 4**).

We hope that these improvements contribute to a deeper understanding of the underlying mechanisms driving soil EPS at large scales. We further hope these improvements help address your concerns and more clearly demonstrate the contribution our study makes beyond existing work.

[Comment 2] There is no questioning of the data in relation to a mechanistic understanding. In my opinion, the authors should have an idea of possible connections, which they then test in order to really take a step towards a better understanding. Everything was related to everything, but I miss the mechanistic idea behind it and it's explanation. It would be better if the authors state clear hypothesis and test them.

[Response 2] Thank you very much for your critical comment. We have revised the **Introduction** to more clearly indicate the research gaps we aim to address and the key questions we seek to answer (**please see our responses to Reviewer 4's comment 22, R4-22, and comment 24, R4-24**) (**Pages 5-6, 113-125; 129-132**). In addition, based on more comprehensive data analyses, we have revised the **Discussion** to better articulate mechanistic insights into how bedrock type and land use may influence EPS content and the EPS-C/MBC ratio, reducing inappropriate interpretations and strengthening the discussion with deeper analyses of these mechanisms (see our responses **R1-22, R4-45, and R4-57, Pages 9-11, Lines 259-307 and Supplementary Discussion 4**).

[Comment 3] Furthermore, I have several doubts in terms of the methodology used. Some examples

In the methodology section it is not becoming clear, which data have been taken from other studies and which data were analyzed on which kind of samples (Stored over how long!?) with which kind of purpose, several methods are just mentioned and the methodology is just referenced. But as a reader I expect to find the most important information within the paper.

As the authors did EPS extractions on dry soil stored for an unknown period of time between 2017 and 2025, which I would highly question to produce the same results as other studies of EPS usually extracting EPS from fresh soil samples (could be one reason why the authors gained higher EPS concentrations!?), I am also questioning how and under which conditions the other data have been gained, e.g. CUE!? Same with the turnover time (microbial biomass turnover time) – how was determined on the sample taken along the transect and why was it related to EPS – what is the expectation behind?

[Response 3] Thank you very much for your valuable and thoughtful comment. We appreciate your concern regarding the clarity and reproducibility of the methodology. In response, we have made the following clarifications and revisions:

(1) Clarification of data sources and sample types:

We have added these sentence “A comprehensive set of soil physicochemical, plant, and microbial parameters was used in the analyses, compiled from both previously published sources⁶¹ and unpublished datasets within the same project framework. Soil physicochemical properties were analyzed on either air-dried or fresh samples depending on the parameter. Soil texture (sand, silt, and clay), cation exchange capacity (CEC), exchangeable Ca^{2+} (Ca_e) and Mg^{2+} (Mg_e) were determined on air-dried soils by the Austrian Agency for Health and Food Safety (AGES) according to European and international standards (ÖNORM). Fe oxyhydroxides and Al oxyhydroxides were determined on air-dried soils by extraction with acid ammonium oxalate (Fe_o and Al_o) and Na-dithionite extracts (Fe_d and Al_d) at the Institute of Soil Research (IBF, University of Natural Resources and Life Sciences, Vienna, Austria)”(Page 15, Lines 446-454), “Roots, collected during soil sieving, were washed and oven-dried at 60 °C to determine fine root biomass (FRB). Dried root and air-dried soil samples were ground and analyzed for root carbon (C) and nitrogen (N) contents, and for soil organic carbon (SOC) and total nitrogen (TN) using an elemental analyzer. Root phosphorus (P) was determined by dry ashing, acid extraction, and colorimetric measurement, and root C:N:P ratios were subsequently calculated. Soil pH was measured in water (1:5 (w:v)) using an ISFET pH sensor (Sentron, Leek, the Netherlands). Soil total P (TP) and inorganic P (TIP) were determined from 0.5 M H_2SO_4 extracts of ignited and control samples, with reactive phosphate quantified colorimetrically using malachite green. Organic P (TOP) was calculated as $\text{TP} - \text{TIP}$, and soil C:N:P ratios were subsequently calculated. Dissolved organic C (DOC) and total dissolved N (TDN) were measured in 1 M KCl extracts (1:5, w:v; 1 h) after filtration through ash-free cellulose filters (Whatman) using a TOC/TN analyzer (TOC-VCPH/TNM-1, Shimadzu, Austria). Ammonium (NH_4^+) and nitrate (NO_3^-) were determined colorimetrically. Soil water holding capacity (WHC) was determined by saturating 10 g of field-moist soil with deionized water and allowing it to drain through an ash-free cellulose filter in a funnel for 2.5 h. Soil moisture content (SMC) was determined gravimetrically as the mass loss upon drying fresh soil. For more details on the measurement procedures, see Noll et al.⁶¹” (Page 15, Lines 455-470), and “Microbial process data, such as microbial growth (C_{growth}) and carbon use efficiency (CUE), exoenzymes, and microbial C, N, and P were measured on fresh soils directly after collection. Specifically, microbial biomass carbon (MBC), nitrogen (MBN), and phosphorus (MBP) were determined via chloroform fumigation extraction⁶², from which the MBC:MBN:MBP ratios were calculated. Microbial nitrogen use efficiency (NUE) was calculated based on concurrent

measurements of microbial growth and gross N mineralization rates⁶³. CUE, C_{growth} , growth normalized to microbial biomass carbon (q_{Growth}), and microbial biomass turnover time (MBT) were estimated from measurements of MBC, basal respiration, and ¹⁸O incorporation into dsDNA, following Zheng et al.⁶⁴” (Pages 15-16, Lines 474-482) to the **Method Section** to offer more detailed information on the analytical procedures. Please also see **R3-5** and **R3-23**.

(2) EPS extraction method and concerns about air-dried samples and long-term storage of samples:

We thank you for your critical and insightful comment regarding the use of air-dried soil for EPS extraction. We acknowledge that EPS is typically extracted from fresh soils in many previous studies, but not in all (see Zhang et al.³⁷ for a tabular compilation of studies including information on EPS extraction method and fresh/air-dry soil use, while Crouzet et al.⁴¹ extracted EPS polysaccharides from dried soils after microcosm incubation). We fully agree that ensuring the extraction method does not alter the native state of EPS is paramount.

In our study, soil samples were collected from a vast geographical region, making immediate processing and fresh soil extraction logistically unfeasible. Air-drying followed by standardized storage is a widely adopted practice in soil science to preserve samples and ensure compositional stability before analysis. Furthermore, air-drying facilitates the homogenization of soils, which significantly improves the reproducibility of our extraction protocol.

To address concerns regarding the potential effects of long-term storage and air-drying, we conducted additional experiments comparing EPS extractions from fresh soils, air-dried soils, and soils stored for approximately four years. These experiments indicated that while absolute EPS concentrations may be slightly affected by storage conditions, the changes were minor: for air-dried soils, EPS polysaccharides decreased by $-1\% \pm 3\%$ (mean \pm SE), EPS proteins increased by $9\% \pm 1\%$, and total EPS (sum of polysaccharides and proteins) changed by $4\% \pm 3\%$. For long-term stored soils, EPS polysaccharides increased by 4% on average, EPS proteins increased by 9%, and total EPS changed by 6% on average. Importantly, the relative patterns across samples remained consistent (see **Supplementary Tables 3 and 4**).

Specifically, we collected a total of seven soil samples from forests, croplands, and grasslands. After collection, soils were immediately sieved through a 2 mm mesh and homogenized. Each sample was split into two subsamples: one analyzed immediately (fresh), and the other air-dried prior to analysis. For EPS polysaccharides, extraction from air-dried soils showed a mean difference of $-1\% \pm 3\%$ (mean \pm SE, $n = 7$; range: -10% to +11%) relative to fresh soils. In contrast, EPS protein extraction from air-dried soils showed a mean difference of $+9\% \pm 1\%$ (mean \pm SE, $n = 5$; range: +7% to +14%) relative to fresh soils across cropland, grassland, and forest soils (**Please see below**).

Zhang et al. ³⁷ also compared EPS contents measured in a paddy red clay soil in a fresh state, in fresh soils after 2-weeks pre-incubation, and in air-dried soils after rewetting and 2-weeks pre-incubation, showing changes (-13% to +26%) in contents of EPS protein and EPS-polysaccharides in air-dried soils, whereas the total EPS content varied by only about 2%.

Soil types	Soil status	EPS polysaccharide	EPS uronic acid	EPS protein	EPS sum
		(μg g ⁻¹ soil)			
Ultisol; arable land	Moist	122.1 ± 16.7	41.9 ± 2.6	86.6 ± 4.8	250.6
Ultisol; arable land	Dried	153.7 ± 12.9	25.7 ± 5.5	75.5 ± 5.1	254.9
Change due to drying		↑ 26%	↓ 39%	↓ 13%	↑ 2%

Long-term storage (4 years; from 2021 to 2025) did not change soil EPS after four years in an air-dried state, as in two soils both EPS components increased by only 1-10% measured relative to fresh soil (Supplementary Table 4). Please also see our responses to similar critiques in **R3-11** and particularly **R4-78** for added analysis of storage and drying effects on soil EPS contents.

Due to logistical constraints, we were unable to perform a direct measurement of fresh soils for this specific study. However, we opted for air-drying to ensure sample stability and homogeneity across all our samples, which was critical for our large-scale comparative analysis. We have now explicitly described this in the **Methods section**: “Soil EPS (including polysaccharides and proteins) was extracted from air-dried soil samples in 2024 using cation exchange resin (CER)³⁶” (**Page 16, Lines 490-491**) and “Furthermore, to ensure the robustness of EPS measurements from air-dried samples used in this study, we additionally tested whether soil moisture status and storage duration influenced EPS extraction. The procedures and results are presented in Supplementary Tables 3 and 4 and discussed in Supplementary Discussion 2” (**Page 16, Lines 499-502**) and acknowledged this limitation in the **Discussion**: “This difference partly reflects differences in bedrock and land-use composition, sampling season, and methodological factors (Supplementary Discussion 1). Due to sample availability, we used air-dried soils, whereas most previous studies used fresh soils. Our preliminary experiment showed minor effects of air-drying: EPS polysaccharides decreased by 1 ± 3%, EPS proteins increased by 9 ± 1%, and total EPS (polysaccharides + proteins) increased by 4 ± 3%. Long-term soil storage had similarly no impact on EPS content (Supplementary Discussion 2)” (**Page 9, Lines 253-258**). Due to word count limitations, additional discussions have been included in **Supplementary Discussion 2**. Please see below for details:

Supplementary Discussion 2 - Effects of soil drying and long-term storage on EPS

Due to sample availability, we used air-dried soils, whereas most previous studies used fresh soils. Zhang et al.³⁷ reported that EPS contents of air-dried remoistened soils typically differed by -13% to +26% from those of fresh soils, depending on EPS composition, which may contribute to minor differences in absolute values. In our additional experiment comparing seven forest, cropland, and grassland soils, air-drying caused only minor changes: EPS polysaccharides decreased by $1\% \pm 3\%$, EPS proteins increased by $9\% \pm 1\%$, and total EPS (sum of EPS polysaccharides and EPS proteins) changed by $4\% \pm 3\%$ (Supplementary Table 3).

EPS was not measured when the original samples were collected (2017); the soils were air-dried and stored, and EPS was later determined in 2024. As a result, direct assessment of long-term storage effects on these soils was not possible. To evaluate potential storage impacts on EPS, we conducted a comparison experiment using two forest soils collected in 2021 in China. The first soil was sampled in Dongtai City (reported previously⁴), and the second in Nanjing in autumn 2021 for an unpublished earthworm microcosm experiment. After four years of air-dry storage (2021 - 2025), both soils showed only minor changes in EPS contents: EPS polysaccharides increased by 4% on average, EPS proteins increased by 9%, and total EPS changed by 6% on average (Supplementary Table 4). These results indicate that long-term air-dry storage had negligible effects on EPS measurements.

Moreover, necromass contamination appears to have only a minor impact on EPS quantification, as shown by limited muramic acid carryover from bacterial cellular necromass ($\sim 10\%$)¹, while other amino sugars such as galactosamine and mannosamine were shown to be integral constituents of EPS, and glucosamine distributes between EPS and necromass, which complicates clear attribution at compound-specific levels.

We agree that future studies comparing both methods would be valuable to standardize the protocol. While the absolute values might be influenced by air-drying, the relative differences observed among experimental groups are considered reliable and informative for the purposes of this study.

Supplementary Table 3 | Comparison of EPS extraction between fresh and air-dried soils from different land use types in China.

Site	Land use	Soil type	Soil dry/moist	EPS	EPS	Total EPS	EPS	EPS	Total EPS
				polysaccharides	proteins		polysaccharides	proteins	
				(μg g ⁻¹ soil)		dry vs fresh			
Dongtai City, Jiangsu Province	Forest	Fluvisol (Sandy loam)	dry	156.64	197.75	354.39			
Dongtai City, Jiangsu Province	Forest	Fluvisol (Sandy loam)	fresh	168.60	184.23	352.83	-7%	7%	0%
Pukou District, Nanjing	Cropland	Yellow brown soil	dry	140.10	135.88	275.98			
Pukou District, Nanjing	Cropland	Yellow brown soil	fresh	135.16	119.05	254.20	4%	14%	9%
Laoshan, Nanjing	Forest	Calcareous soil	dry	313.20	326.90	640.10			
Laoshan, Nanjing	Forest	Calcareous soil	fresh	300.11	300.23	600.35	4%	9%	7%
Yunnan Province	Grassland	Red soil	dry	62.78	NA	NA			
Yunnan Province	Grassland	Red soil	fresh	65.89	NA	NA	-5%	NA	NA
Cixi, Zhejiang Province	Cropland	Coastal alluvial sediments	dry	132.67	NA	NA			
Cixi, Zhejiang Province	Cropland	Coastal alluvial sediments	fresh	140.91	NA	NA	-6%	NA	NA
Pukou District, Nanjing	Forest	Yellow brown soil	dry	220.96	146.25	367.21			
Pukou District, Nanjing	Forest	Yellow brown soil	fresh	245.02	136.03	381.06	-10%	8%	-4%
Sihong County, Suqian	Forest	Alluvial deposits	dry	355.26	293.08	648.34			
Sihong County, Suqian	Forest	Alluvial deposits	fresh	320.02	273.38	593.40	11%	7%	9%
Purple Mountain Park, Nanjing	Forest	Humic cambisol	dry	336.02	NA	NA			
Purple Mountain Park, Nanjing	Forest	Humic cambisol	fresh	323.99	NA	NA	4%	NA	NA

EPS, extracellular polymeric substance. Total EPS refers to the sum of EPS polysaccharides and EPS proteins. NA indicates that the data are not available.

Supplementary Table 4 | Effects of long-term storage on EPS polysaccharides and EPS proteins in soils.

Soil type	2025			2021			2025 vs 2021	
	EPS polysaccharides ($\mu\text{g g}^{-1}$ soil)	Average EPS polysaccharides ($\mu\text{g g}^{-1}$ soil)	EPS proteins ($\mu\text{g g}^{-1}$ soil)	Average EPS proteins ($\mu\text{g g}^{-1}$ soil)	Average EPS polysaccharides ($\mu\text{g g}^{-1}$ soil)	Average EPS proteins ($\mu\text{g g}^{-1}$ soil)	EPS polysaccharides	EPS proteins
Earthworm microcosm soil	964.87	837.84	NA	NA	828.85	NA	1.09%	NA
Earthworm microcosm soil	710.82		NA	NA				
Fluvisol; Forest soil	122.07	107.75	44.48	37.53	99.94	34.33	7.82%	9.33%
Fluvisol; Forest soil	112.27		NA					
Fluvisol; Forest soil	101.13		38.16					
Fluvisol; Forest soil	97.63		23.00					
Fluvisol; Forest soil	123.39		44.49					
Fluvisol; Forest soil	90.03		NA					

EPS, extracellular polymeric substances.

(3) Clarification on microbial parameters such as CUE and turnover time:

Microbial processes such as growth, turnover, exoenzymes, microbial biomass and carbon use efficiency (CUE) were measured shortly after sample collection in 2017, all determined on fresh soil samples. Those data are part of a previous study of large-scale environmental controls of soil microbial carbon processing and CUE, which is in the process of being published by the main author (Qing Zheng, et al., unpublished). Other soil physicochemical characteristics as potential drivers of soil biogeochemistry were measured on air-dried soil samples in 2017 and 2018, such as soil pH, texture, SOC, TN, TP, exchangeable cations, CEC, and others. We have now explicitly described this in the **Methods section (Pages 15-16, Lines 474-482)**.

[Comment 4] Furthermore, important calculations such as EPS-C is not described in detail – but used as a kind of indicator for EPS stabilization or accumulation (all of the terms used vague without clear definition). Since the calculation is not clearly explained in the method section, it feels to me as if the authors interpret the concentration as one or the other, depending on their discretion - as it suits the story better.

[Response 4] Thank you very much for your thoughtful comment. We acknowledge that the definition and calculation of EPS-C were not clearly described in the original submission. In response, we have now added a new subsection in the **Methods Section** specifically explaining how EPS-C was calculated (please see below):

Estimation of EPS-C

The C content of EPS polysaccharides (39.1% C) was estimated by elemental analysis in our laboratory as the average of four microbial and plant exopolysaccharide standards, including hyaluronic acid (37.5% C), xylan (40.5% C), pectin (38.3% C), and arabinogalactan (40.2% C).

The C content of EPS proteins (50.7% C) was estimated from the global average protein N content (16% N), multiplied by a weighted mean C:N ratio of proteinogenic amino acids. Given that individual proteinogenic amino acids show variable C:N ratios, ranging from 1.5:1 to 9:1, and since proteins show marked deviations in their amino acid composition, we refrained from the use of a mean proteinogenic amino acid C:N ratio. Instead, we here applied a weighted mean amino acid C:N ratio calculated from the amino acid sequences of all genome-encoded polypeptides comprising the full proteome of the yeast *Saccharomyces cerevisiae* as an environmental microbe representative⁵¹. This gives a proteome-average of 5.03 mol C and 1.36 mol N per mol protein-derived amino acids⁵¹, a molar C:N ratio of 3.70 ($5.03 / 1.36 = 3.70$) and a mass-based C:N ratio of 3.17 ($5.03 \text{ mol C} \times 12 / 1.36 \text{ mol N} \times 14 = 3.17$). Using a global average protein N content of 16% by mass^{65, 66}, and a weighted mass-based amino acid

C:N in microbial proteins of 3.17, the corresponding protein C content can be approximated by formula (1):

$$\text{protein C content [\%C]} = 16.0 (\%N) \times 3.17 (\text{C:N}) = 50.7\% \quad (1)$$

These conversion factors were applied to measured EPS polysaccharide and EPS protein contents (in mg g⁻¹ soil dw) to estimate their respective C contents.

The final formula (2) for summed EPS-C is:

$$\text{EPS - C (g C kg}^{-1} \text{ soil dw)} = (\text{EPS polysaccharides} \times 39.1\%) + (\text{EPS proteins} \times 50.7\%) \quad (2)$$

which provides an approximate estimate based on representative C content. To our knowledge, this represents an important attempt to quantify soil EPS-C contents, as such estimations were previously hindered by the lack of representative data on the C content of EPS components. In this study, we addressed this gap by determining average C content values from standard compounds representative of microbial EPS polysaccharides and by calculating protein C from proteome-averaged protein C:N ratios and published global protein N contents. While these estimates provide a useful approximation, we acknowledge that they do not fully capture the chemical diversity of EPS in soils. (Pages 17-18, Lines 541-568). Please also see our response R4-65/66/67.

We also recognize that the terms "EPS stabilization" and "EPS accumulation" were previously used without sufficiently clear definitions. To address this, we have revised the manuscript to define these terms more precisely and consistently. Specifically, we now refer to "EPS-C content" as the measurable pool of extractable extracellular polymeric substances in the soil, which reflects the steady-state outcome of concurrent microbial production, degradation, and potential stabilization of EPS via interactions with soil minerals. We avoid interpreting EPS-C solely as an indicator of EPS "accumulation" and EPS-C/MBC as proxy of EPS "production efficiency" and instead focus on its variation and correlations with soil and environmental factors. See also our response R1-1, R1-11 and R3-2 on the same topic.

These clarifications aim to improve the transparency and rigor of our interpretations. We thank you again for pointing out this important issue.

[Comment 5] Overall, the authors draw conclusions that they do not prove and rely on speculation and, force and back correlation. In addition, they use words without defining them and defining their analyses and are therefore appearing very very vague.

[Response 5] Thank you for your critical and constructive feedback. We acknowledge that some of our previous interpretations may have lacked sufficient clarity. In response, we have carefully revised the manuscript to ensure that all conclusions are more tightly aligned with the data and do not overreach the scope of our analyses. See also responses

R1-22 and **R4-45** on renewed the revised interpretation of bedrock effects on EPS and **R4-57** on land use effects on EPS.

In particular, we have revised the use of potentially misleading terms such as “accumulation” and “stabilization,” which imply processes that we did not directly measure. Instead, we now refer to “EPS contents” to more accurately describe the observed values. Additionally, we have clarified that the EPS-C/MBC ratio cannot be interpreted as a measure of production efficiency, since EPS levels in natural soils are influenced by the balance of production, degradation, and stabilization (**R1-11**, **R3-2**), as well as extraction limitations (**R1-22**). We now refer to this ratio more cautiously as a potential indicator of microbial C allocation under steady-state conditions, and we explicitly acknowledge its limitations (Page 5, Lines 84-93; Page 10, Lines 283-287; Page 12; Lines 358-362). We hope these changes improve the clarity and scientific rigor of our manuscript and appropriately address your concerns.

[Comment 6] Line 24-25: EPS is part of the SOC, or? What do you want to say in regard to the “contribution”? Specifically, as you relate to it within the next sentence.

[Response 6] We agree that the term “contribution” was initially too vague and could cause confusion. To clarify: EPS is a component of SOC, contributing directly as part of the soil carbon pool. In addition, EPS plays an indirect role in SOC stabilization by acting as a “glue” that promotes aggregate formation and organo-mineral associations^{8, 28}. Although the role of EPS in C dynamics has been proposed in previous studies, these have generally been based on limited datasets. The goal of our study is to systematically assess the levels of EPS across soils and to identify the environmental and microbial factors that govern its content at large scales. We regret that this motivation was not clearly conveyed in the original abstract. Accordingly, we have revised the sentence from “Extracellular polymeric substances (EPS) are a vital component of microbial residues which contribute to soil organic carbon (SOC)” to “**Extracellular polymeric substances (EPS) are key microbial residues that contribute directly to soil organic carbon (SOC) and indirectly stabilize it by enhancing soil aggregation**”. We hope this revision improves clarity and more accurately reflects the intended meaning (Page 3, Lines 25-26).

[Comment 7] Line 26-27: At that point in time it is not clear to which theories you refer too. Please rewrite. What is the specific knowledge gap you want to fulfill, “contribution” to SOC storage – drivers/controls of EPS?

[Response 7] We agree that the original sentence lacked clarity. To address this and combined with other reviewer’s comments (e.g., **R1-2**), we have revised the sentence from “However, despite various conjectures and hypotheses regarding soil EPS controls, empirical research and experimental evidence to validate these theories have remained

highly limited” to the clearer and more precise: “**Yet, their abundance and large-scale environmental and microbial controls remain poorly quantified**”. This revision better reflects the specific knowledge gap we aim to address regarding the contents and drivers of soil EPS at large scales (**Page 3, Lines 26-27**).

[Comment 8] Line 29: Do you think that large scale controls matter for EPS? What is meant here by large-scale controls?

[Response 8] Thank you for your helpful comment and the opportunity to clarify. By “large-scale controls,” we refer to broad environmental gradients (e.g., climate, bedrock type, land use, and soil properties) that vary across continental scales and may influence soil EPS contents and dynamics. Current empirical data on soil EPS are primarily derived from local- or small-scale field studies, with data at regional to continental scales being particularly limited. Therefore, we believe investigating such large-scale drivers is critical for understanding how EPS distribution patterns and their controlling mechanisms differ across ecosystems. Accordingly, we have revised the original sentence from “In this study, we addressed this knowledge gap by conducting extensive soil sampling across Europe, encompassing diverse climates and bedrock and land use types, to systematically investigate soil EPS contents and large-scale controls” to “**We addressed this gap by conducting extensive soil sampling across a European transect spanning diverse climates, bedrock types, and land uses**” to more accurately reflect the spatial scope of our study (**Page 3, Lines 27-29**).

[Comment 9] Line 42: Unclear, what the authors mean with EPS accumulation!/? Include as for EPS production efficiency... I don't think that prompted is the right verb here – rewrite

[Response 9] Thank you for your critical comment. In response, we have revised this sentence from “On a large scale, soil EPS accumulation was promoted by its production efficiency and by soil factors promoting the sorption and stabilization of EPS, such as clay content, exchangeable Ca and Fe oxides” to “**Across the transect, EPS increased with microbial biomass as a proxy for EPS production potential, while soil properties such as clay content and exchangeable calcium enhanced EPS retention**”, to provide a clearer and more mechanistic explanation (**Page 3, Lines 34-36**). See also response **R1-5**.

[Comment 10] Line 45: In my opinion the authors should not write “critical component of the soil-stable C pool” – as at least to my knowledge it is not proven that it is part of the soil-stable C pool – and the authors do not prove that either in their manuscript – rewriting necessary

[Response 10] Thank you for your helpful comment. We apologize for the imprecise

wording in the original sentence. It is true that CER does not extract the stable EPS fraction and that much of the EPS determined is cycling relatively fast, which has been indicated by additions of ¹³C- and ¹⁴C-labelled EPS components to soils. EPS is therefore not necessarily part of the stable SOC fraction but it can contribute to stabilize SOC through aggregation. See the related responses **R1-22** and **R3-18**. We have revised the sentence from “These findings underscore the significant yet overlooked role of EPS as a critical component of the soil-stable C pool, as it influences microbial C allocation and SOC stabilization and should be further studied to better understand soil C cycling” to “Here, we show that EPS is a functionally important microbial residue, regulated by both environmental filters and microbial allocation strategies, with significant implications for soil carbon cycling and sequestration” (Page 3, Lines 36-38).

[Comment 11] Line 50 “elements” - remove by components

[Response 11] Done (Page 4, Line 42).

[Comment 12] Line 52-53: When I came across the number of 30-60% of SOC for amino sugars, is a contribution of 1.6% as mentioned for EPS substantial?

[Response 12] Thank you for your thoughtful comment. While our data indicate that EPS contributes only ~1.6% to SOC, we consider EPS a relevant and functionally important component of SOC for several reasons:

(1) EPS as a microbial investment: EPS is a costly microbial product secreted in response to environmental stressors (e.g., drought and salinity). It plays critical roles in microbial survival and adaptation, especially under shifting climate conditions ^{5, 8}.

(2) Direct and indirect contributions of EPS to SOC: Beyond its direct contribution to SOC, EPS facilitates soil aggregation and organo-mineral associations ^{5, 8}, which enhance the stabilization of other (non-EPS) organic matter. These indirect effects are likely substantial and relevant for long-term carbon storage. See response **R4-6**.

(3) Potential underestimation of EPS: The actual EPS content in soils may be considerably higher than current values suggest, mainly due to methodological limitations ²⁸. See response **R1-31** The CER extraction method primarily targets the more labile fraction of EPS (bound by multivalent cation bridging and electrostatic binding forces), and current techniques are not yet capable of fully recovering the more persistent, potentially stabilized forms of EPS in soil ²⁸. See also our response **R1-22**. Moreover, the extraction efficiency of CER is low—estimated to capture only about 1/2 to 1/20 of total soil EPS—further contributing to EPS underestimation ^{36, 38, 47}. In contrast, estimates of microbial necromass carbon (MNC) based on amino sugar biomarkers may be inflated, since these markers are not exclusive to necromass ²⁹. For example, galactosamine can be found in substantial concentrations in plant residues,

and other non-necromass materials, even EPS itself²⁹. Therefore, while measures of EPS-C are understood to be underestimates, measures of MNC are known to be overestimates³⁰ and perhaps with a larger margin for error and variability. This means that while on face value our results suggest that MNC was the larger pool of C, true values might be more comparable in size.

Therefore, even if the measured quantitative contribution of EPS to SOC appears small, both methodological constraints and functional evidence indicate that EPS plays a critical and previously overlooked role in soil carbon cycling and stabilization.

We have added the following sentence in the **Introduction** to more specifically highlight the importance of EPS in soil, and we hope this clarifies your concern: “Independent of the presence of soil biofilms, EPS plays multifunctional roles in soil ecosystems^{5, 8}. EPS enhances water retention, mediates nutrient storage, and supports microbial resilience under environmental stress⁸. Beyond favourable effects of EPS on microbial performance, EPS contributes to SOC dynamics by serving as a microbial-derived direct C input and by promoting soil aggregation and mineral-organic associations, which are critical processes for physical and chemical stabilization of SOC^{5, 6}. Furthermore, EPS facilitates microbial attachment to soil particles, potentially enhancing the association between microbial cellular necromass and mineral surfaces following microbial death. Such interactions between EPS and other microbial residues may accelerate soil microbial-derived C stabilization and thus represent an additional, underexplored pathway through which EPS can promote long-term C storage⁵⁰. These diverse functions underscore the ecological relevance of EPS in structuring soils and regulating C cycling” (Page 4, Lines 59-70).

[Comment 13] Line 55: “a natural mixed polymer” – as I understood EPS composition, it is composed by a variety of different constituents – please adapt

[Response 13] Thank you for your helpful comment. We have revised this sentence from “EPS is a natural mixed polymer secreted by microorganisms that supports them to resist environmental stresses and absorb nutrients and other resources^{6, 23}” to “EPS comprises a heterogeneous mixture of biopolymers (exoproteins, exopolysaccharides) targeted for their extracellular role and secreted by microorganisms that support them to resist environmental stresses and absorb nutrients and other resources^{6, 23}” to improve accuracy (Page 4, Lines 48-50).

[Comment 14] Line 70: I would wish that the authors elaborate a bit more on the “potential outstanding role of EPS... for SOC dynamics...” it is mentioned for the microbial functioning but for SOC dynamics it remains unclear to me at that point in time, specifically if their existence is debated overall in soil. That points need more

preparation in my point of view, specifically if that is a main topic for the whole manuscript.

[Response 14] Thank you for your important comment. We agree that our original framing did not sufficiently elaborate on the potential implications of EPS for SOC dynamics. While our previous version briefly mentioned that EPS may contribute to soil structure through its viscosity and water-holding properties, we recognize the need to make the relevance of EPS to SOC dynamics more explicit. While the continuous presence of biofilms in soils remains debated, the presence of EPS in soils is well established and not in question^{8, 67}. To address this, we have clarified that EPS may influence SOC dynamics through both direct and indirect mechanisms. Specifically, EPS contributes directly to the SOC pool as a form of microbial residue, and indirectly by promoting soil aggregation and mineral–organic associations—processes that are essential for the stabilization of SOC. Accordingly, we have revised this sentence (**Page 4, Lines 59-70**), please see response **R4-12**).

***[Comment 15]** Line 72: In regard of the existing literature the authors should check the current literature a bit stronger – e.g. Zhang et al. 2023 Chemical geology a whole table summarizing studies of EPS measurements from different soil types, including different land uses as well.*

[Response 15] Thank you for your valuable suggestion. We apologize for the oversight in our previous literature coverage. We have now re-examined the literature on soil EPS using the CER extraction method and updated our database accordingly (Please see the **Supplementary Dataset 2**). Specifically, through a comprehensive literature review, we collected 22 studies using the CER method for soil EPS extraction, comprising 216 EPS measurements. After filtering to retain data from bulk topsoil, including some pre-incubated samples without added C or N, 18 studies with 83 measurements remained. Mean values from these datasets were calculated (**Supplementary Table 1**) and plotted in the revised **Fig. 1**.

Figure 1. Comparison of EPS polysaccharides and EPS proteins across the European transect and with data derived from previous studies. a Comparison of EPS polysaccharide contents. **b** Comparison of EPS protein contents. Data were collected from 18 previous studies that used the cation exchange resin method to extract soil EPS, totaling 30 site \times land use combinations. The numbers in the figure indicate frequency. Specific data are listed in Supplementary Table 1, and the complete dataset is provided in Supplementary Dataset 2. EPS, extracellular polymeric substances.

Supplementary Table 1 | Previously published quantitative data on extracellular polymeric substances (EPS) extracted using the cation exchange resin method.

Reference	Land use type	EPS polysaccharides	EPS proteins ($\mu\text{g g}^{-1}$ soil)	Total EPS
37	Cropland	151.08	86.56	237.64
	NA	614.29	117.22	731.52
45	NA	121.02	47.67	168.69
	NA	115.72	12.52	128.24
	Grass-Grass	346.02	213.17	559.19
	Grass-Arable	339.55	194.56	534.11
	Grass-Fallow	293.99	156.96	450.95
	Arable-Grass	434.87	198.24	633.11
28	Arable-Arable	317.79	184.64	502.43
	Arable-Fallow	374.66	175.67	550.33
	Fallow-Grass	381.50	191.27	572.77
	Fallow-Arable	235.55	110.67	346.22
	Fallow-Fallow	286.96	126.43	413.39
36	Fallow	169.00	43.00	212.00
	Grassland	401.00	163.00	564.00
	NA	398.00	122.00	520.00
46	NA	239.00	42.00	281.00
	NA	264.00	60.00	324.00
	Grassland	1120.00	NA	NA
47	Cropland	800.00	NA	NA
	Woodland	830.00	NA	NA
	Cropland	743.54	NA	NA
	Woodland	252.01	52.36	304.36
	Cropland	447.44	273.29	720.74
9	Woodland	215.99	60.00	276.00
	Cropland	332.80	194.10	526.90
	Woodland	278.31	80.18	358.49
	Cropland	272.88	145.05	417.93
48	Grassland	191.33	94.90	286.23
4	Woodland	158.31	79.28	237.58
2	Grass-Arable	467.33	177.76	645.08
40	Grassland	456.41	NA	NA
	Cropland	262.78	NA	NA
43	Shrub land	52.19	NA	NA
	Shrub land	76.30	NA	NA
27	Shrub land	130.37	NA	NA
	Shrub land	120.79	NA	NA
44	Woodland	158.60	44.90	203.50
	Woodland	160.30	42.30	202.60

	Woodland	98.30	15.70	114.00
41	Cropland	367.95	NA	NA
39	Fallow	175.43	78.97	254.40
	Cropland	260.78	133.86	394.64
49	Woodland	317.20	301.82	619.01
50	Cropland	471.25	205.19	676.44
	Woodland	691.82	364.28	1056.10

EPS, extracellular polymeric substances; Total EPS refers to the sum of EPS polysaccharides and EPS proteins. NA indicates that the data are not available.

In response, we have revised the sentence from “So far, only about a dozen (~10) sites differing in climate and geology have been tested globally, with limited measurements primarily focused on differences in land use²⁸, soil types⁹, and plantation ages⁴” to “So far, the climatic and geological coverage of soil EPS measurements was relatively restricted, with studies confined to smaller-scale and/or local mechanistic questions, e.g., targeting soil EPS responses to current and legacy land use²⁸, the interaction of land use and geology⁹, land reclamation³⁹, tree plantation age^{4, 50}, agricultural management^{40, 41}, and root-soil aggregate interactions in forests and grasslands^{27, 43, 44}, which has hindered understanding of large-scale controls of soil EPS” (Page 4, Lines 72-77). Please also see our response R4-41.

[Comment 16] Line 74-75: The authors should introduce microbial necromass in regard to the before mentioned cellular and extracellular residues. Please better pave the way why EPS could/should affect necromass - it is not becoming clear to me where the differences between the components are.

[Response 16] Thank you for your constructive suggestion. In response, we have revised the beginning of the **Introduction** from “Soil microbial residues are essential elements of the stable carbon (C) pool, and can be classified as cellular and extracellular residues^{4, 68}. Cellular residues are widely quantified using amino sugars as biomarkers” to “Soil microbial residues are essential components of the soil carbon (C) pool and can be classified as cellular and extracellular residues^{4, 68}. Cellular residues-also known as microbial necromass-mainly consist of microbial cell wall fragments and are widely quantified using amino sugars as biomarkers⁵⁹” (Page 4, Lines 42-44). This revision allows us to explicitly introduce the concept of microbial necromass early in the manuscript. Additionally, we revised the sentence from “For example, the high viscosity and water retention capacity of EPS facilitate soil particle binding and aggregation and improve soil moisture and soil structure⁸” to “Furthermore, EPS facilitates microbial attachment to soil particles, potentially enhancing the association between microbial cellular necromass and mineral surfaces following microbial death. Such interactions between EPS and other microbial residues may accelerate soil

microbial-derived C stabilization and thus represent an additional, underexplored pathway through which EPS can promote long-term C storage⁵⁰” (Page 4, Lines 65-69)” to better clarify the potential relationship between soil EPS and microbial necromass.

[Comment 17] Line 76-77: Is this statement assumed or verified by the papers referenced? I screened the article shortly and I could find a prove of it – Please be as precise as possible in your statements ... throughout the manuscript and adapt accordingly.

[Response 17] Thank you for your careful observation. We have revised this sentence from “The contribution of EPS to SOC depends on (i) the microbial secretion of EPS and (ii) its long-term protection by binding with soil particles^{3, 4}” to “**The content of soil EPS may depend on (i) the microbial secretion of EPS and (ii) its association with soil particles^{5, 6, 7}, as well as (iii) its decomposition and recycling**” to more accurately reflect the current state of knowledge and the available evidence (Page 5, Lines 80-81).

[Comment 18] Line 87-95: Repetitive – can be more condensed and targeted

[Response 18] We have revised this sentence from “This indicates that the soil environment and C and nutrient availability are significant triggers for EPS secretion by microorganisms⁸. Finally, when microorganisms reproduce and microbial communities expand, microbial cells can increase their adhesion capacities to the surfaces of soil particles by secreting EPS, which not only strengthens their attachment but also serves as an effective barrier to block invasion from competing microorganisms^{9, 23}. In summary, in soils microbial EPS secretion is contingent on microbial biomass synthesis (e.g., efficacy of microbial C and N use, microbial growth) and environmental stress⁸, which depend on soil pH, water content, and cation exchange capacity (CEC)^{56, 69} and on the supply of soil nutrients (e.g., SOC, N, and phosphorus (P))^{35, 70}, but also are related to other factors such as population survival strategies⁹” to “**Finally, as microbial communities expand, EPS secretion enhances cell adhesion on soil particles and helps protect against microbial competition^{9, 23}. Collectively, these findings underscore that EPS secretion is governed by microbial growth and environmental stress⁸- both influenced by soil properties, nutrient availability^{56, 69, 35, 70}, and population-level survival strategies⁹**” to be more condensed and targeted (Page 5, Lines 99-103).

[Comment 19] Line 97: As indicated beforehand, there is no prove of long-term preservation of EPS in soils – please add the prove to be able to make that statement or adapt accordingly

Screened shortly - Craig et al. did not include EPS measurements and the article by

Chenu – no prove of preservation – assumption due was done due to the formed structures... and that needs be cleared up....

[Response 19] We acknowledge that there is currently no direct evidence supporting the long-term preservation of microbial EPS in soils. Externally added EPS sugars and proteins decompose rapidly in soils (see **R3-18**). However, the lack of evidence for long-term preservation of EPS is partially explained by the specific EPS extraction method (CER) used, which only extracts weakly bound EPS (sorbed via Ca²⁺ bridges) but not more strongly sorbed EPS, if present (see response **R1-22**). More stringent extraction procedures, on the other hand, co-extract more non-EPS compounds, making the quantification of EPS challenging. In response, we have revised this sentence “Conversely, soils enhance the physical protection of EPS that is secreted by microorganisms^{8, 28}, where mineral binding of EPS allows for its long-term preservation in the soil and contributes to the SOC pool⁴” to “**Additionally, the content of soil EPS is influenced by its interactions with soil particles, which offer protection against degradation⁷¹, potentially affecting its contribution to the SOC pool**” (**Page 5, Lines 104-105**).

[Comment 20] Line 106: The authors need to clarify for my better understanding what they mean by “accumulation” of EPS – are the authors using it simultaneously for preservation, stabilization and why do the authors think so.... EPS are from their composition relatively easy to degrade (proteins / sugars) – I would assume that EPS occurrence in soils is quite dynamic (as microbes use it as well as nutrient storage), so why an accumulation!? Terms need to be clarified.

[Response 20] Thank you for your insightful comment. We agree that the term “accumulation” may be ambiguous in this context. Our intention was to quantify soil EPS levels and identify their environmental and microbial drivers. Therefore, we have revised the sentence “Various hypotheses have been put forward regarding the controls of EPS secretion and accumulation” to “**Various hypotheses have been proposed to explain the environmental controls on microbial EPS secretion and soil EPS content, but most have been tested based on laboratory or small-scale field studies**” (**Pages 5-6, Lines 113-115**). But it is important to note that “labile” compounds such as proteins and complex sugars can accumulate in soils to levels of 10% of SOC (carbohydrates) and 80% of SON (proteins) while recalcitrant compounds such as lignin almost completely decompose and disappear in mineral soils^{72, 73, 74}. Therefore, though EPS is likely relatively easy to degrade and most of it will show rapid turnover, a small fraction of it might “accumulate” in soils and contribute to the long-term C preservation.

[Comment 21] Line 107-108: What is the relation of the reference to the sentence? – it's starting to get exhausting – the authors studied one bedrock – what do the authors

want to say – and is it really the best reference?

[Response 21] Thank you for your helpful comment. In response, we have revised the sentence from “Soils derived from different bedrock types offer varying levels of protection for EPS ⁷⁵” to “**Soils formed from different bedrocks thus may vary in their capacity to bind with EPS⁵**” (**Page 6, Lines 119-120**) and have replaced the original citation with a more relevant one. This revision is also aligned with the clarification made in our response to **R4-22** below. Additionally, we have thoroughly reviewed the entire manuscript to ensure that all citations are accurate and appropriately support the corresponding statements.

***[Comment 22]** 105-111: It would be preferred if the authors stated hypotheses which they tested in their study instead of saying that there are several out there. What specific hypotheses and procedures (compared to the other studies) guided their science? – What is new?*

[Response 22] Thank you for your helpful comment. In this section, we originally aimed to highlight the research gap regarding the environmental controls of soil EPS. However, we agree that our specific hypotheses were not clearly articulated.

To clarify, our study was guided by the hypothesis that soil EPS levels are governed by two key processes:

(1) microbial EPS secretion, which is influenced by land-use-induced changes in nutrient availability and microbial activity; and

(2) the short-term retention of EPS, which depends on its interactions with soil particles and minerals, governed by texture and mineralogy inherited from the bedrock.

This dual-process framework distinguishes our approach from previous studies that typically focused on a single factor (e.g., land use) or limited spatial scales. To better reflect this, we have revised the paragraph as follows:

“**Various hypotheses have been proposed to explain the environmental controls on microbial EPS secretion and soil EPS content, but most have been tested based on laboratory or small-scale field studies. For example, Redmile-Gordon et al.²⁸ showed that EPS acts as a transient binding agent, with protein levels higher in grasslands than in arable or fallow soils, likely reflecting land-use-driven shifts in microbial EPS secretion. However, forest soils—characterized by distinct microbial communities—were not included. Beyond microbial production, soil EPS content also depends on its interaction with soil minerals⁷¹, shaped by texture and mineralogy derived from bedrock. Soils formed from different bedrocks thus may vary in their capacity to bind with EPS⁵, though recently land use was shown to be the more important effector of soil EPS content than bedrock geochemistry in a tropical wet climate⁹. Despite increasing interest, no field study has systematically assessed the combined effects of land use and bedrock across varying climates on soil EPS. Large-scale investigations**

spanning forests, grasslands, and croplands on contrasting bedrock types across a large climate gradient are therefore critical to disentangle the biotic and abiotic controls on soil EPS-C pools” (Pages 5-6, Lines 113-125).

This revision introduces our research aim and conceptual framework more clearly and defines the novelty of our approach in comparison with previous work. We have revised the research questions as requested. Please refer to **R4-24** for the detailed changes regarding the research questions.

[Comment 23] Line 113: Does that mean that the samples were stored from this study frozen?

[Response 23] Thank you for raising this point. The samples used in this study were not frozen. Instead, air-dried soils were used for the EPS extractions. We have now clarified this point more explicitly in the **Methods Section** by stating that “**Soil EPS (including polysaccharides and proteins) was extracted from air-dried soil samples in 2024 using cation exchange resin (CER)³⁶**” (Page 16, Lines 490-491).

[Comment 24] Line 115-116: “What is the distribution of EPS on global scale” – this question is not well prepared in my point of view. I would assume that the EPS concentrations and compositions change with the microbes present in soil and their abundance – probably major differences on the small scale – so what would be the gain of information from a global distribution?

[Response 24] We agree that EPS content and composition closely reflect microbial community structure and abundance, which vary greatly at small spatial scales. Controls of biogeochemical properties and processes are known to change with spatial scale, from the small micro-scale to the large continental scale. For example controls can shift from micro-scale (substrate availability at the pore scale, moisture and O₂ gradients across microsites, and local microbial interactions, viruses, protists) to the regional and continental scale (climate and seasonality driving temperature and moisture constraints, plant input quantity and quality changing across vegetation types, large-scale weathering and atmospheric deposition effects on nutrient availability, and evolutionary and biogeographic effects on dominant microbial taxa). Our aim in raising the question about “global” EPS distribution is to highlight the current knowledge gap on large-scale patterns and environmental drivers of soil EPS across diverse land use systems and geologies, including a wide climatic gradient. Macroclimatic effects on soil EPS have yet to be determined, and this was at the heart of our study. Given EPS’s crucial role in soil health and structure, understanding broad-scale patterns is essential to identify generalizable controls and inform ecosystem management beyond site-specific studies. In response, we have revised the **Research Question 1** from “(1) What is the distribution of EPS concentration in soils on the continental scale, and what are

its physicochemical and biological drivers?” to “**What is the distribution of EPS content in soils along a European climate transect, and how do land use and bedrock types influence this distribution?**” to explicitly focus on the combined effects of climate, land-use and bedrock on soil EPS (**Page 6, Lines 129-131**).

[Comment 25] Line 118-119: Again I don't get the specification on the continental scale – needs clarification!

[Response 25] In response, we have revised this sentence from “What is the contribution of EPS-C to the SOC pool, and what causes potential differences at the continental scale?” to “**What is the contribution of EPS-C to the SOC pool, and what causes potential differences along a European climate transect?**” to more accurately reflect the spatial scope of our study (**Page 6, Lines 131-132**).

[Comment 26] Line 136-137: What was analyzed, why and what was the result? Instead of referring that more detailed analysis can be found in the supplement....

[Response 26] The main findings of this section are already presented in **Fig. 2** and described in detail in **Result Section 2.1** of the main text. In addition, we conducted supplementary linear model and linear mixed-effects model analyses to further assess the effects of bedrock type, land-use type, environmental and microbial factors on EPS-related variables.

To improve clarity and better integrate the supplementary information with the relevant main text sections, we have split the original sentence “More detailed analyses of bedrock, land-use types, and environmental factors influencing EPS variables can be found in Supplementary Tables S3–S14” into three separate and context-specific sentences (see also our response **R3-23**):

- (1) “**Supplementary Tables 6-8 provide additional results from linear models and linear mixed-effects models assessing the influence of bedrock, land-use types, environmental, and microbial factors on total EPS**” (**Page 7, Lines 174-176**).
- (2) “**Supplementary Tables 9-14 provide additional results from linear models and linear mixed-effects models assessing the influence of bedrock, land-use types, environmental, and microbial factors on EPS polysaccharides and EPS proteins**” (**Page 8, Lines 193-196**).
- (3) “**Supplementary Tables 15-17 provide additional results from linear models and linear mixed-effects models assessing the influence of bedrock, land-use types, environmental, and microbial factors on EPS-C/MBC ratio**” (**Pages 8-9, Lines 224-226**).

In addition, we have provided a “**Description of Additional Supplementary Files**” document, which lists the names and main contents of all supplementary tables and datasets, to improve clarity and accessibility.

[Comment 27] Line 139: What is the soil total EPS? Why did the authors include it? is it sound to just sum up sugars and proteins? Could they not overlap? Please elaborate on that!

[Response 27] In this study, we used the sum of EPS polysaccharides and EPS proteins to approximate total EPS. This approach is based on findings from previous research^{76, 77}, which consistently demonstrate that polysaccharides and proteins are the two major components of microbial EPS, while other components, such as lipids and extracellular DNA, typically contribute only a minor fraction (usually less than 10%)^{76, 77}. This is also supported by Steward et al.⁷⁸ who analysed organic carbon (OC) compounds from different EPS extracts using liquid chromatography-organic carbon detection–organic nitrogen detection (LC-OCD-OND), and total protein and polysaccharide content and concluded that “protein and polysaccharides represented the two major components of EPS and, when combined, accounted for the measured DOC in extracts”. Therefore, many EPS studies including soil studies have adopted the approach of summing CER-extracted polysaccharide and protein contents to represent total EPS, such as Li et al.⁷⁹, Xu et al.⁸⁰, Xu et al.⁸¹, and Geyik and Çeçen⁸².

As the reviewer rightly pointed out, some compositional overlap may exist between polysaccharides and proteins in EPS, such as in the form of glycoproteins⁷⁶, which could lead to a slight overestimation of total EPS when simply summing these two components. However, given that polysaccharides and proteins are widely recognized as the dominant constituents of microbial EPS (~90%) and that other components (e.g., DNA, lipids) typically represent a minor fraction (~10%), this summation remains a broadly accepted and practical proxy for total EPS^{76, 77}. The potential overestimation due to overlap is likely small and acceptable for the purpose of comparative analysis across different environments. In a thought experiment, the overestimate would depend on the contribution of proteins to total EPS, and the percentage of glycoproteins in EPS proteins. With 10% glycoproteins in the protein fraction and 50% of total EPS being proteins, we would arrive at total EPS = (45% EPS proteins + 5% EPS glycoproteins) + (50% EPS polysaccharides + 5% EPS glycoproteins) = 105%. This certainly is a gross oversimplification, also assuming for glycoproteins 1 part glycan and 1 part peptide (glycan contributes 50% of mass to glycoprotein). A quick literature survey indicates that (i) in bacterial proteomes glycoproteins are rare (<1%), but more common in the exoproteome (several %), and the glycan portion contributes ~5–15% of glycoprotein mass; (ii) fungi show greater abundance of glycoproteins (10–20% overall proteome; >50% of exoproteome), the glycan portion contributing ~10–30% of mass. The bias, therefore, strongly relates to the share of bacterial versus fungal contributions to EPS proteins in soils.

Following the reviewer's suggestion, we have added a more detailed explanation of this to the **Methods Section**: "Total EPS was approximated as the sum of EPS polysaccharides and EPS proteins, following previous studies^{78,79}. This approximation is based on the fact that EPS polysaccharides and EPS proteins are the major components of microbial EPS, together contributing approximately 90% of total EPS^{76,77}. This approach has been widely employed for estimating total EPS levels. However, we acknowledge that this method may slightly overestimate total EPS due to potential overlaps between components—for example, glycoproteins which would be double-counted in the protein and the polysaccharide fraction⁷⁶. Despite this, the potential bias is likely small and considered acceptable for comparative purposes across sites. The core data used in this study have been provided as Supplementary Dataset 1" (**Page 17, Lines 532-540**).

[Comment 28] Line 145: turnover time!?! Turnover time of what!?! Same with the figures – clear that up!

[Response 28] Thank you for pointing out this ambiguity. We agree that the original phrasing of "turnover time" lacked clarity. To address this, we have revised the term throughout the manuscript to explicitly state "microbial biomass turnover time (MBT)". In addition, we have updated **Figs. 2 and 7** to use the abbreviation "MBT" instead of the generic "turnover time" and we now clearly define MBT as microbial biomass turnover time in the respective figure captions to ensure clarity for readers. MBT is not causally linked to EPS formation, decomposition, or stabilisation. Given that MBT is calculated as the inverse of microbial growth per microbial biomass (specific growth, q_{growth}), we in most instances refrain from using MBT and use q_{growth} instead, which is a physiological proxy for how fast a microbial community grows and divides, and therefore to investigate potential C allocation trade-offs between EPS formation and growth, or whether in circumstances of growth limitation (stress-induced reduction in q_{growth}) EPS production may increase as a stress response (see also **R1-41** and **R4-3**).

[Comment 29] Line 139-147: Instead of writing that you related everything to everything – it would be good if the authors stick to what they expected and what they found. A sentence over 9 line without any information – move to methods. By the way, Ca and Mg are nutrients and not really soil minerals!

[Response 29] In response to your comment, we have removed the detailed list of environmental and microbial variables from the **Results Section**. In addition, we have corrected the terminology: exchangeable Ca^{2+} (Ca_e) and Mg^{2+} (Mg_e) are now appropriately referred to as soil nutrients, not soil minerals.

[Comment 30] Line 196-198: Might that be due to the EPS extraction method. There also different extraction method?

[Response 30] Thank you for your valuable comment. Indeed, different EPS extraction methods may yield varying results, and methodological choices can influence the observed patterns. For a review table of EPS extraction methods, see Huang et al. ¹¹, also highlighting the mechanism of EPS extraction, important features of methods, and disadvantages, including appropriate references. Given the large heterogeneity among soils derived from different bedrock and land-use types in our study, we conducted a preliminary experiment to ensure that the chosen method – the cation exchange resin (CER) extraction approach - was effective across all soil types encountered in our study, which included carbonate, sediment, and silicate soils (Supplementary Fig. 4). The CER method has been established as the main current EPS extraction method, though it cannot quantitatively recover all soil EPS ^{36, 38, 47}, and therefore underestimates soil EPS (see also responses **R1-22**).

The description of the preliminary experiment is provided in **Method Section**: “A preliminary experiment was conducted to optimize EPS extraction using different CER:soil ratios across soils varying in bedrock type and SOC content, showing that 1 g of air-dried soil with 10 g of CER provided a good balance for determining both EPS polysaccharides and EPS proteins (Supplementary Method 1; Supplementary Fig. 4)” (Page 16, Lines 496-499):

Supplementary Method 1 - Optimizing EPS extraction across bedrock and SOC gradients

Before EPS quantification, a preliminary experiment was conducted to test EPS extraction efficiency with different CER:soil ratios across bedrock types, considering the expected higher exchangeable Ca^{2+} levels in carbonate soils and the wide range of SOC concentration variations due to the extensive sampling range. Specifically, three carbonate soil samples and three non-carbonate soil samples with high, medium, and low SOC contents were selected. For each sample, 0.5 g, 1 g, 2 g, and 3 g of air-dried soil were weighed, and then 10 g of CER was added for extraction, to demonstrate that the CER method and dosage used are suitable for different types of soil samples with varying SOC contents. This experiment showed that using 1 g of air-dried soil with 10 g of CER provided a good balance for determining soil EPS, as 3 g was more effective for polysaccharides and 0.5 g or 1 g for proteins, regardless of bedrock type or SOC level (Supplementary Fig. 4).

This preliminary test helped us ensure that the CER extraction protocol used in this study was broadly applicable across diverse soil types, thereby minimizing potential methodological biases and ensuring comparability across samples. This was particularly important in this large-scale study: exchangeable Ca^{2+} differs by magnitudes between limestone soils and acidic silicate soils, causes EPS binding

through multivalent cation bridging, and this exchangeable Ca^{2+} needs to be quantitatively extracted by CER to release EPS bound through this Ca^{2+} bridging mechanism (see **R1-22**).

We also appreciate the reviewer's valuable suggestion concerning the potential influence of extraction methods, particularly with respect to the observed significant responses of EPS polysaccharides to bedrock type, contrasted with their non-significant responses to land-use type. While some variability in EPS polysaccharide extraction efficiency cannot be fully excluded, the same method (CER) was uniformly applied and optimized across all samples. Notably, EPS proteins did not show similar patterns of variation with bedrock type, suggesting that the contrasting responses of EPS polysaccharides and EPS proteins to large-scale variation in environmental drivers are unlikely to be solely due to methodological bias, but rather reflect underlying environmental or microbial controls.

In response, to better focus on the main objective of our study—soil EPS dynamics and their driving factors—we have moved the discussion of the differential responses of EPS polysaccharides and proteins to bedrock and land-use types to **Supplementary Discussion 5**. Within this section, we have included the following sentence to acknowledge methodological considerations: “**Although a preliminary experiment was conducted to optimize the extraction procedure across soil types, variability in extraction efficiency cannot be fully excluded. However, since only EPS polysaccharides, but not EPS proteins, showed significant variation with bedrock type, this indicates that the contrasting patterns are unlikely to result solely from methodological bias, but rather reflect differences in environmental or microbial regulation of EPS components**”. A brief statement “**Moreover, distinct environmental dependencies and biosynthetic pathways of EPS polysaccharides and EPS proteins caused their contents to be differentially by bedrock and land use, respectively (Supplementary Discussion 5)**” (**Page 11, Lines 311-314**) has been retained in the main text for clarity.

[Comment 31] Line 211: The authors should be bit careful in their statements. For me it is not becoming clear how the authors come from a correlation to the guess that “indicating that EPS promotes soil water holding capacity more than bulk SOC or texture” – please revise or explain

Wild guesses and interpretations belong in the discussion – move it.

[Response 31] Thank you for highlighting the inaccurate expression in our original manuscript. In response, we have revised this sentence from “At the same time, both EPS compounds were strongly positively related to soil water holding capacity (particularly EPS polysaccharides), indicating that EPS promotes soil water holding capacity more than bulk SOC or texture do (Fig. 4)” to “**Both EPS compounds were**

strongly related to soil WHC (Fig. 4a)” to improve the scientific accuracy and better reflect our observed data (Page 8, Lines 188-189).

[Comment 32] Line 227-228: Unclear how this numbers have been generated!

[Response 32] In our study, EPS-C values were estimated by converting the contents of EPS polysaccharides and EPS proteins into EPS-C equivalents using conversion factors. Specifically, EPS-C was calculated as the sum of the C content from both EPS polysaccharides and EPS proteins. The C content of EPS polysaccharides was estimated at 39.1%, based on the average C content determined by elemental analysis of several representative exopolysaccharide standards (hyaluronic acid: 37.5%, xylan: 40.5%, pectin: 38.3%, and arabinogalactan: 40.2%), while the C content of EPS proteins was estimated to be 50.7% based on literature values of global protein N percentages and predicted average amino acid C:N ratios across microbial proteins⁵¹. A more detailed explanation has now been added to **Method Section (Pages 17-18, Lines 541-571)** (please see response **R4-4**).

[Comment 33] Line 235-237: Could it be that the authors extracted necromass, due to the soil storage and EPS extraction from dry soil?

[Response 33] Thank you for raising this important point. As only air-dried soil samples were available for this study, EPS was extracted from air-dried soils. We acknowledge that this may increase the risk of co-extracting non-EPS metabolites, such as microbial necromass.

To minimize such potential interference, we used the cation exchange resin (CER) method, which has been widely validated for its low cell lysis and high selectivity in extracting microbial EPS^{36, 38, 83}. Furthermore, preliminary tests across soils with different bedrock types (limestone soils and non-limestone soils) and SOC levels confirmed that CER-extracted EPS reflected consistent and ecologically meaningful patterns across environmental gradients, supporting the comparability of results (Supplementary Fig. 4). Microbial necromass did not significantly co-extract with EPS in two recent studies by Rebeca Oliva Leme^{1, 84}. They showed a small “contamination” of CER-extracted EPS from bacterial cultures with muramic acid, demonstrating that partially decomposed peptidoglycan from bacterial cell walls (cellular necromass) does not significantly contaminate the EPS fraction (~10% of total muramic acid in bacterial culture extracted in EPS fraction; Oliva et al. 2024). However, specific amino sugars such as galactosamine and mannosamine (and to a smaller extent glucosamine) were shown to be integral parts of EPS, yet neither being confined to microbial EPS or necromass.

We have now acknowledged this limitation in the revised **Discussion section** by revising the sentence from “This difference may be due to variations in geographical

area, soil management practices, microbial communities, climate, and soil organic matter content” to “**This difference partly reflects differences in bedrock and land-use composition, sampling season, and methodological factors (Supplementary Discussion 1). Due to sample availability, we used air-dried soils, whereas most previous studies used fresh soils. Our preliminary experiment showed minor effects of air-drying: EPS polysaccharides decreased by $1 \pm 3\%$, EPS proteins increased by $9 \pm 1\%$, and total EPS (polysaccharides + proteins) increased by $4 \pm 3\%$. Long-term soil storage had similarly no impact on EPS content (Supplementary Discussion 2)**” (Page 9, Lines 253-258). See also responses **R3-11**, **4-3**, and **4-78** on the topic of procedural bias by air-drying. A more detailed discussion—including comparisons of dry and wet soils from the literature, our dry–wet soil experimental results, measurements from long-term stored samples, and potential necromass biases—has been added to **Supplementary Discussion 2** (see response **R4-3**).

In an ongoing work, we also hydrolysed the extracted EPS fractions from the same soil samples and analysed their neutral sugar, amino sugar and uronic acid composition (unpublished data). We could not detect muramic acid in any EPS extract across the whole transect.

[Comment 34] Line 237-239: Interpretation into the discussion

[Response 34] In response to your comment, we have revised this sentence from “In addition, notably strong positive correlations were observed among SOC, MBC, MNC, and EPS-C, underscoring their interconnected roles within soil C dynamics (Figs. 5, 6)” to “**In addition, notably strong positive correlations were observed among SOC, MBC, MNC, and EPS-C (Fig. 6)**” (Page 8, Lines 208-209).

To provide a deeper interpretation, we have retained the sentence “In addition, EPS is produced by microbes as a “glue” for attachment of their cells and colonies to soil surfaces⁷⁶, which, after death, cause necromass fragments to be cemented via EPS to soil minerals or organic surfaces, which has been shown microscopically^{85, 86}. This also promotes the strong collinearity between EPS-C and MNC, MBC and SOC (Fig. 6)” in the **Discussion Section (Page 12, Lines 349-353)**.

[Comment 35] Line 267: How did the authors investigate EPS stabilization!?

[Response 35] We acknowledge that the term “EPS stabilization” was not appropriately used in this context, as our study did not directly measure or investigate EPS stabilization. In response, we have revised the sentence from “To assess direct versus indirect effect pathways between environmental drivers and soil EPS-C content, and to evaluate the importance of these effects via driving EPS production efficiency or EPS stabilization, we performed SEM analysis (Fig. 8)” to “**To assess direct versus indirect effect pathways between environmental and microbial drivers and both soil EPS-C**

content and EPS-C/MBC ratio, we performed SEM analysis” to more accurately reflect the results (Page 9, Lines 227-228).

[Comment 36] Line 268-270: Sorry, I don't get it. Ca was positively correlated with EPS and now it has a negative effect on EPS-C!? Clarify! Heatmap of EPS driving factors – for both EPS polysaccharides and proteins compared to SEM? – to me it makes no sense

The authors base all of the presented data on EPS concentrations determined at one point in time. How was an EPS-C accumulation or stabilization determined? As it is not clearly

[Response 36] Thank you for pointing out this issue, and we apologize for the lack of clarity in our original wording. In the SEM, we used PCA-derived composite factors to represent complex soil physicochemical variables, and Ca^{2+} (Ca_e) was one of the contributing variables to this composite factor (see also response **R3-4** on PCA data handling). As shown in the loading plot below the SEM diagram (Original Fig. 8b), Ca^{2+} had a negative loading on principal component (PC1) and a near-zero positive loading on PC2. Meanwhile, this composite factor had a negative path coefficient to EPS-C (Fig. 8a). Therefore, the indirect effect of Ca^{2+} on EPS-C is double-negative, resulting in a net positive association, which aligns well with the direct positive correlation observed in the heatmap (Fig. 2c). Because the SEM originally included both PC1 and PC2, it was somewhat difficult to interpret. In this revision, we optimized the SEM to use only PC1. Based on the updated results, we have revised the corresponding sentence from “Soil physicochemical variables (a composite factor negatively loaded by clay, Ca_e , and Fe_d , and SOC, Supplementary Fig. S1) had a direct negative effect on EPS-C. This aligns with expectations, as higher clay and Ca_e contribute to EPS-C accumulation and stabilization at lower PC (soil) values” to “**Soil physicochemical variables were summarized as a composite factor derived from PC1 of a PCA (Fig. 8b; Supplementary Fig. 1). Specifically, PC1 (soil) was positively loaded by the soil C:N ratio and near-zero positively loaded by SOC, while clay content and Ca_e , had negative loadings. In the SEM, this composite soil factor showed a negative path coefficient to EPS-C, indicating that lower scores on this PCA-derived soil factor (i.e., higher clay, and Ca_e ; lower soil C:N ratio) were associated with higher EPS-C content (Fig. 8a)**” (Page 9, Lines 228-234).

To better explain the effects of the PCA-derived microbial composite factor on EPS-C and EPS-C/MBC, we revised the original sentence from “In contrast, the microbial composite factor (negatively loaded by CUE, C_{growth} , qGrowth , and MBN, Supplementary Fig. S1) exerted a direct positive effect on EPS production efficiency and a direct negative effect on EPS-C (Fig. 8). This is consistent with the observed trade-off between microbial growth and EPS production efficiency, as low microbial”

biomass and growth (at high PC (microbe) values) are associated with high EPS-C production efficiency” to “**In contrast, the microbial composite factor, derived from PCA of CUE, C_{growth}, q_{Growth}, and MBN (Fig. 8c; Supplementary Fig. 1), exerted a direct positive effect on EPS-C/MBC ratio and a negative effect on EPS-C (Fig. 8a). In the PCA loading, C_{growth}, q_{Growth}, and CUE had negative loadings on PC1 (microbial), while MBN had a near-zero positive loading. Thus, the SEM paths indicate that lower values of this PCA-derived microbial factor (i.e., higher CUE, C_{growth}, and q_{Growth}) were associated with lower EPS-C/MBC ratio, supporting a potential trade-off between microbial growth and EPS investment**” (Page 9, Lines 234-240).

Additionally, concerning the use of “EPS-C accumulation” or “stabilization”: we agree that our data are based on a single time point and do not allow direct inference of long-term EPS accumulation or stabilization. We have now removed the term “stabilization” and “accumulation” in the main text.

[Comment 37] Line 270: Which kind of expectations? And what do the authors mean by lower PC (soil) values?

[Response 37] Our expectation was that high contents of Ca_e and clay increase the organic matter and EPS binding capacity of soils and therefore show a positive relationship with soil EPS. In response to your comment, we have revised the sentence for clarity (also in line with response **R4-36**)

[Comment 38] Line 276-277: refer to the data

[Response 38] Thank you for your careful comment on this point. Together with the related suggestion from Reviewer 1 (Response **R1-18**), we realized that this statement was somewhat circular and self-evident. Therefore, to avoid redundancy and improve clarity, we have removed the sentence “Moreover, EPS-C scaled positively with EPS production efficiency, showing that EPS production efficiency promotes EPS accumulation” from the revised manuscript.

[Comment 39] Discussion: It would be overall better if the authors stick to their data and what they can prove or reject instead of wild interpretations and assumptions!

[Response 39] We fully agree that scientific discussion should be grounded in data. In response (see also **R4-2**, **R4-5**, and **R4-22** on speculation, data grounding, hypotheses, and testing), we have carefully re-evaluated the entire discussion section and removed or revised interpretations that were not directly supported by our dataset. Overall, the revised discussion now more strictly adheres to what our data demonstrate or statistically support, avoiding broad assumptions or unproven causal explanations. We appreciate your guidance in improving the rigor and clarity of our interpretations. For

the specific revisions made to the discussion section, please see our detailed responses below.

[Comment 40] Line 309: The author don't do a spatial analysis, so please be precise and write that the study has been conducted along a transect (not even covering Europe, but just some parts see Map of Noll et al). Clarify!

[Response 40] Thank you for this important comment. We have revised the sentence from “This study found a high spatial variability in soil EPS concentrations across the European transect (minimum 149 $\mu\text{g g}^{-1}$ / maximum 2495 $\mu\text{g g}^{-1}$ (16 x the minimum value); Fig. 1)” to “**Across the European transect, soil EPS contents varied widely (149 – 2495 $\mu\text{g g}^{-1}$; 16 \times difference; Fig. 2)**” to improve accuracy and clarity (**Page 9, Lines 246-247**).

[Comment 41] Line 315-316: See my comment above – the authors overlooked quite some studies....

[Response 41] Thank you for your important comment. We have now re-examined the literature on soil EPS using the CER extraction method and updated our database accordingly (Supplementary Table 1 and Supplementary dataset 2). In response, we have revised the sentence from “There are only a few reports of soil EPS contents based on the CER extraction method available in the literature, which we compiled in a data synthesis effort (Supplementary Table S1). Total EPS, EPS polysaccharides, and EPS proteins levels in our study were higher than those previously reported at ~10 sites differing in climate and geology: total EPS (mean: 393 $\mu\text{g g}^{-1}$), EPS polysaccharides (mean: 266 $\mu\text{g g}^{-1}$), and EPS proteins (mean: 126 $\mu\text{g g}^{-1}$)^{2, 4, 9, 28, 36, 38}” to “**An extensive literature survey showed lower EPS contents (~30 sites in 18 studies differing in climate, geology, and land use), with 429 $\mu\text{g g}^{-1}$ for total EPS, 335 $\mu\text{g g}^{-1}$ for EPS polysaccharides, and 131 $\mu\text{g g}^{-1}$ for EPS proteins (Supplementary Table 1). This difference partly reflects differences in bedrock and land-use composition, sampling season, and methodological factors (Supplementary Discussion 1)**” (**Page 9, Lines 250-254**) and replotted **Fig. 1**. Further discussion has been moved to **Supplementary Discussion 1**. Please see below:

Supplementary Discussion 1 - Comparison with published EPS datasets

To compare our EPS data with previously published values, we compiled 216 EPS data points from 22 studies using the CER extraction method, with all data available in Supplementary Dataset 2. After screening for surface bulk soils and excluding data with added substrates, 83 data points from 18 studies were retained. Across ~30 sites differing in climate, geology, and land use, previously reported mean values were 429 $\mu\text{g g}^{-1}$ for total EPS (sum of EPS polysaccharides and EPS proteins), 335 $\mu\text{g g}^{-1}$ for EPS

polysaccharides, and $131 \mu\text{g g}^{-1}$ for EPS proteins (Supplementary Table 1), which were generally lower than the levels observed in our study^{2, 4, 9, 27, 28, 36, 37, 38, 39, 40, 41, 42, 43, 44, 45, 46, 47, 48, 49, 50}.

The higher EPS contents in our dataset likely reflect a combination of ecological and methodological factors. Ecologically, our study included a higher proportion of grassland sites (27/92 vs. 7/46 in previous studies) and calcareous bedrock (27/92 vs. 2/46), both associated with higher EPS contents. In addition, ~67% of previously published data were sampled in January–April and October, whereas our samples were collected from May to August, when microbial activity is elevated, potentially enhancing EPS production. Methodologically, soil storage conditions may influence the amount of extractable EPS. Together, these factors likely account for the elevated mean and median EPS levels observed in our study compared to the literature. Please see also response **R4-15** on the literature data study.

[Comment 42] Line 319-320: (a) Maybe due to differences in EPS extraction from dry/wet soil and storage and EPS polysaccharide determinations!? Please include (b) If referring to explanations of different studies (I assume that previous studies can be checked in more detail in regard to your studied soils (ADI, carbonates and SOC – for example to check if your message can be underpinned by the other studies or not, instead of referring to a bunch of potential differences without a closer look, which is not advancing our knowledge.

[Response 42] Thank you very much for your valuable suggestion.

(a) We reviewed the literature and found that Zhang et al.³⁷ conducted a comparison of EPS extractions from air-dried soils after rewetting and 2-week pre-incubation and fresh soils. Their results showed -13% to 26% difference between the two treatments, suggesting that drying may have limited effects on EPS quantification. The same, i.e., a negligible effect of air-drying and of long-term storage was confirmed by our own supplementary experiment. We collected seven soil samples from forests, croplands, and grasslands. After collection, soils were immediately sieved through a 2 mm mesh and homogenized. Each sample was split into two subsamples: one analyzed immediately (fresh), and the other air-dried prior to analysis. For EPS polysaccharides, extraction from air-dried soils showed a mean difference of $-1\% \pm 3\%$ (mean \pm SE, $n = 7$; range: -10% to +11%) relative to fresh soils. In contrast, EPS protein extraction from air-dried soils showed a mean difference of $+9\% \pm 1\%$ (mean \pm SE, $n = 5$; range: +7% to +14%) relative to fresh soils across cropland, grassland, and forest soils (Supplementary Table 3).

In addition, to assess the effect of long-term storage, we re-analyzed EPS in two air-dried forest soils collected in China. The first was sampled in 2021 in Dongtai City and

reported previously⁴, and the second was collected in autumn 2021 from Nanjing for an unpublished earthworm microcosm experiment. After four years of air-dry storage at room temperature (2021–2025), both soils showed only minor changes, with EPS polysaccharides and EPS proteins increasing by 1–10% relative to fresh soils (Supplementary Table 4). These results indicate that long-term storage had negligible effects on EPS measurements. See also responses **R3-11**, **R4-3** and **R4-78** for more details on air-drying and soil storage effects on EPS.

(b) Based on these new experimental results and the literature evidence, we also revisited and compared our EPS concentrations with those reported in other studies. We have revised the **Discussion** accordingly; for detailed modifications, please see our response **R4-41**, and for further discussion, refer to **Supplementary Discussion 1**.

We also compared our results to land use effects on EPS found by Redmile-Gordon et al.²⁸ and Kidinda et al.⁹. Since the effect of land use on soil EPS was not significant in our study, we have moved this discussion to **Supplementary Discussion 4**, where we also included comparisons with their findings, please see below:

Supplementary Discussion 4 - Land use effects on soil EPS contents and EPS-C/MBC ratios

Among the three land-use types, EPS contents were highest in grassland soils compared to cropland and woodland soils, although the differences did not reach statistical significance (Fig. 2b). To identify potential drivers of EPS content variation, we found that MBC was significantly affected by land use and showed a pattern consistent with EPS (Supplementary Table 5). This, together with the positive correlation between MBC and EPS (Fig. 2c), suggests that higher MBC may indicate larger microbial populations, thereby representing a greater potential for EPS production and serving as a primary control on EPS content across land-use types. In addition, land-use exerted a stronger influence on soil nutrient status (e.g., SOC, TOP, TP, Mg_e, and NO₃⁻) and fine root traits (FRB and Root N) than bedrock (Supplementary Table 5). **Among these, SOC and TOP were significantly higher in grassland than in cropland and positively correlated with EPS, while root N was also highest in grasslands, whereas FRB peaked in woodlands but remained relatively high in grasslands. These indicators suggest that improved nutrient availability and greater plant inputs (root turnover and exudation) may enhance microbial EPS production⁸⁷. This interpretation is consistent with previous findings: Redmile-Gordon et al.²⁸ observed the highest EPS contents in grasslands, while Kidinda et al.⁹ reported higher EPS in fertilized croplands than in forests on nutrient-poor tropical soils. Together, these studies suggest that land-use effects on EPS are primarily mediated by plant inputs and nutrient availability.** Conversely, the lowest MBC and SOC observed in cropland soils may reflect soil degradation or organic matter depletion caused by long-term intensive management practices such as tillage, which could in turn contribute to the

reduced EPS levels. For instance, frequent tillage can disrupt soil structure and aggregates, thereby reducing soil organic matter content¹⁰. On the other hand, although croplands had the highest clay and exchangeable Ca²⁺ contents—factors potentially favouring short-term EPS retention—their EPS levels were lowest. This suggests that limited labile C availability, linked to lower SOC and FRB in croplands, may play a more decisive role in constraining microbial EPS content across the investigated soils. Substantial differences in the composition and function of soil microbial communities under different land use types⁸⁸ may also cause differences in EPS formation and decomposition⁸⁹. While microbial diversity was not investigated in this study, in accordance with investigations of legacy vs current management effects on EPS²⁸, it is primarily the interplay between plant inputs and anthropogenic management that causes these differences in total EPS and EPS protein contents. Interestingly, the ranking of EPS-C/MBC did not mirror that of EPS contents, with croplands showing the highest EPS-C/MBC, followed by grasslands and woodlands (Fig. 7b). Given that both EPS and MBC were lowest in cropland soils, this highest EPS-C/MBC ratio suggests a greater microbial investment in EPS production per unit biomass. This may be a response to increased water stress in croplands, as indicated by significantly lower WHC and SMC compared to woodlands and grasslands (Supplementary Table 5).

We hope these revisions provide a clearer and more informative interpretation of the variability in EPS values across studies.

[Comment 43] Line 322-323: Ok, and now go a step further – what does that mean for your interpretations!? Formatting of the reference!

[Response 43] Thank you for your helpful comment. This sentence — “Generally, soil EPS concentrations are governed by the dynamic balance between its generation, stabilization and decomposition (Shi et al., 2024)” — was originally included to support our interpretation of the patterns observed in our results: specifically, that EPS contents were highest in carbonate soils, whereas EPS-C/MBC was highest in sedimentary soils. We initially interpreted this as reflecting greater EPS stabilization in carbonate soils, and potentially enhanced microbial allocation (EPS production efficiency) to EPS under nutrient-rich conditions in sedimentary soils.

However, based on your comment and also insights from another reviewer (see **R1-1**, **R1-11**, and **R3-2**), we now recognize that this interpretation may not be appropriate. Extracted EPS may not directly reflect EPS stabilization, and EPS-C/MBC in our study is not a reliable proxy for EPS production efficiency. Therefore, we have removed the sentence and revised the corresponding interpretation to better reflect the limitations of our data.

[Comment 44] Line 326-327: The authors should elaborate a bit more on their assumption.

[Response 44] Thank you for your helpful comment. We have revised this sentence “This suggested that high EPS concentrations in carbonate soils may depend on their stabilization processes¹⁴, whereas the EPS in sedimentary rocks (despite their high EPS production efficiencies) may be rapidly decomposed due to the presence of adapted microbial communities efficiently utilizing EPS¹⁵” and rewritten the **Discussion Section** to clarify our reasoning (please see response **R4-45** below).

[Comment 45] Line 337-339: But an overall higher microbial biomass would also mean more potential EPS producers, or? The more MBC the more EPS-C see your own figure – clarify

[Response 45] Thank you very much for your valuable comment. Your suggestion, along with your comment 51, inspired us to conduct a more comprehensive analysis. We performed two-way ANOVA and variance partitioning to compare the effects of bedrock and land use on environmental and microbial factors, and related these results to the correlations between EPS and environmental and microbial parameters (Supplementary Table 5). This allowed us to reassess the differences in EPS content among the three bedrock types. See also response **R1-22** and **R4-22** on refined statistics and on bedrock effects.

Indeed, we found that bedrock type significantly influenced MBC and MBN, which showed a positive correlation with EPS content, more so than land use affected microbial biomass. Therefore, we decided to delete the original sentence “However, greater microbial activities may result in the recycling and degradation of EPS¹⁵, thus, eventually limiting its accumulation²⁴”.

Based on our revised interpretation, the high EPS content in carbonate soils is mainly attributed to two factors:

(1) Higher microbial biomass (such as MBC), which likely reflects a greater potential for EPS production.

(2) Higher soil cation exchange capacity (CEC) and elevated levels of clay content, and Ca²⁺ in carbonate soils, which may promote (a) short-term EPS retention and (b) improve soil conditions, thereby supporting increased microbial biomass and EPS production (feedback on (1) increasing MBC).

We have updated the **Discussion Section** accordingly, please see below and response **R1-22**:

“Among the three bedrock types, EPS contents were significantly higher in carbonate than in silicate and sedimentary soils. This pattern was mirrored in microbial (e.g., MBC and MBN) and soil properties (e.g., CEC, clay, and Ca_e), known to promote EPS, all of which were significantly influenced by bedrock and peaked in carbonate

soils. Conversely, sand content – which was lowest in carbonate soils – was negatively correlated with EPS contents (Supplementary Table 5). Based on these findings, we propose that the highest EPS content in carbonate soils is primarily driven by two aspects: (1) an indirect pathway, where highest soil pH and CEC (particularly Ca_e) modulates the soil microenvironment in ways that stimulate microbial activity²⁷, increase microbial biomass (e.g., MBC and MBN) and thereby causes greater overall capacity for EPS production⁹; and (2) soil minerals (clay) and cations (Ca_e) may contribute to the selective retention of EPS within the soil matrix through multivalent cation bridging between negatively charged sites of EPS and other polymers and minerals^{5, 11}. However, CER-based extraction primarily isolates loosely bound or exchangeable EPS²⁸, stabilized through reversible Ca^{2+} bridging and electrostatic attraction (Supplementary Discussion 3). These fractions are unlikely to represent long-term stabilization but may provide transient protection from microbial degradation, thereby promoting short-term EPS persistence²⁸. Additionally, although our study does not directly include microbial community diversity or composition data, previous research has shown that lithology-driven geochemical processes – such as rock weathering and mineral-associated protection – can indirectly regulate the accumulation of microbial extracellular products (such as glomalin-related soil protein) via microbial functional traits¹². In particular, carbonate soils have been reported to harbor higher microbial diversity¹², which, given the positive relationship between diversity and EPS production¹³, may reflect a greater EPS secretion potential¹². Moreover, the relatively higher WHC in carbonate soils – may reflect the ecological function of soil EPS in improving soil water retention⁵ (Fig. 9). Interestingly, while carbonate soils exhibited the highest EPS contents, their EPS-C/MBC ratios followed a different pattern, with soils on sedimentary rocks demonstrating the highest EPS-C/MBC ratios (Fig. 7a). Although this ratio does not directly reflect EPS production efficiency – since EPS contents in natural soils result from the balance of microbial production and degradation, modulated by mineral interactions and extraction limitations – it may serve as a proxy for the relative allocation of microbial C assimilates to extracellular non-growth products under steady state conditions. Notably, sedimentary soils had the lowest EPS contents, but the highest EPS-C/MBC ratios, which likely stems from reduced MBC rather than increased EPS contents. Supporting this interpretation, microbial growth-related parameters (e.g., C_{growth} , MBC, and MBN) were consistently lowest in sedimentary soils, which also had the lowest WHC and SMC (Supplementary Table 5). Given the decrease in MBC at lower WHC and SMC (Supplementary Fig. 2), we infer that limited water availability in sedimentary soils may constrain microbial growth and shift C allocation towards EPS formation. This shift may reflect an adaptive microbial strategy to enhance survival and maintain extracellular protection under water-stressed conditions⁵. Although silicate soils

exhibited intermediate levels of both EPS and MBC (Supplementary Table 5), they had the lowest EPS-C/MBC ratio. This suggests that microbes in silicate soils may have allocated proportionally less C toward EPS production or are less conducive for intermittent EPS retention. Three possible explanations can be proposed: (1) Environmental stress in silicate soils may be relatively mild – for example, these soils exhibited the highest SMC and WHC – whereas more severe moisture-related stress in sedimentary soils might have triggered greater EPS production; (2) the lowest levels of C_{ae} and clay (highest sand content) in silicate soils may have provided the least protection for EPS (Supplementary Table 5); (3) differences in microbial community composition or metabolic strategies may also contribute to the observed pattern. However, we currently lack direct evidence to confirm the third explanation. Overall, bedrock influenced EPS through (1) promoting microbial biomass and EPS secretion via bedrock-driven environmental conditions, and (2) shaping soil texture and mineralogy, which governs EPS retention and intermittently protects it from microbial degradation” (Pages 9-11, Lines 259-307).

We have added descriptions of the newly incorporated statistical methods in the **Statistical Analyses section**: “To investigate the effects of bedrock and land use type on environmental and microbial factors, we performed two-way ANOVA for each environmental and microbial variable. Variance partitioning analysis (VPA) was then conducted to quantify the relative contributions of bedrock and land use type to the variation in these factors. Redundancy analysis (RDA) was used to test the statistical significance of the partitions identified by VPA” (Pages 18-19, Lines 590-594)

[Comment 46] Line 339-341: To be able to follow your train of thoughts I would like to check the statement that the soils developed on sediment bedrock “were intermediate in properties promoting EPS stabilization” – I guess meaning low clay, low silt, Ca and Fe contents – that’s what I would like to prove it.

[Response 46] Thank you for pointing this out. We have now added a supplementary Table 5 showing the values and statistical differences of environmental and microbial factors across the three bedrock types. See other response **R1-22**, **R4-45**, **R4-47**, **R4-50** and **R4-51** on refined statistics and bedrock effect interpretation. Specifically:

- (1) Clay content: Carbonate > Sediment > Silicate
- (2) Silt content: Carbonate > Sediment > Silicate
- (3) C_{ae} : Carbonate > Sediment > Silicate
- (4) Fe_c : Carbonate > Sediment > Silicate
- (5) Fe_d : Carbonate > Sediment > Silicate

As EPS extracted using the CER method is not considered long-term stable, we agree that the previous interpretation suggesting “EPS stabilization” was not fully supported. Therefore, we have removed this sentence “In the studied soils, those

deriving from sedimentary bedrock were intermediate in properties promoting EPS stabilization (e.g., clay, silt, Ca_e, Fe_d) but had highest EPS production efficiencies, which may explain the second highest soil EPS concentrations”.

Additionally, we acknowledge that our previous statement indicating that sedimentary soils had the “second-highest soil EPS contents” was imprecise. This was based on the visual interpretation of the median values in the boxplot (Fig. 2), where the median EPS content in sedimentary soils appeared higher than in silicate soils. However, the mean EPS content was lowest in sedimentary soils. To clarify this, we have updated the **Fig. 2** to include orange dots representing the mean values for each bedrock type, and we have revised the figure legend accordingly.

[Comment 47] Line 335: I would assume that the larger difference in nutrient can be found based on the land use instead of the bedrocks. Usually cropland and even grassland is fertilized. So, I would expect to find lower nutrients mainly in woodland. You have the data for the parent material and the land use – prove it, instead of referring to some references

[Response 47] Following your suggestion, we conducted a two-way ANOVA to assess the effects of both bedrock type and land use on soil physicochemical and microbial parameters potentially controlling EPS contents in soils. In addition, we performed variance partitioning to determine which factor had a stronger influence on different environmental and microbial parameters. The full results have been added as **Supplementary Table 5**. See also responses **R1-22**, **R4-45**, and below **R4-50** and **R4-51** on refined statistical treatment of bedrock and land use effects and their interpretation

Specifically, the bedrock type had a greater impact on microbial biomass nitrogen (MBN), microbial biomass carbon (MBC), microbial biomass phosphorus (MBP), β -glucosidase activity (BG), β -xylosidase (BX), exoglucanase-cellobiosidase (CEL), leucine aminopeptidase (LAP), N-acetyl- β -glucosaminidase (NAG), vector angles (VectorA), carbon use efficiency (CUE), nitrogen use efficiency (NUE), cation exchange capacity (CEC), exchangeable Ca²⁺ (Ca_e), clay content, sand content, aridity index (ADI), water holding capacity (WHC), soil moisture content (SMC), and pH. In contrast, land use strongly affected soil organic carbon (SOC), total organic phosphorus (TOP), dissolved organic carbon (DOC), total phosphorus (TP), exchangeable Mg²⁺ (Mg_e), NO₃⁻, C:N ratio, C:P ratio, N:P ratio, NH₄⁺, Al oxyhydroxides extracted with Na-dithionite (Al_d), Al oxyhydroxides extracted with acid ammonium oxalate (Al_o), Fe oxyhydroxides extracted with acid ammonium oxalate (Fe_o), root carbon and nitrogen, fine root biomass (FRB), enzyme vector length (VectorL), root C:N and C:P, microbial growth (C_{growth}), microbial growth normalized to microbial biomass carbon (q_{growth}), bacterial, gram-negative bacteria (GNB), and gram-positive bacteria (GPB). Relatively,

land use had a greater impact on soil nutrient levels than bedrock type. Therefore, we have removed the previous inaccurate statement: “Sedimentary rocks are rich in various nutrients (such as NH₄⁺)²² that may stimulate the microbial secretion of EPS. This, in turn, can stimulate microbial resource utilization efficiency and microbial activities through a positive feedback mechanism²³”.

[Comment 48] Line 341-343: Also in the croplands, I would highly doubt that and you have data to prove it – do it!

[Response 48] In our original manuscript, we proposed that silicate bedrock might limit microbial activity and EPS production due to its slow weathering rate and limited nutrient release. However, based on further analysis—including variance partitioning and evaluation of resource availability across different land-use types—we found that this explanation is not well supported by our data (e.g., higher MBC than sediment). Therefore, we have removed this sentence “In contrast, silicate rocks, due to their slower weathering rates and limited nutrient release capacities, may lack certain essential soil microbial elements, thus, limiting microbial activities and the generation of EPS^{25, 26}” from the revised manuscript to ensure that our discussion remains fully grounded in evidence from our dataset.

[Comment 49] Line 344: The authors should precisely use and explain which parameters they assume to influence EPS stabilization and then use them all accordingly, not once including more (Line 340), next time less (Line 344)

[Response 49] Thank you for your valuable comment. We have carefully reviewed the entire **Discussion Section** and ensured that all environmental drivers potentially influencing EPS contents are listed consistently throughout. To improve clarity and scientific rigor, we now refer only to variables that show significant relationships with EPS contents based on correlation analysis and results from two-way ANOVA and variance partitioning. This approach helps to maintain consistency in our interpretation.

[Comment 50] Line 345-346: Prove it! Is a low pH going along with reduced microbial parameters lower MBC, lower CUE,

[Response 50] We examined the relationships among soil pH, microbial parameters, and EPS content across the three bedrock types and summarized the key findings as follows (see also response **R4-46**):

- (1) EPS content: Carbonate > Silicate > Sediment
- (2) EPS-C/MBC: Sediment > Carbonate > Silicate
- (3) pH: Carbonate > Sediment > Silicate
- (4) MBC: Carbonate > Silicate > Sediment
- (5) CUE: Silicate > Carbonate > Sediment

- (6) WHC: Silicate > Carbonate > Sediment
- (7) SMC: Silicate > Carbonate > Sediment
- (8) Ca_e , clay, and silt contents: Carbonate > Sediment > Silicate

Notably, no significant correlation was observed between soil pH and EPS content (**Supplementary Fig. 3**), suggesting that pH alone may not directly explain the variation in EPS between bedrock types. Interestingly, although silicate soils had the second-highest EPS and MBC levels, their EPS-C/MBC ratio was the lowest. In contrast, sedimentary soils exhibited both the lowest EPS and MBC values but the highest EPS-C/MBC ratio.

Supplementary Figure 3 | Relationship between EPS content, EPS-C/MBC ratio and soil pH. a Relationship between extracellular polymeric substances (EPS) and soil pH. **b** Relationship between extracellular polymeric substances carbon to microbial biomass carbon ratio (EPS-C/MBC) and soil pH. Different colors represent land use types, while different point shapes represent bedrock types. The solid line indicates the fitted quadratic regression.

This pattern may be attributed to stronger water stress in sedimentary soils—indicated by the lowest WHC and SMC—which could trigger greater microbial investment in EPS production. Sedimentary soils have intermediate sand content and the lowest SOC amount the three bedrock types, which may further limit soil water retention and nutrient availability, reinforcing the need for microbial EPS production. Conversely, silicate soils had the highest WHC and SMC, potentially reducing the need for EPS-mediated stress protection and leading to lower carbon allocation to EPS. Additionally, silicate soils had the lowest Ca_e , clay, and silt contents, possibly providing the least protection for EPS and thus reducing its short-term retention (Supplementary Table 5).

Accordingly, we have revised the original sentence from “Silicate soils had the lowest levels of potentially EPS-binding properties (clay, silt, and, Ca_e), and the lowest EPS production efficiencies which were accompanied by lowest soil pH values”

eventually reducing microbial activities and decomposition processes” to “Although silicate soils exhibited intermediate levels of both EPS and MBC (Supplementary Table 5), they had the lowest EPS-C/MBC ratio. This suggests that microbes in silicate soils may have allocated proportionally less C toward EPS production or are less conducive for intermittent EPS retention. Three possible explanations can be proposed: (1) Environmental stress in silicate soils may be relatively mild – for example, these soils exhibited the highest SMC and WHC – whereas more severe moisture-related stress in sedimentary soils might have triggered greater EPS production; (2) the lowest levels of Ca_e and clay (highest sand content) in silicate soils may have provided the least protection for EPS (Supplementary Table 5); (3) differences in microbial community composition or metabolic strategies may also contribute to the observed pattern. However, we currently lack direct evidence to confirm the third explanation” (Pages 10-11, Lines 294-304). We have also added **Fig. 9** and **Fig. 10** and supporting analyses in the supplementary materials (**Supplementary Table 5**).

Figure 9. Linear regressions between EPS and soil water properties. a Relationship between EPS and water holding capacity. **b** Relationship between EPS and soil moisture content. Blue lines indicate linear regression fits with 95% confidence intervals (shaded). The R^2 and p -value for each regression are shown on each panel. Each point represents an individual soil sample. EPS, extracellular polymeric substances.

Figure 10. Linear regressions between EPS-C/MBC and key soil and microbial parameters. **a** Relationship between EPS-C/MBC and soil moisture content. **b** Relationship between EPS-C/MBC and aridity index. **c** Relationship between EPS-C/MBC and microbial growth. **d** Relationship between EPS-C/MBC and microbial carbon use efficiency. Blue lines indicate linear regression fits with 95% confidence intervals (shaded). The R^2 and p -value for each regression are shown on each panel. Each point represents an individual soil sample. EPS, extracellular polymeric substances; MBC, microbial biomass carbon; C_{growth} , microbial growth.

[Comment 51] Line 346-349: The authors did not prove the first step in my point of view, that the bedrock is making the difference in soil properties (not differences in bedrock chemistry – please be precise) – I can find that data. The authors found significant different EPS concentrations, the rest is speculation.... Revise it accordingly

[Response 51] Thank you for your valuable comment. We fully understand and appreciate your concern regarding whether the observed differences in soil EPS are truly driven by bedrock type, rather than simply reflecting differences in bedrock chemistry. In response, we have carefully revised the relevant part of the **Discussion Section** to avoid unsupported assumptions and to base our interpretation more strictly

on the data (see response **R4-45**).

Specifically, we performed two-way ANOVA and variance partitioning analyses to quantitatively assess the respective influences of bedrock type and land use on soil physicochemical properties (see Supplementary Table 5). Our results show that bedrock type significantly influences soil properties that are strongly correlated with EPS content, including MBC, MBN, CEC, SMC, Ca_e, and clay content. These findings, together with the observed correlations between EPS and these environmental factors, provide data-supported evidence that bedrock-driven differences in soil properties may underlie the variation in EPS content.

Additionally, we revised the **Discussion Section (Pages 9-11, Lines 259-307)** to remove inappropriate language and to ensure consistency with the observed statistical results. We believe this has improved the clarity and reliability of our interpretation. Thank you again for your constructive feedback.

[Comment 52] Line 462-463: The authors found a weak negative correlation line 261-262 – so is it worse pushing it here – or would be better to try to discuss instead of refreshing what others propagated but have been seldom proven.

[Response 52] Thank you very much for your insightful comment. We agree that the observed correlation between the EPS-C/MBC ratio and ADI is relatively weak and therefore should be interpreted with caution. Moreover, responses to aridity and drought do not align directly, see response **R3-19**. To avoid overstating our findings and to better reflect the data strength, we have revised this sentence from “Although this has not yet been demonstrated for soil microbial communities, we demonstrate this adaptive phenomenon on a large scale i.e., reflected by increasing EPS production efficiency with increasing aridity (indicated by a decrease in the aridity index, ADI) and with worsening drought conditions. This same trend was apparent by the negative correlation between EPS production efficiency and soil moisture content across Europe” to “**Although not previously demonstrated for soil microbial communities, we show this adaptive phenomenon at a large scale, reflected by increasing EPS-C/MBC along an aridity (ADI) gradient, suggesting microbial adaptation to long-term water-limited environments. This pattern is further supported by the negative correlation between EPS-C/MBC and SMC (Fig. 10). Microbial communities therefore invest in EPS secretion to retain soil moisture under increasing water deficit. However, as measured EPS levels are influenced by production, decomposition, and stabilization processes, the true microbial response requires further investigation**” to adopt a more cautious tone. Rather than claiming to demonstrate a confirmed adaptive mechanism, we now describe the observed pattern as a potential trend that warrants further investigation (**Pages 12-13, Lines 369-375**).

[Comment 53] Line 467-468: refer to the data – where can I find it as a reader – such a figure with this correlation would be nice to see in the supplement!

[Response 53] Thank you for your valuable comment. We apologize for the earlier oversight of not including the relationship between EPS and WHC in Fig. 2c. We have now added this correlation to the heatmap in **Fig. 2c**, provide full visualisation of the relation between EPS and WHC in the new **Fig. 9** (see response **R3-10**) and revised this sentence to directly refer to the relevant data.

[Comment 54] Line 331-333: What do the authors mean by “promote adsorption”? I thought the EPS is bound via Ca-bridging!? Clarify

[Response 54] Thank you for pointing this out. We acknowledge the ambiguity in our original phrasing. To clarify, we have revised the sentence from “This provides a suitable living environment for microorganisms, but also promotes the adsorption of EPS by minerals and thereby aids with its stabilization and accumulation^{20, 21}” to “**soil minerals (clay) and cations (Ca_e) may contribute to the selective retention of EPS within the soil matrix through multivalent cation bridging between negatively charged sites of EPS and other polymers and minerals^{5, 11}**” (**Page 10, Lines 267-270**). More details on the mechanisms underlying soil EPS stability have been added to **Supplementary Discussion 3**. Please see below:

Supplementary Discussion 3 - Mechanisms of EPS stabilization in soils

EPS in soils is stabilized via multiple mechanisms^{5, 8, 11}, including (i) physical occlusion inside microaggregates, (ii) co-precipitation and complexation with Fe/Al, (iii) hydrogen bonding, (iv) hydrophobic interactions, (v) inner-sphere ligand exchange to Fe/Al (oxyhydr)oxides and clays, (vi) outer-sphere electrostatic attraction, and (vii) multivalent cation bridging between negatively charged sites of EPS and those on other polymers and minerals. The importance of these mechanisms differs between main EPS compounds, with EPS polysaccharides mainly bound/stabilized by (ii, iii, v, vii) and EPS proteins via (ii, iii, iv, v, vi, vii). The cation bridging mechanism via Ca²⁺ (vii) and electrostatic attraction (vi) causes a loosely bound (not strongly stabilized) association, which is reversible; it is this EPS fraction that is targeted by the CER method due to removing Ca²⁺ by CER and the extraction buffer, weakening electrostatic forces. Binding of EPS by other mechanisms causes weaker binding and stabilization (iii) or stronger binding/immobilization and stabilization (ii, iv, v), but these latter EPS fractions are not targeted or extracted by CER. Since comprehensive studies across EPS components, sources, and soil types are still lacking, this remains poorly understood. In culture experiments, it is also the loosely bound fraction of EPS that is extracted by CER, whereas the tightly bound fraction remains unextracted.

[Comment 55] Line 333-335: How did the authors study sorption and stabilization controls on EPS concentrations (maybe I missed that point?) – it can't be evident from a correlation!!!! Refer to the data that all these properties are highest in the carbonate soils – I would wish to prove that.

[Response 55] We fully agree that correlations alone are not sufficient to demonstrate sorption and stabilization controls on EPS contents. In response, we revised the **Discussion Section** to provide a more data-driven explanation. Specifically, we conducted two-way ANOVA and variance partitioning analyses to evaluate the effects of bedrock type and land use on environmental and microbial factors. Our results showed that carbonate soils exhibited significantly higher levels of Ca_e , and clay content compared to soils developed on silicate and sedimentary bedrock (Supplementary Table 5). These properties are known to influence EPS retention via electrostatic interactions and cation bridging, and were also positively correlated with EPS contents in our study.

Therefore, we have revised the sentence from “Sorption and stabilization controls on EPS concentrations were also evident from positive correlations between EPS concentration and Ca_e , clay, silt and Fe_d , and these properties were also highest in carbonate soils” to “**This pattern was mirrored in microbial (e.g., MBC and MBN) and soil properties (e.g., CEC, clay, and Ca_e), known to promote EPS, all of which were significantly influenced by bedrock and peaked in carbonate soils. Conversely, sand content – which was lowest in carbonate soils – was negatively correlated with EPS contents (Supplementary Table 5). Based on these findings, we propose that the highest EPS content in carbonate soils is primarily driven by two aspects: (1) an indirect pathway, where highest soil pH and CEC (particularly Ca_e) modulates the soil microenvironment in ways that stimulate microbial activity²⁷, increase microbial biomass (e.g., MBC and MBN) and thereby causes greater overall capacity for EPS production⁹; and (2) soil minerals (clay) and cations (Ca_e) may contribute to the selective retention of EPS within the soil matrix through multivalent cation bridging between negatively charged sites of EPS and other polymers and minerals^{5, 11}” (Pages 9-10, Lines 260-270). See also response **R1-22, R4-45/46, R4-50** on sorption and bedrock controls.**

[Comment 56] Line 337: The authors should better let us participate in their nebulous explanations, what is meant here by “positive feedback mechanisms”.

[Response 56] In this context, the “positive feedback mechanism” refers to the following process: nutrients released from sedimentary rocks (such as NH_4^+)²² stimulate microbial EPS production; the increased EPS enhances microbial resource acquisition and protection (e.g., by retaining moisture and stabilizing enzymes), which in turn further promotes microbial growth and activity - leading to even more EPS

production. However, upon further data analysis, we arrived at a new interpretation for the high EPS-C/MBC ratios observed in sedimentary soils: “Interestingly, while carbonate soils exhibited the highest EPS contents, their EPS-C/MBC ratios followed a different pattern, with soils on sedimentary rocks demonstrating the highest EPS-C/MBC ratios (Fig. 7a). Although this ratio does not directly reflect EPS production efficiency – since EPS contents in natural soils result from the balance of microbial production and degradation, modulated by mineral interactions and extraction limitations – it may serve as a proxy for the relative allocation of microbial C assimilates to extracellular non-growth products under steady state conditions. Notably, sedimentary soils had the lowest EPS contents, but the highest EPS-C/MBC ratios, which likely stems from reduced MBC rather than increased EPS contents. Supporting this interpretation, microbial growth-related parameters (e.g., C_{growth} , MBC, and MBN) were consistently lowest in sedimentary soils, which also had the lowest WHC and SMC (Supplementary Table 5). Given the decrease in MBC at lower WHC and SMC (Supplementary Fig. 2), we infer that limited water availability in sedimentary soils may constrain microbial growth and shift C allocation towards EPS formation. This shift may reflect an adaptive microbial strategy to enhance survival and maintain extracellular protection under water-stressed conditions⁵” (Page 10, Lines 281-294).

[Comment 57] Line 350-364: Condense and stick to the data you have.

[Response 57] To improve clarity and better align with the data, we have revised the section on land use from “Among the three land-use types, EPS concentrations were highest in grassland soils compared to cropland and woodland soils. Interestingly, as observed with the different types of bedrock, the rankings for EPS production efficiencies and EPS concentrations did not align consistently across land use types. The EPS production efficiencies of croplands and grasslands were similar and higher than those of woodlands. This divergence may have been attributed to large differences in plant belowground C inputs (root system development) accompanied by external nutrient inputs (fertilization) and disturbance (tillage, grazing) in agricultural systems (Supplementary Table S16)⁹⁰. Soil EPS increased with fine root biomass along the European transect, indicating that part of the land use driven patterns in EPS concentration were plant input driven. Plant fine root biomass was similar in grasslands and woodlands, and much greater than in croplands, but root morphologies also differ dramatically⁹⁰. Grass roots are known to be much thinner and have greater specific root length (length per mass of fine roots), globally leading to much greater fine root length and fine root area in grasslands than in forests, where trees produce thicker roots with less fine root length and area⁹⁰. Thinner roots turnover faster and thereby produce larger root necromass inputs in topsoil which adds to greater root exudate inputs in grasslands”

with their markedly greater root surface area⁹⁰ to “Among the three land-use types, EPS contents were highest in grassland soils compared to cropland and woodland soils, although the differences did not reach statistical significance (Fig. 2b). To identify potential drivers of EPS content variation, we found that MBC was significantly affected by land use and showed a pattern consistent with EPS (Supplementary Table 5). This, together with the positive correlation between MBC and EPS (Fig. 2c), suggests that higher MBC may indicate larger microbial populations, thereby representing a greater potential for EPS production and serving as a primary control on EPS content across land-use types. In addition, land-use exerted a stronger influence on soil nutrient status (e.g., SOC, TOP, TP, Mg_e, and NO₃⁻) and fine root traits (FRB and Root N) than bedrock (Supplementary Table 5). Among these, SOC and TOP were significantly higher in grassland than in cropland and positively correlated with EPS, while root N was also highest in grasslands, whereas FRB peaked in woodlands but remained relatively high in grasslands. These indicators suggest that improved nutrient availability and greater plant inputs (root turnover and exudation) may enhance microbial EPS production⁸⁷. This interpretation is consistent with previous findings: Redmile-Gordon et al.²⁸ observed the highest EPS contents in grasslands, while Kidinda et al.⁹ reported higher EPS in fertilized croplands than in forests on nutrient-poor tropical soils. Together, these studies suggest that land-use effects on EPS are primarily mediated by plant inputs and nutrient availability. Conversely, the lowest MBC and SOC observed in cropland soils may reflect soil degradation or organic matter depletion caused by long-term intensive management practices such as tillage, which could in turn contribute to the reduced EPS levels. For instance, frequent tillage can disrupt soil structure and aggregates, thereby reducing soil organic matter content¹⁰. On the other hand, although croplands had the highest clay and exchangeable Ca²⁺ contents—factors potentially favouring short-term EPS retention—their EPS levels were lowest. This suggests that limited labile C availability, linked to lower SOC and FRB in croplands, may play a more decisive role in constraining microbial EPS content across the investigated soils. Substantial differences in the composition and function of soil microbial communities under different land use types⁸⁸ may also cause differences in EPS formation and decomposition⁸⁹. While microbial diversity was not investigated in this study, in accordance with investigations of legacy vs current management effects on EPS²⁸, it is primarily the interplay between plant inputs and anthropogenic management that causes these differences in total EPS and EPS protein contents”. As the effect of land use on EPS was not significant, this part of the discussion has accordingly been moved to **Supplementary Discussion 4**.

We have retained a brief description in the main text: “Meanwhile, EPS contents did not differ significantly among land use types but tended to be highest in grasslands,

with microbial biomass, nutrient availability, and plant inputs mediating the land-use-based variation; in contrast, EPS-C/MBC ratios peaked in croplands, likely reflecting enhanced microbial EPS investment under water limitation (Supplementary Discussion 4)” (Page 11, Lines 308-311). See also responses R1-27, R3-13, and R4-47.

[Comment 58] 366-368: Please prove it and how does that fit with the discussion beforehand? Did you have cropland on silicate bedrock!?

[Response 58] Following your suggestion in comment R4-39 and your specific point in R4-57 to “stick to our data,” we have removed vague explanations that were not directly supported by our dataset (e.g., long-term fertilization effects). Importantly, we do not have accurate fertilization data for cropland sites. Therefore, we re-analyzed the data using two-way ANOVA and variance partitioning, and reassessed the values of environmental and microbial factors under different land-use types and their correlations with EPS (Supplementary Table 5). Based on these results, we have now provided a revised explanation for the observation that croplands had the highest EPS-C/MBC. Specifically, we have replaced the original sentence “Despite lower plant root C inputs in croplands due to the reduced root system development in cropping systems, on a microbial biomass basis, EPS production efficiency was highest, likely due to relaxed nutrient constraints of microbial activities through long-term fertilization” with “**Interestingly, the ranking of EPS-C/MBC did not mirror that of EPS contents, with croplands showing the highest EPS-C/MBC, followed by grasslands and woodlands (Fig. 7b). Given that both EPS and MBC were lowest in cropland soils, this highest EPS-C/MBC ratio suggests a greater microbial investment in EPS production per unit biomass. This may be a response to increased water stress in croplands, as indicated by significantly lower WHC and SMC compared to woodlands and grasslands (Supplementary Table 5)**” (Supplementary Discussion 4).

[Comment 59] Line 368-370: Do you prove if the lowest plant root input in cropland caused the lowest MBC levels!? I always thought it is due to the overall management (mechanical and chemical interventions)!

Maybe the croplands are just lowest in EPS stabilization properties? Just sarcastic joke... please revise your discussion along your data and what you can prove and not prove. Instead of interpreting your data once like that and then the other way around as it better suits your story.

[Response 59] We acknowledge that in the original manuscript, we did not provide direct evidence showing that the low fine root biomass (FRB) in croplands led to the reduced microbial biomass carbon (MBC). Indeed, as you pointed out, the overall land management practices in cropland, such as mechanical disturbance, may also contribute

to the lower MBC levels. We have revised the discussion accordingly to reflect this possibility. See also our response to land use related effects on EPS in **R1-27**, **R3-13** and **R4-57**.

In addition, after deeper analysis of the data, we now attribute the highest EPS-C/MBC ratio in croplands — despite both EPS and MBC being the lowest among the land-use types — to potential microbial stress responses to limited soil water availability. This is supported by significantly lower water holding capacity (WHC) and soil moisture content (SMC) in cropland soils compared to the other land-use types.

As for your suggestion regarding potential differences in EPS retention, we also explored this aspect. Interestingly, although cropland soils exhibited highest clay and exchangeable Ca^{2+} (Ca_e) contents — both commonly considered important for EPS retention — their EPS contents remained low. This may indicate that, in this case, stimulatory effects of soil nutrient status on microbial EPS production may outweigh those of the retention of EPS.

We have accordingly added this sentence “**Conversely, the lowest MBC and SOC observed in cropland soils may reflect soil degradation or organic matter depletion caused by long-term intensive management practices such as tillage, which could in turn contribute to the reduced EPS levels. For instance, frequent tillage can disrupt soil structure and aggregates, thereby reducing soil organic matter content¹⁰. On the other hand, although croplands had the highest clay and exchangeable Ca^{2+} contents—factors potentially favouring short-term EPS retention—their EPS levels were lowest. This suggests that limited labile C availability, linked to lower SOC and FRB in croplands, may play a more decisive role in constraining microbial EPS content across the investigated soils. Substantial differences in the composition and function of soil microbial communities under different land use types⁸⁸ may also cause differences in EPS formation and decomposition⁸⁹. While microbial diversity was not investigated in this study, in accordance with investigations of legacy vs current management effects on EPS²⁸, it is primarily the interplay between plant inputs and anthropogenic management that causes these differences in total EPS and EPS protein contents” in **Supplementary Discussion 4** and revised this sentence from “However, low plant root C inputs in croplands caused the lowest MBC levels and therefore, even at the highest microbial EPS production efficiencies, the lowest EPS concentrations” to “**Interestingly, the ranking of EPS-C/MBC did not mirror that of EPS contents, with croplands showing the highest EPS-C/MBC, followed by grasslands and woodlands (Fig. 7b). Given that both EPS and MBC were lowest in cropland soils, this highest EPS-C/MBC ratio suggests a greater microbial investment in EPS production per unit biomass. This may be a response to increased water stress in croplands, as indicated by significantly lower WHC and SMC compared to woodlands and grasslands (Supplementary Table 5)**” in **Supplementary Discussion 4**.**

[Comment 60] Line 373-374: How is agricultural management reducing stabilization properties? And if you find overall lower MBC, how will the management increase EPS degradation?

[Response 60] We acknowledge that the previous version of the manuscript did not sufficiently explore the underlying data to support the claim that agricultural management reduces EPS stabilization and accelerates EPS degradation. After conducting a two-way ANOVA and variance partitioning, and examining the environmental and microbial factor values across the three land-use types, we found that this explanation is not well supported by the data. In fact, cropland soils exhibited relatively high clay content (which generally favours EPS stabilization) alongside low MBC. Therefore, we have removed the sentence “These factors contribute to reduced formation and stabilization and accelerated degradation of EPS⁹¹” from the manuscript. Instead, we added the **Supplementary Discussion 4** to provide new explanations for the observed variations in EPS content and EPS-C/MBC ratio among the land-use types based on our data analysis. See responses **R1-27**, **R4-57** and **R4-59** above.

[Comment 61] Line 382-385: Which difference reflects what! Explain!

[Response 61] What we intended to convey is that the content of EPS polysaccharides responds more strongly to changes in bedrock type, while EPS protein content is more responsive to land-use change. We realize that our original wording may have caused confusion, and we have revised the sentence from “This difference reflects the different elemental demands (C, or C plus N) and controls of the biosynthetic pathways of EPS polysaccharides and EPS proteins in microbial ecosystems²³, and the different interaction mechanisms with organic and mineral surfaces of EPS polysaccharides and EPS proteins promoting their binding, stabilization and accumulation” to **“Moreover, distinct environmental dependencies and biosynthetic pathways of EPS polysaccharides and EPS proteins caused their contents to be differentially by bedrock and land use, respectively (Supplementary Discussion 5)” (Page 11, Lines 311-314)**. Further discussion of the mechanisms behind the contrasting responses of EPS polysaccharides and proteins to land-use type and bedrock has been included in **Supplementary Discussion 5**. Please see below:

Supplementary Discussion 5 - Bedrock vs. land use: drivers of specific EPS components

A more detailed analysis revealed that it was primarily the bedrock type that affected the contents of EPS polysaccharides, while the type of land use influenced the contents of EPS proteins (Fig. 3). This divergence likely stems from their distinct environmental dependencies and biosynthetic pathways. Based on our data, bedrock

type significantly influenced microbial biomass (MBC and MBN), soil texture (clay content), CEC, and exchangeable Ca^{2+} (Supplementary Table 5), all of which are positively associated with EPS polysaccharides content. In contrast, land use significantly influenced MBN, and had a stronger effect on microbial q_{Growth} and soil nutrient availability, such as soil NH_4^+ , NO_3^- , and TOP, as well as on SOC (Supplementary Table 5), which plays a critical role in microbial protein synthesis and was positively related to EPS protein. This is consistent with our random forest analysis showing that EPS polysaccharides were mainly driven by Ca_e , MBN, and SOC, whereas EPS proteins were predominantly influenced by MBN, SOC, and q_{Growth} (Fig. 4b, c). These results indicate that while both EPS polysaccharides and EPS proteins are influenced by microbial abundance and nitrogen availability (MBN) as well as soil C status (SOC), EPS polysaccharides are closely associated with soil Ca^{2+} , highlighting their role in reinforcing soil structure, whereas EPS proteins are more strongly linked to microbial biosynthetic activity (q_{Growth}). Additionally, based on the principle of the CER method, variations in Ca^{2+} concentrations induced by different bedrock types may influence EPS extraction efficiency and thus affect the measured EPS contents³⁶. Although a preliminary experiment was conducted to optimize the extraction procedure across soil types, variability in extraction efficiency cannot be fully excluded. However, since only EPS polysaccharides, but not EPS proteins, showed significant variation with bedrock type, this indicates that the contrasting patterns are unlikely to result solely from methodological bias, but rather reflect differences in environmental or microbial regulation of EPS components.

[Comment 62] Line 386: How can the authors come up with an EPS accumulation!? You did not study that, or?

[Response 62] Thank you for your valuable suggestion. As our study did not include multi-time-point measurement of EPS, we agree that we cannot provide direct evidence for EPS accumulation. To improve the accuracy and clarity of our statement, we have removed the sentence “The phenomenon of differential controls of EPS polysaccharide and EPS protein accumulation is therefore triggered by the selective regulation of microbial metabolic dynamics leading to EPS compound biosynthesis and secretion and differences in their stabilization mechanisms” accordingly.

[Comment 63] Line 388-393: here the authors open up another discussion box – which they also can't prove – delete – unnecessary for the study

[Response 63] Thank you for your helpful comment. We have removed this sentence “This differential accumulation mechanisms are thought to have strong repercussions on the properties of EPS like its stickiness and adhesion, mechanical stability”

(Supplementary Fig. S2), ion binding capacity and hydration potential which are related to the ratio of EPS protein: EPS polysaccharides^{92, 93}. However, studies on the EPS protein-to-polysaccharide ratio have primarily been conducted in marine environments, with no research available in the context of soils” from the revised manuscript to maintain focus and clarity.

[Comment 64] Line 398-400: It would help if the authors include a precise description of the calculation.

[Response 64] Thank you for your helpful suggestion. We have now included a clear and detailed description of the EPS-C calculation in the **Methods Section (Pages 17-18, Lines 541-571)**. For further explanation regarding the derivation of the conversion factors, please also refer to our response **R4-4**.

[Comment 65] Line 401: Prove that number!

[Response 65] The C content of EPS polysaccharides (39.1%) used in our calculation was derived as the arithmetic mean of the C content of four representative standard compounds commonly assigned as microbial and/or plant EPS compounds: hyaluronic acid (37.5% C), xylan (40.5% C), pectin (38.3% C), and arabinogalactan (40.2% C). While we acknowledge that this estimation cannot fully reflect the structural and compositional diversity of microbial EPS in soils, it provides a reproducible and reasonable approximation for quantifying EPS-C at larger scales.

We have clarified this point in the **Methods Section (Pages 17-18, Lines 541-571)**, see our response **R4-4**.

[Comment 66] Line 401-404: But microbes produce a variety of EPS sugars and proteins – even single microbes can produce different EPS constituents – how does that relate to your estimation based on one species!?

[Response 66] Thank you for your insightful comment. Indeed, we fully acknowledge that soil microbes produce a wide variety of EPS polysaccharides and EPS proteins, and even individual strains can synthesize diverse EPS polysaccharide forms and likely 100s of EPS-related exoproteins. In our study, the estimation of the carbon content of EPS proteins (50.7%) was based on the average amino acid composition and weighed amino acid C:N ratio of microbial proteins, using the amino acid composition of the yeast proteome as a representative dataset⁵¹, and assuming a global average nitrogen content of 16% in proteins^{65, 66}. This allowed us to calculate the corresponding carbon content based on average proteinogenic amino acid C:N ratios. See also response in **R4-4**.

This method provides a reproducible estimation. However, we fully agree that it cannot capture the full biochemical diversity and complexity of microbial EPS in soils. Given the growing interest in soil EPS and its roles in soil functioning, our intention was to provide an approximate but practical estimate of total EPS amounts and associated carbon pools in soil systems. We believe such estimates, while approximate, are tractable and valuable for advancing early understanding and guiding future, more detailed studies.

To reflect this limitation, we have also added an explicit explanation:

“The final formula (2) for summed EPS-C is:

$$\text{EPS} - \text{C} \text{ (g C kg}^{-1} \text{ soil dw) =} \\ (\text{EPS polysaccharides} \times 39.1\%) + (\text{EPS proteins} \times 50.7\%) \quad (2)$$

which provides an approximate estimate based on representative C content. To our knowledge, this represents an important attempt to quantify soil EPS-C contents, as such estimations were previously hindered by the lack of representative data on the C content of EPS components. In this study, we addressed this gap by determining average C content values from standard compounds representative of microbial EPS polysaccharides and by calculating protein C from proteome-averaged protein C:N ratios and published global protein N contents. While these estimates provide a useful approximation, we acknowledge that they do not fully capture the chemical diversity of EPS in soils” in the **Methods section** and clarified in the text that this represents an approximation (**Page 18, Lines 560-568**). We hope this clarification addresses your concern.

[Comment 67] Line 404: Still unclear how you then come up with EPS-C (summing up!?)?

[Response 67] We sincerely apologize for the lack of clarity that may have led to confusion. A more detailed explanation has now been added to **Method Section (Pages 17-18, Lines 541-571)**. Please see our previous explanations in response **R4-4**, and **R4-65/66**, which depict how EPS-C was calculated from summing up C equivalents of EPS proteins and EPS polysaccharides.

[Comment 68] Line 413: what is meant by highly active soil C pool? How can a highly active C pool promote SOC stabilization – seem contradicting to me?

[Response 68] We agree that the phrase “highly active soil C pool” may cause confusion when used alongside “SOC stabilization”. Our intention was to emphasize that EPS-C, as a microbial product, is dynamic in nature but can contribute to stabilization through indirect physical mechanisms, such as promoting aggregation and mineral-organic associations (see also response **R4-6**, **R4-12** and **R4-20** on related

aspects). To clarify this, we have revised this sentence from “As a direct product of microbial anabolic metabolism, EPS-C represents a highly active soil C pool and promotes SOC storage and stabilization by improving soil structure through aggregation and via promoting the formation of mineral-organic complexes²⁸” to “**As a dynamic, microbially derived fraction of the soil C pool, CER-extracted EPS likely represents a transient and labile portion of SOC²⁸, yet it promotes SOC stabilization by enhancing soil aggregation and facilitating mineral-organic associations—functioning as a ‘glue’ that improves soil structure⁵” (Page 11, Lines 323-326).**

[Comment 69] Line 415-416: “amplifying effect on the soil C cycle”? – please stick to your data!

[Response 69] We acknowledge that the original expression “amplifying effect on the soil C cycle” was vague and overstated in the context of our dataset, and we have now revised the sentence accordingly to better reflect the evidence and avoid implying causality beyond what our data support.

Our intention was to highlight a possible functional role of EPS in modulating soil enzymatic activity (see also response **R1-13**, **R3-3**, and **R3-16**), which may in turn influence soil C cycling processes. For instance, in our study, we found slight positive correlations between EPS and β -glucosidase (BG) activity and exoglucanase-cellobiosidase (CEL) activity. In line with this, a review by De Beeck et al.³⁵ discusses how fungal EPS matrices can enhance the stability and activity range of extracellular enzymes under varying soil conditions (e.g., pH, temperature, and denaturing agents), potentially affecting biogeochemical processes. However, such interactions are highly enzyme- and species-specific, and do not always enhance activity.

Since we only observed a slight positive correlation, we revised this sentence from “Furthermore, EPS-C has an amplifying effect on the soil C cycle and soil functions, and plays key roles in soil erosion resistance” to “**EPS may also influence soil C cycling indirectly by enhancing the stability of extracellular enzymes and modulating their activity³⁵, consistent with the weak positive correlations observed with BG and CEL**”(Page 11, Lines 326-328) to briefly mention this potential function of EPS.

This revised version avoids the ambiguous term “amplifying effect” and better aligns with both our own data and the current state of knowledge.

[Comment 70] Line 416-419: Sorry, I can't follow this line – I can't find a prove of this statement within this manuscript!

[Response 70] We apologize for the confusion. In the original version of the manuscript, we only presented the relationship between EPS and soil water holding capacity in the Supplementary information, which made it difficult to verify the statement in the main text. To address this, we have now moved the relevant results into the main text and

included them in the heatmap of **Fig. 2c** and as a visualisation in the new **Fig. 9**. Please also see our previous response **R3-10** and **R4-53** on this. Correspondingly, we have revised this sentence from “One important functional aspect of EPS was shown here to be the key positive driver of soil water holding capacity, compared to weaker associations to SOC or soil texture (in the absence of direct measurements of soil porosity and pore size distribution)” to “**A key functional role was evident in the strong positive association between EPS and soil water holding capacity (Fig. 9a), highlighting its contribution to soil moisture retention**” (**Page 11, Lines 328-330**).

[Comment 71] Line 419- 420: Coming up with such statement after demonstrating such a low contribution to SOC is really confusing!

[Response 71] Thank you for your important observation. We agree that the original statement may appear contradictory given the relatively low contribution of EPS-C to total SOC shown in our results. Our intention was to emphasize that, despite its low quantitative contribution, EPS plays a disproportionate functional role in regulating soil structure and water retention, which can be critical under climate change scenarios. However, to avoid confusion, we have now revised the sentence from “Its regulatory impacts on SOC and soil water storage are especially important in view of climate change effects in agricultural systems⁹⁴” to “**Despite its constrained quantitative contribution to SOC, EPS may exert a disproportionate influence on soil structural properties and water dynamics, making its role particularly relevant under climate change²⁸**” (**Page 11, Lines 330-332**). See also our previous responses **R4-6**, **R4-12** and **R4-68**.

[Comment 72] Line 421-440: Why is this part of the discussion here – separate section – if the authors want to point out that necromass is important!

[Response 72] Thank you for your helpful comment. Our intention in this section was to compare the relative contributions of EPS and microbial necromass carbon (MNC) to SOC, as both represent distinct microbial pathways of SOC formation. However, in line with the valuable suggestions from you and reviewer 1 and 3 (see response **R1-31** and **R3-18**), we have substantially revised this paragraph from “In contrast to extracellular microbial residues, the contribution of cellular residues in the form of MNC to SOC was ~10 x that of estimated EPS-C on a continental scale, which is mainly due to differences in C allocation to extracellular versus cellular residue formation mechanisms and differences in their lability or stability⁴. EPS is an active pool that is selectively secreted by microorganisms in response to adverse environmental conditions, being limited in quantity, but otherwise exhibits relatively rapid decomposition^{6,56}. As the inevitable product of microbial turnover and death, MNC is rather recalcitrant against degradation, and by binding to soil mineral surfaces is

preserved in soil for prolonged periods to become an important source of SOC^{57, 58, 59}. This passivity and stability makes the contribution of MNC to SOC significantly higher than that of EPS-C. Interestingly, we found a strong positive relationship between EPS-C and MNC, which may be due to them being both outputs of microbial anabolic metabolism, active microbes producing EPS and replicating, with microbial cells turning into necromass by different processes⁹⁵. In addition, EPS is produced by microbes as a glue for attachment of their cells and colonies to soil surfaces⁷⁶, which after death cause that necromass fragments are cemented via EPS to soil minerals or organic surfaces which has been shown microscopically^{85, 86}. This also promotes the collinearity between EPS-C and MNC, and with SOC. Overall, EPS therefore likely plays a double ‘sticky’ role, on the one hand promoting cell and necromass attachment to (mineral) surfaces stabilizing SOC via necromass accumulation, and on the other hand by stimulating the aggregation of particulate organic matter and of mineral associated organic matter (including MNC-laden mineral particles) into micro- and macroaggregates, thereby also promoting SOC stabilization on a higher level” to “In contrast to extracellular microbial residues, cellular residues (as MNC) contributed roughly 10 times more to SOC than estimated EPS-C across the large-scale transect. Incomplete extraction of EPS partly explains the difference between MNC and EPS-C observed here. Perhaps more importantly, amino sugar biomarkers do not exclusively represent necromass²⁹. Galactosamine, for example, can be found in substantial concentrations in plant residues, and other non-necromass materials, even EPS itself²⁹. Therefore, while measures of EPS-C are understood to be underestimates, measures of MNC are known to be overestimates³⁰ and perhaps with a larger margin for error and variability. This means that while on face value our results suggest that MNC was the larger pool of C, true values might be more comparable in size. More research is needed to elucidate with confidence whether EPS-C or MNC contributes more to SOC formation and stabilization.

EPS is a transient pool that is selectively secreted by microorganisms in response to adverse environmental conditions, with no measurable legacy effects observed beyond 2.5 years²⁸. In contrast, MNC as the inevitable product of microbial turnover and death, is more resistant to degradation, with turnover times ranging from multiple years to decades^{55, 60}. Interestingly, we found a strong positive relationship between EPS-C and MNC (Fig. 6g), which may be due to them being both outputs of microbial anabolic metabolism, active microbes producing EPS and replicating, with microbial cells turning into necromass via different pathways⁹⁵. In addition, EPS is produced by microbes as a “glue” for attachment of their cells and colonies to soil surfaces⁷⁶, which, after death, cause necromass fragments to be cemented via EPS to soil minerals or organic surfaces, which has been shown microscopically^{85, 86}. This also promotes the strong collinearity between EPS-C and MNC, MBC and SOC (Fig. 6). Overall, EPS

therefore likely plays a double ‘sticky’ role, on the one hand promoting cell and necromass attachment to (mineral) surfaces stabilizing SOC via necromass accumulation, and on the other hand by stimulating the aggregation of particulate organic matter and of mineral associated organic matter (including MNC-laden mineral particles) into micro- and macroaggregates, thereby also promoting SOC stabilization on a higher level” and fully integrated it into the existing discussion section rather than presenting it as a separate part. We believe this integration improves the flow and coherence of the argument (Pages 11-12, Lines 333-357).

[Comment 73] Line 421-446: Stick to your data and not conclude assumptions! Rewrite and condense

[Response 73] In response to your comment, we have carefully and substantially revised and condensed the paragraph as indicated in our response **R4-72** above. Please refer to our response above for details (Pages 11-12, Lines 333-357).

[Comment 74] Line 477-478: Why do the authors think so? If EPS is a kind of survival strategy C would rather allocate into EPS, then growth, or? How do the authors come up with their assumption – based on what?

[Response 74] Thank you for your thoughtful comment. We acknowledge the need to clarify our reasoning. Our intention was to convey that EPS is a costly high-molecular-weight compound that microbes tend to produce as a survival mechanism under environmental stress. In contrast, under favourable conditions characterized by high carbon use efficiency (CUE) and rapid microbial growth, microorganisms may reduce their investment in EPS and keep EPS production at minimal levels necessary for cell attachment and cell-cell communication. Instead, they may prioritize carbon allocation to biosynthesis and proliferation. This interpretation is supported by studies indicating that microbial investment in extracellular products, including EPS, may decrease when growth conditions are favourable and carbon and nutrient availability is high³¹. Vice versa, microbial investment into EPS may increase under stress (e.g. water deficit) when growth declines because of biotic or environmental limitations. This means that a negative association between two traits or here between two microbial processes can be triggered by (i) a carbon allocation trade-off where the investment into one process limits the investment into another one, due to limited resources for C allocation, or (ii) that stress induces one process and at the same time impairs another one, the responses not driven by resource limitation and reciprocal allocation trade-offs. A positive association between traits or microbial processes suggests (iii) trait covariance, or trait coordination where investment in one trait tends to be coupled with investment in another. In the stress response mechanism, microbial growth would decrease while EPS secretion (not measured here) would increase, both in absolute terms (on a soil dry mass

basis) as well as on relative terms (on a microbial biomass basis). This cannot be distinguished from a C allocation trade-off, where environments favouring one process would reduce the allocation to the other process. C allocation trade-offs are thought to only come into play when each process individually can consume a large fraction of available C resource and thereby reduce allocation to another process, a topic that is true for microbial growth but this is as yet unknown for EPS-C allocation (does it account for a substantial fraction of microbial C uptake under specific environmental conditions). Without real measurements of *in situ* EPS production rates by soil microbes using isotope tracing and without paired quantitative estimates of microbial growth or microbial C uptake as the 100% influx to be partitioned between competing C allocation processes this will not be resolved.

We have revised the sentence from “For instance under conditions of high growth and high CUE, microbes may prioritize C allocation to biosynthesis for growth processes, eventually reducing the C flow to EPS formation and secretion” to “**In line with EPS secretion being typically stress-induced⁵, EPS-C/MBC ratios were lower under non-stressed conditions (Fig. 10), where high microbial growth favours C allocation toward biomass rather than EPS. The negative relationships between EPS-C/MBC and microbial growth (C_{growth}) and CUE thus indicate a shift in microbial C allocation from growth to non-growth anabolic processes such as EPS production³¹. This strategy, adopted by soil microorganisms to secrete EPS or invest in growth or other C allocation processes is therefore likely influenced by their physiological state and C allocation trade-offs. However, this trade-off does not contradict the positive correlation between EPS and MBC: while the relative allocation to EPS may decrease (lower EPS-C/MBC), absolute EPS production can remain high when microbial biomass is large, as more cells collectively produce more EPS, forming a matrix for cell attachment and biofilm formation**” to reflect this nuance more clearly and to ensure that it is grounded in existing literature rather than speculative interpretation (**Page 13, Lines 388-398**). Please also see our responses on EPS and microbial C allocation, stress responses, and trade-offs versus trait covariance in **R1-41** and **R4-3**, and below in **R4-75**.

[Comment 75] Line 484-485 and Line 492: Prove that – refer to data and how do the authors interpret the positive correlation between MBC and EPS, the higher the microbial biomass the higher EPS – as shown by your data – how is that fitting to your argumentation?

[Response 75] Thank you for your insightful comment. We agree that the positive correlation between microbial biomass carbon (MBC) and extracellular polymeric substances (EPS) observed in our data reflects that higher microbial biomass generally

corresponds to greater overall EPS production potential. This makes intuitive sense, as more microbes can secrete more EPS in total.

However, we also found that under stressful conditions (low WHC and SMC), such as in cropland soils where MBC is relatively low (**Supplementary Fig. 2**), the ratio of EPS-C to MBC (EPS-C/MBC) was significantly higher (Supplementary Table 5). This suggests that microbes facing environmental stress allocate a larger proportion of their C towards EPS production as a survival strategy, even when their overall abundance is limited. Thus, EPS-C/MBC ratio increases under stress despite lower total biomass.

Supplementary Figure 2 | Linear regressions between microbial biomass carbon (MBC) and soil water properties. a Relationship between MBC and soil moisture content. **b** Relationship between MBC and water holding capacity. Blue lines indicate linear regression fits with 95% confidence intervals (shaded). The R^2 and p -value for each regression are shown on each panel. Each point represents an individual soil sample.

Based on this, we have revised the original sentences from “EPS production efficiency was highest under stressful conditions, where microbial growth, microbial biomass and CUE were low” to “**Although our data show a positive correlation between MBC and total EPS, suggesting that greater microbial biomass increases the overall potential for EPS production, we also observed higher EPS-C/MBC ratios under stressful conditions, such as in soils on sedimentary bedrock. This indicates that despite lower microbial abundance, microbes may allocate a greater proportion of their C to EPS as a survival strategy under environmental stress**” to better reflect these findings (**Page 13, Lines 382-386**).

Regarding Line 492, based on another reviewer’s suggestion (**R1-45** and **R3-21**), we have removed the sentence “Instead, C and nutrient limitation promote EPS production in aquatic and soil microbial communities, though the mechanistic

underpinning remains to be resolved in soils” to avoid unsupported assumptions and to better align the discussion with the data.

[Comment 76] Line 544-546: It is still unclear how many samples per site are analyzed. What forms the replicates? How where the cluster build, based on which assumptions? It is not enough to give citation in regard to its importance.

[Response 76] We have now revised the sentence from “Each soil sample was composited from five replicates. In total, soils from 92 sites were sampled, with 23 site clusters (69 soils) including woodland, grassland, and cropland soils ⁶¹” to “**Soil samples were taken from May to August 2017, at the peak of the vegetation season across a European climate transect. The sampled soils differed in terms of soil parent material, land use, and vegetation. At each site, samples of mineral topsoil (0-15 cm) were taken with a soil corer (5 cm). Each bulk soil sample consisted of five sub-samples taken at ~5 m distance from each other. These five sub-samples were composited and then sieved to give one mixed mineral soil sample per site. Therefore, no within-site replicates were analysed. Where possible, more than one and up to all three land uses (woodland/forest, grassland, and cropland) were sampled in close vicinity, termed a “region” or site cluster. In total, we sampled 92 mineral topsoils (92 sites) from 39 geographically distinct regions (a region characterized by the same climate and bedrock but sites with different land use). In 23 regions (site clusters), we were able to sample soils from woodland (forest), grassland, and cropland areas in close proximity (Supplementary Dataset 1)**” to more clearly explain the number of samples per site, how replicates were taken and composited, and how site clusters were constructed **(Page 14, Lines 430-440).**

[Comment 77] Line550: Which data have been taken within these study and which data come from other studies? Needs clarification.

[Response 77] Thank you for pointing this out. In this study, we used environmental and microbial parameters together with soil EPS. EPS was extracted and quantified from air-dried soil samples, which were collected in 2017, stored in air-dried form, and analyzed for EPS in 2024. Other microbial parameters were measured immediately after soil sampling. To clarify the sources of data, we have added these sentence “**Soil samples were taken from May to August 2017, at the peak of the vegetation season across a European climate transect**” **(Page 14, Lines 430-431)**, “**A comprehensive set of soil physicochemical, plant, and microbial parameters was used in the analyses, compiled from both previously published sources⁶¹ and unpublished datasets within the same project framework**” **(Pages 14-15, Lines 446-448)**, and “**Microbial process data, such as microbial growth (C_{growth}) and carbon use efficiency (CUE), exoenzymes, and microbial C, N, and P were measured on fresh soils directly after collection**” **(Page 15,**

Lines 474-476), and “Soil EPS (including polysaccharides and proteins) was extracted from air-dried soil samples in 2024 using cation exchange resin (CER)³⁶” (Page 16, Lines 490-491) to the **Methods Section** to clarify this point. See also responses **R3-5**, **R3-23**, and **R4-3**.

[Comment 78] Line 543-544: The authors need to clarify when the analyses were done (sampling in 2017) and if the storage of the soil samples can change EPS concentrations. Microbial parameters are variable depending on the soil storage. Please clarify! How comparable are EPS extraction from dry and wet soil samples? For dry samples usually provide higher SOC extracted due to the dead organisms – don’t you extract completely different substances?

[Response 78] The soil samples used for EPS measurements were originally collected in 2017 and stored air-dried at room temperature until EPS analysis (2024).

Thank you for raising this important point regarding the storage duration and the influence of soil air-drying on EPS quantification, which we also addressed in responses **R3-11** and **R4-3**. To address this concern, we conducted both a literature review and supplementary experiments. Zhang et al.³⁷ investigated EPS extraction from both fresh (moist) and air-dried remoistened soils and found that EPS contents extracted from dried soils were lower than those from moist soils, with reductions ranging approximately from 6% to 23% depending on the EPS component.

Soil types	Soil status	EPS polysaccharide	EPS uronic acid	EPS protein	EPS sum
		(μg g ⁻¹ soil)			
Ultisol; arable land	Moist	122.1 ± 16.7	41.9 ± 2.6	86.6 ± 4.8	250.6
Ultisol; arable land	Dried	153.7 ± 12.9	25.7 ± 5.5	75.5 ± 5.1	254.9
Change due to drying		↑ 26%	↓ 39%	↓ 13%	↑ 2%

To address concerns regarding the potential effects of air-drying and long-term storage, we conducted additional experiments comparing EPS extractions from fresh soils, air-dried soils, and soils stored for approximately four years. We found that: for EPS polysaccharides, extraction from air-dried soils showed a mean difference of -1% ± 3% (mean ± SE, n = 7; range: -10% to +11%) relative to fresh soils. In contrast, EPS protein extraction from air-dried soils showed a mean difference of +9% ± 1% (mean ± SE, n = 5; range: +7% to +14%) relative to fresh soils across cropland, grassland, and forest soils (see Supplementary Table 3).

Unfortunately, EPS was not measured in 2017 when the samples in this study were originally collected, so we could not directly assess long-term storage effects on these exact soils. However, we re-analyzed EPS in two air-dried forest soils collected in

China. The first was sampled in 2021 in Dongtai City and reported previously ⁴, and the second was collected in autumn 2021 from Nanjing for an unpublished earthworm microcosm experiment. After four years of air-dry storage at room temperature (2021-2025), both soils showed only minor changes, with EPS polysaccharides and EPS proteins increasing by 1–10% relative to fresh soils (Supplementary Table 4). These results indicate that long-term storage had negligible effects on EPS measurements. See also our response **R3-11** and **R4-3**.

These findings, together with the supporting literature, suggest that the EPS contents extracted from our air-dried soils are reliable and comparable.

In response, we have added a sentence to the end of the **Methods Section** to clarify this and refer readers to the **Supplementary Information** where full details of the procedures and results are provided. The added sentence is: “**Furthermore, to ensure the robustness of EPS measurements from air-dried samples used in this study, we additionally tested whether soil moisture status and storage duration influenced EPS extraction. The procedures and results are presented in Supplementary Tables 3 and 4 and discussed in Supplementary Discussion 2**” (**Page 16, Lines 499-502**).

[Comment 79] Line 592-596: I shortly checked the method by Redmile-Gordon and the amount of CER added to fresh soil is depending on the SOC content. Is it correct that the authors haven't done that!?

[Response 79] As correctly pointed out, the original method by Redmile-Gordon et al. adjusts the amount of CER added to fresh soil based on SOC content ³⁶. We followed this recommendation and initially calculated the required CER amount for all our samples based on their measured SOC concentrations. We confirmed that the use of 10 g CER per 1 g dry soil provided sufficient exchange capacity even for soils with the highest SOC levels in our dataset. Therefore, we standardized this amount (10 g CER per 1 g soil) across all samples to ensure comparability, while maintaining extraction efficiency. Please also refer to response **R4-30**.

We have now clarified this point more explicitly in the manuscript by adding the following sentence to the **Methods Section**: “**Before extraction, we also calculated the required amount of CER for each soil sample based on its SOC content following the method proposed by Redmile-Gordon et al.³⁶. These calculations showed that 10 g CER per 1 g dry soil provided sufficient ion-exchange capacity even for the samples with the highest SOC levels, and importantly for soils with the highest exchangeable Ca²⁺ concentrations. Therefore, we applied this amount uniformly across all soils to ensure methodological consistency across SOC-rich and SOC-poor soils and limestone and non-limestone soils**” (**Page 16, Lines 502-508**).

[Comment 80] Line 622-626: The estimation of EPS-C is a central part of the

manuscript. The authors need to give more information on how the EPS-C estimation (calculation) has been done instead of referring to a reference. It is necessary to a reader to be able to follow and understand the manuscript.

[Response 80] Thank you for this important suggestion. In response, we have revised **Methods section (Pages 17-18, Lines 541-571)** to include a detailed description of the EPS-C estimation procedure, with conversion factors and formula specified (see also response **R4-4**).

Cited literature:

1. Oliva RL, Vogt C, Bublitz TA, Camenzind T, Dyckmans J, Joergensen RG. Galactosamine and mannosamine are integral parts of bacterial and fungal extracellular polymeric substances. *ISME Commun* **4**, ycae038 (2024).
2. Redmile-Gordon MA, Evershed RP, Hirsch PR, White RP, Goulding KWT. Soil organic matter and the extracellular microbial matrix show contrasting responses to C and N availability. *Soil Biol Biochem* **88**, 257-267 (2015).
3. Lehmann A, Zheng W, Rillig MC. Soil biota contributions to soil aggregation. *Nat Ecol Evol* **1**, 1828-1835 (2017).
4. Shi K, *et al.* Accumulation of soil microbial extracellular and cellular residues during forest rewilding: Implications for soil carbon stabilization in older plantations. *Soil Biol Biochem* **188**, 109250 (2024).
5. Zhang M, Wu Y, Qu C, Huang Q, Cai P. Microbial extracellular polymeric substances (EPS) in soil: From interfacial behaviour to ecological multifunctionality. *Geo-Bio Interfaces* **1**, e4 (2024).
6. Flemming H-C. The perfect slime. *Colloids Surf, B* **86**, 251-259 (2011).
7. Flemming H-C, Wingender J. The biofilm matrix. *Nat Rev Microbiol* **8**, 623-633 (2010).
8. Costa OY, Raaijmakers JM, Kuramae EE. Microbial extracellular polymeric substances: ecological function and impact on soil aggregation. *Front Microbiol* **9**, 1636 (2018).
9. Kidinda LK, Babin D, Doetterl S, Kalbitz K, Mujinya BB, Vogel C. Extracellular polymeric substances are closely related to land cover, microbial communities, and enzyme activity in tropical soils. *Soil Biol Biochem* **187**, 109221 (2023).
10. Wang Z, *et al.* Human-induced erosion has offset one-third of carbon emissions from land cover change. *Nat Clim Change* **7**, 345-349 (2017).
11. Huang L, *et al.* A review of the role of extracellular polymeric substances (EPS) in wastewater treatment systems. *Int J Environ Res Public Health* **19**, 12191 (2022).
12. Ling Q, *et al.* Bedrock geochemistry regulates glomalin-related soil protein accrual in subtropical karst forest soils, Southwest China. *Ecol Indic* **176**, 113680 (2025).
13. Wu Y, *et al.* Soil biofilm formation enhances microbial community diversity and

- metabolic activity. *Environ Int* **132**, 105116 (2019).
14. Henao LJ, Mazeau K. Molecular modelling studies of clay–exopolysaccharide complexes: soil aggregation and water retention phenomena. *Materials Science and Engineering: C* **29**, 2326-2332 (2009).
 15. Wang X, Sharp CE, Jones GM, Grasby SE, Brady AL, Dunfield PF. Stable-isotope probing identifies uncultured Planctomycetes as primary degraders of a complex heteropolysaccharide in soil. *Appl Environ Microbiol* **81**, 4607-4615 (2015).
 16. Folk RL. The natural history of crystalline calcium carbonate; effect of magnesium content and salinity. *J Sediment Res* **44**, 40-53 (1974).
 17. Ott R, Gallen SF, Helman D. Erosion and weathering in carbonate regions reveal climatic and tectonic drivers of carbonate landscape evolution. *Earth Surf Dyn* **11**, 247-257 (2023).
 18. Jones DL, Cooledge EC, Hoyle FC, Griffiths RI, Murphy DV. pH and exchangeable aluminum are major regulators of microbial energy flow and carbon use efficiency in soil microbial communities. *Soil Biol Biochem* **138**, 107584 (2019).
 19. Heinze S, Raupp J, Joergensen RG. Effects of fertilizer and spatial heterogeneity in soil pH on microbial biomass indices in a long-term field trial of organic agriculture. *Plant Soil* **328**, 203-215 (2010).
 20. Zeng L, *et al.* Bedrock and climate jointly control microbial necromass along a subtropical elevational gradient. *Appl Soil Ecol* **189**, 104902 (2023).
 21. Chenu C. Extracellular polysaccharides: an interface between microorganisms and soil constituents. *Environmental impact of soil component interactions* **1**, 217-233 (1995).
 22. Littke R, Zieger L. Formation of organic-rich sediments and sedimentary rocks. *Hydrocarbons, Oils and Lipids: Diversity, Origin, Chemistry and Fate*, 475-492 (2020).
 23. Flemming H-C, Neu TR, Wozniak DJ. The EPS matrix: the “house of biofilm cells”. *J Bacteriol* **189**, 7945-7947 (2007).
 24. Dittrich F, Klaes B, Brandt L, Groschopf N, Thiele-Bruhn S. The stonesphere in agricultural soils: A microhabitat associated with rock fragments bridging rock and soil. *Eur J Soil Sci* **75**, e70025 (2024).
 25. White AF, Brantley SL. The effect of time on the weathering of silicate minerals: why do weathering rates differ in the laboratory and field? *Chem Geol* **202**, 479-506 (2003).
 26. Penman DE, Rugenstein JKC, Ibarra DE, Winnick MJ. Silicate weathering as a feedback and forcing in Earth's climate and carbon cycle. *Earth Sci Rev* **209**, 103298 (2020).
 27. Zethof JH, *et al.* Prokaryotic community composition and extracellular polymeric substances affect soil microaggregation in carbonate containing semiarid grasslands.

- Front Environ Sci* **8**, 51 (2020).
28. Redmile-Gordon M, Gregory AS, White RP, Watts CW. Soil organic carbon, extracellular polymeric substances (EPS), and soil structural stability as affected by previous and current land-use. *Geoderma* **363**, 114143 (2020).
 29. Salas E, *et al.* Reevaluation and novel insights into amino sugar and neutral sugar necromass biomarkers in archaea, bacteria, fungi, and plants. *Sci Total Environ* **906**, 167463 (2024).
 30. Meng D, *et al.* Refining Amino Sugar-Based Conversion Factors for Quantification of Microbial Necromass Carbon in Soils. *Global Change Biol* **31**, e70443 (2025).
 31. Bölscher T, *et al.* Beyond growth: The significance of non-growth anabolism for microbial carbon-use efficiency in the light of soil carbon stabilisation. *Soil Biol Biochem* **193**, 109400 (2024).
 32. Wingender J, Neu TR, Flemming H-C. *What are bacterial extracellular polymeric substances?* Springer (1999).
 33. Zhang P, *et al.* Extracellular protein analysis of activated sludge and their functions in wastewater treatment plant by shotgun proteomics. *Sci Rep* **5**, 12041 (2015).
 34. Dubé C-D, Guiot SR. Characterization of the protein fraction of the extracellular polymeric substances of three anaerobic granular sludges. *AMB Express* **9**, 23 (2019).
 35. De Beeck MO, Persson P, Tunlid A. Fungal extracellular polymeric substance matrices—highly specialized microenvironments that allow fungi to control soil organic matter decomposition reactions. *Soil Biol Biochem* **159**, 108304 (2021).
 36. Redmile-Gordon MA, Brookes PC, Evershed RP, Goulding KWT, Hirsch PR. Measuring the soil-microbial interface: extraction of extracellular polymeric substances (EPS) from soil biofilms. *Soil Biol Biochem* **72**, 163-171 (2014).
 37. Zhang M, *et al.* Characterising soil extracellular polymeric substances (EPS) by application of spectral-chemometrics and deconstruction of the extraction process. *Chem Geol* **618**, 121271 (2023).
 38. Wang S, *et al.* Extraction of extracellular polymeric substances (EPS) from red soils (Ultisols). *Soil Biol Biochem* **135**, 283-285 (2019).
 39. Vuko M, Cania B, Vogel C, Kublik S, Schloter M, Schulz S. Shifts in reclamation management strategies shape the role of exopolysaccharide and lipopolysaccharide-producing bacteria during soil formation. *Microb Biotechnol* **13**, 584-598 (2020).
 40. Hale L, Curtis D, Leon N, McGiffen Jr M, Wang D. Organic amendments, deficit irrigation, and microbial communities impact extracellular polysaccharide content in agricultural soils. *Soil Biol Biochem* **162**, 108428 (2021).
 41. Crouzet O, *et al.* Soil photosynthetic microbial communities mediate aggregate stability: influence of cropping systems and herbicide use in an agricultural soil. *Front Microbiol* **10**, 1319 (2019).

42. Luo Y, *et al.* Effects of different soil organic amendments (OAs) on extracellular polymeric substances (EPS). *Eur J Soil Biol* **121**, 103624 (2024).
43. Bettermann A, *et al.* Importance of microbial communities at the root-soil interface for extracellular polymeric substances and soil aggregation in semiarid grasslands. *Soil Biol Biochem* **159**, 108301 (2021).
44. Baumert VL, *et al.* Root-induced fungal growth triggers macroaggregation in forest subsoils. *Soil Biol Biochem* **157**, 108244 (2021).
45. Chen Y, *et al.* Quantifying the trade-off between yield and contamination in soil EPS extraction using cation exchange resin. *J Soils Sediments*, 1-9 (2025).
46. Bublitz TA, Oliva RL, Hupe A, Joergensen RG. Optimization of the bicinechonic acid assay for quantifying carbohydrates of soil extracellular polymeric substances. *Plant Soil* **498**, 699-709 (2024).
47. Bérard A, *et al.* Exopolysaccharides in the rhizosphere: A comparative study of extraction methods. Application to their quantification in Mediterranean soils. *Soil Biol Biochem* **149**, 107961 (2020).
48. Redmile-Gordon M, Chen L. Zinc toxicity stimulates microbial production of extracellular polymers in a copiotrophic acid soil. *Int Biodeterior Biodegrad* **119**, 413-418 (2017).
49. Liao J, *et al.* Unreported role of earthworms as decomposers of soil extracellular polymeric substance. *Appl Soil Ecol* **197**, 105325 (2024).
50. Liu C, *et al.* Arbuscular mycorrhizal fungi hyphal density rather than diversity stimulates microbial necromass accumulation after long-term Robinia pseudoacacia plantations. *Soil Biol Biochem* **206**, 109817 (2025).
51. Bragg JG, Wagner A. Protein carbon content evolves in response to carbon availability and may influence the fate of duplicated genes. *Proceedings of the Royal Society B: Biological Sciences* **274**, 1063-1070 (2007).
52. Zunino H, Borie F, Aguilera S, Martin JP, Haider K. Decomposition of ¹⁴C-labeled glucose, plant and microbial products and phenols in volcanic ash-derived soils of Chile. *Soil Biol Biochem* **14**, 37-43 (1982).
53. Martin JP, Haider K, Farmkr W, Fustec-Mathon E. Decomposition and distribution of residual activity of some ¹³C-microbial polysaccharides and cells, glucose, cellulose and wheat straw in soil. *Soil Biol Biochem* **6**, 221-230 (1974).
54. Stott DE, Kassim G, Jarrell W, Martin JP, Haider K. Stabilization and incorporation into biomass of specific plant carbons during biodegradation in soil. *Plant Soil* **70**, 15-26 (1983).
55. Throckmorton HM, Bird JA, Dane L, Firestone MK, Horwath WR. The source of microbial C has little impact on soil organic matter stabilisation in forest ecosystems. *Ecol Lett* **15**, 1257-1265 (2012).
56. Flemming H-C, Wingender J, Szewzyk U, Steinberg P, Rice SA, Kjelleberg S.

- Biofilms: an emergent form of bacterial life. *Nat Rev Microbiol* **14**, 563-575 (2016).
57. Wu H, *et al.* Soil microbial necromass: The state-of-the-art, knowledge gaps, and future perspectives. *Eur J Soil Biol* **115**, 103472 (2023).
58. Zhu X, Jackson RD, DeLucia EH, Tiedje JM, Liang C. The soil microbial carbon pump: From conceptual insights to empirical assessments. *Global Change Biol* **26**, 6032-6039 (2020).
59. Liang C, Amelung W, Lehmann J, Kästner M. Quantitative assessment of microbial necromass contribution to soil organic matter. *Global Change Biol* **25**, 3578-3590 (2019).
60. Liu X, *et al.* Linking microbial immobilization of fertilizer nitrogen to in situ turnover of soil microbial residues in an agro-ecosystem. *Agric Ecosyst Environ* **229**, 40-47 (2016).
61. Noll L, Zhang S, Zheng Q, Hu Y, Hofhansl F, Wanek W. Climate and geology overwrite land use effects on soil organic nitrogen cycling on a continental scale. *Biogeosciences* **19**, 5419-5433 (2022).
62. Brookes PC, Landman A, Pruden G, Jenkinson DS. Chloroform fumigation and the release of soil nitrogen: a rapid direct extraction method to measure microbial biomass nitrogen in soil. *Soil Biol Biochem* **17**, 837-842 (1985).
63. Zhang S, Zheng Q, Noll L, Hu Y, Wanek W. Environmental effects on soil microbial nitrogen use efficiency are controlled by allocation of organic nitrogen to microbial growth and regulate gross N mineralization. *Soil Biol Biochem* **135**, 304-315 (2019).
64. Zheng Q, *et al.* Growth explains microbial carbon use efficiency across soils differing in land use and geology. *Soil Biol Biochem* **128**, 45-55 (2019).
65. Jones DB. *Factors for converting percentages of nitrogen in foods and feeds into percentages of proteins*. US Department of Agriculture Washington, DC (1941).
66. Mariotti F, Tomé D, Mirand PP. Converting nitrogen into protein—beyond 6.25 and Jones' factors. *Crit Rev Food Sci Nutr* **48**, 177-184 (2008).
67. Baveye PC. “Soil biofilms”: Misleading description of the spatial distribution of microbial biomass in soils. *Soil Ecology Letters* **2**, 2-5 (2020).
68. Buckeridge KM, Creamer C, Whitaker J. Deconstructing the microbial necromass continuum to inform soil carbon sequestration. *Funct Ecol* **36**, 1396-1410 (2022).
69. Adessi A, de Carvalho RC, De Philippis R, Branquinho C, da Silva JM. Microbial extracellular polymeric substances improve water retention in dryland biological soil crusts. *Soil Biol Biochem* **116**, 67-69 (2018).
70. Ates O. Systems biology of microbial exopolysaccharides production. *Front Bioeng Biotechnol* **3**, 200 (2015).
71. Mikutta R, Zang U, Chorover J, Haumaier L, Kalbitz K. Stabilization of extracellular polymeric substances (*Bacillus subtilis*) by adsorption to and coprecipitation with Al forms. *Geochim Cosmochim Acta* **75**, 3135-3154 (2011).

72. Knicker H, Schmidt MW, Kögel-Knabner I. Nature of organic nitrogen in fine particle size separates of sandy soils of highly industrialized areas as revealed by NMR spectroscopy. *Soil Biol Biochem* **32**, 241-252 (2000).
73. Gunina A, Kuzyakov Y. Sugars in soil and sweets for microorganisms: review of origin, content, composition and fate. *Soil Biol Biochem* **90**, 87-100 (2015).
74. Gleixner G, Bol R, Balesdent J. Molecular insight into soil carbon turnover. *Rapid Commun Mass Spectrom* **13**, 1278-1283 (1999).
75. Hubbert K, Graham R, Anderson M. Soil and weathered bedrock: components of a Jeffrey pine plantation substrate. *Soil Sci Soc Am J* **65**, 1255-1262 (2001).
76. More TT, Yadav JSS, Yan S, Tyagi RD, Surampalli RY. Extracellular polymeric substances of bacteria and their potential environmental applications. *J Environ Manage* **144**, 1-25 (2014).
77. Tsuneda S, Aikawa H, Hayashi H, Yuasa A, Hirata A. Extracellular polymeric substances responsible for bacterial adhesion onto solid surface. *FEMS Microbiol Lett* **223**, 287-292 (2003).
78. Stewart TJ, Traber J, Kroll A, Behra R, Sigg L. Characterization of extracellular polymeric substances (EPS) from periphyton using liquid chromatography-organic carbon detection–organic nitrogen detection (LC-OCD-OND). *Environ Sci Pollut Res* **20**, 3214-3223 (2013).
79. Li J, Chen Y, Qi J, Zuo X, Meng F. Characterization of EPS subfractions from a mixed culture predominated by partial-denitrification functional bacteria. *Water Res X* **24**, 100250 (2024).
80. Xu C, *et al.* Comparison of microgels, extracellular polymeric substances (EPS) and transparent exopolymeric particles (TEP) determined in seawater with and without oil. *Mar Chem* **215**, 103667 (2019).
81. Xu C, *et al.* The role of microbially-mediated exopolymeric substances (EPS) in regulating Macondo oil transport in a mesocosm experiment. *Mar Chem* **206**, 52-61 (2018).
82. Geyik AG, Çeçen F. Production of protein-and carbohydrate-EPS in activated sludge reactors operated at different carbon to nitrogen ratios. *J Chem Technol Biotechnol* **91**, 522-531 (2016).
83. Sheng G-P, Yu H-Q, Li X-Y. Extracellular polymeric substances (EPS) of microbial aggregates in biological wastewater treatment systems: A review. *Biotechnol Adv* **28**, 882-894 (2010).
84. Oliva RL, Khadka UB, Camenzind T, Dyckmans J, Joergensen RG. Constituent of extracellular polymeric substances (EPS) produced by a range of soil bacteria and fungi. *BMC Microbiol* **25**, 298 (2025).
85. Liu S, *et al.* Early Triassic stromatolites from the Xingyi area, Guizhou Province, southwest China: geobiological features and environmental implications.

- Carbonates Evaporites* **32**, 261-277 (2017).
86. Dohnalkova AC, *et al.* Molecular and microscopic insights into the formation of soil organic matter in a red pine rhizosphere. *Soils* **1**, 4 (2017).
 87. Rillig MC, Mummey DL. Mycorrhizas and soil structure. *New Phytol* **171**, 41-53 (2006).
 88. Drenovsky RE, Steenwerth KL, Jackson LE, Scow KM. Land use and climatic factors structure regional patterns in soil microbial communities. *Global Ecol Biogeogr* **19**, 27-39 (2010).
 89. Donot F, Fontana A, Baccou J, Schorr-Galindo S. Microbial exopolysaccharides: main examples of synthesis, excretion, genetics and extraction. *Carbohydr Polym* **87**, 951-962 (2012).
 90. Jackson RB, Mooney HA, Schulze E-D. A global budget for fine root biomass, surface area, and nutrient contents. *Proceedings of the National Academy of Sciences* **94**, 7362-7366 (1997).
 91. Kabir Z. Tillage or no-tillage: impact on mycorrhizae. *Can J Plant Sci* **85**, 23-29 (2005).
 92. Santschi PH, *et al.* Can the protein/carbohydrate (P/C) ratio of exopolymeric substances (EPS) be used as a proxy for their 'stickiness' and aggregation propensity? *Mar Chem* **218**, 103734 (2020).
 93. Schwehr KA, *et al.* Protein: Polysaccharide ratio in exopolymeric substances controlling the surface tension of seawater in the presence or absence of surrogate Macondo oil with and without Corexit. *Mar Chem* **206**, 84-92 (2018).
 94. Jensen JL, Schjønning P, Watts CW, Christensen BT, Peltre C, Munkholm LJ. Relating soil C and organic matter fractions to soil structural stability. *Geoderma* **337**, 834-843 (2019).
 95. Camenzind T, Mason-Jones K, Mansour I, Rillig MC, Lehmann J. Formation of necromass-derived soil organic carbon determined by microbial death pathways. *Nat Geosci* **16**, 115-122 (2023).

Reviewer #1 – second-round comments

[Comment 1-1] Line 27: To say factors influencing EPS are "poorly quantified" can be interpreted in several ways. To avoid the derogatory sense and misunderstanding, I would suggest a change to "Yet, their abundance and large-scale environmental and microbial controls have only begun to be investigated." ... the subsequent sentence can then simply continue with "We conducted extensive soil sampling...". This solves the problems and emphasises the novelty of your work - without adding potential for misunderstanding.

[Response 1-1] Thank you very much for this excellent and constructive suggestion. Following this comment, together with the related suggestions raised in **Reviewer 1's Comment 1-7 on R4-7 and Reviewer 1's Comment 1-3 on R1-3 (Please see below)**, we have revised this sentence from "Yet, their abundance and large-scale environmental and microbial controls remain poorly quantified. We addressed this gap by conducting extensive soil sampling across a European transect spanning diverse climates, bedrock types, and land uses" to "Yet, their abundance and large-scale environmental and microbial controls have only begun to be investigated. We conducted extensive soil sampling across a European transect spanning diverse climates, bedrock types, and land uses" **(Please see the revised manuscript (clean version), Page 3, Lines 23-26, the same applies hereafter).**

[Comment 1-2] Line 92: No need to be negative here: "which has hindered understanding of large-scale controls of soil EPS". These works have not "hindered" understanding, on the contrary, your study is a logical progression from these sensible forerunning investigations. Please delete this unnecessary critique (also, the scale limitations of others' works are already made clear in lines 86 to 89).

[Response 1-2] Thank you for this constructive point. We agree that the wording could be perceived as overly critical of prior work. As suggested, we have deleted the sentence "which has hindered understanding of large-scale controls of soil EPS" to avoid any negative implication toward previous studies.

[Comment 1-3] Line 107: *There are several studies underway. This sentence does not make sense – presumably after the preceding changes were made. To correct, please change “remain largely unconstrained” to “are much needed” – highlighting the importance of this body of work.*

[Response 1-3] We have revised the wording from “remain largely unconstrained” to “**are much needed**” to emphasize the importance of ongoing studies better, as suggested (**Page 5, Line 76**).

[Comment 1-4] Lines 183 to 184 state: *“EPS proteins differed significantly across land use types (grassland > woodland 184 / cropland; Fig. 3d), while showing no significant variation with bedrock (Fig. 3c)”. Correct. However, currently the discussion appears to suggest that land-use has *no effect* which contradicts the present statement, and the state-of-the-art. This result (effect of land-use on EPS protein content) is significant and should be presented as being in accordance with ref#16 (effects of land-use on EPS protein and soil properties) in the discussion (proposed below):*

Line 308 – 309: currently in error, and can be improved. “Meanwhile, EPS contents did not differ significantly among land use types but tended to be highest in grasslands”. This is followed by speculation on mechanisms and a link to Supplementary Discussion 5 without clarifying the main effects (or comparing to relevant literature in the main discussion). Figure 3 shows 1) a statistically significant effect of [bedrock] on [EPS polysaccharide], and 2) a statistically significant effect of [land-use] on [EPS protein]. This should be much clearer in the discussion.

To resolve all of the issues above, please replace the following text “Meanwhile, EPS contents did not differ significantly among land use types but tended to be highest in grasslands, with microbial biomass, nutrient availability, and plant inputs mediating the land-use based variation; in contrast,” with “We found land-use was not a significant driver of EPS polysaccharide, but was a highly significant driver of EPS protein (Figure 3). This adds important depth to land-use research. Redmile-Gordon et al., (2020) 16 found land-use change to perennial grassland increased soil EPS and aggregate stability with EPS protein being far more influential for soil structural stability than EPS polysaccharide16. Our findings, that both EPS protein and EPS-C are elevated under grassland (across contrasting soil bedrocks) supports their assertion that CER-extracted EPS protein is more physically influential than polysaccharides in the stabilisation of soil aggregates16 - and thus also C stabilisation.” The narrative can then smoothly continue with “EPS-C/MBC ratios peaked in croplands, likely reflecting enhanced microbial EPS investment under water limitation (Supplementary Discussion 4)...

[Response 1-4] Thank you for this critical and constructive observation. In the original manuscript, we discussed the lack of significant effect of land use on EPS content as shown in Fig. 2b, but overlooked the significant effect of land use on EPS protein presented in Fig. 3d. We have revised the sentence from “Meanwhile, EPS contents did not differ significantly among land use types but tended to be highest in grasslands, with microbial biomass, nutrient availability, and plant inputs mediating the land-use based variation; in contrast” to “**We found**

land-use was not a significant driver of EPS polysaccharide, but was a highly significant driver of EPS protein (Fig.3). This adds important depth to land-use research. Redmile-Gordon et al.¹⁵ found that land-use change to perennial grassland increased soil EPS and aggregate stability, with EPS protein being far more influential for soil structural stability than EPS polysaccharides¹⁵. Our findings, that both EPS protein and EPS-C are elevated under grassland (across contrasting land management types), support their assertion that CER-extracted EPS protein is more physically influential than EPS polysaccharides in shaping soil aggregates¹⁵ - and thus also C stabilization (Supplementary Discussion 2)” (Page 11, Lines 296-303).

In addition, following *Reviewer 4’s Comment 4-6*, we have removed the sentence “EPS-C/MBC ratios peaked in croplands, likely reflecting enhanced microbial EPS investment under water limitation (Supplementary Discussion 4)”, which attributed elevated EPS-C/MBC ratios in croplands solely to water limitation. This removal avoids overinterpretation and ensures that the discussion consistently focuses on the contrasting responses of EPS polysaccharides and EPS proteins to bedrock and land-use types.

[Comment 1-5] Line 343 to 344: There has been a small mistake, currently misleading: “EPS is a transient pool that is selectively secreted by microorganisms in response to adverse environmental conditions, with no measurable legacy effects observed beyond 2.5 years¹⁶. The cited work (16) found higher EPS protein where there was no disturbance, certainly not “adverse conditions” – however, the study was not set up to determine effects of adversity. Also, there were no measurements conducted between 2.5 years and 50 – so these EPS could still have been measurable after 10 years, 20 years (we simply don’t know more than they are undetectable after 50). So the above statement is incorrect.

Accordingly, please delete all of the following, including the unfounded speculation about MNC being more resistant to decomposition (we simply don’t know yet until measurements of EPS at 10, 20, 30 years are conducted):

“EPS is a transient pool that is selectively secreted by microorganisms in response to adverse environmental conditions, with no measurable legacy effects observed beyond 2.5 years¹⁶. In contrast, MNC as the inevitable product of microbial turnover and death, is more resistant to degradation, with turnover times ranging from multiple years to decades^{43, 44}”.

Please also delete the paragraph break so that this discussion on EPS-C, MNC, and SOC formation is together.

[Response 1-5] Thank you for these meticulous comments. We fully understand the concern raised here. While this comment suggested removing the sentence, we note that in *Reviewer 1’s Comment 1-16 on R4-72 (Please see below)*, you provided detailed guidance on how it could be revised. Following this later suggestion, we have revised the sentence from “EPS is a transient pool that is selectively secreted by microorganisms in response to adverse environmental conditions, with no measurable legacy effects observed beyond 2.5 years¹. In contrast, MNC as the inevitable product of microbial turnover and death, is more resistant to degradation, with turnover times ranging from multiple years to decades^{2, 3}” to “EPS is a transient pool that is selectively secreted by microorganisms where sufficient labile C is available, with the majority of EPS either stabilising or being turned over in less than 2.5

years¹⁵” in the main text (Page 12, Lines 332-334), thereby retaining the key information more accurately and carefully. We have removed the paragraph break so that the discussion on EPS-C, MNC, and SOC formation is continuous.

[Comment 1-6] Finally: Supplementary discussion 5 has unsupported speculation on roles in soil structure: “EPS polysaccharides are closely associated with soil Ca²⁺, highlighting their role in reinforcing soil structure, whereas EPS proteins are more strongly linked to microbial biosynthetic activity (q_{Growth})”. While the association might be suggestive of something, such as dependency on Ca²⁺ (for production, structure, or otherwise) proteins are typically more formative in soil structural stability - and don’t require increased Ca²⁺ availability to fulfil that function¹⁶. For simplicity, please just delete the speculation “highlighting their role in soil structure” as it would not be appropriate to use an association with Ca²⁺ to infer structural relevance of EPS polysaccharide over EPS protein in this way.

[Response 1-6] Thank you for your helpful comment. We have revised this sentence from: “These results indicate that while both EPS polysaccharides and EPS proteins are influenced by microbial abundance and nitrogen availability (MBN) as well as soil C status (SOC), EPS polysaccharides are closely associated with soil Ca²⁺, highlighting their role in reinforcing soil structure, whereas EPS proteins are more strongly linked to microbial biosynthetic activity (q_{Growth})” to “These results indicate that while both EPS polysaccharides and EPS proteins are influenced by microbial abundance and nitrogen availability (MBN) as well as soil C status (SOC), EPS polysaccharides are closely associated with soil Ca²⁺, and EPS proteins are more strongly linked to microbial biosynthetic activity (q_{Growth})” in **Supplementary discussion 3** (Supplementary discussion 5 in the previous version).

Reviewer #3 – second-round comments

[Comment 3-1] I reviewed an earlier version of this work and found that the revisions have improved it. I have three remaining comments (and a few specific ones), listed below:

[Response 3-1] Thank you for your positive feedback on the revised manuscript and for providing your valuable comments. We have carefully considered all specific points and addressed them below.

[Comment 3-2] Soil moisture content (SMC) as predictor of EPS content: SMC at the time of measurement (L469-470) might not be representative of the soil moisture regime, or even the moisture levels in the previous week, so I am not sure it is a meaningful predictor of EPS. Also, SMC likely correlates with silt and clay content, as fine texture soils hold more water than sandy soils. I suspect the correlations found between EPS and SMC might be mirroring those with clay content (or water holding capacity, which might also be correlated with clay content). I get the impression that removing the EPS-SMC correlations (Figures 9-10) would not change the main messages.

[Response 3-2] Thank you for this insightful comment. We agree that SMC is highly contingent upon meteorological conditions, particularly recent rainfall, and can vary over hours or days. At the subcontinental scale, however, SMC still shows a robust relationship with the climatic moisture regime and with EPS-C/MBC (Fig. 9c). Based on the rationale that water deficit can be a key driver stimulating EPS-C/MBC, explained by EPS retaining water as a microbial response under drought⁴, and our findings with both positive relationships between EPS-C/MBC and both, ADI and SMC (Fig. 9c, 9d), we therefore suggest that SMC may influence EPS at the large scale. Nevertheless, we acknowledge your concern that SMC may be linked with soil clay content (and SOC content). Therefore, we have toned down the implication of SMC in the **Discussion** and now mostly relate it to ADI, which more stably reflects long-term climatic water deficit. Specifically, we have revised the sentence from “Although our data show a positive correlation between MBC and total EPS, suggesting that greater microbial biomass increases the overall potential for EPS production, we also observed higher EPS-C/MBC ratios under stressful conditions, such as in soils on sedimentary bedrock” to “**Although our data show a positive correlation between MBC and total EPS, higher EPS-C/MBC ratios were observed under stressful conditions, such as in soils developed from sedimentary bedrock. These soils are characterized by low WHC and SMC, likely resulting from their relatively low SOC content and relatively high sand content, and by relatively low ADI values, indicating drier climatic conditions (Supplementary Table 4)**” (Please see the revised manuscript (clean version), Page 13, Lines 371-375, the same applies hereafter). We have also revised the sentence from “Supporting this interpretation, microbial growth-related parameters (e.g., C_{growth} , MBC, and MBN) were consistently lowest in sedimentary soils, which also had the lowest WHC and SMC (Supplementary Table 5)” to “**Supporting this interpretation, microbial growth-related parameters (e.g., C_{growth} , MBC, and MBN) were consistently lowest in sedimentary soils, which were also characterized by the lowest SMC and relatively low ADI in our study (Supplementary Table 4), potentially reflecting water limitation, a factor suggested by previous studies to be associated with increased soil EPS²⁷**” (Page 10, Lines 281-285).

Additionally, following your comment and *Reviewer 4’s Comment 4-10*, we reduced the number of figures in the main manuscript by merging **Figs. 9 and 10** into a single figure (**new Fig. 9**), showing (i) the relationship between EPS and SMC and between EPS and WHC, and (ii) the relationship of EPS-C/MBC with SMC, ADI, C_{growth} , and CUE (Please see below). Furthermore, following *your Comment 3-4* (Please see below), EPS and EPS-C/MBC in the figure have been log-transformed, as indicated in the figure caption.

Figure 9. Linear regressions showing relationships between EPS and EPS-C/MBC and key climatic, soil, and microbial parameters. EPS and EPS-C/MBC values were log-transformed before regression. **a** Relationship between EPS and soil moisture content. **b** Relationship between EPS and soil water holding capacity. **c** Relationship between EPS-C/MBC and soil moisture content. **d** Relationship between EPS-C/MBC and aridity index. **e** Relationship between EPS-C/MBC and microbial growth. **f** Relationship between EPS-C/MBC and microbial carbon use efficiency. Blue lines indicate linear regression fits with 95% confidence intervals (shaded). The R^2 and p -value for each regression are shown on each panel. Each point represents an individual soil sample. EPS, extracellular polymeric substances. MBC, microbial biomass carbon (Page 23, Lines 718-726).

[Comment 3-3] Aridity index: since potential evapotranspiration (PET) has already been calculated for all the sites, I wonder why not defining aridity index in the usual way as ratio of mean annual precipitation (MAP) to mean annual PET? The current definition ($ADI=MAP/(MAT+33)$) yields a quantity that has strange dimensions of length per degree of temperature. (Please also note that the units should be reported when values of ADI defined in this way are reported or shown, as it is not a non-dimensional quantity.) I would suggest to use the usual ADI definition instead, to make comparisons with other studies simpler—though I doubt such a change would alter the results.

[Response 3-3] Thank you very much for this constructive comment. Following your suggestion, we recalculated the aridity index using the standard definition ($ADI = MAP / PET$) for all sites, where MAP is mean annual precipitation (mm yr^{-1}) and PET is mean annual potential evapotranspiration (mm yr^{-1}), calculated from daily PET values. All statistical analyses were repeated accordingly (**Figs. 2, 4, 7, 8, 9; Supplementary Figs. 5 and 6; Supplementary Tables 4-16**). The main statistical trends (directions and magnitude) and conclusions of our study remained essentially unchanged.

Specifically, we have revised the sentence from “An aridity index (ADI) was calculated as MAP divided by (MAT + 33) according to Quan et al⁵” to “**An aridity index (ADI) was calculated as the ratio of MAP to mean annual PET⁶⁰**” (**Page 14, Line 416**).

*[Comment 3-4] Figure 6: I wonder if power law relations might better represent the patterns in this figure, while also reducing heteroscedasticity issues (at least by eye it seems variations are higher at higher values of SOC, MBC, MNC). Also, power laws $y=a*x^b$ pass through the origin of the axes, while the linear regressions now predict non-zero MBC, MNC, and EPS-C even at zero SOC, which is not very meaningful.*

[Response 3-4] Thank you for this helpful suggestion. Following your advice, we re-analyzed the relationships in **Fig. 6** using power-law models by fitting linear regressions to log-transformed variables (log-log space). This has now been explicitly described in the figure caption (**Please see below**).

Figure 6. Linear relationships among extracellular polymeric substance carbon (EPS-C), microbial biomass carbon (MBC), microbial necromass carbon (MNC), and soil organic carbon (SOC). **a** Relationship between MBC and SOC. **b** Schematic illustration of the relationships among EPS-C, MNC, SOC, and MBC. **c** Relationship between MNC and SOC. **d** Relationship between MNC and MBC. **e** Relationship between EPS-C and SOC. **f** Relationship between EPS-C and MBC. **g** Relationship between EPS-C and MNC. Each point represents an individual soil sample. Different colors indicate different land use types, and different shapes indicate different bedrock types. **Black regression lines represent power-law relationships fitted as linear models in ln-ln space.** Corresponding regression equations, R^2 , and p -values are displayed on the plots (Page 22, Lines 685-686).

We also revised the description in the **Statistical analyses** section, changing it from “Using general linear models, we examined the relationships between EPS-C, MNC, MBC, and SOC (lm function in R)” to “**We examined the relationships between EPS-C, MNC, MBC, and SOC by fitting linear regressions to ln-transformed variables using the *lm* function in R**” (Page 19, Lines 577-579).

[Comment 3-5] L83-93: this part of the paragraph seems out of place, as it presents definitions of EPS metrics, while the rest of the paragraph is on EPS functions.

[Response 3-5] Thank you for pointing this out. We agree that this original part of the paragraph contained excessive explanation of EPS metrics (e.g., EPS/MBC ratio) and was not well aligned with the introduction of EPS functions.

To improve clarity and coherence, we have moved the following section from the **Introduction** to the **Methods**.

“The EPS production efficiency represents the amount of EPS secreted by microorganisms in soils and was calculated as EPS: microbial biomass carbon (MBC) ratio⁷. However, variable stabilization and decomposition processes occur after EPS production, potentially altering the ratio. Thus, while “EPS production efficiency” is a useful concept for characterizing short-term EPS responses of the soil microbial biomass (where stabilization/decomposition processes are assumed to be negligible)⁸ in studies of natural soils in comparative equilibrium (with time for soil-specific stabilization and decomposition processes to occur), this metric is better termed and viewed as the EPS/MBC ratio. This not only represents the current state of equilibrium more accurately, but is consistent with the findings that current methods of soil EPS measurement (by cation exchange resin (CER) extraction) do not quantify the decomposed or stabilized EPS fractions remaining after 2.5 years or more, and instead represent a transient-but more active- fraction of SOC^{1”}.

In the revised **Methods**, we describe how the EPS-C/MBC ratio was calculated and note its limitations. Specifically, we revised the text from: “Based on EPS-C, we calculated the ratio of EPS-C to MBC, while the EPS-C/SOC ratio was computed to represent the contribution of EPS-C to total soil organic carbon” to: “Based on EPS-C, we calculated the ratio of EPS-C to MBC. This EPS-C:MBC ratio has previously been termed “EPS production efficiency”, referring to the amount of EPS secreted by microorganisms in soils¹⁶. However, post-production processes, including EPS-soil associations and decomposition, can modify this ratio. While the concept is useful for characterizing short-term microbial EPS responses under conditions where these processes are negligible⁵¹, in natural soils, the metric is more appropriately referred to as the EPS-C/MBC ratio and should be interpreted with caution. Additionally, we calculated the EPS-C/SOC ratio to represent the contribution of EPS-C to total SOC” (Page 18, Lines 553-560). This revision simplifies the explanation, integrates the calculation with interpretation, and aligns the discussion with EPS functions more clearly.

In addition, to improve the logical flow of the **Introduction**, we revised the preceding sentence from: “The secretion of EPS typically occurs under specific environmental conditions and can be approximated by microbial EPS production efficiency” to: “The secretion of EPS is typically stimulated under specific environmental conditions^{1”} (Page 5, Lines 78-79).

[Comment 3-6] L94: not sure what “template” means in this context.

[Response 3-6] We have revised the sentence by replacing the term “template” with “**structural matrix**”, clarifying that EPS serves as a structural matrix that facilitates the activity of extracellular enzymes (Page 5, Line 80).

[Comment 3-7] L152: typo “not significantly”.

[Response 3-7] Done (Page 6, Line 134).

[Comment 3-8] L277: language here is a bit vague—what are the functional traits involved? They are listed thereafter, but as written, the statement ending in L277 appears not conclusive.

[Response 3-8] To improve clarity and precision, we revised the sentence from “Additionally, although our study does not directly include microbial community diversity or composition data, previous research has shown that lithology-driven geochemical processes – such as rock weathering and mineral-associated protection – can indirectly regulate the accumulation of microbial extracellular products (such as glomalin-related soil protein) via microbial functional traits”⁹ to “**Additionally, although our study does not directly include microbial community diversity or composition data, previous research has shown that lithology-driven geochemical processes – such as rock weathering and mineral-associated protection – can indirectly regulate microbial extracellular products (such as glomalin-related soil protein) by affecting microbial diversity and resource elemental stoichiometry**⁴¹” **(Page 10, Lines 268-272).**

[Comment 3-9] L358: I would refer to EPS-C/MBC ratio instead of “EPS production efficiency” as elsewhere in the manuscript.

[Response 3-9] Thank you for the helpful comment. In this sentence, “Changes in short-term microbial EPS production efficiency were previously calculated by standardizing soil EPS to soil microbial ATP⁸, a proxy for microbial biomass”, we intended to describe how EPS production efficiency has been quantified in previous short-term incubation studies. Specifically, Redmile-Gordon et al.⁸ defined EPS production efficiency as soil EPS standardized by microbial ATP. To ensure an accurate representation of the referenced work, we therefore retained the term “EPS production efficiency” in this context.

Regarding the term “EPS production efficiency”, we have carefully checked the entire manuscript. This term now appears only in four specific contexts **(Please see below)**:

- (1) “**In our study, this ratio does not directly reflect EPS production efficiency – since EPS contents in natural soils result from the balance of microbial production and degradation, modulated by mineral interactions and extraction limitations**” **(Page 10, Lines 277-279)**. Here, the term is used to clarify that the EPS-C/MBC ratio in our study cannot be interpreted as a direct measure of EPS production efficiency.
- (2) “**Changes in short-term microbial EPS production efficiency were previously calculated by standardizing soil EPS to soil microbial ATP**⁵¹, as an independent indicator of the soil microbial biomass” **(Page 12, Lines 346-348)**. Here, the term is used to accurately cite previous literature.
- (3) “Therefore, we present EPS-C/MBC ratios “as is” without inferring EPS production efficiency” **(Page 12, Lines 353-354)**. This emphasizes that EPS-C/MBC should not be interpreted as EPS production efficiency in our study.

- (4) “Based on EPS-C, we calculated the ratio of EPS-C to MBC. This EPS-C:MBC ratio has previously been termed “EPS production efficiency”, referring to the amount of EPS secreted by microorganisms in soils¹⁶. However, post-production processes, including EPS-soil associations and decomposition, can modify this ratio. While the concept is useful for characterizing short-term microbial EPS responses under conditions where these processes are negligible⁵¹, in natural soils, the metric is more appropriately referred to as the EPS-C/MBC ratio and should be interpreted with caution” (Page 18, Lines 553-559). Here, the use of the term serves to clarify that EPS-C/MBC in natural soils is not a direct measure of EPS production efficiency.

In all other parts of the manuscript, we consistently use the term “EPS-C/MBC ratio”.

[Comment 3-10] L384: why would conditions be more stressful in soils developed from sedimentary bedrock? I would remind of the possible reasons.

[Response 3-10] Thank you for this valuable comment. We have revised the sentence from “Although our data show a positive correlation between MBC and total EPS, suggesting that greater microbial biomass increases the overall potential for EPS production, we also observed higher EPS-C/MBC ratios under stressful conditions, such as in soils on sedimentary bedrock” to “~~Although our data show a positive correlation between MBC and total EPS, higher EPS-C/MBC ratios were observed under stressful conditions, such as in soils developed from sedimentary bedrock. These soils are characterized by low WHC and SMC, likely resulting from their relatively low SOC content and relatively high sand content, and by relatively low ADI values, indicating drier climatic conditions (Supplementary Table 4)~~” to provide a clearer explanation (Page 13, Lines 371-375).

[Comment 3-11] L570: even at steady state the ratio of the EPS and microbial biomass contents does not always reflect the ratio of the rates of EPS production over biomass production (unless the turnover rates of EPS and biomass are exactly the same).

[Response 3-11] Thank you for this important clarification. We agree that even under steady-state conditions, ratios of standing stocks do not necessarily reflect ratios of production rates unless turnover rates are identical. To avoid any potential ambiguity, we have removed the phrase “as a proxy for the relative allocation of microbial C assimilates to extracellular non-growth products under steady-state conditions”. Following *your Comment 3-5 (Please see above)*, we have revised the sentence from “Based on EPS-C, the EPS-C/MBC ratio was calculated as a proxy for the relative allocation of microbial C assimilates to extracellular non-growth products under steady-state conditions, while the EPS-C/SOC ratio was computed to represent the contribution of EPS-C to total soil organic carbon” to “~~Based on EPS-C, we calculated the ratio of EPS-C to MBC. This EPS-C:MBC ratio has previously been termed “EPS production efficiency”, referring to the amount of EPS secreted by microorganisms in soils¹⁶. However, post-production processes, including EPS-soil associations and decomposition, can modify this ratio. While the concept is useful for characterizing short-term microbial EPS responses under conditions where these processes are negligible⁵¹, in natural~~

soils, the metric is more appropriately referred to as the EPS-C/MBC ratio and should be interpreted with caution. Additionally, we calculated the EPS-C/SOC ratio to represent the contribution of EPS-C to total SOC” (Page 18, Lines 553-560).

Reviewer #4 – second-round comments

[Comment 4-1] First of all, and most importantly, the methodology. EPS has been extracted from dry soil samples, usually done on fresh soil samples, although it is known that microbial processes change on short-timescale and also based on sample preparation and storage. Furthermore, the EPS extraction method used is not solely specific in its extraction and we know that soils flash C, getting available, after drying soil samples. Overestimation is quite likely; authors show higher EPS-sugar data than other studies (Figure 1). So, what do the authors study!?

Sorry, but I still don't find the response convincing: In a reference used in the response to my caution with regard to the method, the authors showed 25 % more sugar's – so a strong overestimation of C within the EPS after extracting from dry soil material – authors tried to convince that the reduction in the other measured components (e.g. proteins going down) would compensate for this, as the sum is not changing much – not convincing; Their own analysis (no reference to the data in the manuscript – see below) did not show large variation, in the response the results are given as mean and SE, could you give it as SD and refer to data and discussion in the supplement.

Overall, I strongly question whether it makes sense to combine the various EPS components, as there could also be overlaps between the different components.

I would recommend discussing the topic even more openly. Line 318..... : This differences could either come from... or could be due to sample treatment....

Line 323: Refer to data in the supplement

[Response 4-1] Thank you for the detailed and rigorous evaluation of the EPS methodology. We acknowledge that EPS extraction from dried versus fresh soils can introduce methodological biases. In our initial revision, we therefore attempted to address this concern by including additional, supplementary experiments comparing air-dried and fresh soils, as well as storage effects. However, following further consideration and the guidance of **Reviewer 1's Comment 1-1 on R4-1 and Comment 1-5 on R4-3 (Please see below)**, we recognized that these supplementary experiments, given their limited scope and underlying assumptions, could potentially introduce confusion or unintended bias, rather than clarifying the interpretation of EPS measurements. Accordingly, and in line with the reviewer's recommendation, we have removed the additional supplementary information (**Supplementary Tables 3 and 4 in the previous version**) and related discussions (**Supplementary Discussion 1 and Supplementary Discussion 2 in the previous version**).

Instead, following **Reviewer 1's Comment 1-4 on R4-1 (Please see below)**, we have replaced the original text in Lines 318–323 — “This difference partly reflects differences in bedrock and land-use composition, sampling season, and methodological factors (Supplementary Discussion 1). Due to sample availability, we used air-dried soils, whereas

most previous studies used fresh soils. Our preliminary experiment showed minor effects of air-drying: EPS polysaccharides decreased by $1 \pm 3\%$, EPS proteins increased by $9 \pm 1\%$, and total EPS (polysaccharides + proteins) increased by $4 \pm 3\%$. Long-term soil storage had similarly no impact on EPS content (Supplementary Discussion 2)” with the following sentences in the **Discussion**, as suggested, to explicitly acknowledge the methodological limitations associated with EPS extraction and to clearly define the interpretative boundaries of our study.

“To keep sampling efforts manageable at the continental scale, we extracted EPS from air-dried aggregates. It is important to note that air-drying can cause artefacts: triggering cell-lysis³⁵ and increasing EPS production¹ in the short period before the dry state is reached. However, the majority of increased C-release from soils subjected to drying is removed during the EPS pre-extraction step³⁶. While our use of dried aggregates prevents direct comparison with the majority of EPS studies, which extract moist soils (e.g. Zhang et al.,¹³, Luo et al.,³⁷; Bölscher et al.,³⁸), our large dataset enables the relative effects of geological, geographical, and edaphic factors on soil EPS concentrations to be investigated. Other researchers have also performed ad hoc changes to the standard soil EPS extraction protocol. For example, Bublitz et al.³⁹ claimed that most soils do not contain measurable amounts of non-EPS carbohydrate. However, this unsubstantiated claim is at odds with the majority of soil EPS research where a purification (or “pre-extraction”) step is included to remove confounding non-EPS proteins and polysaccharides². Zhang et al.³⁶ deconstructed the EPS extraction process and investigated the quality of both pre-extracts and EPS-extracts. They found that artefacts from EPS extraction of air-dried soil were much smaller than the large amounts of easily detectable non-EPS proteins and polysaccharides removed during the purifying, “pre-extraction” step. This emphasizes that while EPS extraction from dry aggregates invites limitations, these are far smaller than the problems occurring when the pre-extraction step is omitted. With some of our soils receiving organic inputs and/or significant root exudates over the continent, confidence between different managements was vital. Therefore, we guarded against EPS overestimation by including the purification steps described by Redmile-Gordon et al.², which largely eliminates any errors from overlooking hydration status. Nonetheless, our data should not be compared directly with those of others who ensure full physico-chemical integrity of EPS by avoiding desiccation, as described in the original and complete method²” **(Please see the revised manuscript (clean version), Pages 9-10, Lines 233-255, the same applies hereafter).**

[Comment 4-2] Secondly, the authors still use vague expressions – which are misleading and new terms without real definitions - and thus without background and verifiability. In addition, it remains unclear to an inexperienced reader what has been verified or is just assumptions, as this is not made clear. Just some examples from the Abstract: “EPS retention” – some parameters increase EPS retention, but what is EPS retention!? “environmental filters” – regulated by environmental filters – no clue; it is still not clear if EPS has a significant implication on Soil carbon cycling and sequestration – that should be formulated like that and not just claimed - The authors show themselves, EPS-C contribution of 0,3-3,9% of SOC against 30-60% necromass contribution to SOC – significance?I would recommend the authors to critically review their text once again in this regard.

[Response 4-2] Thank you for highlighting the need for clearer terminology and a stricter distinction between observation, inference, and assumption. We have revised the manuscript to improve conceptual precision, remove vague expressions, and avoid introducing undefined terms.

(1) First, we clarify that our study quantifies EPS-C pools across a broad environmental gradient and relates them to soil physicochemical and microbial properties. We do not directly measure EPS production, stabilization, or decomposition. Any mechanistic interpretations are therefore framed explicitly as inferred processes, not demonstrated ones.

(2) Regarding terminology:

a. “EPS retention” has been replaced by “**association of EPS with soil surfaces and in aggregates**”. By this, we mean the transient physical and chemical interactions between EPS and minerals or organic matter, which may influence EPS residence time without implying long-term stabilization or accumulation.

Specifically, we have revised the sentence from “Based on these findings, we propose that the higher EPS content observed in carbonate soils is primarily associated with two aspects: (1) an indirect pathway, where high soil CEC supports microbial activity¹⁷, increases microbial biomass (e.g., MBC and MBN) and thereby promotes the capacity for EPS production⁷; and (2) soil minerals (clay) and cations (Ca_e) may contribute to the selective retention of EPS within the soil matrix through multivalent cation bridging between negatively charged sites of EPS and other polymers and minerals^{12, 18}” to “**Based on these findings, we propose that the higher EPS content observed in carbonate soils is primarily associated with two aspects: (1) an indirect pathway, where high soil CEC supports microbial activity²², increases microbial biomass (e.g., MBC and MBN) and thereby promotes the capacity for EPS production¹⁶ (Supplementary Table 3); and (2) soil minerals (clay) and cations (Ca_e) may enhance the association of EPS with soil surfaces and within aggregates through multivalent cation bridging between negatively charged sites of EPS and other polymers and minerals^{13, 40}” (Page 10, Lines 260-266). We have revised the sentence from “This suggests that microbes in silicate soils may have allocated proportionally less C toward EPS production or are less conducive for EPS retention” to “**This suggests that microbes in silicate soils may have allocated proportionally less C toward EPS production or are less conducive to the association of EPS with the soil surface and within aggregates**” (Pages 10-11, Lines 286-288).**

b. “Environmental filters” has been removed. We now refer explicitly to climatic, edaphic, and land-use factors that are statistically associated with EPS-C patterns. Specifically, we have revised the sentence from “Here, we show that EPS is a functionally important microbial residue, regulated by both environmental filters and microbial allocation strategies, with significant implications for soil carbon cycling and sequestration” to “**Our results demonstrate that EPS represents a functionally important microbial residue, regulated by geological, climatic, edaphic, microbial, and land-use factors, with significant implications for soil carbon cycling and sequestration**” (Page 3, Lines 32-35).

- c. Terms implying long-term persistence (e.g., accumulation, stabilization, preservation) are no longer used. Instead, we refer to apparent EPS-C pools and EPS-soil associations, consistent with the labile nature of CER-extractable EPS. Specifically, we have revised the sentence from “Furthermore, soil minerals such as Fe- and Al-oxyhydroxides and exchangeable Ca and Mg interact with EPS functional groups to strongly bind it to soil particles, also causing soil aggregation and improving the stability of soil structures, and thereby stimulating further accumulation of EPS^{19, 20}” to “**Furthermore, soil minerals such as Fe- and Al-oxyhydroxides and exchangeable Ca and Mg interact with EPS functional groups^{31, 32}, promoting the association of EPS with soil particles**” (Page 5, Lines 93-95). We have also revised the sentence from “As mentioned above, the accumulation of soil EPS is contingent on the dynamic balance between its generation, stabilization, and degradation” to “**As mentioned above, apparent soil EPS pools reflect a dynamic balance among EPS production, association with soil surfaces and aggregates, and degradation**” (Page 12, Lines 349-351). We have revised the sentence from “Distinct bedrock and land-use types have shaped diverse ecological environments, and no doubt influenced the decomposition and accumulation of EPS” to “**Distinct bedrock and land-use types have shaped diverse ecological environments and have undoubtedly influenced EPS production, degradation, and its associations with soil particles**” (Page 12, Lines 351-353). We have revised the sentence from “Soil mineral composition and particle size distribution, therefore, can affect the stabilization of EPS. For instance, silt- and clay-rich soils can tightly bind EPS, which reduces the EPS decomposition rate and enhances its preservation in the soil²¹” to “**Soil mineral composition and particle size distribution can therefore affect how EPS associates with soil particles and aggregates. For instance, silt- and clay-rich soils can promote these associations, potentially reducing EPS accessibility to decomposition¹¹**” (Page 5, Lines 90-93). We have revised the sentence from “However, as measured EPS levels are influenced by production, decomposition, and stabilization processes, the true microbial response requires further investigation” to “**However, as measured EPS levels are influenced by production, decomposition, and their association with soil particles and aggregates, the true microbial response requires further investigation**” (Page 13, Lines 363-365). Correspondingly, we have revised the descriptions in the **Supplementary Information**. Please see the updated **Supplementary Discussion 1 – Mechanisms of EPS - soil particle associations**, in which all terms previously referring to “stabilization” have been replaced with the clearer expression EPS-soil associations.
- (3) **Concerning the role of EPS in soil C cycling:** Before our study, EPS-C had not been quantified (only EPS) and therefore not been set in relation to SOC, so its contribution to SOC was unknown. In the meantime, very recent publications^{22, 23} also indicate low contributions, but they were published after we submitted our paper to *Ncomms*. We have revised this sentence from “Across the European transect, soil EPS-C averaged 0.41 g C kg⁻¹ (0.06 – 1.07 g C kg⁻¹), contributing 1.6% to SOC (0.3 – 3.9%)” to “**Across the European transect, soil EPS-C averaged 0.41 g C kg⁻¹ (0.06 – 1.07 g C kg⁻¹), contributing 1.6% to SOC (0.3 – 3.9%). Recent small-scale studies have reported similar values,**

consistent with our results^{27, 43}” (Page 11, Lines 306-308), and the references have been added to the **Discussion**.

- (4) To emphasize our study’s novelty, we revised the sentence in the **introduction** from “Given the potentially outstanding role of EPS for microbial function and SOC dynamics, it is interesting to note that quantitative measurements of soil EPS contents remain constrained” to “**Given the potentially outstanding role of EPS for microbial function and SOC dynamics, it is interesting to note that quantitative measurements of soil EPS contents remain limited, and the lack of accepted EPS-C conversion factors leaves its contribution to SOC largely unknown**” (Page 4, Lines 68-70).
- (5) We now explicitly state that EPS-C represents a small fraction of total SOC (0.3-3.9%), especially compared to microbial necromass (30-60%). Our intention is not to claim that EPS is a major contributor to SOC stocks (though this was not known before), but rather to highlight that EPS is a largely overlooked C pool at regional scales and may play a functional role in soil aggregation, microbial habitat formation, and short-term C dynamics, even if its quantitative contribution to SOC stocks is limited. Accordingly, we have revised the introduction sentence from “Soil microbial residues are essential components of the soil carbon (C) pool and can be classified as cellular and extracellular residues^{24, 25}Cellular residues-also known as microbial necromass-mainly consist of microbial cell wall fragments and are widely quantified using amino sugars as biomarkers²⁶. They have been extensively investigated, are reasonably well understood^{26, 27}, and contribute approximately 30-60% of the total soil organic carbon (SOC)²⁶. However, our knowledge of extracellular residues, such as microbial extracellular polymeric substances (EPS) and their contribution to SOC, remains highly limited” to “**Soil microbial extracellular polymeric substances (EPS) are essential components of the soil carbon (C) pool^{1, 2}. In contrast to EPS, microbial necromass mainly consists of microbial cell wall fragments and is widely quantified using amino sugars as biomarkers³. They have been extensively investigated, are reasonably well understood^{3, 4}, and contribute approximately 30-60% of the total soil organic carbon (SOC)³. However, EPS remains a largely overlooked C pool at regional scales and may play important functional roles in soil aggregation, microbial habitat formation, and short-term C dynamics^{1, 5}. Its relationships with geological, land use, climatic, edaphic, plant, and microbial factors are not yet fully understood**” in the **introduction** (Page 4, Lines 38-45) to avoid overstatement.

We appreciate your guidance, which has helped us substantially improve the clarity and conceptual rigor of the manuscript.

[Comment 4-3] Line 338-339 – prove it for MBC/MBN – you have the data.

[Response 4-3] Thank you for pointing this out. Following your suggestion, we explicitly tested the relationships between soil pH, CEC, and exchangeable Ca²⁺ (Ca_e) with microbial biomass carbon (MBC) and microbial biomass nitrogen (MBN), using linear models. The results are now provided in a new **Supplementary Table 3 (please see below)**. These analyses show that CEC was a significant positive predictor of both MBC and MBN, while pH showed

a significant negative effect, and Ca_e was not significant when considered alongside CEC and pH.

Based on these results, we have revised the original statement to avoid over-generalization regarding the role of pH and Ca_e , and added **Supplementary Table 3** to the **Supplementary Information**. The sentence has been changed from: “Based on these findings, we propose that the highest EPS content in carbonate soils is primarily driven by two aspects: (1) an indirect pathway, where highest soil pH and CEC (particularly Ca_e) modulates the soil microenvironment in ways that stimulate microbial activity¹⁷, increase microbial biomass (e.g., MBC and MBN) and thereby causes greater overall capacity for EPS production⁷” to “**Based on these findings, we propose that the higher EPS content observed in carbonate soils is primarily associated with two aspects: (1) an indirect pathway, where high soil CEC supports microbial activity²², increases microbial biomass (e.g., MBC and MBN) and thereby promotes the capacity for EPS production¹⁶ (Supplementary Table 3)**” (Page 10, Lines 260-264).

Supplementary Table 3 | Linear models explaining microbial biomass carbon and microbial biomass nitrogen

Response variables	Predictor	Estimate	Std. Error	t value	p value
MBC	Intercept	1153.60	334.82	3.45	0.001
	pH	-144.48	52.97	-2.73	0.008
	CEC	67.27	25.53	2.64	0.010
	Ca_e	-73.47	68.66	-1.07	0.287
MBN	Intercept	172.76	74.80	2.31	0.023
	pH	-24.78	11.83	-2.09	0.039
	CEC	11.26	5.70	1.97	0.052
	Ca_e	-4.45	15.34	-0.29	0.772

MBC, microbial biomass carbon; MBN, microbial biomass nitrogen; CEC, cation exchange capacity; Ca_e , exchangeable calcium (Ca^{2+}).

[Comment 4-4] Line 351-353: It’s going too far in my point of view; WHC is influenced by SOC and texture – both parameters driving EPS – so is it not just a false correlation; which occurs when two variables show a statistical relationship, but one does not cause the other; this can be due to a third confounding variable influencing both, such as SOC and texture!?

[Response 4-4] Thank you for this valuable comment. We agree that soil WHC is strongly influenced by SOC and texture, both of which may also co-vary with EPS. To avoid over-interpretation, we have therefore removed this sentence: “Moreover, the relatively higher WHC in carbonate soils – may reflect the ecological function of soil EPS in improving soil water retention¹² (Fig. 9)” from the manuscript.

[Comment 4-5] Line 364: double prove your interpretation by checking if the sedimentary soils are more often found in arid conditions – I would have thought that carbonate-rich soils are more often found under arid conditions – thus it is counterintuitive – please prove.

[Response 4-5] Thank you for this insightful comment. As our dataset does not allow us to directly assess long-term moisture regimes across bedrock types, we are unable to robustly

evaluate this interpretation. Therefore, we have removed the sentence “Given the decrease in MBC at lower WHC and SMC (Supplementary Fig. 2), we infer that limited water availability in sedimentary soils may constrain microbial growth and shift C allocation towards EPS formation. This shift may reflect an adaptive microbial strategy to enhance survival and maintain extracellular protection under water-stressed conditions¹²” from the manuscript.

[Comment 4-6] Line 390-391: Is water limitation the only stress potentially triggering microbes to produce EPS!?

[Response 4-6] Thank you for this question. We agree that attributing the elevated EPS-C/MBC ratios in croplands solely to water limitation was inappropriate. To avoid overinterpretation and to maintain consistency with the revised discussion, we have therefore removed the sentence “EPS-C/MBC ratios peaked in croplands, likely reflecting enhanced microbial EPS investment under water limitation” from the manuscript.

[Comment 4-7] Line 408-409: see my comment above about false correlations – prove what is driving WHC – you have the data, and check autocorrelations.

[Response 4-7] Thank you for this suggestion. In line with your previous *Comment 4-4* and the concern raised here regarding potential false correlations, we have removed the sentence “A key functional role was evident in the strong positive association between EPS and soil water holding capacity (Fig. 9a), highlighting its contribution to soil moisture retention”.

[Comment 4-8] Line 451: Check the Bublitz et al.

[Response 4-8] Thank you for your helpful reminder. We have checked the recent study by Bublitz et al.²⁹, which demonstrated that field-induced drought significantly increased EPS-carbohydrates and EPS-proteins (by ~20%) while microbial biomass showed a different response. We have now cited this study and revised the sentence from “Under harsh conditions, microorganisms produce EPS to enhance survival³⁰; for example, EPS helps retain water and protect cells from drought damage³¹. Although not previously demonstrated for soil microbial communities, we show this adaptive phenomenon at a large scale, reflected by increasing EPS-C/MBC along an aridity (ADI) gradient, suggesting microbial adaptation to long-term water-limited environments” to “**Under harsh conditions, microorganisms produce EPS to enhance survival⁸; for example, EPS helps retain water and protect cells from drought damage⁵². Recent field experiments have shown that drought can significantly increase soil EPS⁵³. Moving beyond these localized observations, we show at a large scale that EPS-C/MBC increases along an ADI gradient, indicating microbial adaptation to long-term water-limited environments**” (**Pages 12-13, Lines 356-360**).

[Comment 4-9] Overall, I would recommend subheadings in the discussion based on the questions set now for the manuscript, which gives the authors the chance to come back to them more clearly in the discussion.

[Response 4-9] In the original manuscript and in the first revision, the **Discussion** was structured using subheadings corresponding to the main research questions of the study. However, according to the formatting requirements of *Nature Communications*, subheadings are not permitted in the Discussion section. Therefore, in the current version, we have removed the subheadings and instead carefully reorganized the **Discussion** into clearly structured paragraphs, ensuring that the main research questions are revisited and discussed in a clear and logical manner. We appreciate your understanding.

[Comment 4-10] *I would also recommend to reduce the amount of figures (10 until now) within the manuscript and move figures which might not as necessary to the supplement.*

[Response 4-10] Thank you for this helpful suggestion. Following your recommendation and also considering **Reviewer 3's Comment 3-2**, we have reduced the number of figures in the main manuscript by merging Figs. 9 and 10 into a single figure (new **Fig. 9**) (Please see above).

Reviewer #1's comments on Reviewer #4's (first- and second-round) comments

[Comment 1-1 on R4-1] *I share the majority of R4's concerns around the use of dried aggregates, and interfering C. Nonetheless, I think the Authors' original data, and experience here is still of high value (including state-of-the-art learnings, and broad environmental relevance). Their expansive study area, and attention to EPS as a potential source of stabilized C, warrant attention. However, their focus should stay within that scope. This means several of the added revisions referencing the additional work will need removing. They should simply present the caveats more clearly, as suggested, below.*

R4's comments and with regard to additional work:

1) *most importantly, regarding the methodology: EPS has been extracted from dry soil samples, usually done on fresh soil samples, and it is known that microbial processes change on a short-timescale based on sample preparation and storage.*

2) *Overestimation from a flush of C from drying, is quite likely, as is biological response for greater EPS exudation.*

Ad-hoc adaptation of the method -whether omitting the pre-extraction (Reference #46), or using dried aggregates (current study)- means that results are not directly comparable with the work of most others. The data remain useful as a self-contained dataset for analysis. While the use of dry aggregates has limitations mentioned by R4 – many of these are largely nullified by pre-extraction (Zhang at al., 2024; evidence given below). However, there are much larger weaknesses in the additional work conducted. Weaknesses include confounding 1) laboratory change effects over 4 years 2) rewetting soils stored in artificial conditions 3) assuming factors identical to a natural soil at state of recent sampling, and 4) microbial community and function both compromised by drying and storage.

Accordingly, my further recommendations are:

Delete extra work: Supplementary Tables 3 & 4 and all references to them. There are too many assumptions behind the data, and they are potentially misleading – they imply false

equivalence to freshly sampled (moist) soil – when rewetted (Table 4) and embody a very small dataset with insufficient meaning (Table 3).

[Response 1-1 on R4-1] Thank you very much for your insightful comments. Following your suggestion, we realized that these supplementary analyses could indeed be potentially misleading due to the underlying assumptions and that the limited sample size may constrain the robustness and interpretability of the results. Therefore, as recommended, we have removed **Supplementary Tables 3 and 4** and deleted all related descriptions from the manuscript, including the statement: “Furthermore, to ensure the robustness of EPS measurements from air-dried samples used in this study, we additionally tested whether soil moisture status and storage duration influenced EPS extraction. The procedures and results are presented in Supplementary Tables 3 and 4 and discussed in Supplementary Discussion 2” **(Please see the revised manuscript (clean version), the same applies hereafter).**

[Comment 1-2 on R4-1] Investigate cited studies and remove all references to other studies in all text and tables who i) skipped the pre-extraction step, or ii) were performed in cultures. This includes Reference #46 needing removal from Supplementary Table 1. Please could the authors ensure they remove any others (they are rare).

[Response 1-2 on R4-1] Thank you for pointing this out. We have carefully re-investigated all cited studies and revised the manuscript accordingly. First, we removed the publication “Optimization of the bicinchoninic acid assay for quantifying carbohydrates of soil extracellular polymeric substances” from our database, as this study did not include a CaCl₂ pre-extraction step before EPS extraction. In addition, the study “Biological soil crust elicits microbial community and extracellular polymeric substances restructuring to reduce the soil erosion on tropical island, South China Sea” was also excluded, as no pre-extraction step was applied.

We then systematically re-checked all remaining studies included in the text and tables to ensure that (i) EPS was extracted from field-collected soil samples rather than culture-based experiments, and (ii) a CaCl₂ pre-extraction step was applied before EPS extraction. In addition, we included four newly published soil EPS studies (published between August 2025 and the present) that meet these criteria. Accordingly, **Supplementary Dataset 2** and **Supplementary Table 1** have been updated **(Please see below)**.

Supplementary Table 1 | Previously published quantitative data on extracellular polymeric substances (EPS) extracted using the cation exchange resin method.

Reference	Land use type	EPS	EPS	Total EPS
		polysaccharides	proteins	
(µg g ⁻¹ soil)				
(Zhang et al., 2023) ¹¹	Cropland	151.08	86.56	237.64
	NA	614.29	117.22	731.52
(Chen et al., 2025) ³²	NA	121.02	47.67	168.69
	NA	115.72	12.52	128.24
(Redmile-Gordon et al., 2020) ¹	Grass-Grass	346.02	213.17	559.19

	Grass-Arable	339.55	194.56	534.11
	Grass-Fallow	293.99	156.96	450.95
	Arable-Grass	434.87	198.24	633.11
	Arable-Arable	317.79	184.64	502.43
	Arable-Fallow	374.66	175.67	550.33
	Fallow-Grass	381.50	191.27	572.77
	Fallow-Arable	235.55	110.67	346.22
	Fallow-Fallow	286.96	126.43	413.39
(Redmile-Gordon et al., 2014) ¹⁶	Fallow	169.00	43.00	212.00
	Grassland	401.00	163.00	564.00
	Grassland	1120.00	NA	NA
(Bérard et al., 2020) ³³	Cropland	800.00	NA	NA
	Woodland	830.00	NA	NA
	Cropland	743.54	NA	NA
	Woodland	252.01	52.36	304.36
	Cropland	447.44	273.29	720.74
(Kidinda et al., 2023) ⁷	Woodland	215.99	60.00	276.00
	Cropland	332.80	194.10	526.90
	Woodland	278.31	80.18	358.49
	Cropland	272.88	145.05	417.93
(Redmile-Gordon et al., 2017) ³⁴	Grassland	191.33	94.90	286.23
(Shi et al., 2024) ²⁵	Woodland	158.31	79.28	237.58
(Redmile-Gordon et al., 2015) ⁸	Grass-Arable	467.33	177.76	645.08
(Hale et al., 2021) ³⁵	Grassland	456.41	NA	NA
	Cropland	262.78	NA	NA
(Bettermann et al., 2021) ³⁶	Shrub land	52.19	NA	NA
	Shrub land	76.30	NA	NA
(Zethof et al., 2020) ¹⁷	Shrub land	130.37	NA	NA
	Shrub land	120.79	NA	NA
	Woodland	158.60	44.90	203.50
(Baumert et al., 2021) ³⁷	Woodland	160.30	42.30	202.60
	Woodland	98.30	15.70	114.00
(Crouzet et al., 2019) ³⁸	Cropland	367.95	NA	NA
(Vuko et al., 2020) ³⁹	Fallow	175.43	78.97	254.40
	Cropland	260.78	133.86	394.64
(Liao et al., 2024) ⁴⁰	Woodland	317.20	301.82	619.01
(Liu et al., 2025) ⁴¹	Cropland	471.25	205.19	676.44
	Woodland	691.82	364.28	1056.10
(Peng et al., 2025) ⁴²	Cropland	297.48	37.23	334.71
(G. Bogar et al., 2025) ⁴³	Cropland	113.00	NA	NA
	Dryland	393.32	114.69	508.01
(Feng et al., 2026) ²²	Dryland	245.47	43.84	289.31
	Woodland	284.00	100.70	384.70

	Woodland	65.74	50.65	116.39
	Cropland	411.89	104.01	515.90
	Cropland	84.10	54.48	138.58
(Li et al., 2026) ²³	Grassland	489.08	229.30	718.38

Total EPS refers to the sum of EPS polysaccharides and EPS proteins. NA indicates that the data are not available.

Furthermore, based on the updated database, we have also updated **Fig. 1** in the main text (**Please see below**) and revised the corresponding sentence from “Across the European transect, total soil EPS content (sum of EPS polysaccharides and EPS proteins) ranged from 149 to 2495 $\mu\text{g g}^{-1}$ soil, with a grand mean of $956 \pm 55 \mu\text{g g}^{-1}$ soil (SE; n = 92), higher than previously reported values (mean: $429 \mu\text{g g}^{-1}$ soil) (Figs. 1, 2; Supplementary Table 1)” to “**Across the European transect, total soil EPS content (sum of EPS polysaccharides and EPS proteins) ranged from 149 to 2495 $\mu\text{g g}^{-1}$ soil, with a grand mean of $956 \pm 55 \mu\text{g g}^{-1}$ soil (SE; n = 92), higher than previously reported values (mean: $423 \mu\text{g g}^{-1}$ soil) (Figs. 1, 2; Supplementary Table 1)**”(Page 6, Lines 123-125). We also revised the sentence from “An extensive literature survey showed lower EPS contents (~30 sites in 18 studies differing in climate, geology, and land use), with $429 \mu\text{g g}^{-1}$ for total EPS, $335 \mu\text{g g}^{-1}$ for EPS polysaccharides, and $131 \mu\text{g g}^{-1}$ for EPS proteins (Supplementary Table 1)” to “**An extensive literature survey showed lower EPS contents (~37 sites in 22 studies differing in climate, geology, and land use), with $423 \mu\text{g g}^{-1}$ for total EPS, $325 \mu\text{g g}^{-1}$ for EPS polysaccharides, and $128 \mu\text{g g}^{-1}$ for EPS proteins (Supplementary Table 1)**”(Page 9, Lines 230-233).

Figure 1. Comparison of EPS polysaccharides and EPS proteins across the European transect and with data derived from previous studies. a Comparison of EPS polysaccharide contents. **b** Comparison of EPS protein contents. The numbers in the figure indicate frequency. Specific data are listed in Supplementary Table 1, and the complete dataset is provided in Supplementary Dataset 2. EPS, extracellular polymeric substances.

[Comment 1-3 on R4-1] Further recommendations regarding the issue of dry aggregate extractions follow below.

R4 also questioned: whether it makes sense to combine the various EPS components, as there could also be overlaps between the different components.

This is laudable. However, some small overlap seems reasonable considering the state-of-the-art. The authors have excessively conceded in response to R4 (R4-27; addressed in sequence below). I'd be happy to review the final version – provided the authors double-check all their responses, and keep them minimal, accurate, and manageable.

To save others from pitfalls of over-interpreting/comparing data gathered from dried aggregates – AND worse ad hoc changes such as skipping pre-extraction, it is important to establish early in The Discussion that results should be contained to the present study, and

present reasonable boundaries of the approaches. LINES 318-323: are the perfect place to resolve this. This is where the core value and clarity need to be established so that what follows makes sense. Accordingly,

I suggest authors delete: “This difference partly reflects differences in bedrock and land-use composition, sampling season, and methodological factors (Supplementary Discussion 1). Due to sample availability, we used air-dried soils, whereas most previous studies used fresh soils. Our preliminary experiment showed minor effects of air-drying: EPS polysaccharides decreased by $1 \pm 3\%$, EPS proteins increased by $9 \pm 1\%$, and total EPS (polysaccharides + proteins) increased by $4 \pm 3\%$. Long-term soil storage had similarly no impact on EPS content (Supplementary Discussion 2).”

[Response 1-3 on R4-1] Thank you for this important and valuable comment and recommendation. Following your suggestion, we agree that discussing air-drying effects and comparisons with fresh-soil studies may encourage over-interpretation beyond the scope of this work. Accordingly, we have deleted the relevant sentence and now restrict the **Discussion** to the present study only.

[Comment 1-4 on R4-1] And INSERT verbatim:-at line 318-

“To keep sampling efforts manageable at continental scale, we extracted EPS from air-dried aggregates. It is important to note that air-drying can cause artefacts: triggering cell-lysis (Khan et al., 2019) and increasing EPS production (Costa et al., 2018) in the short period before the dry state is reached. However, the majority of increased C-release from soils subjected to drying is removed during the EPS pre- extraction step (Zhang et al. 2023). While our use of dried aggregates prevents direct comparison with the majority of EPS studies which extract moist soils (e.g. Zhang et al., 2024[ref 5], Luo et al., 2024[ref 42]; Bolscher et al., 2024[relevant citation to add]), our large dataset enables the relative effects of geological, geographical and edaphic factors on soil EPS concentrations to be investigated. Other researchers have also performed ad hoc changes to the standard soil EPS extraction protocol. For example, Bublitz et al. (2024) claimed that most soils do not contain measurable amounts of non-EPS carbohydrate. However, this unsubstantiated claim is at odds with the majority of soil EPS research where a purification (or “pre-extraction”) step is included to remove confounding non-EPS proteins and polysaccharides (Redmile- Gordon et al., 2014). Zhang et al. (2023) deconstructed the EPS extraction process, and investigated the quality of both pre-extracts and EPS-extracts. They found that artefacts from EPS extraction of air-dried soil were much smaller than the large amounts of easily detectable non-EPS proteins and polysaccharides removed during the purifying, “pre-extraction” step. This emphasizes that while EPS extraction from dry aggregates invites limitations, these are far smaller than the problems occurring when the pre-extraction step is omitted. With some of our soils receiving organic inputs and/or significant root exudates over the continent - confidence between different managements was vital. Therefore, we guarded against EPS overestimation by including the purification steps described by Redmile-Gordon et al. (2014), which largely eliminates any errors from overlooking hydration status. Nonetheless, our data should not be compared directly with those of others who ensure full physico-chemical integrity of EPS by

avoiding desiccation, as described in the original and complete method (Redmile-Gordon et al. 2014) ”.

[Response 1-4 on R4-1] We sincerely thank you for providing this clear and constructive wording. We have adopted your suggested text to replace the original sentence, as it more effectively establishes the scope of the study and clarifies the boundaries of interpretation. This revision has substantially improved the clarity and focus of the **Discussion (Pages 9-10, Lines 233-255)**.

***[Comment 1-5 on R4-3]** R4-3 Delete Supplementary Discussion 1 and Supplementary Discussion 2 which add to the confusion on further analyses conducted (the arguments are rightly not accepted by R4 anyway). Weaknesses include confounding 1) laboratory change effects over 4 years 2) rewetting soils stored in artificial conditions 3) assuming factors identical to a natural soil at state of recent sampling, and 4) microbial community and function both compromised by drying and storage. In preference, the addition to the discussion above (Suggestion 4) deals with the approach chosen, the limitations, the context, the strength and the clarity for newcomers all in one. NOTE: This removal of the confounding additional work requires also rolling back several of the current additions which the authors made in response to R4:*

[Response 1-5 on R4-3] We have followed this recommendation and deleted **Supplementary Discussion 1 and Supplementary Discussion 2**, as well as all references to them throughout the manuscript. Consistent with your suggestion, we have also rolled back the related additions made in our previous revision and ensured that, apart from the concise statement on methodological boundaries introduced in response to the *Reviewer’s Comment 1-4 on R4-1*), no further extended discussion of methodological differences is included.

***[Comment 1-6 on R4-5]** R4-5:Not satisfactory. Resolved by my suggestion in response to R1-22 (part 2 below).*

[Response 1-6 on R4-5] Thank you for the valuable comment. As suggested, this issue has now been fully addressed by incorporating the revision proposed in response to *Comment 1-6 on R1-22 and Comment 1-7 on R1-22 (please see below)*. In addition, we have carefully reviewed the manuscript to identify and revise any inappropriate wording or expressions related to this issue. Detailed changes are described in our response to *Reviewer 4’s Comment 4-2 (please see above)*, *Reviewer 1’s Comment 1-8 on 4-9 (please see below)*, and the *General Comment from Reviewer #1’s comments on Reviewer #1’s (first-round) comments (please see below)*.

***[Comment 1-7 on R4-7]** R4-7: Misleading. EPS are not “poorly quantified” Resolved by my suggestion in response to R1-5 (part 2 below). I requested to replace this sentence with “However, systematic evaluation of this claim has yet to be reported”. Please do as asked.*

[Response 1-7 on R4-7] Thank you for your valuable and detailed comments. Following your suggestion in *Comment 1-1 (Please see above)*, we have replaced the sentence “Yet, their

abundance and large-scale environmental and microbial controls remain poorly quantified” with “**Yet, their abundance and large-scale environmental and microbial controls have only begun to be investigated**” (Page 3, Lines 23-24). We hope that this revision addresses your concern.

[Comment 1-8 on R4-9] R4-9: Not satisfactory. Resolved by my suggestion in response to R1-5 (part 2 below).

[Response 1-8 on R4-9] As suggested, and following this comment as well as *Reviewer’s Comment 1-5 on R1-5*, we have revised the sentence from “Across the transect, EPS increased with microbial biomass as a proxy for EPS production potential, while soil properties such as clay content and exchangeable calcium enhanced EPS retention” to a purely descriptive statement: “**Across the transect, EPS was positively associated with microbial biomass (MBC) and also showed positive associations with soil clay content and exchangeable calcium**” reporting observed associations only, without implying mechanisms or causality (Page 3, Lines 31-32).

In addition, we have removed or revised all inappropriate mechanistic wording, such as “EPS production potential” and “EPS retention”, throughout the manuscript. Detailed revisions related to **EPS retention** are described in our response to *Comment 4-2 (please see above)*. With respect to “**EPS production potential**”, we specifically revised the sentence from “In particular, carbonate soils have been reported to harbor higher microbial diversity⁹, which, given the positive relationship between diversity and EPS production⁴⁴, may reflect a greater EPS secretion potential⁹” to “**In particular, carbonate soils have been reported to harbor higher microbial diversity⁴¹. Given the reported positive association between microbial diversity and EPS⁴², this pattern aligns with the elevated EPS levels observed in carbonate soils**” (Page 10, Lines 272-275). We also revised the sentence from “Although our data show a positive correlation between MBC and total EPS, suggesting that greater microbial biomass increases the overall potential for EPS production, we also observed higher EPS-C/MBC ratios under stressful conditions, such as in soils developed from sedimentary bedrock” to “**Although our data show a positive correlation between MBC and total EPS, higher EPS-C/MBC ratios were observed under stressful conditions, such as in soils developed from sedimentary bedrock**” (Page 13, Lines 371-372). We also revised the sentence from “Finally, microbial community composition may affect EPS production and degradation, as taxa differ markedly in their genomic potential for EPS secretion⁴⁵ and degradation⁷” to “**Finally, microbial community composition may influence observed EPS patterns, as taxa differ markedly in genomic traits related to EPS synthesis²⁹ and degradation¹⁶**” (Page 13, Lines 389-390).

[Comment 1-9 on R4-15] R4-15 I agree, the author's response is not satisfactory. Resolved by insertion to discussion, Point 4 above.

[Response 1-9 on R4-15] Thank you for this helpful suggestion. Following your guidance, this issue has now been addressed by inserting the revised text into the **Discussion**, as outlined in *Response 1-4 on R4-1* above (Pages 9-10, Lines 233-255).

[Comment 1-10 on R4-18] R4-18: correct the logic and rewrite (L99-103) to: “EPS secretion is governed by microbial growth and environmental stress, soil properties, nutrient availability” (a C source for EPS is not stress; remove the nested syntax).

[Response 1-10 on R4-18] Thank you for this helpful suggestion. We have revised the sentence from “Finally, as microbial communities expand, EPS secretion enhances cell adhesion on soil particles and helps protect against microbial competition^{7, 28}. Collectively, these findings underscore that EPS secretion is governed by microbial growth and environmental stress⁴- both influenced by soil properties, nutrient availability^{46, 47, 45, 48}, and population-level survival strategies⁷” to “**Finally, as microbial communities expand, EPS secretion enhances cell adhesion on soil particles and helps protect against microbial competition^{5, 16}. Collectively, these findings underscore that EPS secretion is governed by microbial growth and environmental stress¹, soil properties, nutrient availability^{25, 26, 28, 29}, and population-level survival strategies¹⁶” (Page 5, Lines 85-88).**

[Comment 1-11 on R4-27] R4-27: some small overlap between protein and carbohydrate seems reasonable considering the state-of-the-art, and the authors approach remains useful. I expect the overlap is likely minor (majority of C residing in either a polysaccharide monomer, or an amino acid in a protein) However, the authors seem to have oversimplified in response to R4 with “for example, glycoproteins which would be doublecounted”. However, “double-counting” is a gross overstatement. Glycoproteins are likely vulnerable to partial degradation or fractionation during cation exchange resin (CER) extraction of EPS and the assumption that they are simply “double-counted” in both protein and polysaccharide assays is an oversimplification. The acidic microenvironment around the resin beads can lower local pH, potentially leading to partial hydrolysis of glycosidic or peptide bonds. Furthermore, Colorimetric assays (e.g., Lowry, Bradford for proteins; phenol-sulfuric acid for carbohydrates) detect specific reactive groups, not intact macromolecules. In summary Assays do not truly double-count glycoproteins; rather, they may partially detect their components, leading to incomplete or skewed quantification, but nothing is perfect. This is the state-of-the-art. Replace “for example, glycoproteins which would be doublecounted” with “for example, measurement of glycoproteins which are partially detected in both colorimetric protein and polysaccharide assays”.

[Response 1-11 on R4-27] Thank you for the valuable suggestion and insightful clarification. Following your recommendation, we have replaced the original sentence with the revised formulation you proposed, which more clearly reflects the partial overlap arising from the analytical detection of glycoproteins in both colorimetric protein and polysaccharide assays (Page 17, Lines 522-523).

[Comment 1-12 on R4-33] R4-33: The authors current response is superseded by the suggested addition to main discussion above. Importantly, the authors response to R4-33 shows another area where confusion is being propagated: The 1st reference in the manuscript (Oliva et al. 2024) conflates soil EPS with culture-derived EPS and does not separate the confounding

carbon that R4 refers to. The cited authors omitted any pre-extraction steps and therefore included soluble metabolites, broth residues, and necromass products. The paper reports that new (necromass) indicators (GalN and MurN) show “100 % contribution from the EPS fraction”, but this figure arises because both their “EPS extract” and “total extract” measured largely the same soluble pool—an artefact of non-independent fractions, certainly not proof of EPS exclusivity, or even being “an integral parts of EPS”. This may explain some of the current confusion and references to Olivia et al. (2024). Such work should not open a study on soil EPS; other established soil-based studies provide a more appropriate foundation. Accordingly, please promote the following reference (Costa, currently #8) to citation number 1, and remove Olivia et al. (2024).

Costa OY, Raaijmakers JM, Kuramae EE. Microbial extracellular polymeric substances: ecological function and impact on soil aggregation. *Front Microbiol* 9, 1636 (2018).

Promote to share position with the work already currently at citation #2

Redmile-Gordon MA, Evershed RP, Hirsch PR, White RP, Goulding KWT. Soil organic matter and the extracellular microbial matrix show contrasting responses to C and N availability. *Soil Biol Biochem* 88, 257-267 (2015).

Furthermore, since the current manuscript does not investigate or measure “extracellular residues” (a subjective and arguable definition; with only necromass, microbial biomass, and EPS measured here) To reflect the state-of-the-art more accurately....

Please replace LINES 57-58: “Soil microbial residues are essential components of the soil carbon (C) pool and can be classified as cellular and extracellular residues (1, 2)”. WITH “Soil microbial extracellular polymeric substances (EPS) are essential components of the soil carbon (C) pool (Costa et al., 2018; Redmile-Gordon et al., 2014) (1, 2).

[Response 1-12 on R4-33] Thank you for this thoughtful suggestion. We have revised the sentence as recommended, replacing the general description of microbial residues with a more specific statement focusing on soil microbial EPS, to more accurately introduce EPS as the primary research focus of this study. We have also updated the citations accordingly, now referencing Costa et al.⁴ and Redmile-Gordon et al.¹⁶ (**Page 4, Lines 38-39**).

[Comment 1-13 on R4-36] Please Check: R4-36 “In contrast, the microbial composite factor, derived from PCA of CUE, Cgrowth, qGrowth, and MBN (Fig. 8c; Supplementary Fig. 1), exerted a direct positive effect on EPS-C/MBC ratio and a negative effect on EPS-C (Fig. 8a). If the effect on EPS-C is negative (and more negative than on MBC) then the effect on EPS-C/MBC is also negative, no? Are these the right way around?”

[Response 1-13 on R4-36] Thank you for pointing this out. While the microbial composite factor exerts a negative effect on EPS-C, this does not inherently necessitate a negative effect on the EPS-C/MBC ratio. The ratio can increase if MBC declines more sharply than EPS-C. Furthermore, in our SEM, EPS-C/MBC was treated as an independent response variable. Consequently, the standardized path coefficient reflects the net effect of the microbial factor on the ratio itself, rather than being a mathematical derivation from the individual effects on EPS-C or MBC alone. We hope this clarification addresses your concern.

[Comment 1-14 on R4-41] R4-41: Delete Lines 253-258 (in favour of the discussion inclusion, point 4): “This difference partly reflects differences in bedrock and land-use composition, sampling season, and methodological factors (Supplementary Discussion 1). Due to sample availability, we used air-dried soils, whereas most previous studies used fresh soils. Our preliminary experiment showed minor effects of air-drying: EPS polysaccharides decreased by $1 \pm 3\%$, EPS proteins increased by $9 \pm 1\%$, and total EPS (polysaccharides + proteins) increased by $4 \pm 3\%$. Long-term soil storage had similarly no impact on EPS content (Supplementary Discussion 2)” (Page 9, Lines 253-258).

[Response 1-14 on R4-41] Thank you for your valuable comment. Following this comment and **Comment 1-3 on R4-1 (please see above)**, the corresponding paragraph has been removed from the manuscript.

[Comment 1-15 on R4-45] RESPONSE TO R4-45 (same for R1-22) Delete “intermittent” and then delete the following overreach/speculation which is presented as fact: “Overall, bedrock influenced EPS through (1) promoting microbial biomass and EPS secretion via bedrock-driven environmental conditions, and (2) shaping soil texture and mineralogy, which governs EPS retention and intermittently protects it from microbial degradation”. You don’t have such granularity of data.

[Response 1-15 on R4-45] Thank you for this helpful comment. Following your suggestion, we have deleted the term “intermittent” and removed the statement that inappropriately described bedrock-driven mechanisms beyond the support of our data.

[Comment 1-16 on R4-72] EDIT R4-72: “EPS is a transient pool that is selectively secreted by microorganisms in response to adverse environmental conditions, with no measurable legacy effects observed beyond 2.5 years²⁸. This is wrong, and is being taken out of context. EPS were not measured at the same timescales as necromass, - so the comparison with necromass is misleading. Also, the referenced study showed most EPS under grass. Not stressful or “adverse” conditions. CHANGE TO “EPS is a transient pool that is selectively secreted by microorganisms where sufficient labile C is available, with the majority of EPS either stabilising, or being turned over in less than 2.5 years²⁸”.

[Response 1-16 on R4-72] Thank you for this valuable comment and for the detailed clarification. We recognized that the original statement was inappropriate and potentially misleading. Following your recommendation in this comment, together with **Comment 1-5 (Please see above)**, we have revised the original sentence from “EPS is a transient pool that is selectively secreted by microorganisms in response to adverse environmental conditions, with no measurable legacy effects observed beyond 2.5 years¹. In contrast, MNC as the inevitable product of microbial turnover and death, is more resistant to degradation, with turnover times ranging from multiple years to decades^{2, 3}” to “**EPS is a transient pool that is selectively secreted by microorganisms where sufficient labile C is available, with the majority of EPS**

either stabilising or being turned over in less than 2.5 years¹⁵” in the main text (Page 12, Lines 332-334). We appreciate your insightful guidance.

[Comment 1-17 on R4-77 and R4-78] Update response to R4-77, R4-78, these relate to Supplementary Discussion (suggested delete).

[Response 1-17 on R4-77 and R4-78] Thank you very much for your careful and detailed comments. Following your suggestion, and in line with *Comments 1-1 on R4-1* and *Comment 1-5 on R4-3*, we have deleted **Supplementary Discussion 1 and Supplementary Discussion 2** and removed all related references from the manuscript. In addition, we have deleted the sentence “Furthermore, to ensure the robustness of EPS measurements from air-dried samples used in this study, we additionally tested whether soil moisture status and storage duration influenced EPS extraction. The procedures and results are presented in Supplementary Tables 3 and 4 and discussed in Supplementary Discussion 2” to avoid further confusion.

[Comment 1-18 on R4-79] Good responses to R4: R4-79: This is a fair approach: “Before extraction, we also calculated the required amount of CER for each soil sample based on its SOC content following the method proposed by Redmile-Gordon et al.³⁶. These calculations showed that 10 g CER per 1 g dry soil provided sufficient ion-exchange capacity even for the samples with the highest SOC levels, and importantly for soils with the highest exchangeable Ca²⁺ concentrations. Therefore, we applied this amount uniformly across all soils to ensure methodological consistency across SOC-rich and SOC-poor soils and limestone and non-limestone soils” (Page 16, Lines 502- 508).

[Response 1-18 on R4-79] Thank you for this positive assessment. We appreciate your acknowledgement that this approach is appropriate.

[Comment 1-19 on other R4 comments] No objections: R4-80, 79, 76, 75, 74, 73, 71, 70, 69, 68, 67, 66, 65, 64, 63, 62, 61, 60, 59, 58, 57, 56, 55, 54, 53, 52, 51, 50, 49, 48, 47, 46, 43, 42, 40, 39, 38, 37, 35, 34, 32, 31, 30, 29, 28, 26, 25, 24, 23, 22, 21, 20, 19, 17, 16, 14, 13, 12, 11, 10, 8, 6, 4, 2, 1. Problematic responses in red are either addressed above, or in PART 2 below (responses to my own comments, R1): R4-78, 77, 72, 45, 41, 36, 33, 27, 18, 15, 9, 7, 5, 3.

[Response 1-19 on other R4 comments] Thank you for your careful examination and thorough review. We appreciate the time and effort you invested in providing these detailed comments. We have carefully revised the manuscript throughout and provided detailed responses accordingly.

Reviewer #1’s comments on Reviewer #1’s (first-round) comments

[General comment] *Some changes have not been fully actioned. For example, there are still incorrect statements around the subject of “EPS production efficiency” and the importance of some points has been missed. These can be resolved simply. If the authors are content to accept*

my suggestions here -verbatim- and satisfy all requests of The Editor, then I am happy to recommend acceptance following a rapid final round of review.

[Response] Thank you for your careful review. We have revised the manuscript more thoroughly following your comments and appreciate your guidance. We hope the revisions now address your concerns.

Regarding **EPS production efficiency**, we have carefully checked the entire manuscript. This term now only appears in the following contexts:

- (1) “In our study, this ratio does not directly reflect EPS production efficiency – since EPS contents in natural soils result from the balance of microbial production and degradation, modulated by mineral interactions and extraction limitations” (Please see the revised manuscript (clean version), Page 10, Lines 277-279, the same applies hereafter). Here, the term is used to clarify that the EPS-C/MBC ratio in our study cannot be interpreted as a direct measure of EPS production efficiency.
- (2) “Changes in short-term microbial EPS production efficiency were previously calculated by standardizing soil EPS to soil microbial ATP⁵¹, as an independent indicator of the soil microbial biomass” (Page 12, Lines 346-348). Here, the term is used to accurately cite previous literature.
- (3) “Therefore, we present EPS-C/MBC ratios “as is” without inferring EPS production efficiency” (Page 12, Lines 353-354). This emphasizes that EPS-C/MBC should not be interpreted as EPS production efficiency in our study.
- (4) “Based on EPS-C, we calculated the ratio of EPS-C to MBC. This EPS-C:MBC ratio has previously been termed “EPS production efficiency”, referring to the amount of EPS secreted by microorganisms in soils¹⁶. However, post-production processes, including EPS-soil associations and decomposition, can modify this ratio. While the concept is useful for characterizing short-term microbial EPS responses under conditions where these processes are negligible⁵¹, in natural soils, the metric is more appropriately referred to as the EPS-C/MBC ratio and should be interpreted with caution” (Page 18, Lines 553-559). Here, the use of the term serves to clarify that EPS-C/MBC in natural soils is not a direct measure of EPS production efficiency.

In all other instances, EPS production efficiency does not appear in the manuscript.

[Comment 1-1 on R1-1] R1-1: *The changes to EPS-C/MBC ratio are noted and appreciated. However, the first response to my first comment ends on “it may serve as a proxy for the relative allocation of microbial C assimilates to extracellular non-growth products under steady state conditions” (Pages 10, Lines 283-287). This is not satisfactory. The authors have gone ‘full-circle’ in some cases describing an “EPS production efficiency” in different words. The steps to bear in mind (from historic, to time of sampling) are:*

EPS production/allocation > soil retention/stabilisation/degradation > observed EPS.

Therefore, one cannot quantify any of the previous steps here from the last using this dataset. Accordingly, Lines 283-287: Please Change “Although this ratio does not directly reflect EPS production efficiency – since EPS contents in natural soils result from the balance

of microbial production and degradation, modulated by mineral interactions and extraction limitations – it may serve as a proxy for the relative allocation of microbial C assimilates to extracellular non-growth products under steady state conditions” TO: In our study, this ratio does not directly reflect EPS production efficiency – since EPS contents in natural soils result from the balance of microbial production and degradation, modulated by mineral interactions and extraction limitations”.

Deleting “it may serve as a proxy for the relative allocation of microbial C assimilates to extracellular non-growth products under steady state conditions” because EPS allocation precedes other factors affecting measurements. (‘Thought experiment’ for consideration: if a soil type causes disintegration of EPS, it might appear as if EPS were simply not produced; that would not be discernable from this data).

Please Change: “Changes in short-term microbial EPS production efficiency were previously calculated by standardizing soil EPS to soil microbial ATP², a proxy for microbial biomass. This EPS-C/MBC ratio may be used to assess the potential of microorganisms to secrete EPS under various environmental conditions². However, as mentioned above, the accumulation of soil EPS is contingent on the dynamic balance between its generation, stabilization, and degradation” (Page 12, Lines 358-362). TO: “Changes in short-term microbial EPS production efficiency were previously calculated by standardizing soil EPS to soil microbial ATP⁸, as an independent indicator of the soil microbial biomass. The EPS-C/MBC ratio of soils can also be used, and in natural soils, represents net result of all historic influences contributing to the current state (whether steady or dynamic). As mentioned above, the accumulation of soil EPS is contingent on the dynamic balance between its generation, stabilization, and degradation” (Page 12, Lines 358-362).

[Response 1-1 on R1-1] Thank you for this careful and constructive comment. We acknowledge that the previous wording could still be interpreted as implying EPS production or allocation efficiency. Following your guidance, we have revised or removed the relevant statements to avoid this implication (**Page 10, Lines 277-279; Page 12, Lines 346-351**).

[Comment 1-2 on R1-2] R1-2 – *The authors have missed the point. EPS are not “poorly quantified” in all senses. I requested to replace this sentence with “However, systematic evaluation of this claim has yet to be reported” – which is far more informative, and far less wrong. Instead of making the recommended improvement, the authors made the original problem worse. Please do as asked.*

[Response 1-2 on R1-2] Thank you for this important comment. We apologize that our previous revision did not adequately address this issue and, instead, resulted in less precise wording. We appreciate your careful and detailed guidance on this sentence in both **Comment R1-1 and Comment 1-7 on R4-7**. Following your recommendation in **Comment 1-1 (Please see above)**, we have now replaced the original sentence verbatim with the proposed wording (**Page 3, Lines 23-24**).

[Comment 1-3 on R1-3] R1-3 – *the resistance on the correction is continued. Please action R1-3 as asked.*

[Response 1-3 on R1-3] This has been addressed. We have revised the sentence from “Yet, their abundance and large-scale environmental and microbial controls remain poorly quantified. We addressed this gap by conducting extensive soil sampling across a European transect spanning diverse climates, bedrock types, and land uses” to “**Yet, their abundance and large-scale environmental and microbial controls have only begun to be investigated**” (Page 3, Lines 23-24). Please also refer to our *Response to Comment 1-1, Comment 1-7 on R4-7, and Comment 1-2 on R1-2 (Please see above)*.

[Comment 1-4 on R1-5] R1-5 – Not satisfied. Confounded logic and clinging to concepts not supported by the data: “EPS production potential” suffers from similar limitations to “EPS production efficiency” in this context. “While soil properties such as clay content and exchangeable calcium enhanced EPS retention”. Again, how do the authors think there is evidence for this level of detail and confidence (rhetorical). I request the authors desist from unsubstantiated speculation in all further correspondence.

[Response 1-4 on R1-5] Following your comment, we removed all references to “EPS production potential” and “EPS retention” in the main manuscript. Detailed revisions regarding “EPS production potential” can be found in our response to *Comment 4-2 (Please see above)*, and those regarding “EPS retention” are provided in our response to *Comment 1-8 on 4-9 (Please see above)*. Accordingly, we have revised the sentence from “Across the transect, EPS increased with microbial biomass as a proxy for EPS production potential, while soil properties such as clay content and exchangeable calcium enhanced EPS retention” to a purely descriptive statement: “**Across the transect, EPS was positively associated with microbial biomass (MBC) and also showed positive associations with soil clay content and exchangeable calcium**” reporting observed associations only, without implying mechanisms or causality (Page 3, Lines 31-32).

[Comment 1-5 on R1-6] R1-6 – “EPS increased with microbial biomass as a proxy for EPS production potential?” No. MBC is MBC. And we know nothing of the “EPS production potential”. Please refer to my original request.

[Response 1-5 on R1-6] Thank you for this clarification. We apologize for the previous wording, which did not accurately reflect our results. We have revised the sentence from “Across the transect, EPS increased with microbial biomass as a proxy for EPS production potential, while soil properties such as clay content and exchangeable calcium enhanced EPS retention” to a purely descriptive statement: “**Across the transect, EPS was positively associated with microbial biomass (MBC) and also showed positive associations with soil clay content and exchangeable calcium**” reporting observed associations only, without implying mechanisms or causality (Page 3, Lines 31-32). Please refer to our *Response on Comment 1-4 on R1-5* for details (Please see above).

[Comment 1-6 on R1-22] R1-22A: Refer to my requested change for R1-1 AND delete “intermittent” later in the paragraph, and then delete the following imagined overreach/speculation which is presented as fact: “Overall, bedrock influenced EPS through

(1) promoting microbial biomass and EPS secretion via bedrock-driven environmental conditions, and (2) shaping soil texture and mineralogy, which governs EPS retention and intermittently protects it from microbial degradation”. You simply don’t have such granularity of data. (Pages 9-11, Lines 259-307).

[Response 1-6 on R1-22] Thank you for this valuable comment and for the clear guidance. Following your recommendation and **Comment 1-15 on R4-45 (Please see above)**, we have deleted the term “*intermittent*” and removed the concluding statement.

[Comment 1-7 on R1-22] R1-22B: “In culture experiments, it is also the loosely bound fraction of EPS that is extracted by CER, whereas the tightly bound fraction remains unextracted”. Where is the reference for this? and what is your definition of “loosely” and “tightly bound”? What value is this subjective definition? Why assume culture findings apply to soils? What about other multivalent metals, why assume some EPS are not occluded in aggregates when this has not been investigated? I strongly recommend to delete the above statement at the end of the paragraph - and leave it to The Editors discretion as to whether Supplementary Discussion 3 should be included at all. In the very least, the language should be less confident. Eg “EPS in soils is thought stabilized via multiple mechanisms”, then adding appropriate “thoughts” and “claimed” to each of the existing statements, ensuring a reference is given to support each - without adding any new statements.

[Response 1-7 on R1-22] Thank you for this detailed and thoughtful clarification. We acknowledge that the statement in our previous response letter — “In culture experiments, it is also the loosely bound fraction of EPS that is extracted by CER, whereas the tightly bound fraction remains unextracted” — was imprecisely defined. We also agree that the mechanistic understanding of EPS stabilization in soils remains incomplete and that our original wording was overly confident. We have therefore revised the paragraph to use more cautious language (e.g., “thought to”, “suggested”, “proposed”, “considered”) and ensured that each statement is supported by appropriate literature references. No new claims were added, and the list of proposed stabilization mechanisms was retained as requested. We believe this revised wording more accurately reflects the current state of knowledge and associated uncertainties. **Please see below:**

Original Supplementary Discussion 3:

Supplementary Discussion 3 - Mechanisms of EPS stabilization in soils

EPS in soils is stabilized via multiple mechanisms^{4, 12, 18}, including (i) physical occlusion inside microaggregates, (ii) co-precipitation and complexation with Fe/Al, (iii) hydrogen bonding, (iv) hydrophobic interactions, (v) inner-sphere ligand exchange to Fe/Al (oxyhydr)oxides and clays, (vi) outer-sphere electrostatic attraction, and (vii) multivalent cation bridging between negatively charged sites of EPS and those on other polymers and minerals. The importance of these mechanisms differs between main EPS compounds, with EPS polysaccharides mainly bound/stabilized by (ii, iii, v, vii) and EPS proteins via (ii, iii, iv, v, vi, vii). The cation bridging mechanism via Ca²⁺ (vii) and electrostatic attraction (vi) causes a loosely bound (not strongly stabilized) association, which is reversible; it is this EPS fraction that is targeted by the CER method due to removing Ca²⁺ by CER and the extraction buffer.

weakening electrostatic forces. Binding of EPS by other mechanisms causes weaker binding and stabilization (iii) or stronger binding/immobilization and stabilization (ii, iv, v), but these latter EPS fractions are not targeted or extracted by CER. Since comprehensive studies across EPS components, sources, and soil types are still lacking, this remains poorly understood. In culture experiments, it is also the loosely bound fraction of EPS that is extracted by CER, whereas the tightly bound fraction remains unextracted.

Revised Supplementary Discussion 1 (Supplementary Discussion 3 in the previous version):

Supplementary Discussion 1 - Mechanisms of EPS - soil particle associations

EPS in soils is thought to associate with soil particles through multiple mechanisms, including (i) physical occlusion inside microaggregates, (ii) co-precipitation and complexation with Fe/Al, (iii) hydrogen bonding, (iv) hydrophobic interactions, (v) inner-sphere ligand exchange to Fe/Al (oxyhydr)oxides and clays, (vi) outer-sphere electrostatic attraction, and (vii) multivalent cation bridging between negatively charged sites of EPS and those on other polymers and minerals^{4, 12, 18}. The relative importance of these mechanisms is suggested to differ among major EPS compound classes, with EPS polysaccharides primarily binding via mechanisms (ii, iii, v, vii), and both can promote aggregate formation and are therefore affected (i) and can be occluded in microaggregates^{4, 12, 18}.

Cation bridging via Ca^{2+} (vii) and electrostatic attraction (vi) are generally considered to result in relatively loose and reversible associations. This EPS fraction is therefore assumed to be preferentially targeted by the CER extraction method, as Ca^{2+} removal and changes in ionic strength weaken these interactions. In contrast, EPS associated with soil particles via other mechanisms is thought to result in weaker associations (e.g., hydrogen bonding; iii) or stronger interactions (e.g., co-precipitation, hydrophobic interactions, and inner-sphere complexation; ii, iv, v), but these fractions are not expected to be efficiently extracted by CER^{4, 12, 18}.

Because comprehensive studies covering different EPS components, sources, and soil types are still lacking, the relative contribution of these EPS-soil association mechanisms remains poorly constrained.

We will await the Editor's decision regarding whether **Supplementary Discussion 1** should be retained.

[Comment 1-8 on R1-27] R1 – 27 A) You need to include the reference (16) B) your statement that the effect of land use on soil EPS was not significant (Fig. 2b) is in error See Fig 3d. This detail: that EPS polysaccharide and EPS protein are related to bedrock, and land-management, respectively, is a key part of the findings! Please reinstate to the main discussion.

[Response 1-8 on R1-27] Thank you for your helpful comment. Following your suggestion and in line with *Comment 1-4 (Please see above)*, we have reinstated the land-use-related discussion in the main text and corrected our previous statement regarding the significance of land-use effects on soil EPS.

We have added the following sentences to the **Discussion** following your suggestion: **We found land-use was not a significant driver of EPS polysaccharide, but was a highly significant driver of EPS protein (Fig.3). This adds important depth to land-use research. Redmile-Gordon**

et al.¹⁵ found that land-use change to perennial grassland increased soil EPS and aggregate stability, with EPS protein being far more influential for soil structural stability than EPS polysaccharides¹⁵. Our findings, that both EPS protein and EPS-C are elevated under grassland (across contrasting land management types), support their assertion that CER-extracted EPS protein is more physically influential than EPS polysaccharides in shaping soil aggregates¹⁵ - and thus also C stabilization (Supplementary Discussion 2). (Page 11, Lines 296-303).

In addition, we now cite Redmile-Gordon et al. (2020) to support the revised interpretation of land-use effects on EPS: *Redmile-Gordon, M., Gregory, A. S., White, R. P. & Watts, C. W. Soil organic carbon, extracellular polymeric substances (EPS), and soil structural stability as affected by previous and current land-use. Geoderma 363, 114143 (2020).*

We have also revised **Supplementary Discussion 4** to ensure consistency with these changes and to avoid any misleading statements regarding land-use effects.

Original Supplementary Discussion 4:

Supplementary Discussion 4 - Land use effects on soil EPS contents and EPS-C/MBC ratios

Among the three land-use types, EPS contents were highest in grassland soils compared to cropland and woodland soils, although the differences did not reach statistical significance (Fig. 2b). To identify potential drivers of EPS content variation, we found that MBC was significantly affected by land use and showed a pattern consistent with EPS (Supplementary Table 5). This, together with the positive correlation between MBC and EPS (Fig. 2c), suggests that higher MBC may indicate larger microbial populations, thereby representing a greater potential for EPS production and serving as a primary control on EPS content across land-use types. In addition, land-use exerted a stronger influence on soil nutrient status (e.g., SOC, TOP, TP, Mg_e, and NO₃⁻) and fine root traits (FRB and Root N) than bedrock (Supplementary Table 5). Among these, SOC and TOP were significantly higher in grassland than in cropland and positively correlated with EPS, while root N was also highest in grasslands, whereas FRB peaked in woodlands but remained relatively high in grasslands. These indicators suggest that improved nutrient availability and greater plant inputs (root turnover and exudation) may enhance microbial EPS production⁴⁹. This interpretation is consistent with previous findings: Redmile-Gordon et al.¹ observed the highest EPS contents in grasslands, while Kidinda et al.⁷ reported higher EPS in fertilized croplands than in forests on nutrient-poor tropical soils. Together, these studies suggest that land-use effects on EPS are primarily mediated by plant inputs and nutrient availability. Conversely, the lowest MBC and SOC observed in cropland soils may reflect soil degradation or organic matter depletion caused by long-term intensive management practices such as tillage, which could in turn contribute to the reduced EPS levels. For instance, frequent tillage can disrupt soil structure and aggregates, thereby reducing soil organic matter content⁵⁰. On the other hand, although croplands had the highest clay and exchangeable Ca²⁺ contents—factors potentially favouring short-term EPS retention—their EPS levels were lowest. This suggests that limited labile C availability, linked to lower SOC and FRB in croplands, may play a more decisive role in constraining microbial EPS content across the investigated soils. Substantial differences in the composition and function of soil microbial communities under different land use types⁵¹ may also cause differences in EPS

formation and decomposition⁵². While microbial diversity was not investigated in this study, in accordance with investigations of legacy vs current management effects on EPS¹, it is primarily the interplay between plant inputs and anthropogenic management that causes these differences in total EPS and EPS protein contents. Interestingly, the ranking of EPS-C/MBC did not mirror that of EPS contents, with croplands showing the highest EPS-C/MBC, followed by grasslands and woodlands (Fig. 7b). Given that both EPS and MBC were lowest in cropland soils, this highest EPS-C/MBC ratio suggests a greater microbial investment in EPS production per unit biomass. This may be a response to increased water stress in croplands, as indicated by significantly lower WHC and SMC compared to woodlands and grasslands (Supplementary Table 5).

Revised Supplementary Discussion 2 (Supplementary Discussion 4 in the previous version):

Supplementary Discussion 4 - Land use effects on soil EPS contents and EPS-C/MBC ratios

Among the three land-use types, EPS protein content differed significantly, being highest in grassland soils, whereas EPS polysaccharide and total EPS contents did not differ significantly (Figs. 1, 2). To identify potential drivers of EPS content variation, we found that MBC was significantly affected by land use and showed a pattern consistent with EPS (Supplementary Table 4). This, together with the positive correlation between MBC and EPS (Fig. 2c), suggests that higher MBC is associated with higher EPS content across land-use types. In addition, land-use exerted a stronger influence on soil nutrient status (e.g., SOC, TOP, TP, Mg_e, and NO₃⁻) and fine root traits (FRB and Root N) than bedrock (Supplementary Table 4). Among these, SOC and TOP were significantly higher in grassland than in cropland and positively correlated with EPS, while root N was also highest in grasslands, whereas FRB peaked in woodlands but remained relatively high in grasslands. These indicators, such as nutrient availability and plant inputs (root turnover and exudation), are linked to higher EPS content as observed by others⁴⁹. This interpretation is consistent with previous findings: Redmile-Gordon et al.¹ observed the highest EPS contents in grasslands, while Kidinda et al.⁷ reported higher EPS in fertilized croplands than in forests on nutrient-poor tropical soils. Together, these studies suggest that land-use effects on EPS are primarily mediated by plant inputs and nutrient availability. Conversely, the lowest MBC and SOC observed in cropland soils may reflect soil degradation or organic matter depletion caused by long-term intensive management practices such as tillage, which could, in turn, contribute to the reduced EPS levels. For instance, frequent tillage can disrupt soil structure and aggregates, thereby reducing soil organic matter content⁵⁰. On the other hand, although croplands had the highest clay and exchangeable Ca²⁺ contents—factors increasing the sorptive surface area and the sorption potential for EPS—their EPS levels were lowest. This suggests that limited labile C availability, linked to lower SOC and FRB in croplands, may play a more decisive role in constraining microbial EPS content across the investigated soils. Substantial differences in soil microbial communities under different land use types⁵¹ may also be associated with the observed EPS pool⁵². While microbial diversity was not investigated in this study, in accordance with investigations of legacy vs current management effects on EPS¹, it is primarily the interplay

between plant inputs and anthropogenic management that causes these differences in total EPS and EPS protein contents¹.

Following the reviewers' comments, we deleted EPS-C/MBC related content in Supplementary Discussion 4.

[Comment 1-9] *Once the authors have addressed these structural and methodological points, I would be pleased to provide a brief follow-up review to verify that the key corrections have been implemented, rather than line-editing the entire text at this stage.*

[Response 1-9] Thank you for your constructive comments and tremendous help in improving our manuscript quality. We have revised the manuscript to address the structural and methodological points raised. We appreciate your willingness to provide a follow-up review to verify that the key corrections have been implemented.

References

1. Redmile-Gordon M, Gregory AS, White RP, Watts CW. Soil organic carbon, extracellular polymeric substances (EPS), and soil structural stability as affected by previous and current land-use. *Geoderma* **363**, 114143 (2020).
2. Throckmorton HM, Bird JA, Dane L, Firestone MK, Horwath WR. The source of microbial C has little impact on soil organic matter stabilisation in forest ecosystems. *Ecol Lett* **15**, 1257-1265 (2012).
3. Liu X, *et al.* Linking microbial immobilization of fertilizer nitrogen to in situ turnover of soil microbial residues in an agro-ecosystem. *Agric Ecosyst Environ* **229**, 40-47 (2016).
4. Costa OY, Raaijmakers JM, Kuramae EE. Microbial extracellular polymeric substances: ecological function and impact on soil aggregation. *Front Microbiol* **9**, 1636 (2018).
5. Quan C, Han S, Utescher T, Zhang C, Liu Y-SC. Validation of temperature–precipitation based aridity index: Paleoclimatic implications. *Palaeogeogr Palaeoclimatol Palaeoecol* **386**, 86-95 (2013).
6. UNEP. World Atlas of Desertification. In: <https://wedocs.unep.org/20.500.11822/42137> (1992).
7. Kidinda LK, Babin D, Doetterl S, Kalbitz K, Mujinya BB, Vogel C. Extracellular polymeric substances are closely related to land cover, microbial communities, and enzyme activity in tropical soils. *Soil Biol Biochem* **187**, 109221 (2023).

8. Redmile-Gordon MA, Evershed RP, Hirsch PR, White RP, Goulding KWT. Soil organic matter and the extracellular microbial matrix show contrasting responses to C and N availability. *Soil Biol Biochem* **88**, 257-267 (2015).
9. Ling Q, *et al.* Bedrock geochemistry regulates glomalin-related soil protein accrual in subtropical karst forest soils, Southwest China. *Ecol Indic* **176**, 113680 (2025).
10. Khan SU, Hooda PS, Blackwell MS, Busquets R. Microbial biomass responses to soil drying-rewetting and phosphorus leaching. *Frontiers in Environmental Science* **7**, 133 (2019).
11. Zhang M, *et al.* Characterising soil extracellular polymeric substances (EPS) by application of spectral-chemometrics and deconstruction of the extraction process. *Chem Geol* **618**, 121271 (2023).
12. Zhang M, Wu Y, Qu C, Huang Q, Cai P. Microbial extracellular polymeric substances (EPS) in soil: From interfacial behaviour to ecological multifunctionality. *Geo-Bio Interfaces* **1**, e4 (2024).
13. Luo Y, *et al.* Effects of different soil organic amendments (OAs) on extracellular polymeric substances (EPS). *Eur J Soil Biol* **121**, 103624 (2024).
14. Bölscher T, *et al.* Beyond growth: The significance of non-growth anabolism for microbial carbon-use efficiency in the light of soil carbon stabilisation. *Soil Biol Biochem* **193**, 109400 (2024).
15. Bublitz TA, Oliva RL, Hupe A, Joergensen RG. Optimization of the bicinchoninic acid assay for quantifying carbohydrates of soil extracellular polymeric substances. *Plant Soil* **498**, 699-709 (2024).
16. Redmile-Gordon MA, Brookes PC, Evershed RP, Goulding KWT, Hirsch PR. Measuring the soil-microbial interface: extraction of extracellular polymeric substances (EPS) from soil biofilms. *Soil Biol Biochem* **72**, 163-171 (2014).
17. Zethof JH, *et al.* Prokaryotic community composition and extracellular polymeric substances affect soil microaggregation in carbonate containing semiarid grasslands. *Front Environ Sci* **8**, 51 (2020).
18. Huang L, *et al.* A review of the role of extracellular polymeric substances (EPS) in wastewater treatment systems. *Int J Environ Res Public Health* **19**, 12191 (2022).
19. Hong Z, Chen W, Rong X, Cai P, Dai K, Huang Q. The effect of extracellular polymeric substances on the adhesion of bacteria to clay minerals and goethite. *Chem Geol* **360**, 118-125 (2013).

20. Huang P. Soil mineral-organic matter-microorganism interactions: fundamentals and impacts. *Adv Agron* **82**, 393-472 (2004).
21. Chenu C. Clay—or sand—polysaccharide associations as models for the interface between micro-organisms and soil: water related properties and microstructure. In: *Soil Structure/Soil Biota Interrelationships*. Elsevier (1993).
22. Feng M, *et al.* Increased microbial extracellular polymeric substances as a key factor in deep soil organic carbon accumulation. *Soil Biol Biochem* **212**, 109998 (2026).
23. Li H, *et al.* Increased drought intensity stimulates the extracellular polymeric substance accumulation and their contribution to soil organic carbon rather than microbial necromass. *Soil Biol Biochem*, 110044 (2025).
24. Buckeridge KM, Creamer C, Whitaker J. Deconstructing the microbial necromass continuum to inform soil carbon sequestration. *Funct Ecol* **36**, 1396-1410 (2022).
25. Shi K, *et al.* Accumulation of soil microbial extracellular and cellular residues during forest rewilding: Implications for soil carbon stabilization in older plantations. *Soil Biol Biochem* **188**, 109250 (2024).
26. Liang C, Amelung W, Lehmann J, Kästner M. Quantitative assessment of microbial necromass contribution to soil organic matter. *Global Change Biol* **25**, 3578-3590 (2019).
27. Ni X, *et al.* A quantitative assessment of amino sugars in soil profiles. *Soil Biol Biochem* **143**, 107762 (2020).
28. Flemming H-C, Neu TR, Wozniak DJ. The EPS matrix: the “house of biofilm cells”. *J Bacteriol* **189**, 7945-7947 (2007).
29. Bublitz TA, *et al.* Soil extracellular polymeric substances and microbial biomass react differently to field induced drought stress in contrasting cropping systems at different wheat developmental stages. *Biol Fertil Soils*, 1-13 (2025).
30. Flemming H-C, Wingender J. The biofilm matrix. *Nat Rev Microbiol* **8**, 623-633 (2010).
31. Roberson EB, Firestone MK. Relationship between desiccation and exopolysaccharide production in a soil *Pseudomonas* sp. *Appl Environ Microbiol* **58**, 1284-1291 (1992).
32. Chen Y, *et al.* Quantifying the trade-off between yield and contamination in soil EPS extraction using cation exchange resin. *J Soils Sediments*, 1-9 (2025).

33. Bérard A, *et al.* Exopolysaccharides in the rhizosphere: A comparative study of extraction methods. Application to their quantification in Mediterranean soils. *Soil Biol Biochem* **149**, 107961 (2020).
34. Redmile-Gordon M, Chen L. Zinc toxicity stimulates microbial production of extracellular polymers in a copiotrophic acid soil. *Int Biodeterior Biodegrad* **119**, 413-418 (2017).
35. Hale L, Curtis D, Leon N, McGiffen Jr M, Wang D. Organic amendments, deficit irrigation, and microbial communities impact extracellular polysaccharide content in agricultural soils. *Soil Biol Biochem* **162**, 108428 (2021).
36. Bettermann A, *et al.* Importance of microbial communities at the root-soil interface for extracellular polymeric substances and soil aggregation in semiarid grasslands. *Soil Biol Biochem* **159**, 108301 (2021).
37. Baumert VL, *et al.* Root-induced fungal growth triggers macroaggregation in forest subsoils. *Soil Biol Biochem* **157**, 108244 (2021).
38. Crouzet O, *et al.* Soil photosynthetic microbial communities mediate aggregate stability: influence of cropping systems and herbicide use in an agricultural soil. *Front Microbiol* **10**, 1319 (2019).
39. Vuko M, Cania B, Vogel C, Kublik S, Schloter M, Schulz S. Shifts in reclamation management strategies shape the role of exopolysaccharide and lipopolysaccharide-producing bacteria during soil formation. *Microb Biotechnol* **13**, 584-598 (2020).
40. Liao J, *et al.* Unreported role of earthworms as decomposers of soil extracellular polymeric substance. *Appl Soil Ecol* **197**, 105325 (2024).
41. Liu C, *et al.* Arbuscular mycorrhizal fungi hyphal density rather than diversity stimulates microbial necromass accumulation after long-term Robinia pseudoacacia plantations. *Soil Biol Biochem* **206**, 109817 (2025).
42. Peng Y, Zhang H, Lv Z, Zhang J, Li G. Microbial inoculation improves soil aggregation by enhancing exopolysaccharides and lipopolysaccharides-related gene abundance in saline soil. *Appl Soil Ecol* **214**, 106388 (2025).
43. Bogar G, Lennon J, Vander Stel H, Evans S. Microscale assay for the quantification of total polysaccharides to estimate extracellular polymeric substances (EPS) in soil. *J Microbiol Methods*, 107324 (2025).
44. Wu Y, *et al.* Soil biofilm formation enhances microbial community diversity and metabolic activity. *Environ Int* **132**, 105116 (2019).

45. De Beeck MO, Persson P, Tunlid A. Fungal extracellular polymeric substance matrices—highly specialized microenvironments that allow fungi to control soil organic matter decomposition reactions. *Soil Biol Biochem* **159**, 108304 (2021).
46. Adessi A, de Carvalho RC, De Philippis R, Branquinho C, da Silva JM. Microbial extracellular polymeric substances improve water retention in dryland biological soil crusts. *Soil Biol Biochem* **116**, 67-69 (2018).
47. Flemming H-C, Wingender J, Szewzyk U, Steinberg P, Rice SA, Kjelleberg S. Biofilms: an emergent form of bacterial life. *Nat Rev Microbiol* **14**, 563-575 (2016).
48. Ates O. Systems biology of microbial exopolysaccharides production. *Front Bioeng Biotechnol* **3**, 200 (2015).
49. Rillig MC, Mummey DL. Mycorrhizas and soil structure. *New Phytol* **171**, 41-53 (2006).
50. Wang Z, *et al.* Human-induced erosion has offset one-third of carbon emissions from land cover change. *Nat Clim Change* **7**, 345-349 (2017).
51. Drenovsky RE, Steenwerth KL, Jackson LE, Scow KM. Land use and climatic factors structure regional patterns in soil microbial communities. *Global Ecol Biogeogr* **19**, 27-39 (2010).
52. Donot F, Fontana A, Baccou J, Schorr-Galindo S. Microbial exopolysaccharides: main examples of synthesis, excretion, genetics and extraction. *Carbohydr Polym* **87**, 951-962 (2012).

ROUND 2 REVIEWER
1 ATTACHMENT:
Part 1: R4's comments

I share the majority of R4's concerns around the use of dried aggregates, and interfering C. Nonetheless, I think the Authors' original data, and experience here is still of high value (including state-of-the-art learnings, and broad environmental relevance). Their expansive study area, and attention to EPS as a potential source of stabilized C, warrant attention. However, their focus should stay within that scope. This means several of the added revisions referencing the additional work will need removing. They should simply present the caveats more clearly, as suggested, below.

R4's comments and with regard to additional work:

- 1) *most importantly, regarding the methodology: EPS has been extracted from dry soil samples, usually done on fresh soil samples, and it is known that microbial processes change on a short-timescale based on sample preparation and storage.*
- 2) *Overestimation from a flush of C from drying, is quite likely, as is biological response for greater EPS exudation.*

Ad-hoc adaptation of the method -whether omitting the pre-extraction (Reference #46), or using dried aggregates (current study)- means that results are not directly comparable with the work of most others. The data remain useful as a self-contained dataset for analysis. While the use of dry aggregates has limitations mentioned by R4 – many of these are largely nullified by pre-extraction (Zhang at al., 2024; evidence given below). However, there are much larger weaknesses in the **additional work conducted**. Weaknesses include confounding **1)** laboratory change effects over 4 years **2)** rewetting soils stored in artificial conditions **3)** assuming factors identical to a natural soil at state of recent sampling, and **4)** microbial community and function both compromised by drying and storage.

Accordingly, my further recommendations are:

- 1) Delete extra work: Supplementary Tables 3 & 4 and all references to them.**
There are too many assumptions behind the data, and they are potentially misleading – they imply false equivalence to freshly sampled (moist) soil – when rewetted (Table 4) and embody a very small dataset with insufficient meaning (Table 3).
- 2) Investigate cited studies and remove all** references to other studies in all text and tables who i) skipped the pre-extraction step, or ii) were performed in

cultures. **This includes Reference #46** needing removal from Supplementary Table 1. Please could the authors ensure they remove any others (they are rare).

Further recommendations regarding the issue of dry aggregate extractions follow below.

R4 also questioned:

whether it makes sense to combine the various EPS components, as there could also be overlaps between the different components.

This is laudable. However, some small overlap seems reasonable considering the state-of-the-art. The authors have *excessively conceded* in response to R4 (**R4-27**; addressed in sequence below). I'd be happy to review the final version – provided the authors double-check all their responses, and keep them minimal, accurate, and manageable.

To save others from pitfalls of over-interpreting/comparing data gathered from dried aggregates – AND *worse ad hoc changes such as skipping pre-extraction*, it is important to establish early in **The Discussion** that results should be contained to the present study, and present reasonable boundaries of the approaches. **LINES 318-323**: are the perfect place to resolve this. This is where the core value and clarity need to be established so that what follows makes sense. Accordingly,

3) I suggest authors **delete**:

“This difference partly reflects differences in bedrock and land-use composition, sampling season, and methodological factors (Supplementary Discussion 1). Due to sample availability, we used air-dried soils, whereas most previous studies used fresh soils. Our preliminary experiment showed minor effects of air-drying: EPS polysaccharides decreased by 1 % 3%, EPS proteins increased by 9 % 1%, and total EPS (polysaccharides + proteins) increased by 4 % 3%. Long-term soil storage had similarly no impact on EPS content (Supplementary Discussion 2).

4) And **INSERT verbatim**: -at line 318-

“To keep sampling efforts manageable at continental scale, we extracted EPS from air-dried aggregates. It is important to note that air-drying can cause artefacts: triggering cell-lysis (Khan et al., 2019) and increasing EPS production (Costa et al., 2018) in the short period before the dry state is reached. However, the majority of increased C-release from soils subjected to drying is removed during the EPS pre-extraction step (Zhang et al. 2023). While our use of *dried* aggregates prevents direct comparison with the majority of EPS studies which extract *moist* soils (e.g. Zhang et al., 2024[ref 5], Luo et al., 2024[ref 42]; Bolscher et al., 2024[relevant citation to

Commented [A1]: Khan, S.U., Hooda, P.S., Blackwell, M.S.A., & Busquets, R. (2019). *Microbial Biomass Responses to Soil Drying-Rewetting and Phosphorus Leaching*. *Frontiers in Environmental Science*, 7:133.

add]), our large dataset enables the relative effects of geological, geographical and edaphic factors on soil EPS concentrations to be investigated. Other researchers have also performed ad hoc changes to the standard soil EPS extraction protocol. For example, Bublitz et al. (2024) claimed that most soils do not contain measurable amounts of non-EPS carbohydrate. However, this unsubstantiated claim is at odds with the majority of soil EPS research where a purification (or “pre-extraction”) step is included to remove confounding non-EPS proteins and polysaccharides (Redmile-Gordon et al., 2014). Zhang et al. (2023) deconstructed the EPS extraction process, and investigated the quality of both pre-extracts and EPS-extracts. They found that artefacts from EPS extraction of air-dried soil were much smaller than the large amounts of easily detectable non-EPS proteins and polysaccharides removed during the purifying, “pre-extraction” step. This emphasizes that while EPS extraction from dry aggregates invites limitations, these are far smaller than the problems occurring when the pre-extraction step is omitted. With some of our soils receiving organic inputs and/or significant root exudates over the continent - confidence between different managements was vital. Therefore, we guarded against EPS overestimation by including the purification steps described by Redmile-Gordon et al. (2014), which largely eliminates any errors from overlooking hydration status. Nonetheless, our data should not be compared directly with those of others who ensure full physico-chemical integrity of EPS by avoiding desiccation, as described in the original and complete method (Redmile-Gordon et al. 2014)”.

Commented [A2]: Bölscher, T., Vogel, C., Olagoke, F.K., Meurer, K.H.E., Herrmann, A.M., Colombi, T., Brunn, M., & Domeignoz-Horta, L.A. (2024). Beyond growth: The significance of non-growth anabolism for microbial carbon-use efficiency in the light of soil carbon stabilisation. *Soil Biology and Biochemistry*, **193**, 109400. <https://doi.org/10.1016/j.soilbio.2024.109400>

Luo, Y., Gonzalez Lopez, J.B., van Veelen, H.P.J., Kok, D.J.D., Postma, R., Thijssen, D., Sechi, V., ter Heijne, A., Bezemer, T.M., & Buisman, C.J.N. (2024). Effects of different soil organic amendments (OAs) on extracellular polymeric substances (EPS). *European Journal of Soil Biology*, **121**, 103624. <https://doi.org/10.1016/j.ejsobi.2024.103624>

- 5) **R4-3 Delete Supplementary Discussion 1 and Supplementary Discussion 2** which add to the confusion on further analyses conducted (the arguments are rightly not accepted by R4 anyway). Weaknesses include confounding 1) laboratory change effects over 4 years 2) rewetting soils stored in artificial conditions 3) assuming factors identical to a natural soil at state of recent sampling, and 4) microbial community and function both compromised by drying and storage. *In preference, the addition to the discussion above (Suggestion 4) deals with the approach chosen, the limitations, the context, the strength and the clarity for newcomers all in one.* **NOTE:** This removal of the confounding additional work requires also rolling back several of the current additions which the authors made in response to **R4:**
- 6) **R4-5:** Not satisfactory. Resolved by my suggestion in response to **R1-22** (part 2 below).
- 7) **R4-7:** Misleading. EPS are not “poorly quantified” Resolved by my suggestion in response to **R1-5** (part 2 below). I requested to replace this sentence with “*However, systematic evaluation of this claim has yet to be reported*”. Please do as asked.

- 8) **R4-9:** Not satisfactory. Resolved by my suggestion in response to **R1-5** (part 2 below).
- 9) **R4-15** I agree, authors response not satisfactory. Resolved by insertion to discussion, **Point 4** above.
- 10) **R4-18:** correct the logic and rewrite (**L 99-103**) to: “*EPS secretion is governed by microbial growth and environmental stress, soil properties, nutrient availability*” (a C source for EPS is not stress; remove the nested syntax).
- 11) **R4-27:** some small overlap between protein and carbohydrate seems reasonable considering the state-of-the-art, and the authors approach remains useful. I expect the overlap is likely minor (majority of C residing in either a polysaccharide monomer, or an amino acid in a protein) However, the authors seem to have oversimplified in response to R4 with “*for example, glycoproteins which would be doublecounted*”. However, “double-counting” is a gross overstatement. Glycoproteins are likely vulnerable to partial degradation or fractionation during cation exchange resin (CER) extraction of EPS and the assumption that they are simply “double-counted” in both protein and polysaccharide assays is an oversimplification. The acidic microenvironment around the resin beads can lower local pH, potentially leading to partial hydrolysis of glycosidic or peptide bonds. Furthermore, Colorimetric assays (e.g., Lowry, Bradford for proteins; phenol-sulfuric acid for carbohydrates) detect specific reactive groups, not intact macromolecules. In summary **Assays do not truly double-count** glycoproteins; rather, they may **partially detect** their components, leading to **incomplete or skewed quantification**, but nothing is perfect. This is the state-of-the-art. Replace “*for example, glycoproteins which would be doublecounted*” with “*for example, measurement of glycoproteins which are part detected in both colorimetric protein and polysaccharide assays*”

R4-33: The authors current response is superseded by the suggested addition to main discussion above. **Importantly, the authors response to R4-33 shows another area where confusion is being propagated:** The 1st reference in the manuscript (Oliva et al. 2024) conflates **soil EPS** with **culture-derived EPS** *and does not separate the confounding carbon that R4 refers to*. The cited authors omitted any pre-extraction steps and therefore included soluble metabolites, broth residues, and necromass products. The paper reports that new (necromass) indicators (GalN and MurN) show “100 % contribution from the EPS fraction”, but this figure arises because both their “EPS extract” and “total extract” measured largely the same soluble pool—an artefact of non-independent fractions, certainly not proof of EPS exclusivity, or even being “an integral parts of EPS”. This may explain some of the current confusion and references to Oliva et al. (2024). Such work should not open a study on soil EPS; other established soil-based studies provide a more

appropriate foundation. Accordingly, please promote the following reference (Costa, currently #8) to citation number 1, and remove Olivia et al. (2024).

Costa OY, Raaijmakers JM, Kuramae EE. Microbial extracellular polymeric substances: ecological function and impact on soil aggregation. *Front Microbiol* **9**, 1636 (2018).

Promote to share position with the work already currently at citation #2

Redmile-Gordon MA, Evershed RP, Hirsch PR, White RP, Goulding KWT. Soil organic matter and the extracellular microbial matrix show contrasting responses to C and N availability. *Soil Biol Biochem* **88**, 257-267 (2015).

Furthermore, since the current manuscript does not investigate or measure “extracellular residues” (a subjective and arguable definition; with only necromass, microbial biomass, and EPS measured here) To reflect the state-of-the-art more accurately....

12) Please replace LINES 57-58: “Soil microbial residues are essential components of the soil carbon (C) pool and can be classified as cellular and extracellular residues (1, 2)”. **WITH** “Soil microbial extracellular polymeric substances (EPS) are essential components of the soil carbon (C) pool (Costa et al., 2018; Redmile-Gordon et al., 2014) (1, 2).

13) Please Check: R4-36 “In contrast, the microbial composite factor, derived from PCA of CUE, Cgrowth, qGrowth, and MBN (Fig. 8c; Supplementary Fig. 1), exerted a direct positive effect on EPS-C/MBC ratio and a negative effect on EPS-C (Fig. 8a). If the effect on EPS-C is negative (and more negative than on MBC) then the effect on EPS-C/MBC is also negative, no? Are these the right way around?

14) R4-41: Delete Lines 253-258 (in favour of the discussion inclusion, point 4): “This difference partly reflects differences in bedrock and land-use composition, sampling season, and methodological factors (Supplementary Discussion 1). Due to sample availability, we used air-dried soils, whereas most previous studies used fresh soils. Our preliminary experiment showed minor effects of air-drying: EPS polysaccharides decreased by 1 [±] 3%, EPS proteins increased by 9 [±] 1%, and total EPS (polysaccharides + proteins) increased by 4 [±] 3%. Long-term soil storage had similarly no impact on EPS content (Supplementary Discussion 2)” (Page 9, Lines 253-258).

15) RESPONSE TO R4-45 (same for R1-22) Delete “intermittent” and then delete the following overreach/speculation which is presented as fact: *“Overall, bedrock influenced EPS through (1) promoting microbial biomass and EPS secretion via bedrock-driven environmental conditions, and (2) shaping soil texture and mineralogy, which governs EPS retention and intermittently protects it from microbial degradation”*. You don’t have such granularity of data.

16) EDIT R4-72: *“EPS is a transient pool that is selectively secreted by microorganisms in response to adverse environmental conditions, with no measurable legacy effects observed beyond 2.5 years²⁸*. This is wrong, and is being taken out of context. EPS were not measured at the same timescales as necromass, - so the comparison with necromass is misleading. Also, the referenced study showed most EPS under grass. Not stressful or “adverse” conditions. **CHANGE TO** *“EPS is a transient pool that is selectively secreted by microorganisms where sufficient labile C is available, with the majority of EPS either stabilising, or being turned over in less than 2.5 years²⁸*

12) Update response to **R4-77, R4-78**, these relate to **Supplementary Discussion (suggested delete)**

Good responses to R4

R4-79: This is a fair approach: *“Before extraction, we also calculated the required amount of CER for each soil sample based on its SOC content following the method proposed by Redmile-Gordon et al.³⁶. These calculations showed that 10 g CER per 1 g dry soil provided sufficient ion-exchange capacity even for the samples with the highest SOC levels, and importantly for soils with the highest exchangeable Ca²⁺ concentrations. Therefore, we applied this amount uniformly across all soils to ensure methodological consistency across SOC-rich and SOC-poor soils and limestone and non-limestone soils” (Page 16, Lines 502-508).*

Complete list of R4 comments

No objections in black. Problematic responses in red are either addressed above, or in PART 2 below (responses to my own comments, R1) R4-80, 79, 78, 77, 76, 75, 74, 73 72, 71, 70, 69, 68, 67, 66, 65, 64, 63, 62, 61, 60, 59, 58, 57, 56, 55, 54, 53, 52, 51, 50, 49, 48, 47,

46, 45, 43, 42, 41, 40, 39, 38, 37, 36, 35, 34, 33, 32, 31, 30, 29, 28, 27, 26, 25, 24, 23, 22, 21, 20, 19, 18, 17, 16, 15, 14, 13, 12, 11, 10, 9, 8, 7, 6, 5, 4, 3, 2, 1.

PART 2: Authors responses to R1

(mostly satisfactory, with some exceptions)

Some changes have not been fully actioned. For example, there are still incorrect statements around the subject of “EPS production efficiency” and the importance of some points has been missed. These can be resolved simply. If the authors are content to accept my suggestions here *-verbatim-* and satisfy all requests of The Editor, then I am happy to recommend acceptance following a rapid final round of review.

R1-1: The changes to **EPS-C/MBC ratio** are noted and appreciated. However, the first response to my first comment ends on “it may serve as a proxy for the *relative allocation of microbial C assimilates to extracellular non-growth products under steady state conditions*” (**Pages 10, Lines 283-287**). This is not satisfactory. The authors have gone ‘full- circle’ in some cases describing an “EPS production efficiency” in different words. The steps to bear in mind (from historic, to time of sampling) are:

EPS production/allocation > soil retention/stabilisation/degradation > **observed EPS**.

Therefore, one cannot quantify of any of the previous steps here from the last using this dataset. Accordingly, **Lines 283-287: Please Change** “*Although this ratio does not directly reflect EPS production efficiency - since EPS contents in natural soils result from the balance of microbial production and degradation, modulated by mineral interactions and extraction limitations - it may serve as a proxy for the relative allocation of microbial C assimilates to extracellular non-growth products under steady state conditions*” **TO:** *In our study, this ratio does not directly reflect EPS production efficiency - since EPS contents in natural soils result from the balance of microbial production and degradation, modulated by mineral interactions and extraction limitations*”.

Deleting “it may serve as a proxy for the relative allocation of microbial C assimilates to extracellular non-growth products under steady state conditions” because EPS allocation precedes other factors affecting measurements. (‘Thought experiment’ for consideration: if a soil type causes disintegration of EPS, it might appear as if EPS were simply not produced; that would not be discernable from this data).

Please Change: *“Changes in short-term microbial EPS production efficiency were previously calculated by standardizing soil EPS to soil microbial ATP², a proxy for microbial biomass. This EPS-C/MBC ratio may be used to assess the potential of microorganisms to secrete EPS under various environmental conditions². However, as mentioned above, the accumulation of soil EPS is contingent on the dynamic balance between its generation, stabilization, and degradation” (Page 12, Lines 358-362).* **TO:** *“Changes in short-term microbial EPS production efficiency were previously calculated by standardizing soil EPS to soil microbial ATP², as an independent indicator of the soil microbial biomass. The EPS-C/MBC ratio of soils can also be used, and in natural soils, represents net result of all historic influences contributing to the current state (whether steady or dynamic). As mentioned above, the accumulation of soil EPS is contingent on the dynamic balance between its generation, stabilization, and degradation” (Page 12, Lines 358-362).*

R1-2 – The authors have missed the point. EPS are not “poorly quantified” in all senses. I requested to replace this sentence with *“However, systematic evaluation of this claim has yet to be reported”* – which is far more informative, and far less wrong. Instead of making the recommended improvement, the authors made the original problem worse. Please do as asked.

R1-3 – the resistance on the correction is continued. Please action R1-3 as asked.

R1-4 - OK

R1-5 – Not satisfied. Confounded logic and clinging to concepts not supported by the data: *“EPS production potential”* suffers from similar limitations to *“EPS production efficiency”* in this context. *“While soil properties such as clay content and exchangeable calcium enhanced EPS retention”* Again, how do the authors think there is evidence for this level of detail and confidence (rhetorical). *I request the authors desist from unsubstantiated speculation in all further correspondence.*

R1-6 – *“EPS increased with microbial biomass as a proxy for EPS production potential?”* No. MBC is MBC. And we know nothing of the *“EPS production potential”*. Please refer to my original request.

R1-7 Great – OK.

R1-8 Good

R1-9 Good

R1-10 Good

R1-11 Good

R1-12 Good

R1-13 Good

R1-14 Good

R1-15 Good

R1-16 Good

R1-17 Good

R1-18 Good

R1-19 Good

R1-20 Good

R1-21 A good statement has been removed without need, but, OK.

R1-22A: Refer to my requested change for R1-1 AND **delete** “intermittent” later in the paragraph, and then **delete** the following imagined overreach/speculation which is presented as fact: “Overall, bedrock influenced EPS through (1) promoting microbial biomass and EPS secretion via bedrock-driven environmental conditions, and (2) shaping soil texture and mineralogy, which governs EPS retention and intermittently protects it from microbial degradation”. You simply don’t have such granularity of data. **(Pages 9-11, Lines 259-307).**

R1-22B: *“In culture experiments, it is also the loosely bound fraction of EPS that is extracted by CER, whereas the tightly bound fraction remains unextracted”.* Where is the reference for this? and what is your definition of “loosely” and “tightly bound”? What value is this subjective definition? Why assume culture findings apply to soils? What about other multivalent metals, why assume some EPS are not occluded in aggregates when this has not been investigated? I strongly recommend to delete the above statement at the end of the paragraph - and leave it to The Editors discretion as to whether **Supplementary Discussion 3** should be included at all. In the very least, the language should be less confident. Eg “EPS in soils is **thought** stabilized via multiple mechanisms”, then adding appropriate **“thoughts”** and **“claimed”** to each of the existing statements, ensuring a reference is given to support each - *without* adding any new statements.

R1 – 23 GOOD

R1 – 24 GOOD

R1 – 25 GOOD

R1 – 26 GOOD

R1 – 27 A) You need to include the reference (16) B) your statement that **the effect of land use on soil EPS was not significant (Fig. 2b)** is in error **See Fig 3d**. This detail: that EPS polysaccharide and EPS protein are related to bedrock, and land-management, respectively, is a key part of the findings! Please reinstate to the main discussion.

R1 – 28 GOOD

R1 – 29 GOOD

R1 - 30 GOOD

R1 - 31 GOOD

R1 - 32 GOOD

R1 - 33 GOOD

R1 – 34 to 51 ALL GOOD

Once the authors have addressed these structural and methodological points, I would be pleased to provide a brief follow-up review to verify that the key corrections have been implemented, rather than line-editing the entire text at this stage.